# On the Optimization and Generalization of Two-layer Transformers with Sign Gradient Descent

**Bingrui Li**[1], **Wei Huang**[2], **Andi Han**[2], **Zhanpeng Zhou**[4],
**Taiji Suzuki**[3,2], **Jun Zhu**[1], **Jianfei Chen**[1]
[1]Dept. of Comp. Sci. and Tech., Institute for AI, BNRist Center, THBI Lab,
Tsinghua-Bosch Joint ML Center, Tsinghua University
[2]RIKEN AIP [3]University of Tokyo [4]Shanghai Jiao Tong University
lbr22@mails.tsinghua.edu.cn; jianfeic@tsinghua.edu.cn

## Abstract

The Adam optimizer is widely used for transformer optimization in practice, which makes understanding the underlying optimization mechanisms an important problem. However, due to the Adam's complexity, theoretical analysis of how it optimizes transformers remains a challenging task. Fortunately, Sign Gradient Descent (SignGD) serves as an effective surrogate for Adam. Despite its simplicity, theoretical understanding of how SignGD optimizes transformers still lags behind. In this work, we study how SignGD optimizes a two-layer transformer – consisting of a softmax attention layer with trainable query-key parameterization followed by a linear layer – on a linearly separable noisy dataset. We identify four stages in the training dynamics, each exhibiting intriguing behaviors. Based on the training dynamics, we prove the fast convergence but poor generalization of the learned transformer on the noisy dataset. We also show that Adam behaves similarly to SignGD in terms of both optimization and generalization in this setting. Additionally, we find that the poor generalization of SignGD is not solely due to data noise, suggesting that both SignGD and Adam requires high-quality data for real-world tasks. Finally, experiments on synthetic and real-world datasets empirically support our theoretical results.

## 1 Introduction

The transformer architecture (Vaswani et al., 2017) has become ubiquitous across various domains, achieving state-of-the-art results in areas such as language modeling (Devlin et al., 2019; Brown et al., 2020), computer vision (Dosovitskiy et al., 2021; Peebles & Xie, 2023), and reinforcement learning (Chen et al., 2021). Regardless of the specific task or data modality, the Adam optimizer (Kingma & Ba, 2015) is typically employed to train large transformer models, making it the *de facto* choice in practice. This widespread use highlights that understanding the inner mechanism on how Adam optimizes transformers is an important problem. However, the complexity of Adam's formulation presents significant challenges for rigorous analysis. Many of the underlying mechanisms of how Adam optimizes transformers are still poorly understood.

Recent theoretical works (Jelassi et al., 2022; Tarzanagh et al., 2023a; Tian et al., 2023) study the *training dynamics* of transformers across various datasets and objectives. *Training dynamics analysis* allows us to trace the evolution of model parameters throughout the training process. In doing so, it enables a precise description of the optimization process, which can ultimately lead to new insights on convergence and generalization results. However, analyzing the training dynamics of transformers presents many challenges. The transformer architecture is inherently more complex than simpler models like MLPs (Wang & Ma, 2023; Xu & Du, 2023) and CNNs (Cao et al., 2022; Kou et al., 2023), making a detailed analysis of its training dynamics more challenging. To facilitate such analyses, researchers often introduced relaxed assumptions, such as using linear attention (Zhang et al., 2024b) or unrealistic initialization (Li et al., 2023c). A more commonly employed assumption in theoretical works is reparameterizing the query and key matrices into a single joint attention matrix, as seen in many studies (e.g., Tian et al. (2023)). While this assumption simplifies the analysis, it remains

unrealistic in practice. Moreover, existing analyses focus primarily on Gradient Descent (GD) or Stochastic Gradient Descent (SGD), with little attention paid to optimizers like Adam. The analysis of transformer training dynamics remains an active area of research.

Our work addresses a crucial gap by analyzing the training dynamics of Sign Gradient Descent (SignGD), which is an effective surrogate for understanding Adam. SignGD is a simple gradient-based algorithm that updates parameters using only the sign of the gradient, discarding the gradient's magnitude. Over the years, SignGD has been extensively studied (Balles & Hennig, 2018; Balles et al., 2020; Bernstein et al., 2018; 2019), and has inspired the development of optimizers like Adam (Kingma & Ba, 2015) and Lion (Chen et al., 2023). More importantly, SignGD shares many similarities with Adam, making it an effective proxy for gaining insights into Adam's optimization behavior (Balles & Hennig, 2018; Bernstein et al., 2018; Kunstner et al., 2023; 2024; Wu et al., 2020; Zou et al., 2023). For example, Kunstner et al. (2023) has shown that while a performance gap between Adam and GD persists in the full-batch setting on transformers, SignGD can effectively bridge this gap, achieving performance closer to Adam. Despite its simplicity, however, theoretical understanding of how SignGD optimizes transformers remains an open problem.

**Our contributions.** In this work, we study the problem of how SignGD optimizes transformers in a binary classification task with linearly separable datasets with signal and noise.

- We provide a theoretical characterization of the entire training dynamics of SignGD. Specifically, we identify four different stages in the training dynamics, each exhibiting unique behaviors, as summarized in Tab. 1. This detailed four-stage analysis captures the complex yet systematic dynamics within the attention layer, and offers a precise description of how SignGD optimizes transformers in our setting.

- Based on the training dynamics, we prove the convergence and generalization results. On our noisy dataset, SignGD demonstrates *fast convergence but poor generalization*, achieving a linear convergence rate in training loss but maintaining a high constant test loss, leading to a sparse attention matrix through noise memorization. Additionally, we provide evidence that Adam exhibits similar behaviors to SignGD in terms of training dynamics, convergence, and generalization, suggesting that SignGD is a strong proxy for understanding Adam. We also find that the poor generalization of SignGD is not solely due to data noise, but is also related to its inherent algorithmic properties, indicating that SignGD and Adam require higher data quality in practice compared to GD. Our results and findings are further validated through experiments on both synthetic and real-world datasets.

Table 1: Overview of the four-stage dynamics: corresponding behaviors and theoretical results.

| Stage I | *The mean value noise shifts early, then stabilizes.* | **Lemma 4.1** |
|---|---|---|
| Stage II | *The query & key noise align their sign to each other.* | **Lemma 4.2, 4.3** |
| Stage III | *Majority voting determines the sign of query & key signals.* | **Lemma 4.4** |
| Stage IV | *The noise-signal softmax outputs decay fast exponentially,* | **Lemma 4.5** |
| | *then the query & key noise align their sign to signals.* | **Lemma 4.6, 4.7** |

**Technical novelties.** We use the *feature learning* framework (Allen-Zhu & Li, 2023; Cao et al., 2022; Zou et al., 2023; Huang et al., 2023a) for our theoretical analysis. Our technical novelties include: Firstly, we analyze an softmax attention layer with trainable query-key parameterization, which is not carefully studied in the literature. Secondly, we perform a multi-stage analysis for transformers by breaking down the complex dynamics into simple sub-stages. In each sub-stage, only one or two key behaviors dominate. Finally, we cleverly combined SignGD and the sparse data model, greatly simplifying the analysis.

In summary, our work investigates the training dynamics of transformers using SignGD. To the best of our knowledge, this is the first provable result characterizing the training dynamics of transformers with SignGD. Our findings offer valuable insights into the inner workings of both SignGD and Adam, advancing our theoretical understanding of transformers and their optimization.

## 2  PRELIMINARIES

**Notations.** We use lower case letters, lower case bold face letters, and upper case bold face letters to denote scalars, vectors, and matrices respectively. For a vector $\mathbf{v} = [v_1, \ldots, v_d]^\top$, we denote the $\ell_2$ and $\ell_1$ norm by $\|\mathbf{v}\|$ and $\|\mathbf{v}\|_1$, respectively. For two fixed non-negative sequences $\{x_n\}$

and $\{y_n\}$, we denote $x_n = O(y_n)$ if there exist some absolute constant $C > 0$ and $N > 0$ such that $|x_n| \leq C |y_n|$ for all $n \geq N$. We say $x_n = \Omega(y_n)$ if $y_n = O(x_n)$, and say $x_n = \Theta(y_n)$ if $x_n = O(y_n)$ and $x_n = \Omega(y_n)$. We use $\tilde{O}(\cdot)$, $\tilde{\Omega}(\cdot)$ and $\tilde{\Theta}(\cdot)$ to hide logarithmic factors in these notations, respectively. Moreover, we denote $x_n = \text{poly}(y_n)$ if $x_n = O(y_n^D)$ for some constant $D > 0$, and $x_n = \text{polylog}(y_n)$ if $x_n = \text{poly}(\log(y_n))$. We use $[d]$ to denote the set $\{1, 2, \ldots, d\}$. We use $\text{sgn}(x) = x/|x|$ when $x \neq 0$ and $\text{sgn}(0) = 0$. We denote a $n$-dim all-ones vector by $\mathbf{1}_n$.

**Data model.** We consider a binary classification task where each data point contains signal vector and sparse noise vector. The data model is formally defined in Definition 2.1.

**Definition 2.1.** *Let $\boldsymbol{\mu} \in \mathbb{R}^d$ be a fixed vector representing the signal contained in each data point. We assume $\boldsymbol{\mu} = [1, 0, \ldots, 0]^\top$. For each data point $(\mathbf{X}, y)$, the predictor $\mathbf{X} = [\mathbf{x}^{(1)}, \mathbf{x}^{(2)}] \in \mathbb{R}^{d \times 2}$ consists of two patches (or tokens, vectors), where $\mathbf{x}^{(1)}, \mathbf{x}^{(2)} \in \mathbb{R}^d$, and the label $y$ is binary, i.e., $y \in \{\pm 1\}$. The data is generated from a distribution $\mathcal{D}$, which we specify as follows:*

1. *The label $y$ is generated as a Rademacher random variable.*
2. *Randomly select $s$ coordinates from $[d] \setminus \{1\}$ uniformly, denoted as a vector $\mathbf{s} \in \{0, 1\}^d$. Generate each coordinate in $\boldsymbol{\xi}$ from distribution $N(0, \sigma_p^2)$, and then mask off the first coordinate and other $d - s - 1$ coordinates, i.e., $\boldsymbol{\xi} = \boldsymbol{\xi} \odot \mathbf{s}$.*
3. *One of $\mathbf{x}^{(1)}, \mathbf{x}^{(2)}$ is randomly selected and then assigned as $y\boldsymbol{\mu}$, representing the signal, while the other is designated as $\boldsymbol{\xi}$, representing noise.*

The design of the signal patch $y\boldsymbol{\mu}$ and the noise patch $\boldsymbol{\xi}$ can be viewed as a simplification of real-world image classification problems where only certain patches contain useful features that are correlated with the label, e.g., the wheel of a car, while many other patches contain uninformative features or consist solely of noise, e.g., the background of the image. Specifically, $y\boldsymbol{\mu}$ represents useful, label-correlated features (referred to as signal), whereas $\boldsymbol{\xi}$ represents non-informative features or irrelevant noise (referred to as noise).

**Remarks on data assumptions.** We make several assumptions regarding sparsity, orthogonality, and context length. Specifically, we assume $\boldsymbol{\mu}$ is 1-sparse (i.e., it has only one non-zero entry), $\boldsymbol{\xi}$ is $s$-sparse, and that $\boldsymbol{\mu}$ and $\boldsymbol{\xi}$ are orthogonal. The sparsity assumption is essential for analysing optimizers that are not invariant under orthogonal transformations, such as Adam and SignGD. Our results can be easily extended to any $C$-sparse signal vector $\boldsymbol{\mu}$, where $C = O(1)$ is a constant, with non-zero entries in arbitrary positions and constant magnitude. The orthogonality assumption holds with high probability under the sparsity assumption, which confirms its validity (see Lemma C.2 for details). We also assume a context length of $L = 2$ for technical simplification. With additional appropriate assumptions, our analysis can be extended to data with longer contexts (see discussion in Appendix F.2). We empirically validate our theoretical results for non-sparse, non-orthogonal, and multi-patch data in Appendix B. Data models comprised of signal and noise patches with similar assumptions have also been studied in recent works (Allen-Zhu & Li, 2023; Cao et al., 2022; Jelassi et al., 2022; Zou et al., 2023; Huang et al., 2023b; Han et al., 2024; Huang et al., 2024a).

**Two-layer transformers.** Motivated by vision transformers (Dosovitskiy et al., 2021), we consider a two-layer transformer, where the first layer is a single-head softmax attention layer and the second layer is a linear head layer. The attention layer is a sequence-to-sequence mapping, of which the parameters are $\mathbf{W} := (\mathbf{W}_Q, \mathbf{W}_K, \mathbf{W}_{V,j})$, where $\mathbf{W}_Q, \mathbf{W}_K \in \mathbb{R}^{m_k \times d}$ and $\mathbf{W}_{V,j} \in \mathbb{R}^{m_v \times d}$ for $j \in \{\pm 1\}$. The parameters of the second layer are fixed as $1/m_v$ and $-1/m_v$ respectively. We also talk about learnable linear head in Appendix F.3. Then, the network can be written as $f(\mathbf{W}, \mathbf{X}) := F_1(\mathbf{W}, \mathbf{X}) - F_{-1}(\mathbf{W}, \mathbf{X})$, where $F_1(\mathbf{W}, \mathbf{X})$ and $F_{-1}(\mathbf{W}, \mathbf{X})$ are defined as:

$$F_j(\mathbf{W}, \mathbf{X}) := \frac{1}{m_v} \sum_{l=1}^{L} \mathbf{1}_{m_v}^\top \mathbf{W}_{V,j} \mathbf{X} \text{softmax}\left(\mathbf{X}^\top \mathbf{W}_K^\top \mathbf{W}_Q \mathbf{x}^{(l)}\right).$$

Let $\mathbf{w}_{Q,s} := \mathbf{W}_{Q,(\cdot,s)}^\top \in \mathbb{R}^d$, $\mathbf{w}_{K,s} := \mathbf{W}_{K,(\cdot,s)}^\top \in \mathbb{R}^d$, $\mathbf{w}_{V,j,r} := \mathbf{W}_{V,j,(\cdot,r)}^\top \in \mathbb{R}^d$ be the $s$-th or $r$-th row of the parameter $\mathbf{W}_Q, \mathbf{W}_K, \mathbf{W}_{V,j}$, respectively. Let $\bar{\mathbf{w}}_{V,j} := \sum_{r \in [m_v]} \mathbf{w}_{V,j,r}/m_v$ be the *mean value in $F_j$*. Let $\mathbf{v} = \bar{\mathbf{w}}_{V,1} - \bar{\mathbf{w}}_{V,-1}$ be the *mean value*. We can write the model in a simpler form:

$$F_j(\mathbf{W}, \mathbf{X}) = \frac{1}{m_v} \sum_{r \in [m_v]} \left[(s_{11} + s_{21})\left\langle \mathbf{w}_{V,j,r}, \mathbf{x}^{(1)}\right\rangle + (s_{12} + s_{22})\left\langle \mathbf{w}_{V,j,r}, \mathbf{x}^{(2)}\right\rangle\right], \quad (1)$$

where $s_{la} := \text{softmax}\,(z_{l1}, \ldots z_{lL})_a$, and $z_{la} := \sum_{s \in [m_k]} \langle \mathbf{w}_{Q,s}, \mathbf{x}^{(l)} \rangle \langle \mathbf{w}_{K,s}, \mathbf{x}^{(a)} \rangle$. Unless otherwise specified, we set $L = 2$.

We refer to the softmax outputs with the noise vector as the query and the signal vector as the key as noise-signal softmax outputs. Formally, for data point $(\mathbf{X}_i, y_i)$, it is defined as

$$s_{i,21}^{(t)} = \frac{\exp\left(\sum_{s \in [m_k]} \langle \mathbf{w}_{Q,s}^{(t)}, \boldsymbol{\xi}_i \rangle \langle \mathbf{w}_{K,s}^{(t)}, y_i \boldsymbol{\mu} \rangle\right)}{\exp\left(\sum_{s \in [m_k]} \langle \mathbf{w}_{Q,s}^{(t)}, \boldsymbol{\xi}_i \rangle \langle \mathbf{w}_{K,s}^{(t)}, y_i \boldsymbol{\mu} \rangle\right) + \exp\left(\sum_{s \in [m_k]} \langle \mathbf{w}_{Q,s}^{(t)}, \boldsymbol{\xi}_i \rangle \langle \mathbf{w}_{K,s}^{(t)}, \boldsymbol{\xi}_i \rangle\right)}.$$

Similarly, we refer to $s_{i,11}^{(t)}, s_{i,12}^{(t)}, s_{i,22}^{(t)}$ as signal-signal, signal-noise, and noise-noise softmax outputs, respectively. When $L = 2$, a key fact about the softmax function is that $s_{i,l1}^{(t)} + s_{i,l2}^{(t)} \equiv 1$, for $l \in [2]$. The subscript is used only to distinguish between signal and noise, without imposing any restrictions on the permutation of patches. Although we use the symbol $s$ for sparsity, softmax outputs, and the indices of query and key neurons simultaneously, the context clearly indicates which one is being referred to.

**Training algorithm.** We train our transformer model by minimizing the empirical cross-entropy loss function $L_S(\mathbf{W}) := \frac{1}{n} \sum_{i=1}^{n} \ell(y_i \cdot f(\mathbf{W}, \mathbf{X}_i))$, where $\ell(x) := \log\,(1 + \exp(-x))$ is the logistic loss function, and $S := \{(\mathbf{X}_i, y_i)\}_{i=1}^{n}$ is the training dataset. We further define the test loss $L_{\mathcal{D}}(\mathbf{W}) := \mathbb{E}_{(\mathbf{X}, y) \sim \mathcal{D}}[\ell(y \cdot f(\mathbf{W}, \mathbf{X}))]$.

We study SignGD starting from Gaussian initialization, where each entry of $\mathbf{W}_Q, \mathbf{W}_K, \mathbf{W}_{V,j}$ for $j \in \{\pm 1\}$ is sampled from a Gaussian distribution $N(0, \sigma_0^2)$. The SignGD update for the parameters can be written as

$$\mathbf{w}_{V,j,r}^{(t+1)} = \mathbf{w}_{V,j,r}^{(t)} - \eta \, \text{sgn}(\nabla_{\mathbf{w}_{V,j,r}} L_S(\mathbf{W}^{(t)})),$$

$$\mathbf{w}_{Q,s}^{(t+1)} = \mathbf{w}_{Q,s}^{(t)} - \eta \, \text{sgn}(\nabla_{\mathbf{w}_{Q,s}} L_S(\mathbf{W}^{(t)})), \quad \mathbf{w}_{K,s}^{(t+1)} = \mathbf{w}_{K,s}^{(t)} - \eta \, \text{sgn}(\nabla_{\mathbf{w}_{K,s}} L_S(\mathbf{W}^{(t)})),$$

for all $s \in [m_k]$, $j \in \{\pm 1\}$ and $r \in [m_v]$. The expanded gradient formulas and update rules can be seen in Appendix D.

## 3 MAIN RESULTS

In this section, we present our main results. Firstly, we provide a detailed characterization on training dynamics with a four-stage analysis, each exhibiting different behaviors. Then, based on the training dynamics, we analyze the convergence and generalization at the end of the training. We further give an evidence that Adam exhibits similar behaviors to SignGD, and provide new insights that SignGD and Adam requires higher data quality compared to GD.

Before presenting the main results, we state our main condition. All of our theoretical results are based on the following condition.

**Condition 3.1.** *Suppose that*

1. **[Data dimension and Sparsity]** *Data dimension $d$ is sufficiently large with $d = \Omega(\text{poly}(n))$. Sparsity $s$ satisfies: $s = \Theta(d^{1/2} n^{-2})$.*
2. **[Noise strength]** *The standard variance of noise $\sigma_p$ satisfies: $\sigma_p = \Omega(d^{-1/4} n^3)$.*
3. **[Network width and initialization]** *Network width of value $m_v$ and of query & key $m_k$ satisfy: $m_k, m_v = \Omega(\text{polylog}(d))$. Network initialization $\sigma_0$ satisfies: $\sigma_0 = o(\sigma_p^{-1} s^{-1} m_k^{-1/2})$.*
4. **[Training dataset size]** *The training sample size $n$ satisfies: $n = \Omega(m_k^4)$.*
5. **[Learning rate]** *The learning rate $\eta$ satisfies: $\eta = O(\text{poly}(d^{-1}))$ is sufficiently small.*

**Remarks on Condition 3.1.** Our Condition 3.1 is frequently used in the literature and realistic in practice (Cao et al., 2022; Chatterji & Long, 2021; Frei et al., 2022). The conditions on $d$ and $s$ make sure the different noise patches have disjoint support with high probability. The condition on $\sigma_p$ implies $\sigma_p \sqrt{s} = \Omega(n^2 \|\boldsymbol{\mu}\|)$, which indicates the noise in the dataset is strong. The conditions on $m_v$, $m_k$, $n$, $\eta$ are technical and mild. $m_v$ and $m_k$ affect the convergence of mean value noise and softmax outputs, respectively. The size of $n$ affects the concentration of $\|\boldsymbol{\xi}\|_1$. Finally, the conditions on $\sigma_0$ ensures the network weights are small enough at initialization, which makes the learning process fall into the *feature learning* regime. Note if we set $m_k = \Theta(\text{polylog}(d))$, then our condition become $\sigma_0 = \Theta(d^{-1/2})$, which is realistic in practice.

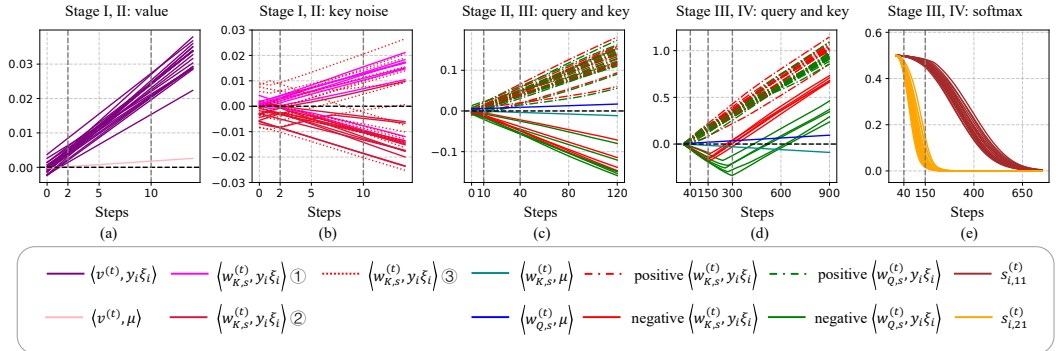

Figure 1: **The training dynamics of two-layer transformers with SignGD. (a)** Dynamics of mean value noise and mean value signals in Stage I, and II. **(b)** Dynamics of key noise in Stage I, and II. We mark different key noise in different colors. ①: $\mathbf{w}_{K,s}^{(t)} \in S_{K+,Q+}^{(0)} := \{\mathbf{w}_{K,s}^{(t)} : \langle \mathbf{w}_{K,s}^{(0)}, y_i\boldsymbol{\xi}_i \rangle > 0, \langle \mathbf{w}_{Q,s}^{(0)} y_i\boldsymbol{\xi}_i \rangle > 0\}$. ②: $\mathbf{w}_{K,s}^{(t)} \in S_{K-,Q-}^{(0)} := \{\mathbf{w}_{K,s}^{(t)} : \langle \mathbf{w}_{K,s}^{(0)}, y_i\boldsymbol{\xi}_i \rangle < 0, \langle \mathbf{w}_{Q,s}^{(0)} y_i\boldsymbol{\xi}_i \rangle < 0\}$. ③: $\mathbf{w}_{K,s}^{(t)} \in (S_{K+,Q+}^{(0)} \cup S_{K-,Q-}^{(0)})^c$. **(c)** Dynamics of query noise, key noise, query signals, key signals in Stage II and III. The dotted lines represent positive (query and key) noise at $t = 40$, and the solid lines represent negative noise at the same point. **(d)** Dynamics of query noise, key noise, query signals, key signals in Stage III and IV. The dotted lines and solid lines have the same meanings in (c). **(e)** Dynamics of softmax outputs in four stages. The dynamics over the whole time horizon is provided in Fig. 18. An illustration explaining the behaviors of all quantities in all stages is provided in Fig. 19.

## 3.1 TRAINING DYNAMICS ANALYSIS: FOUR-STAGE ANALYSIS

Based on Condition 3.1, we aim to explore the underlying optimization mechanism of SignGD through training dynamics. We identify four distinct stages in the training dynamics, where the primary behaviors and theoretical results for each stage are summarized in Tab. 1. This four-stage analysis captures the complex yet systematic dynamics within the attention layer and provides a valuable tool for further analysis of convergence and generalization. In this subsection, we informally describe the core phenomena in each stage, while the formal results are presented in Sec. 4.

To better understand the four-stage training dynamics, the dynamics of key quantities at key timesteps or during the entire dynamics are illustrated in the Fig. 1 and Tab. 2.

From $t = 0$ to $t = 2$ is the **Stage I** of the dynamics, which shows the early shift and stabilization of *mean value noise*, i.e., $\langle \mathbf{v}^{(t)}, y_i\boldsymbol{\xi}_i \rangle$. The mean value noise increases monotonically from the random initialization and becomes positive, then stabilizes into a linear relationship with $t$ by the stage's end (Fig. 1 (a)). This period is so rapid that other quantities, including value signals and query & key noise all remain close to their initialization (Fig. 1 (b) illustrates key noise as an example). This linear behavior of mean value noise make it ordered in the gradients of query and key noise.

From $t = 2$ to $t = 10$ is the **Stage II** of the dynamics, which illustrates the sign alignment between *query & key noise*, i.e., $\langle \mathbf{w}_{Q,s}^{(t)}, y_i\boldsymbol{\xi}_i \rangle$ and $\langle \mathbf{w}_{K,s}^{(t)}, y_i\boldsymbol{\xi}_i \rangle$. At initialization, the signs of the query & key noise are independent. By the end of Stage II, however, the signs of the noise for each neuron align, becoming either jointly positive or jointly negative, and continuous to grow subsequently. Additionally, the number of positive and negative neurons is nearly equal. Fig. 1 (b) shows how key noise aligns their signs. Tab. 2 provides statistics on the signs of query and key noise at initialization and at the end of Stage II ($t = 10$).

From $t = 10$ to $t = 40$ is the **Stage III** of the dynamics, which shows how the sign of *query & key signals*, i.e. $\langle \mathbf{w}_{Q,s}^{(t)}, \boldsymbol{\mu} \rangle$ and $\langle \mathbf{w}_{K,s}^{(t)}, \boldsymbol{\mu} \rangle$, are determined by majority voting. Before Stage III, query and key signals remain close to the initialization, and their gradients are disordered. However, at the start of Stage III, the sum of key noise $\sum_{i=1}^n \langle \mathbf{w}_{K,s}^{(t)}, y_i\boldsymbol{\xi}_i \rangle$ dominates the gradients of query signals, making the update direction of query signals aligned with its sign. For a given neuron $s$, key noise is nearly uniform across all samples, thus this mechanism effectively act as majority voting. The sign of the key signals is determined symmetrically by the sum of the query noise, with an opposite sign to the query signals. The behaviors in Stage III are shown in Fig. 1 (c).

Table 2: Sign alignment between query and key noise in Stage II. $S_{K+,Q+}^{(t)}$, defined as $S_{K+,Q+}^{(t)} := \{(s,i) \in [m_k] \times [n] : \langle \mathbf{w}_{K,s}^{(t)}, y_i \boldsymbol{\xi}_i \rangle > 0, \langle \mathbf{w}_{Q,s}^{(t)}, y_i \boldsymbol{\xi}_i \rangle > 0\}$, represents the number of neurons and samples having positive query noise and positive key noise. The definitions for $S_{K+,Q-}^{(t)}$, $S_{K-,Q+}^{(t)}$, $S_{K-,Q-}^{(t)}$ are similar. Each element in the middle of the table represents the size of the intersection of the corresponding row set and the corresponding column set. For example, $|S_{K+,Q+}^{(0)} \cap S_{K+,Q+}^{(t)}| = 486$. The signs of query and key noise are independent at initialization but aligned at $t = 10$, which can be seen as an estimate of $T_2^{\text{SGN}}$.

| init($t=0$)\$t=10$ | $\lvert S_{K+,Q+}^{(t)}\rvert$ | $\lvert S_{K+,Q-}^{(t)}\rvert$ | $\lvert S_{K-,Q+}^{(t)}\rvert$ | $\lvert S_{K-,Q-}^{(t)}\rvert$ | Row sum |
|---|---|---|---|---|---|
| $\lvert S_{K+,Q+}^{(0)}\rvert$ | 486 | 1 | 0 | 25 | 512 |
| $\lvert S_{K+,Q-}^{(0)}\rvert$ | 244 | 4 | 9 | 250 | 507 |
| $\lvert S_{K-,Q+}^{(0)}\rvert$ | 223 | 10 | 4 | 221 | 458 |
| $\lvert S_{K-,Q-}^{(0)}\rvert$ | 37 | 2 | 3 | 481 | 523 |
| Column sum | 990 | 17 | 16 | 977 | 2000 |

From $t = 40$ to $t = 2000$ is the **Stage IV** of the dynamics. During this stage, *noise-signal softmax outputs*, i.e. $s_{i,21}^{(t)}$ decay exponentially fast in Stage IV. The duration from Stage I to Stage III can be relatively short, with all softmax outputs $s_{i,11}^{(t)}$, $s_{i,21}^{(t)}$ concentrated at $1/2$. However, in Stage IV, $s_{i,21}^{(t)}$ decreases *exponentially* to zero, while $s_{i,11}^{(t)}$ remains stuck at $1/2$. The concentration and fast decay are shown in Fig. 1 (e). The difference between $s_{i,11}^{(t)}$ and $s_{i,21}^{(t)}$ is due to the varying rates of increase between query signals and query noise.

Furthermore, "negative"[*] query and key noise align their sign to signals in Stage IV. We focus on a single neuron with a positive query signal, as shown in Fig.1 (d), though this applies to all neurons. Before the noise-signal softmax outputs decay to zero, the dynamics of all signals and noise remain unchanged. At a critical point ($t = 150$ in Fig.1 (d)), all negative key noise begins aligning with the positive query signal, gradually decreasing in magnitude and crossing zero. Once the negative key noise approaches zero and fully aligns (around $t = 300$ in Fig.1 (d)), the negative query noise begins aligning, eventually becoming positive by the end of the stage. From that point on, the sign of all signals and noise remains unchanged. The sign alignment of negative key and query noise is shown in Fig. 1 (d).

Overall, the dynamics exhibit complex and intriguing sign alignment behaviors within the attention layer, and they ultimately reach stabilization.

## 3.2 CONVERGENCE AND GENERALIZATION ANALYSIS: FAST CONVERGENCE BUT POOR GENERALIZATION

Beyond the training dynamics, we characterize the convergence and generalization result at the end of the training. Additionally, we provide evidence that Adam exhibits similar behaviors to SignGD in optimization and generalization, and suggest that SignGD and Adam requires high data quality.

**Theorem 3.2.** *For any $\epsilon > 0$, under Condition 3.1, with probability at least $1 - n^{-1/3}$, there exists $T = O(\log(\epsilon^{-1})\eta^{-1}\sigma_p^{-1}s^{-1})$, and $T_{\text{attn}} = \tilde{O}(\eta^{-1}m_k^{-1/2}\sigma_p^{-1/2}s^{-1/2}\|\boldsymbol{\mu}\|^{-1/2})$ such that*

1. **[Training loss]** *The training loss converges to $\epsilon$: $L_S(\mathbf{W}^{(T)}) \leq \epsilon$.*
2. **[Test loss]** *The trained transformer has a constant order test loss: $L_{\mathcal{D}}(\mathbf{W}^{(T)}) = \Theta(1)$.*
3. **[Noise memorization of value]** *The value matrix in attention layer memorizes noises in the training data: For all $i \in [n]$, $|\langle \mathbf{v}^{(T)}, \boldsymbol{\xi}_i \rangle| = \Omega(1)$, $|\langle \mathbf{v}^{(T)}, \boldsymbol{\mu} \rangle| = \tilde{O}(\sigma_p^{-1}s^{-1})$.*
4. **[Noise memorization of query & key]** *The softmax outputs of attention layer attends all the weights to the noise patch in the training data: For all $i \in [n]$, $s_{i,11}^{(T_{\text{attn}})} = o(1)$, $s_{i,21}^{(T_{\text{attn}})} = o(1)$.*

Theorem 3.2 outlines the *training and test loss* at the end of training. In this setting, SignGD achieves a fast linear convergence rate in training loss, but test loss remains high, summarizing the behavior as *fast convergence but poor generalization*. Theorem 3.2 also presents new results on *noise*

---

[*]To be more accurate, the "negative" key noise means all the key noise with the sign opposite of query signals. When we focus on a neuron with a positive query signal, the "negative" key noise is exactly negative key noise.

*memorization in the attention layer.* The post-softmax attention matrix concentrates all its weights on the noise patch, resulting in a sparse matrix, which is consistent with previous analyses (Tian et al., 2023; 2024; Li et al., 2023c). In contrast to prior works (Cao et al., 2022; Zou et al., 2023) that focus on noise memorization in linear or convolutional layers, our results address the more complex issue of noise memorization in the attention layer. The proof of Theorem 3.2 is in Appendix E.8.

**Remark.** Theorem 3.2 focus on the logistic test loss. We further give a result about final 0-1 test loss in Appendix F.1 since bad logistic loss doesn't necessarily imply bad 0-1 loss in binary classification task. Interestingly, the size of 0-1 test loss depends on the network initialization $\sigma_0$.

**Fast convergence but poor generalization contradcits the 'train faster, generalize better' argument.** Algorithmic stability (Bousquet & Elisseeff, 2002; Hardt et al., 2016) is a widely used technique in generalization analysis. One typical argument of algorithmic stability is "train faster, generalize better" (Hardt et al., 2016). Our results provide a counterexample to this argument by showing SignGD trains fast but generalizes poorly. Notably, Teng et al. (2023) introduced a measure explaining why SGD trains slower but generalizes better than GD, aligning with our viewpoint.

**Adam exhibits similar behaviors to SignGD in optimization and generalization.** We conducted experiments with Adam on both synthetic and real-world datasets, tracing the dynamics on synthetic data and measuring test loss on noisy MNIST dataset. On the synthetic data, Adam follows a similar four-stage dynamics as SignGD, with Stage III and Stage IV shown in Fig. 2 (a),(b). Further similarities in other stages, and the results across different $\beta_1$ values are given in Appendix B.3. In the noisy MNIST data, Adam also shows high test loss, like SignGD, especially under strong noise. This indicates that Adam shares key similarities with SignGD in training dynamics, convergence, and generalization, further supporting the use of SignGD as a proxy for understanding Adam. However, when $\beta_1$ and $\beta_2$ in Adam are close to 1, which is commonly used in practice, SignGD does not always behave like Adam. In our experiments, Adam did not exhibit the sign alignment of query noise with query signals in Stage IV. We suspect this difference is due to the momentum in Adam.

**SignGD and Adam require higher data quality than GD.** We compare SignGD and GD from both theoretical and empirical perspectives. Theoretically, our results reveal that SignGD achieves a linear convergence rate, while GD typically converges more slowly with a sublinear rate in learning CNNs for the same task (Cao et al., 2022; Kou et al., 2023) or transformers for different tasks (Nichani et al., 2024; Huang et al., 2024b). Empirically, our experiments demonstrate that SignGD trains faster than GD (Fig. 2 (c)), but GD generalizes better consistently across different levels of data noise (Fig. 2 (d)). Furthermore, our experiments show that Adam, like SignGD, is also sensitive to data noise, underperforming in generalization compared to GD in noisy conditions. This indicates that GD is better at learning true useful features from noisy data, while SignGD and Adam are more sensitive to noise. Therefore, *the poor generalization of SignGD, as indicated by our theoretical results, is not solely due to data noise but is also related to its inherent algorithmic properties.* This highlights that both SignGD and Adam require higher data quality than GD. We recommend that practitioners using SignGD or Adam as optimizers consider improving data quality to mitigate their sensitivity to noise. Experiment details and additional results can be found in Appendix B.

In summary, we prove that SignGD achieves fast convergence but poor generalization on a noisy dataset. We provide empirical evidence that Adam exhibits similar behaviors to SignGD in terms of training dynamics, convergence, and generalization, offering new insights into understanding Adam through the lens of SignGD. By comparing it with GD, we find that the poor generalization of SignGD is not solely due to data noise but is related to the inherent properties of SignGD, suggesting that both SignGD and Adam require higher data quality in practice.

## 4 PROOF SKETCH

In this section, we present the proof sketch of the training dynamics, which can subsequently easily lead to the convergence and generalization results.

We use the *feature learning* framework (Allen-Zhu & Li, 2023; Cao et al., 2022), which studies the dynamics of parameter-data inner product. This inner product shows a simpler and clearer pattern compared with parameter itself. Specifically, these quantities are $\langle \mathbf{w}_{Q,s}, \boldsymbol{\mu} \rangle$, $\langle \mathbf{w}_{Q,s}, y_i \boldsymbol{\xi}_i \rangle$, $\langle \mathbf{w}_{K,s}, \boldsymbol{\mu} \rangle$, $\langle \mathbf{w}_{K,s}, y_i \boldsymbol{\xi}_i \rangle$, $\langle \mathbf{w}_{V,j,r}, \boldsymbol{\mu} \rangle$, $\langle \mathbf{w}_{V,j,r}, y_i \boldsymbol{\xi}_i \rangle$, and we name them by query signals, query noise, key signals, key noise,value signals, value noise, respectively. The mean value signals (noise) is the mean of the value signals (noise), denoted by $\langle \mathbf{v}, \boldsymbol{\mu} \rangle$ ($\langle \mathbf{v}, y_i \boldsymbol{\xi}_i \rangle$). Conditional on these quantities, the output $f(\mathbf{W}, \mathbf{X}_i)$ is independent of parameters $\mathbf{W}$.

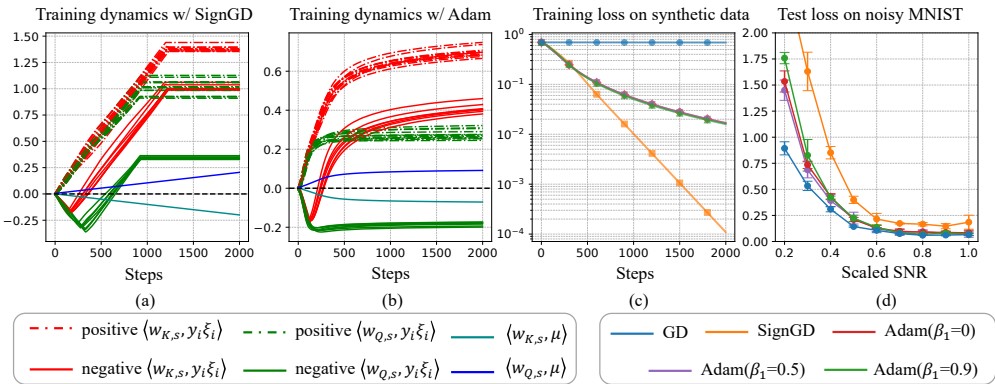

Figure 2: **Comparison of SignGD with Adam and GD on synthetic and real-world datasets. (a)** Dynamics of query noise, key noise, query signals, and key signals with SignGD on the synthetic dataset. **(b)** Dynamics of the same quantities with Adam($\beta_1 = 0.9$). **(c)** Training loss curve (log scale) on the synthetic data for different optimizers. The training loss with SignGD decays exponentially. Note that the training losses for Adam($\beta_1 = 0.9$), Adam($\beta_1 = 0.5$), Adam($\beta_1 = 0.0$) overlap. **(d)** Test loss on the noisy MNIST dataset across varying noise levels. A larger scaled SNR indicates less noise in the dataset.

**Technical Novelties.** We summarize our technical novelties in three points as below.

**Firstly**, we analyze an softmax attention layer with trainable query-key parameterization. In this case, the dynamics of query and key parameters are strongly correlated, and we need to carefully consider the inner interaction between query and key. This is a quite challenging setting, and a detailed comparison with previous works is given in Appendix A.

**Secondly**, we perform a multi-stage analysis for transformers by breaking down the complex dynamics into sub-stages. In each stage, only one or two key behaviors dominate, while other patterns remain mostly unchanged or have minimal impact. This breakdown works because of the varying rates of change in key quantities, which is influenced by the parameters $\sigma_p$, $\sigma_0$, $s$, etc.

**Thirdly**, we cleverly combined SignGD and the sparse data model by observing that the magnitude of the update of any inner products of interest remains constant across all iterations. Formally, for all $s$, $j$, $r$, and $i$, with high probability, we have:

$$\left| \langle \mathbf{w}_{V,j,r}^{(t+1)} - \mathbf{w}_{V,j,r}^{(t)}, \boldsymbol{\mu} \rangle \right|, \left| \langle \mathbf{w}_{K,s}^{(t+1)} - \mathbf{w}_{K,s}^{(t)}, \boldsymbol{\mu} \rangle \right|, \left| \langle \mathbf{w}_{Q,s}^{(t+1)} - \mathbf{w}_{Q,s}^{(t)}, \boldsymbol{\mu} \rangle \right| = \eta \left\| \boldsymbol{\mu} \right\|,$$

$$\left| \langle \mathbf{w}_{V,j,r}^{(t+1)} - \mathbf{w}_{V,j,r}^{(t)}, y_i \boldsymbol{\xi}_i \rangle \right|, \left| \langle \mathbf{w}_{K,s}^{(t+1)} - \mathbf{w}_{K,s}^{(t)}, y_i \boldsymbol{\xi}_i \rangle \right|, \left| \langle \mathbf{w}_{Q,s}^{(t+1)} - \mathbf{w}_{Q,s}^{(t)}, y_i \boldsymbol{\xi}_i \rangle \right| = \eta \left\| \boldsymbol{\xi}_i \right\|_1.$$

This property implies that each iteration's update is determined solely by its sign, allowing us to simplify the analysis by focusing only on the update direction's sign.

In the following four subsections, we present the key theoretical results for each stage, respectively.

### 4.1 STAGE I. MEAN VALUE NOISE EARLY SHIFTS AND STABILIZES.

In **Stage I**, we focus on *mean value noise & signals*, i.e. $\langle \mathbf{v}^{(t)}, y_i \boldsymbol{\xi}_i \rangle$ and $\langle \mathbf{v}^{(t)}, \boldsymbol{\mu} \rangle$. Let $\beta_{\boldsymbol{\xi}} := \max_{i,s,j,r}\{|\langle \mathbf{w}_{Q,s}^{(0)}, \boldsymbol{\xi}_i \rangle|, |\langle \mathbf{w}_{K,s}^{(0)}, \boldsymbol{\xi}_i \rangle|, |\langle \mathbf{w}_{V,j,r}^{(0)}, \boldsymbol{\xi}_i \rangle|\}$ and $T_1 := 4\beta_{\boldsymbol{\xi}} m_v^{-1/2} \eta^{-1} \sigma_p^{-1} s^{-1} = \tilde{\Theta}(\sigma_0 m_v^{-1/2} \eta^{-1} s^{-1/2})$, we call $t \in [0, T_1]$ Stage I. The following lemma is the main result in Stage I.

**Lemma 4.1** (Stage I). *We have (1) (Magnitude). For all $i \in [n]$ and $t \geq T_1$, $\langle \mathbf{v}^{(t)}, y_i \boldsymbol{\xi}_i \rangle = \Theta(t\eta\sigma_p s)$.*
*(2) (Negligibility of mean value signals). For all $i \in [n]$ and $t \geq T_1$, $\langle \mathbf{v}^{(t)}, \boldsymbol{\mu} \rangle = o(\langle \mathbf{v}^{(t)}, y_i \boldsymbol{\xi}_i \rangle)$.*
*(3) (Query & Key noise barely move). For all $s \in [m_k]$, $i \in [n]$ and $t \leq T_1$, $\langle \mathbf{w}_{Q,s}^{(t)}, \boldsymbol{\xi}_i \rangle = \langle \mathbf{w}_{Q,s}^{(0)}, \boldsymbol{\xi}_i \rangle \cdot (1 \pm o(1))$, $\langle \mathbf{w}_{K,s}^{(t)}, \boldsymbol{\xi}_i \rangle = \langle \mathbf{w}_{K,s}^{(0)}, \boldsymbol{\xi}_i \rangle \cdot (1 \pm o(1))$.*

Lemma 4.1 **(1)** shows the mean value noise quickly stabilize into a linear relationship with $t$. Lemma 4.1 **(2)** allows us to disregard the influence of the mean value signals on the query and key gradients, as it is negligible compared to the impact of the mean value noise. Lemma 4.1 **(3)** states that the query & key noise remain close to their initialization before $T_1$, allowing us to approximate them as still being at their initial values at $T_1$. Lemma 4.1 **(1)(3)** jointly show the early shift and stabilization of mean value noise. The proof of this section is in Appendix E.4.

## 4.2 STAGE II. QUERY AND KEY NOISE ALIGN THEIR SIGNS TO EACH OTHER

In **Stage II**, we focus on *query and key noise*, i.e., $\langle \mathbf{w}_{Q,s}^{(t)}, y_i \boldsymbol{\xi}_i \rangle$ and $\langle \mathbf{w}_{K,s}^{(t)}, y_i \boldsymbol{\xi}_i \rangle$. Let $T_2 := 50\sqrt{2}n\beta_{\boldsymbol{\xi}}\eta^{-1}\sigma_p^{-1}s^{-1} = \tilde{\Theta}(\sigma_0 n \eta^{-1} s^{-1/2})$, we call $t \in [T_1, T_2]$ Stage II.

**Lemma 4.2** (Sign Alignment Between Query and Key Noise). *Let* $T_2^{SGN} := 3\sqrt{2}\beta_{\boldsymbol{\xi}}\eta^{-1}\sigma_p^{-1}s^{-1}$. *Then, with probability* $1 - \delta$, $\text{sgn}(\langle \mathbf{w}_{Q,s}^{(T_2^{SGN})}, y_i \boldsymbol{\xi}_i \rangle) = \text{sgn}(\langle \mathbf{w}_{K,s}^{(T_2^{SGN})}, y_i \boldsymbol{\xi}_i \rangle)$. *Specifically: (1) If* $\text{sgn}(\langle \mathbf{w}_{Q,s}^{(0)}, y_i \boldsymbol{\xi}_i \rangle) = \text{sgn}(\langle \mathbf{w}_{K,s}^{(0)}, y_i \boldsymbol{\xi}_i \rangle)$, *then* $\text{sgn}(\langle \mathbf{w}_{Q,s}^{(T_2^{SGN})}, y_i \boldsymbol{\xi}_i \rangle) = \text{sgn}(\langle \mathbf{w}_{Q,s}^{(0)}, y_i \boldsymbol{\xi}_i \rangle)$. *(2) If* $\text{sgn}(\langle \mathbf{w}_{Q,s}^{(0)}, y_i \boldsymbol{\xi}_i \rangle) = -\text{sgn}(\langle \mathbf{w}_{K,s}^{(0)}, y_i \boldsymbol{\xi}_i \rangle)$, *then the conditional probability of the event* $\{\text{sgn}(\langle \mathbf{w}_{Q,s}^{(T_2^{SGN})}, y_i \boldsymbol{\xi}_i \rangle) = j\}$ *are at least* $1/2 - O(\delta)$ *for* $j \in \{\pm 1\}$.

Lemma 4.2 indicates that all query and key noise can be divided into two groups. If they initially have the same sign, they reinforce each other and grow from initialization. However, if not, they first both decay toward zero, evolving in opposite directions. As they approach zero, the smaller value crosses zero, and the larger one flips direction, aligning their signs and growing in the same direction.

**Lemma 4.3** (End of Stage II). *Let* $\beta_{\boldsymbol{\mu}} := \max_{s,j,r}\{|\langle \mathbf{w}_{Q,s}^{(0)}, \boldsymbol{\mu} \rangle|, |\langle \mathbf{w}_{K,s}^{(0)}, \boldsymbol{\mu} \rangle|, |\langle \mathbf{w}_{V,j,r}^{(0)}, \boldsymbol{\mu} \rangle|\}$. *We have: (1) (Magnitude) At* $T_2$, *for all* $s \in [m_k]$ *and* $i \in [n]$, $|\langle \mathbf{w}_{Q,s}^{(t)}, y_i \boldsymbol{\xi}_i \rangle|, |\langle \mathbf{w}_{K,s}^{(t)}, y_i \boldsymbol{\xi}_i \rangle| = \Theta(t\eta\sigma_p s)$. *(2) (Concentration of softmax)* $s_{i,11}^{(t)} = 1/2 \pm o(1)$, $s_{i,21}^{(t)} = 1/2 \pm o(1)$ *for all* $i \in [n]$ *and* $t \le T_2$. *(3) (Sum of noise) With probability at least* $1 - \delta$, *for all* $s \in [m_k]$, $|\sum_{i=1}^{n} \langle \mathbf{w}_{Q,s}^{(T_2)}, y_i \boldsymbol{\xi}_i \rangle|, |\sum_{i=1}^{n} \langle \mathbf{w}_{K,s}^{(T_2)}, y_i \boldsymbol{\xi}_i \rangle| \ge 2n\beta_{\boldsymbol{\mu}}$. *(4) For all* $s \in [m_k]$, *and* $t \le T_2$, $\langle \mathbf{w}_{Q,s}^{(t)}, \boldsymbol{\mu} \rangle = \langle \mathbf{w}_{Q,s}^{(0)}, \boldsymbol{\mu} \rangle \cdot (1 \pm o(1))$, $\langle \mathbf{w}_{K,s}^{(t)}, \boldsymbol{\mu} \rangle = \langle \mathbf{w}_{K,s}^{(0)}, \boldsymbol{\mu} \rangle \cdot (1 \pm o(1))$.

Lemma 4.3 **(1)** shows that query & key noise become linear with $t$ at $T_2$. However, according to Lemma 4.3 **(2)**, the softmax outputs of attention layer remain concentrated around $1/2$ at $T_2$. Lemma 4.3 **(3)** estimates the magnitude of the sum of the query & key noise, i.e., $\sum_{i=1}^{n} \langle \mathbf{w}_{Q,s}^{(t)}, y_i \boldsymbol{\xi}_i \rangle$ and $\sum_{i=1}^{n} \langle \mathbf{w}_{K,s}^{(t)}, y_i \boldsymbol{\xi}_i \rangle$ at $T_2$, which plays a role in Stage III. Lastly, Lemma 4.3 **(4)** states that the query & key signals remain near their initialization before $T_2$. The proof of this section is in Appendix E.5.

## 4.3 STAGE III. MAJORITY VOTING DETERMINES THE SIGN OF QUERY AND KEY SIGNALS.

In **Stage III**, we focus on *query & key signals*, i.e., $\langle \mathbf{w}_{Q,s}^{(t)}, \boldsymbol{\mu} \rangle$ and $\langle \mathbf{w}_{K,s}^{(t)}, \boldsymbol{\mu} \rangle$. Recall $\beta_{\boldsymbol{\mu}} := \max_{s,j,r}\{|\langle \mathbf{w}_{Q,s}^{(0)}, \boldsymbol{\mu} \rangle|, |\langle \mathbf{w}_{K,s}^{(0)}, \boldsymbol{\mu} \rangle|, |\langle \mathbf{w}_{V,j,r}^{(0)}, \boldsymbol{\mu} \rangle|\}$. Let $T_3 := 3\beta_{\boldsymbol{\mu}}\eta^{-1}\|\boldsymbol{\mu}\|^{-1} = \tilde{\Theta}(\sigma_0 \eta^{-1})$, we call $t \in [T_2, T_3]$ Stage III.

**Lemma 4.4** (Stage III). *We have: (1) (Noise determines the sign of signals by majority voting) For all* $s \in [m_k]$ *and* $t \in [T_2, T_3]$, $\text{sgn}(\langle \mathbf{w}_{Q,s}^{(t+1)}, \boldsymbol{\mu} \rangle - \langle \mathbf{w}_{Q,s}^{(t)}, \boldsymbol{\mu} \rangle) = \text{sgn}(\sum_{i \in [n]} \langle \mathbf{w}_{K,s}^{(t)}, y_i \boldsymbol{\xi} \rangle)$, *and* $\text{sgn}(\langle \mathbf{w}_{K,s}^{(t+1)}, \boldsymbol{\mu} \rangle - \langle \mathbf{w}_{K,s}^{(t)}, \boldsymbol{\mu} \rangle) = -\text{sgn}(\sum_{i \in [n]} \langle \mathbf{w}_{Q,s}^{(t)}, y_i \boldsymbol{\xi} \rangle)$. *(2) (Magnitude) At* $T_3$, *for all* $s \in [m_k]$, $\langle \mathbf{w}_{Q,s}^{(T_3)}, \boldsymbol{\mu} \rangle = \text{sgn}(\sum_i \langle \mathbf{w}_{Q,s}^{(T_3)}, y_i \boldsymbol{\xi}_i \rangle) \cdot \Theta(T_3 \eta \|\boldsymbol{\mu}\|)$, *and* $\langle \mathbf{w}_{K,s}^{(T_3)}, \boldsymbol{\mu} \rangle = \text{sgn}(-\sum_i \langle \mathbf{w}_{Q,s}^{(T_3)}, y_i \boldsymbol{\xi}_i \rangle) \cdot \Theta(T_3 \eta \|\boldsymbol{\mu}\|)$. *(3) (Concentration of softmax)* $s_{i,11}^{(t)} = 1/2 \pm o(1)$, $s_{i,21}^{(t)} = 1/2 \pm o(1)$ *for all* $i \in [n]$ *and* $t \in [T_2, T_3]$.

Lemma 4.4 **(1)** states that the update direction during Stage III and the final sign of the query & key signals are determined by sum of key & query noise, respectively, which remains dominant in their gradients throughout this stage. Since query & key noise have roughly the same magnitude for all $i \in [n]$, the sign dictation can be viewed as a majority voting mechanism. Lemma 4.4 **(2)** shows that query & key signals become linear with $t$ at $T_3$. Although much smaller than query & key noise, they will become dominant in Stage IV when noise-signal softmax outputs approach zero. Lemma 4.4 **(3)** also states that the softmax outputs remain around $1/2$ at this point, indicating that Stages I-III can occur within a short time in practice. Additionally, all query & key noise continues to grow throughout Stage III. The proof of this section is in Appendix E.6.

## 4.4 STAGE IV. QUERY AND KEY NOISE ALIGN THEIR SIGN TO SIGNALS BY BY THE FAST DECAY OF NOISE-SIGNAL SOFTMAX OUTPUTS

In **Stage IV**, we focus on the *noise-signal softmax outputs* $s_{i,21}^{(t)}$ and all query & key signals and noise that could be affected by $s_{i,21}^{(t)}$. Let $T_4 := C_3 \log(C_3 \sigma_p s \|\boldsymbol{\mu}\|^{-1})\eta^{-1} m_k^{-1/2} \sigma_p^{-1} s^{-1} = \tilde{\Theta}(m_k^{-1/2}\eta^{-1}\sigma_p^{-1}s^{-1})$, where $C_3 = \Theta(1)$ is a large constant, we call $t \in [T_3, T_4]$ Stage IV.

**Lemma 4.5** (Exponentially Fast Decay of Noise-Signal Softmax Outputs). *Let $T_4^- \geq T_3$ be the last time such that for all $t \in [T_3, T_4^-]$, $s \in [m_k]$ and $i \in [n]$, $\left| \sum_{i=1}^n s_{i,21}^{(t)} s_{i,22}^{(t)} \langle \mathbf{w}_{Q,s}^{(t)}, y_i \boldsymbol{\xi}_i \rangle \right| \geq \frac{1}{2} n \left| \langle \mathbf{w}_{Q,s}^{(t)}, \boldsymbol{\mu} \rangle \right|$ and $s_{i,21}^{(t)} s_{i,22}^{(t)} \left| \langle \mathbf{w}_{Q,s}^{(t)}, y_i \boldsymbol{\xi}_i \rangle \right| \geq 2 s_{i,11}^{(t)} s_{i,12}^{(t)} \left| \langle \mathbf{w}_{Q,s}^{(t)}, \boldsymbol{\mu} \rangle \right|$. Then, we have $T_4^- = \tilde{\Theta}(\eta^{-1} m_k^{-1/2} \sigma_p^{-1} s^{-1})$, and for all $i \in [n]$: (1) $s_{i,21}^{(t)} = \exp(-O(m_k t^2 \eta^2 \sigma_p^2 s^2))$ for $t \in [T_3, T_4^-]$ and $s_{i,21}^{(T_4^-)} = \exp(O(\log(n \|\boldsymbol{\mu}\| / \sigma_p s))) = o(1)$. (2) $s_{i,11}^{(t)} = 1/2 \pm o(1)$, for $t \in [T_3, T_4^-]$.*

Lemma 4.5 states that the noise-signal softmax outputs decay exponentially and approach zero during $[T_3, T_4^-]$, while other softmax outputs stay around $1/2$. All signals and noise continue to grow as in Stage III until just before $T_4^-$. Shortly after $T_4^-$, the final sign alignment of query and key noise begins.

**Lemma 4.6** (Sign Alignment of Key Noise). *There exists a small constant $\theta_c \in (0, 1)$ such that for all $s \in [m_k]$ and $i \in [n]$, we have: (1) $\mathrm{sgn}(\langle \mathbf{w}_{K,s}^{(t+1)}, y_i \boldsymbol{\xi} \rangle - \langle \mathbf{w}_{K,s}^{(t)}, y_i \boldsymbol{\xi} \rangle) = \mathrm{sgn}(\langle \mathbf{w}_{K,s}^{(t)}, y_i \boldsymbol{\xi} \rangle)$, for $t \in [T_3, T_4^-]$. (2) $\mathrm{sgn}(\langle \mathbf{w}_{K,s}^{(t+1)}, y_i \boldsymbol{\xi} \rangle - \langle \mathbf{w}_{K,s}^{(t)}, y_i \boldsymbol{\xi} \rangle) = \mathrm{sgn}(\langle \mathbf{w}_{Q,s}^{(T_3)}, \boldsymbol{\mu} \rangle)$, for $t \geq (1 + \theta_c) T_4^-$.*

**Negative key noise alignment with signals.** Consider a single neuron with a positive query signal. Lemma 4.6 states that after $(1 + \theta_c) T_4^-$, all negative key noise flip direction and begin aligning with query signal. Before $T_4^-$, the dominant gradient terms for key noise are related to both query noise and noise-signal softmax outputs. However, after $(1 + \theta_c) T_4^-$, due to the exponential decay of $s_{i,21}^{(t)}$, the query signal term becomes dominant.

**Lemma 4.7** (Delayed Sign Alignment of Query Noise). *With the same $\theta_c$ in Lemma 4.6, for all $s \in [m_k]$ and $i \in [n]$, we have: (1) $\mathrm{sgn}(\langle \mathbf{w}_{Q,s}^{(t+1)}, y_i \boldsymbol{\xi} \rangle - \langle \mathbf{w}_{Q,s}^{(t)}, y_i \boldsymbol{\xi} \rangle) = \mathrm{sgn}(\langle \mathbf{w}_{Q,s}^{(t)}, y_i \boldsymbol{\xi} \rangle)$, for $t \in [T_3, (2 - \theta_c) T_4^-]$. (2) $\mathrm{sgn}(\langle \mathbf{w}_{Q,s}^{(t+1)}, y_i \boldsymbol{\xi} \rangle - \langle \mathbf{w}_{Q,s}^{(t)}, y_i \boldsymbol{\xi} \rangle) = \mathrm{sgn}(\langle \mathbf{w}_{Q,s}^{(T_3)}, \boldsymbol{\mu} \rangle)$, for $t \geq (2 + 3\theta_c) T_4^-$.*

**Different alignment times for negative query and key noise.** Lemma 4.7 indicates a time gap between the alignment of negative key noise and negative query noise. Negative query noise continues to grow until $(2 - \theta_c) T_4^-$ since $s_{i,21}^{(t)}$ does not directly influence its gradient. However, as key noise approaches or crosses zero, the query signal term begins to dominate the gradient of query noise due to their correlation. Intuitively, $(2 - \theta_c) T_4^-$ is the point where key noise still has significant magnitude, while $(2 + 3\theta_c) T_4^-$ indicates the completion of key noise alignment.

**Lemma 4.8** (End of Stage IV). *(1) For all $t \geq T_4$, $s \in [m_k]$, and $i \in [n]$ $\mathrm{sgn}(\langle \mathbf{w}_{Q,s}^{(t)}, y_i \boldsymbol{\xi}_i \rangle) = \mathrm{sgn}(\langle \mathbf{w}_{K,s}^{(t)}, y_i \boldsymbol{\xi}_i \rangle) = \mathrm{sgn}(\langle \mathbf{w}_{Q,s}^{(t)}, \boldsymbol{\mu} \rangle)$. (2) Loss does not converge. $L_S(\mathbf{W}^{(T_4)}) = \Theta(1)$.*

Lemma 4.8 shows that final sign alignment completes, but the training loss does not convergence at the end of Stage IV. After Stage IV, the dynamics simplify, with all quantities growing continuously, and the training loss decreasing exponentially. This simplified behavior makes it easier to prove convergence and generalization results. The proof of this section is in Appendix E.7.

## 5 CONCLUSION AND LIMITATIONS

In conclusion, we present a theoretical analysis of the training dynamics of a two-layer transformers using SignGD for a binary classification task involving a dataset with both signal and noise. We identify four distinct stages in the training dynamics, characterizing the complex and intriguing behaviors within the attention layer. Our results demonstrate the fast convergence but poor generalization for SignGD. Additionally, we provide new insights into the similarities between SignGD and Adam, suggesting that both require higher data quality compared to GD in practical applications. We hope that our work contributes to a deeper theoretical understanding and aids in the design of more efficient optimization methods for transformers.

**Limitations.** Several relaxed assumptions were made to simplify the theoretical analysis. Specifically, the datasets we considered are linearly separable, signal and noise vectors are sparse, and the context length is limited to 2 in our main theory, creating a significant gap from real-world datasets. Furthermore, our data model is motivated by image data, leaving open the question of how SignGD optimizes transformers on language data, which could be a promising direction for future research. The transformer model we analyzed includes only a single-head self-attention layer, whereas deeper transformers and multi-head attention are more commonly used in practice. We discuss some potential extensions in Appendix F, and discuss differences from real-world setup in Appendix H.

ACKNOWLEDGMENT

This work was supported by the NSFC Project (No. 62376131), Tsinghua Institute for Guo Qiang, and the High Performance Computing Center, Tsinghua University. J.Z is also supported by the XPlorer Prize.

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

# Appendix

## A    DETAILED RELATED WORK

**Training dynamics of transformers.**    Currently, the theoretical analysis of transformers training dynamics is still in a stage of flourishing, and there is no fixed research paradigm. Several recent works studied the training dynamics of transformers with different data, models and focus. Jelassi et al. (2022) showed how a one-layer single-head attention model with only position embedding learns the spatial structures in the data by GD. Tian et al. (2023) studied the training dynamics of a one-layer single-head attention model with a data model for next-token prediction trained with SGD, and Tian et al. (2024) analyzed the joint dynamics of an attention and MLP layer. Li et al. (2023d) studied the training dynamics of a single-head attention layer on the $\ell_2$ loss with the data modeled by topic modeling. Li et al. (2023c) studied a three-layer vision transformer with query and key parameterization trained by SGD on the hinge loss. Oymak et al. (2023) studied a single-head attention layer on the prompt-tuning setting, where the query and key parameters in the attention are fixed during the training process. Tarzanagh et al. (2023a;b) studied the dynamics of a single-head attention layer and a tunable token, and connects the training dynamics to a certain SVM problem. While Tarzanagh et al. (2023a;b) only presented an asymptotic convergence result, Vasudeva et al. (2024) showed the global convergence, provided a convergence rate $t^{-3/4}$ for GD, as well as removing the restriction on a fixed linear head. Also extending Tarzanagh et al. (2023a;b), Sheen et al. (2024) studied this problem with a query-key parameterized transformer, which gives different implicit regularization compared with the single attention matrix parameterization. Huang et al. (2024b) studied the GD dynamics of a single-head attention layer with a self-supervised learning objective. Nichani et al. (2024) studied the GD dynamics of a disentangled two-layer attention-only transformer on random sequences with causal structure and proved that it can learn the causal structure in the first attention layer. Wang et al. (2024) studied the data model introduced by Sanford et al. (2023), showing that an one-layer transformer can efficiently learn this task via GD, while fully-connected networks cannot express the task. Jiang et al. (2024) studied the GD dynamics of a two-layer transformer with a single-head self-attention layer and a fixed linear head on a dataset with signal and noise, which has the similar data and model setting to our work. However, Jiang et al. (2024) targets on the benign overfitting of GD, while our work focus on the optimization and generalization of SignGD.

Recently, there has been some works studying the behaviors of transformers on in-context learning tasks to understand the powerful in-context learning abilities of large language models. We mainly focus on works with convergence guarantee via training dynamics analysis. Ahn et al. (2023); Mahankali et al. (2024) showed that a one-layer transformer implements single-step gradient descent to minimize the pre-training loss for an in-context learning task. Zhang et al. (2024b) studied the Gradient Flow (GF) of a single-head linear attention layer on in-context linear regression task. Huang et al. (2023c) studied how a single-head attention layer trained by GD solves an in-context linear regression task where the the input data are orthogonal. Chen et al. (2024) studied the GF for an one-layer multi-head attention model for an in-context linear regression task, where the groundtruth function admits a multi-task structure. Kim & Suzuki (2024) studied the mean-field dynamics of a transformer with one linear attention layer and one MLP layer on an in-context regression task. Li et al. (2024) studied the SGD dynamics of one-layer transformer with a single-head self-attention layer and a two-layer MLP on an in-context classification task.

**Comparison with other works using query-key parameterization.** Practically, the modern transformer architecture uses query-key parameterization in the attention module (Vaswani et al., 2017; Dao et al., 2022; Zhang et al., 2024a; 2025a;b). However, most of works mentioned above take simplifications that replaces the query and key matrices with a single attention matrix. Among those works, only Li et al. (2023c; 2024); Sheen et al. (2024); Jiang et al. (2024) studied the softmax attention with trainable query-key parameterization, but there are also some limitations. Li et al. (2023c; 2024) introduced relaxed assumptions about initialization, which are too stringent and make softmax outputs not concentrated at $1/2$ at initialization anymore. Sheen et al. (2024) started from diagonal query and key matrices and used a data-correlated "Alignment Property" assumption for the general query and key initialization, which seems hard to verify whether it holds practically. Compared with those works, we study query and key matrix from Gaussian initialization which is commonly used in practice. Finally, Jiang et al. (2024) studied trainable query and key with Gaussian initialization. However, their initializations for the queries, keys, and values differ, whereas our initialization for the queries, keys, and values is the same.

Additionally, all of these studies analyzed the dynamics of (S)GD or GF, while we focus on SignGD.

**Understanding of Adam on transformers.** While Adam may fail to converge in convex objective (Reddi et al., 2018), it performs so well on transformers and is better than SGD, which means Adam converges faster and achieves lower training loss (Ahn et al., 2024; Jiang et al., 2023; Kunstner et al., 2023; 2024; Pan & Li, 2023; Zhang et al., 2024c; 2020b).

Previous works tried to give an explanation about this fact from different perspectives. Liu et al. (2020) observed unbalanced gradients in transformers and Adam can give uniform parameter update. Zhang et al. (2020b) suggested that heavy-tailed stochastic gradient noise in language data on transformers compared with image data on CNN models is the main cause that adaptive methods are good. However, Kunstner et al. (2023) showed that the heavy-tailed stochastic gradient noise may not be the main factor. They compared the performance of deterministic Adam and GD in full-batch settings, and observed that Adam is still better than GD. Jiang et al. (2023) showed that Adam could bias the trajectories towards regions where Hessian has relatively more uniform diagonals while SGD cannot. Zhang et al. (2024c) also studied from the perspective of Hessian. They showed the distances between Hessian spectrum of different parameter blocks are large in transformers which may hamper SGD but can be handled by Adam. Zhang et al. (2025c) and Li et al. (2023a) found that second moment of the Adam optimizer is not sensitive which indicates the advantage over SGD is robust. Pan & Li (2023) showed that Adam can lead to smaller directional smoothness values which may imply better optimization. Kunstner et al. (2024) showed that heavy-tailed class imbalance in language modeling tasks is a difficulty for GD but Adam and SignGD do not suffer from this problem.

Furthermore, many works have focused on proving the convergence rate of Adam in the framework of classical convergence analysis. Zhang et al. (2020a) and Crawshaw et al. (2022) sought alternative relaxed assumptions for Adam. Li et al. (2023b) and Wang et al. (2023) improved the analysis of Adam under those assumptions.

## B EXPERIMENTAL DETAILS AND MORE EXPERIMENTS

### B.1 EXPERIMENTAL SETTINGS

We perform numerical experiments on the synthetic and real-world datasets to verify our main results.

**Experimental setting for synthetic dataset.** The synthetic dataset is generated according to our Definition 2.1. For data hyerparameters, they can be uniquely determined by one row in Tab. 3 and the value of $d$. In the Fig. 1 and Tab. 2 of main text, we use **(a)** with $d = 2000$. We always use 500 samples for computing test loss.

For optimizers, we use following default hyperparameters. For sign gradient descent, we use the learning rate $\eta = $ 1e-4. For Adam, we use the learning rate $\eta = $ 1e-4, $\beta_1 = 0.9$, $\beta_2 = 0.999$, and $\epsilon = $ 1e-15. For gradient descent, we use the learning rate $\eta = $ 1e-1.

Also, we use neuron $s = 0$ in Fig. 1 by default and in following experiments. We use sign gradient descent with the learning rate $\eta = $ 1e-7 in 2000 iterations to simulate sign gradient descent with the learning rate $\eta = $ 1e-4 in 2 iterations in Fig. 1 (a),(b) and in following experiments. We use a learning rate of 1e-4 for all optimizers in Fig. 2 (c) for a fair comparison.

Table 3: Experimental settings of data model. $n$ is the training sample size and there are always equal samples in both classes. $s$ is the noise sparsity level. $\sigma_p$ is the standard deviation of noise. $\sigma_0$ is the network initialization standard deviation. $m_v$ is the value dimension. $m_k$ is the query and key dimension. 'orthogonal' means whether the noise patch and signal patch are orthogonal. If they are not orthogonal, the $s$ coordinates are selcted form $[d]$ instead of $[d] \backslash \{1\}$. 'iters' is the total iteration/epoch number in one run. Signal patch $\boldsymbol{\mu}$ is always $[1, 0, \dots, 0]^\top$.

|  | $n$ | $s$ | $\sigma_p$ | orthogonal | $\sigma_0$ | $m_v$ | $m_k$ | iters |
|---|---|---|---|---|---|---|---|---|
| **(a)** | $0.01d$ | $0.04d$ | $2.0/\sqrt{s}$ | True | $0.1/\sqrt{d}$ | $0.01d$ | $0.05d$ | 2000 |
| **(b)** | $0.01d$ | $0.04d$ | $2.0/\sqrt{s}$ | False | $0.1/\sqrt{d}$ | $0.01d$ | $0.05d$ | 2000 |
| **(c)** | $0.01d$ | $0.4d$ | $2.0/\sqrt{s}$ | True | $0.1/\sqrt{d}$ | $0.01d$ | $0.05d$ | 2000 |
| **(d)** | $0.01d$ | $0.4d$ | $2.0/\sqrt{s}$ | False | $0.1/\sqrt{d}$ | $0.01d$ | $0.05d$ | 2000 |

**Experimental setting for real-world dataset.** We conduct real-world experiments on the MNIST dataset. We introduce the noise to the dataset in the following way. For each image, we first multiply each pixel in the image with a factor $\lambda$, which we call "scaled SNR", and then add gaussian random noises with standard deviation $1 - \lambda$ to the outer regions with a width of 7. Additionally, we only the class 3 and 7 for classification, to make a binary classification task which is consistent with our theoretical settings. To input the data into the transformers, we patchify the data with a size of $7 \times 7$.

We train a two-layer transformer model consistent with our theoretical setting. We set $d = 49$, $m_v = m_k = 10$, $\sigma_0 = 0.1/\sqrt{d}$. We only use 2000 training data points and use deterministic optimizers to train the network. For SignGD and Adam across different values of $\beta_1$, we use learning rate $\eta = 1e - 2$, and train the models for 200 epochs. For GD, we use different learning rates for different SNR since one learning rate of GD cannot adapt to data with noise in different magnitude. Specifically, we use $\eta = 1e - 1$ for SNR in $[0.9, 1.0]$, $\eta = 3e - 1$ for SNR in $[0.7, 0.8]$, $\eta = 6e - 1$ for SNR in $[0.5, 0.6]$, $\eta = 1e0$ for SNR in $[0.2, 0.3, 0.4]$. We train the model for 500 epochs for GD. In all these settings, our training setup can guarantee a training loss smaller than 0.05. We calculate the test losses on the entire test datasets (of the class 3 and 7). Finally, we conduct three runs for each training setup and report the mean and standard deviation.

### B.2 COMPARISON WITH NON-SPARSE AND/OR NON-ORTHOGONAL DATA

In Fig. 3, we run the experiments with the same setting as Fig. 1, but use a different legend. In Fig. 4, 5, 6, we run the experiments with non-orthogonal sparse data, orthogonal non-sparse data, and non-orthogonal non-sparse data, respectively. The legends follow Fig. 3. The figures show that our theoretical results hold empirically in those data settings.

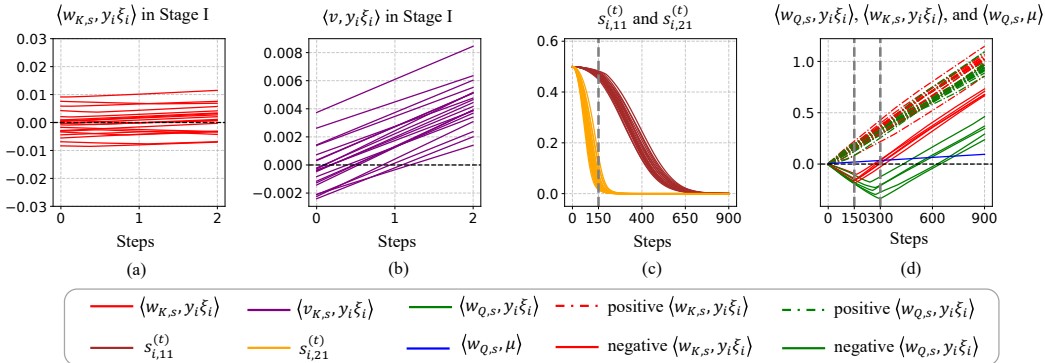

Figure 3: **Data setting (a) in Tab. 3 with** $d = 2000$. **(a)** Key noise dynamics over $t = 0$ to $t = 2$. **(b)** Mean value noise dynamics over $t = 0$ to $t = 2$. While mean value noise stabilizes into a linear relationship with $t$ early, key noise remains close to initialization. **(c)** Softmax output dynamics over $t = 0$ to $t = 900$. The softmax outputs decay exponentially. At $t = 150$, $s_{i,21}^{(t)}$ approaches zero, while $s_{i,11}^{(t)}$ remains close to $1/2$. **(d)** Dynamics of query noise, key noise, and query signals over $t = 0$ to $t = 900$: The dotted lines represent positive query and key noise at $t = 100$, and the solid lines represent negative noise at the same point. By Stage III, the majority of positive noise makes the query signal positive through majority voting. In Stage IV, sign alignment of key noise starts at about $t = 150$, coinciding with $s_{i,21}^{(t)}$ approaching zero, while delayed sign alignment of query noise begins around $t = 300$, about twice as late as the key noise.

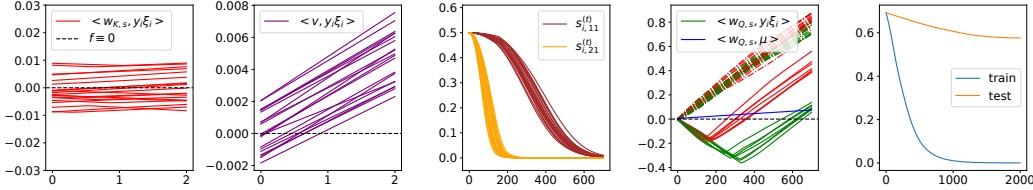

Figure 4: Data setting **(b)** in Tab. 3 with $d = 2000$.

## B.3 COMPARISON WITH ADAM AND GRADIENT DESCENT

In Fig. 7, 8, 9, we plot the dynamics of softmax outputs, query signal, query and key noise, and training and test loss with different optimizers. The figures show that our theoretical results almost hold empirically in Adam except the sign alignment of negative query noise in the Stage IV. This is due to Adam utilizes the information of history gradients to modify the current update. We give an explanation about this. When the key signal becomes dominant in the gradients of query noise in sign gradient descent, it is actually very small in magnitude, which implies the small gradients can be easily dominated by the momentum in Adam. But otherwise, symbolic gradient descent is almost similar to how Adam behaves under different hyperparameters.

Also, we can observe that the convergence of softmax outputs in sign gradient descent and Adam is much faster than that in gradient descent. At around $t = 600$, the softmax outputs just start to leave $1/2$ in gradient descent but is almost converged in sign gradient descent and Adam. It is also noted that gradient descent can leads to small generalization gap in this setting.

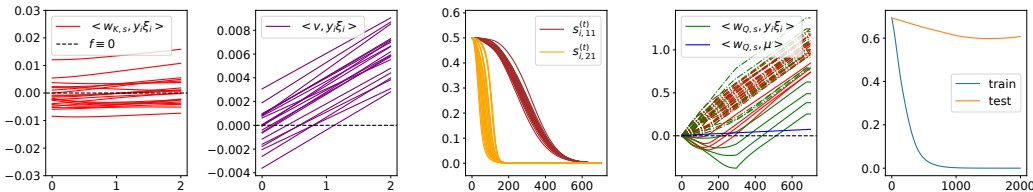

Figure 5: Data setting **(c)** in Tab. 3 with $d = 2000$.

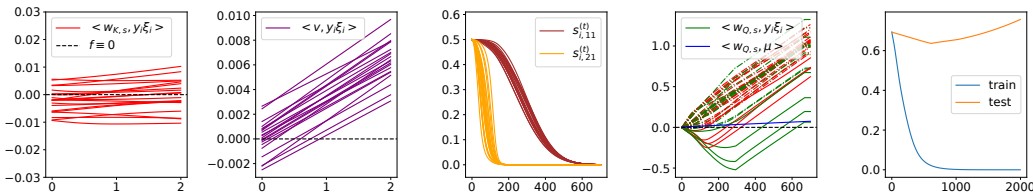

Figure 6: Data setting **(d)** in Tab. 3 with $d = 2000$.

### B.4 COMPARISON WITH GREATER CONTEXT LENGTH

In this section, we investigate when will happen when context length is greater than 2, i.e., $L > 2$. At this time, the model is defined as

$$F_j(\mathbf{W}, \mathbf{X}) := \frac{1}{m_v} \sum_{l=1}^{L} \mathbf{1}_{m_v}^\top \mathbf{W}_{V,j} \mathbf{X} \mathrm{softmax}\left(\mathbf{X}^\top \mathbf{W}_K^\top \mathbf{W}_Q \mathbf{x}^{(l)}\right).$$

For each data point $(\mathbf{X}, y)$, predictor $\mathbf{X} = [\mathbf{x}^{(1)}, \mathbf{x}^{(2)}, \ldots, \mathbf{x}^{(L)}] \in \mathbb{R}^{d \times L}$ have $L$ patches (or tokens), where $\mathbf{x}^{(1)}, \mathbf{x}^{(2)}, \ldots, \mathbf{x}^{(L)} \in \mathbb{R}^d$, and label $y$ is binary, i.e., $y \in \{\pm 1\}$. The data generation is similar to the $L = 2$ case, except that we randomly select $L/2$ patches and assign them by $y\boldsymbol{\mu}$ as signal patches, while the remaining $L/2$ patches are noise patches. The noise patches in one data sample are mutually independent.

In our experiments, we use $L = 10$, and Fig. 10, 11, 12 plot the dynamics of softmax outputs, query signal, query and key noise, and training and test loss with different optimizers. In those figures, for all $i \in [n]$, $a_i$ is defined as

$$a_i = \arg\max_{l \in [L]}\{s_{i,1l}^{(T)}\}.$$

We empirically observe that for all data point, only one element in each line of $L \times L$ post-softmax attention matrix is activated, while other elements in the line are almost zero, which means that the attention attends to only one patch. This patch for each line is also uniform in different lines and therefore uniquely corresponds to one data point, which is exactly $a_i$. In the $L = 2$ case, we have $a_i = 2$ for all $i \in [n]$. But when $L > 2$ and there are many noise patches in one data sample, $a_i$ can be varied across samples but $X^{(a_i)}$ must be a noise patch.

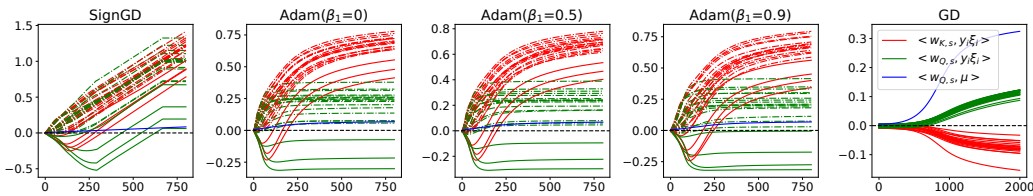

Figure 7: Sign alignment of signal to noise by majority voting in Stage III and sign alignment of negative noise to query signal by decay of noise-signal softmax outputs in Stage IV. We use data setting **(d)** in Tab. 3 with $d = 2000$. We always use $\beta_2 = 0.99$ and $\epsilon = $ 1e-15 in Adam.

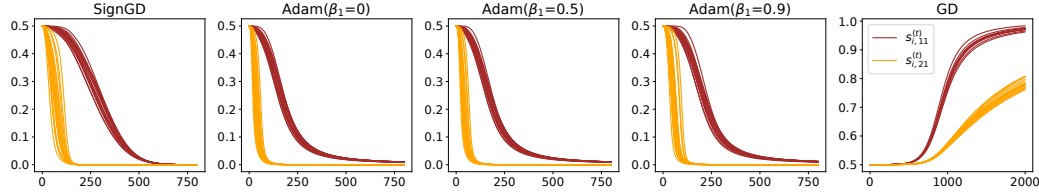

Figure 8: The dynamics of softmax outputs. We use data setting **(d)** in Tab. 3 with $d = 2000$. We always use $\beta_2 = 0.99$ and $\epsilon = 1\text{e-}15$ in Adam.

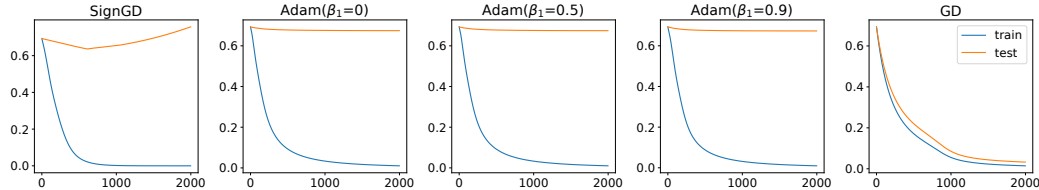

Figure 9: Training and test loss. We use data setting **(d)** in Tab. 3 with $d = 2000$. We always use $\beta_2 = 0.99$ and $\epsilon = 1\text{e-}15$ in Adam.

The greater context length case is consistent with the $L = 2$ case, and thus consistent with our theoretical results in the sense that (1) For all $i \in [n]$, the dominated query noise $\langle \mathbf{w}_{Q,s}, y_i X_i^{(a_i)} \rangle$ and key noise $\langle \mathbf{w}_{K,s}, y_i X_i^{(a_i)} \rangle$ are the same in sign with the query signal $\langle \mathbf{w}_{Q,s}, \boldsymbol{\mu} \rangle$. (2) The convergence of noise-signal/noise softmax outputs is very fast and faster than signal-signal/noise softmax outputs. For gradient descent, while the loss fast converges, but the query and key parameters are basically not learned, even with learning rate $\eta = 1$.

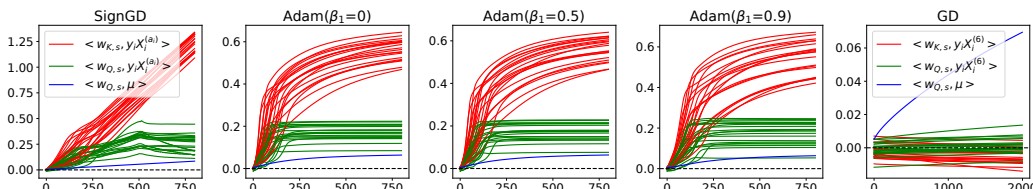

Figure 10: The dynamics of query noise, key noise and query signal. We use data setting **(d)** in Tab. 3 with $d = 2000$. We use $\beta_2 = 0.99$ and $\epsilon = 1\text{e-}15$ in Adam and we use $\eta = 1.0$ for gradient descent.

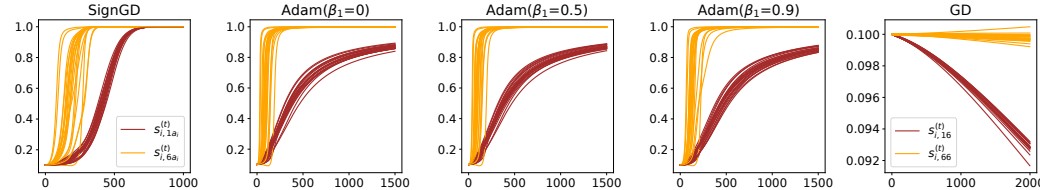

Figure 11: The dynamics of softmax outputs. We use data setting **(d)** in Tab. 3 with $d = 2000$. We use $\beta_2 = 0.99$ and $\epsilon = $ 1e-15 in Adam and we use $\eta = 1.0$ for gradient descent.

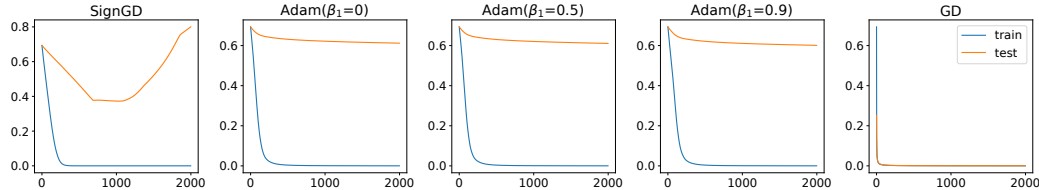

Figure 12: Training and test loss. We use data setting **(d)** in Tab. 3 with $d = 2000$. We use $\beta_2 = 0.99$ and $\epsilon = $ 1e-15 in Adam and we use $\eta = 1.0$ for gradient descent. In this case, gradient descent converges much faster than sign gradient descent and Adam since we use a large learning rate $\eta = 1.0$, which is 1e4 times of the learning rate used in sign gradient descent and Adam.

### B.5    COMPARISON WITH MULTI-HEAD ATTENTION

See Fig. 13 for the full dynamics of a simplified transformer model with 4 attention heads using SignGD. We observe that the softmax outputs of each head exhibit the same behaviors, and all dynamics are consistent with the single-head case.

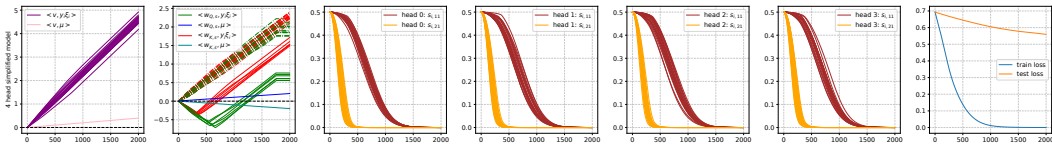

Figure 13: Equivalent of Fig. 18 but using 4 head in the simplified transformer model. The subfigure 3-6 shows the dynamics of softmax outputs in each attention head.

### B.6   COMPARISON WITH MORE COMPLEX ATTENTION MODELS

We have conducted additional experiments on deeper transformers using our synthetic dataset with SignGD, exploring various settings. Specifically, we extend our analysis to models with additional attention layers, MLP layers, and residual connections, which are essential components of modern transformer architectures. Since our theory primarily predicts the behavior of data-parameter inner products, for transformers with multiple attention layers, we focus on the dynamics of the first layer.

To examine how well the key behaviors identified by our theory persist in more complex models, we performed an ablation study. We provide the full dynamics of all relevant quantities in Fig. 14 and Fig. 15 and augment these results with Tables 4-11, which illustrate the sign alignment behavior during Stage II.

**Transformers with Residual Connections.** Firstly, on transformers with residual connections, across all model configurations we tested—including 2-layer transformers without MLPs, 3-layer transformers without MLPs, 2-layer transformers with MLPs, and 3-layer transformers with MLPs—we observe the following behaviors, consistent with our theoretical predictions:

- Stage I: Value noise increases faster than query and key noise, and the value signal remains small relative to the value noise.

- Stage II: Query and key noise exhibit sign alignment behavior early in training.

- Stage III: The query and key signals have opposite signs, determined by query (and key) noise via a majority-voting mechanism.

- Stage IV: Noise-feature softmax outputs firstly decay and decay exponentially, and both negative query and key noise align with the query signal.

However, we remark that in more complex models, the final alignment observed in Stage IV—i.e., the flip of negative query and key noise—often halts midway. This phenomenon becomes more pronounced with the addition of MLP layers, where the final alignment stops earlier. We attribute this behavior to the *rapid shrinking of query and key gradients*. This is partly driven by the decay of softmax outputs (as shown in Lemma D.7). Furthermore, as the number of layers increases and/or MLP layers are introduced, additional layers significantly contribute to this gradient shrinkage, as illustrated in the last column of Figure 14. It is worth noting that this gradient shrinking is a numerical precision issue unrelated to our theory. In theory, the sign operation maps gradients to ±1 regardless of their magnitude. However, in practice, extremely small gradients are rounded to zero, disrupting the alignment process. Despite this, we conclude that the key behaviors predicted by our theory persist in deeper transformers with residual connections.

**Transformers without Residual Connections.** On the other hand, in deeper transformers lacking residual connections, the dynamics become erratic. While some short-term behaviors (e.g., sign alignment between query and key noise in Stage II, and the opposing signs between query and key signal) are preserved (see Tables 8-11, and Figure 15), long-term behaviors deviate significantly from theoretical predictions. For instance:

- Feature-feature softmax outputs start to increase instead of decreasing.

- The dynamics of positive key noise become non-monotonic.

- Value noise exhibits irregular patterns rather than increasing consistently.

Additionally, we remark that the training dynamics of transformers without residual connections are less stable and more irregular compared to those with residual connections. This instability may be linked to the phenomenon of rank collapse in transformers, as discussed in prior works (Dong et al., 2021; Noci et al., 2022).

Based on these findings, we conclude that the key behaviors predicted by our theory persist in deeper transformers with residual connections. Without the residual connections, the key behaviors outlined in our theory are only partially preserved. Understanding the behaviors in subsequent layers and why deeper models make the gradient of first layer shrink faster could be a future direction.

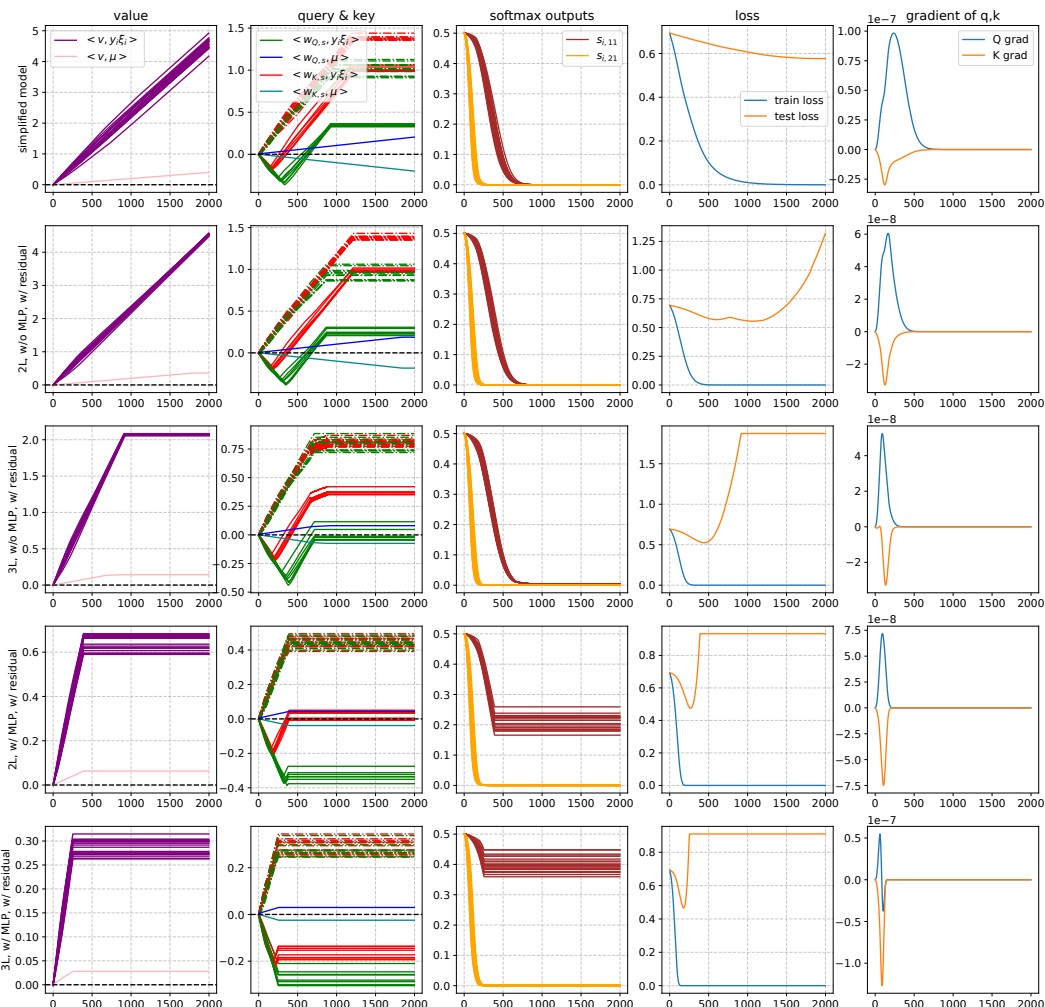

Figure 14: Equivalent of Fig. 18 but with four additional configurations for transformer models: 1) 2 layer attention-only transformer blocks with residual connections; 2) 3 layer attention-only transformer blocks with residual connections; 3) 2 layer attention+MLP transformer blocks with residual connections; 4) 3 layer attention+MLP transformer blocks with residual connections;

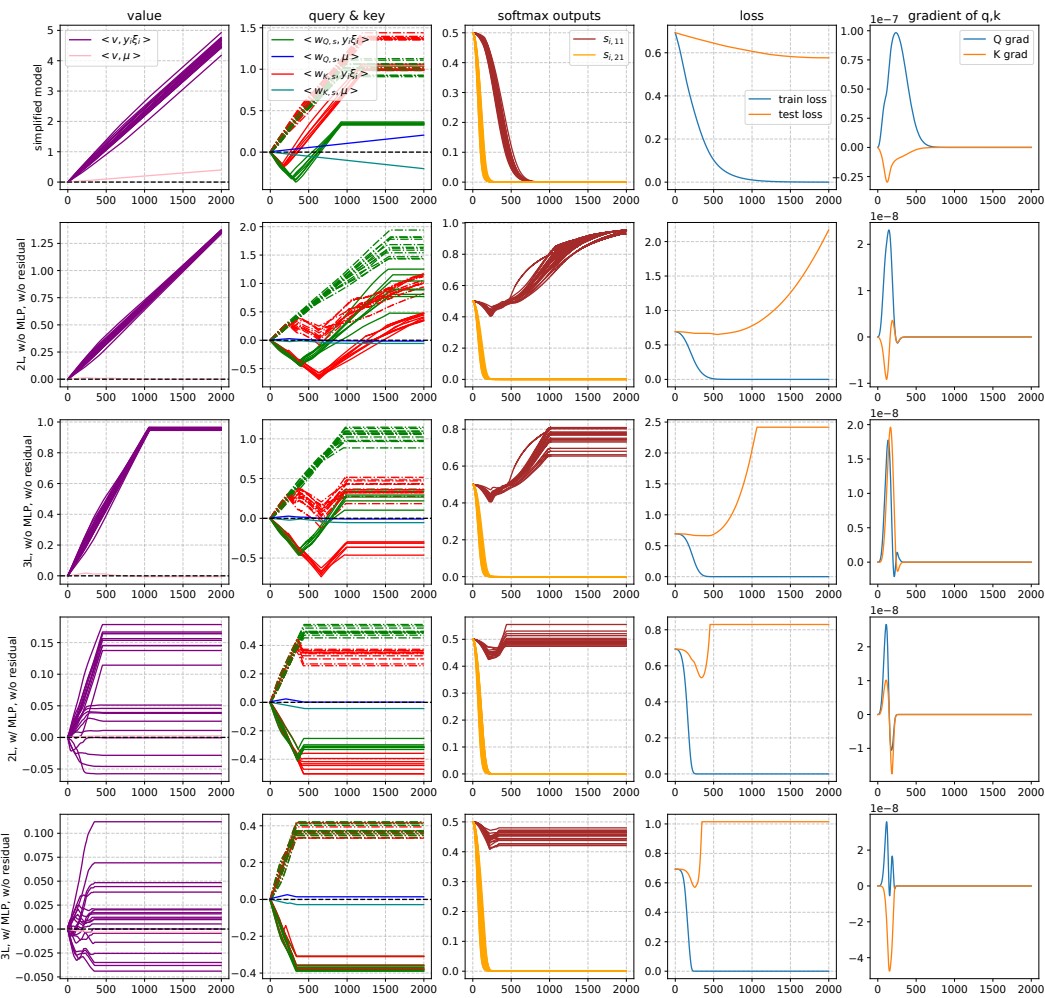

Figure 15: Equivalent of Fig. 18 but with four additional configurations for transformer models: 1) 2 layer attention-only transformer blocks without residual connections; 2) 3 layer attention-only transformer blocks without residual connections; 3) 2 layer attention+MLP transformer blocks without residual connections; 4) 3 layer attention+MLP transformer blocks without residual connections; The last line represents the mean gradient of $\mathbf{W}_Q$ and $\mathbf{W}_K$ parameters in each iteration.

Table 4: Sign alignment between query and key noise in Stage II for model: 2L, w/o MLP, w/ residual. The notation in this table and following tables is the same as Tab. 2.

| init$(t=0)\backslash t=10$ | $|S_{K+,Q+}^{(t)}|$ | $|S_{K+,Q-}^{(t)}|$ | $|S_{K-,Q+}^{(t)}|$ | $|S_{K-,Q-}^{(t)}|$ | Row sum |
|---|---|---|---|---|---|
| $|S_{K+,Q+}^{(0)}|$ | 483 | 2 | 1 | 26 | 512 |
| $|S_{K+,Q-}^{(0)}|$ | 242 | 3 | 9 | 253 | 507 |
| $|S_{K-,Q+}^{(0)}|$ | 224 | 10 | 3 | 221 | 458 |
| $|S_{K-,Q-}^{(0)}|$ | 37 | 3 | 3 | 480 | 523 |
| Column sum | 986 | 18 | 16 | 980 | 2000 |

## B.7 EXPLANATIONS FOR DIFFERENCES BETWEEN SIGNGD AND ADAM

Although SignGD can serve as a proxy for understanding Adam, our experiments reveal notable differences between the two. In Figure 2(a) and (b), SignGD causes the negative query to eventually

Table 5: Sign alignment between query and key noise in Stage II for model: 3L, w/o MLP, w/ residual.

| $\text{init}(t=0)\backslash t=10$ | $|S^{(t)}_{K+,Q+}|$ | $|S^{(t)}_{K+,Q-}|$ | $|S^{(t)}_{K-,Q+}|$ | $|S^{(t)}_{K-,Q-}|$ | Row sum |
|---|---|---|---|---|---|
| $|S^{(0)}_{K+,Q+}|$ | 482 | 2 | 0 | 28 | 512 |
| $|S^{(0)}_{K+,Q-}|$ | 243 | 4 | 7 | 253 | 507 |
| $|S^{(0)}_{K-,Q+}|$ | 222 | 12 | 3 | 221 | 458 |
| $|S^{(0)}_{K-,Q-}|$ | 39 | 1 | 3 | 480 | 523 |
| Column sum | 986 | 19 | 13 | 982 | 2000 |

Table 6: Sign alignment between query and key noise in Stage II for model: 2L, w/ MLP, w/ residual.

| $\text{init}(t=0)\backslash t=10$ | $|S^{(t)}_{K+,Q+}|$ | $|S^{(t)}_{K+,Q-}|$ | $|S^{(t)}_{K-,Q+}|$ | $|S^{(t)}_{K-,Q-}|$ | Row sum |
|---|---|---|---|---|---|
| $|S^{(0)}_{K+,Q+}|$ | 479 | 4 | 0 | 29 | 512 |
| $|S^{(0)}_{K+,Q-}|$ | 241 | 14 | 6 | 246 | 507 |
| $|S^{(0)}_{K-,Q+}|$ | 217 | 9 | 11 | 221 | 458 |
| $|S^{(0)}_{K-,Q-}|$ | 39 | 3 | 2 | 479 | 523 |
| Column sum | 976 | 30 | 19 | 975 | 2000 |

Table 7: Sign alignment between query and key noise in Stage II for model: 3L, w/ MLP, w/ residual.

| $\text{init}(t=0)\backslash t=10$ | $|S^{(t)}_{K+,Q+}|$ | $|S^{(t)}_{K+,Q-}|$ | $|S^{(t)}_{K-,Q+}|$ | $|S^{(t)}_{K-,Q-}|$ | Row sum |
|---|---|---|---|---|---|
| $|S^{(0)}_{K+,Q+}|$ | 481 | 3 | 1 | 27 | 512 |
| $|S^{(0)}_{K+,Q-}|$ | 232 | 25 | 16 | 234 | 507 |
| $|S^{(0)}_{K-,Q+}|$ | 213 | 9 | 20 | 216 | 458 |
| $|S^{(0)}_{K-,Q-}|$ | 36 | 3 | 1 | 483 | 523 |
| Column sum | 962 | 40 | 38 | 960 | 2000 |

Table 8: Sign alignment between query and key noise in Stage II for model: 2L, w/o MLP, w/o residual.

| $\text{init}(t=0)\backslash t=10$ | $|S^{(t)}_{K+,Q+}|$ | $|S^{(t)}_{K+,Q-}|$ | $|S^{(t)}_{K-,Q+}|$ | $|S^{(t)}_{K-,Q-}|$ | Row sum |
|---|---|---|---|---|---|
| $|S^{(0)}_{K+,Q+}|$ | 486 | 2 | 0 | 24 | 512 |
| $|S^{(0)}_{K+,Q-}|$ | 249 | 4 | 4 | 250 | 507 |
| $|S^{(0)}_{K-,Q+}|$ | 224 | 7 | 4 | 223 | 458 |
| $|S^{(0)}_{K-,Q-}|$ | 33 | 2 | 1 | 487 | 523 |
| Column sum | 992 | 15 | 9 | 984 | 2000 |

Table 9: Sign alignment between query and key noise in Stage II for model: 3L, w/o MLP, w/o residual.

| $\text{init}(t=0)\backslash t=10$ | $|S^{(t)}_{K+,Q+}|$ | $|S^{(t)}_{K+,Q-}|$ | $|S^{(t)}_{K-,Q+}|$ | $|S^{(t)}_{K-,Q-}|$ | Row sum |
|---|---|---|---|---|---|
| $|S^{(0)}_{K+,Q+}|$ | 483 | 1 | 2 | 26 | 512 |
| $|S^{(0)}_{K+,Q-}|$ | 238 | 9 | 13 | 247 | 507 |
| $|S^{(0)}_{K-,Q+}|$ | 222 | 8 | 9 | 219 | 458 |
| $|S^{(0)}_{K-,Q-}|$ | 34 | 4 | 5 | 480 | 523 |
| Column sum | 977 | 22 | 29 | 972 | 2000 |

Table 10: Sign alignment between query and key noise in Stage II for model: 2L, w/ MLP, w/o residual.

| init$(t=0)\backslash t=10$ | $|S^{(t)}_{K+,Q+}|$ | $|S^{(t)}_{K+,Q-}|$ | $|S^{(t)}_{K-,Q+}|$ | $|S^{(t)}_{K-,Q-}|$ | Row sum |
|---|---|---|---|---|---|
| $|S^{(0)}_{K+,Q+}|$ | 484 | 1 | 0 | 27 | 512 |
| $|S^{(0)}_{K+,Q-}|$ | 245 | 5 | 12 | 245 | 507 |
| $|S^{(0)}_{K-,Q+}|$ | 213 | 12 | 5 | 228 | 458 |
| $|S^{(0)}_{K-,Q-}|$ | 41 | 0 | 2 | 480 | 523 |
| Column sum | 983 | 18 | 19 | 980 | 2000 |

Table 11: Sign alignment between query and key noise in Stage II for model: 3L, w/ MLP, w/o residual.

| init$(t=0)\backslash t=10$ | $|S^{(t)}_{K+,Q+}|$ | $|S^{(t)}_{K+,Q-}|$ | $|S^{(t)}_{K-,Q+}|$ | $|S^{(t)}_{K-,Q-}|$ | Row sum |
|---|---|---|---|---|---|
| $|S^{(0)}_{K+,Q+}|$ | 473 | 4 | 3 | 32 | 512 |
| $|S^{(0)}_{K+,Q-}|$ | 233 | 12 | 14 | 248 | 507 |
| $|S^{(0)}_{K-,Q+}|$ | 211 | 14 | 9 | 224 | 458 |
| $|S^{(0)}_{K-,Q-}|$ | 36 | 2 | 6 | 479 | 523 |
| Column sum | 953 | 32 | 32 | 983 | 2000 |

become positive, whereas it remains negative with Adam. Additionally, in Figure 2(c), the training loss of SignGD converges linearly, while Adam exhibits sublinear convergence. While we previously suggested that these differences might arise from Adam's momentum term, we did not provide detailed evidence. Here, we try to explain these differences in terms of training dynamics and convergence rates.

To investigate factors influencing Adam's behavior, we vary its $\beta$ parameters and conduct experiments under the same model and dataset as in Figure 2. In Figure 2, we observe that $\beta_1$ values ranging from 0 (no first moment) to 0.9 (commonly used in practice) do not significantly impact training speed. Similarly, in Figure 7, changes in $\beta_1$ have little effect on training dynamics. Thus, our focus shifts to the role of $\beta_2$.

**Convergence rate.** In Figure 16,we observe that when $\beta_2 > 0.9$, the training loss exhibits a sublinear convergence rate. We remark that when the $\beta_2 < 0.9$, the loss curve closely resembles that of SignGD, thus we use a range of [0.9, 0.999] for $\beta_2$. Since the training loss convergence is primarily driven by the growth of mean value noise, we believe this behavior can be approximated through the analysis of a linear model fitting the noise.

**Training Dynamics.** Figure 17 (first row) shows that only small values of $\beta_2$ prevent the negative query noise from turning positive. As $\beta_2$ increases, the dynamics become smoother, and the evolution of query noise halts earlier.

To understand this, we examine the mean gradient and update magnitude in the second and last rows of Figure 17. Unlike deeper transformers, the query and key gradients do not shrink faster. Instead, Adam's update magnitude for query parameters decays to zero before the gradients approach zero. This early decay of the update magnitude (or effective step size) can be attributed to $\beta_2$. As $\beta_2$ increases, the update magnitude decreases earlier, while the gradient shrinkage occurs at the same point.

These observations suggest that $\beta_2$ plays a crucial role in both the convergence rate and training dynamics of Adam, highlighting key differences from SignGD.

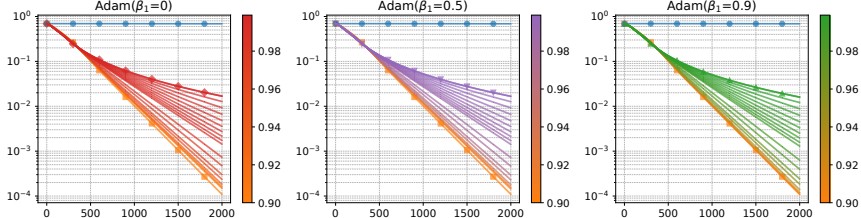

Figure 16: Training loss curve (log scale) on the synthetic dataset for different optimizers. In each plot, $\beta_1$ is fixed to 0, 0.5, or 0.9, respectively, while $\beta_2$ varies from 0.9 to 0.999. The colorbar next to each plot represents the value of $\beta_2$. This range for $\beta_2$ is chosen because we observe that the training loss of Adam with $\beta_2$ values below 0.9 is very similar to that of SignGD.

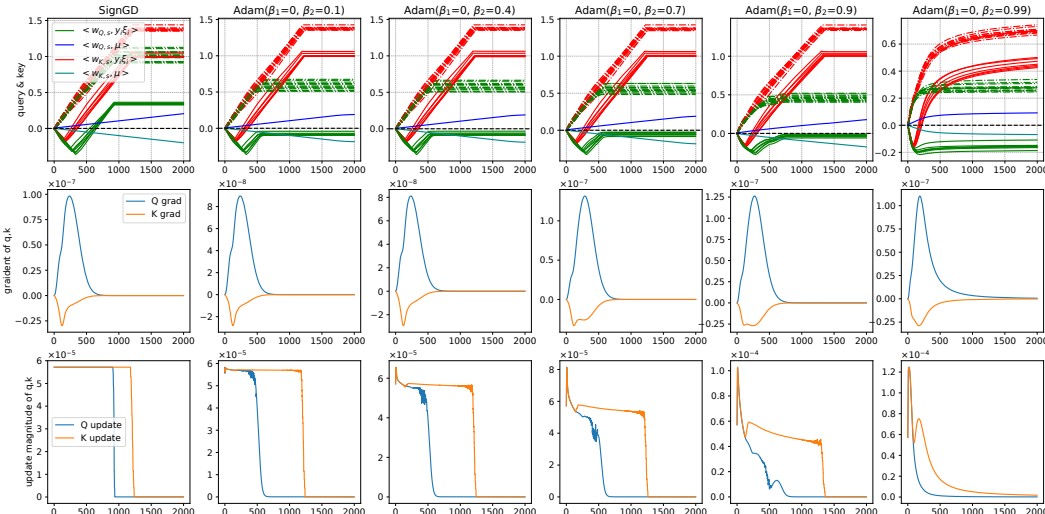

Figure 17: Equivalent of Fig. 18 but using Adam with $\beta_1 = 0$ and varying $\beta_2$. The last row represents the mean update magnitude, i.e., $\ell_1$ norm between next the current iterates, of parameters $\mathbf{W}_Q$, $\mathbf{W}_K$ at all iterations.

## C PRELIMINARY LEMMAS

The following lemma studies non-overlap support property in sparse data model.

**Lemma C.1** (Non-overlapping support of noise, Lemma C.1 in Zou et al. (2023)). *Suppose $s = \Omega(d^{1/2}n^{-2})$. Let $\{(\mathbf{X}_i, y_i)\}_{i=1,\dots n}$ be the training dataset sampled according to Definition 2.1. Moreover, let $\mathcal{B}_i = \operatorname{supp}(\boldsymbol{\xi}_i)$ be the support of $\boldsymbol{\xi}_i$. Then with probability at least $1 - n^{-2}$, $\mathcal{B}_i \cap \mathcal{B}_j = \emptyset$ for all $i, j \in [n]$.*

The following lemma studies the relation between sparsity assumption and orthogonality assumption. Sparsity assumption can imply orthogonality assumption with high probability, which states the generality of orthogonal assumption under sparsity assumption.

**Lemma C.2** (Sparsity implies orthogonality). *Suppose $s = \Theta(d^{1/2}n^{-2})$, $d = \operatorname{poly}(n)$. Suppose that the training datasets are generated following Definition. 2.1 except that the non-zero coordinates of noise vectors are uniformly selected from $[d]$ instead of $[d] \setminus \{1\}$. Then, with probability at least $1 - O(1/(n\sqrt{d}))$, we have $\boldsymbol{\mu}$ is orthogonal to $\boldsymbol{\xi}_i$ for all $i \in [n]$.*

*Proof.* We have

$$\mathbb{P}[\exists i \in [n], (\boldsymbol{\xi}_i)_0 \neq 0] = 1 - \left(1 - \frac{s}{d}\right)^n$$
$$\leq 1 - \exp(-2ns/d)$$
$$\leq 2ns/d$$
$$= O(1/(n\sqrt{d})).$$

Note that this lemma can be extended to the signal vectors with constant non-zero entries without modifying the proof idea. $\square$

Let $S_1 = \{i|y_i = 1\}$ and $S_{-1} = \{i|y_i = 1\}$. We have the following lemmas characterizing their sizes.

**Lemma C.3.** *Suppose that $\delta > 0$ and $n \geq 8\log(4/\delta)$. Then with probability at least $1 - \delta$, $|S_1|, |S_{-1}| \in [n/4, 3n/4]|$.*

*Proof.* Since $|S_1| = \sum_{i \in [n]} \mathbb{1}(y_i = 1)$, $|S_{-1}| = \sum_{i \in [n]} \mathbb{1}(y_i = -1)$, we have $\mathbb{E}[|S_1|] = \mathbb{E}[|S_{-1}|] = n/2$. By Hoeffding's inequality, for arbitrary $t > 0$ the following holds:

$$\mathbb{P}||S_1| - \mathbb{E}|S_1|| \geq t \leq 2\exp\left(-\frac{2t^2}{n}\right), \mathbb{P}||S_{-1}| - \mathbb{E}|S_{-1}|| \geq t \leq 2\exp\left(-\frac{2t^2}{n}\right).$$

Setting $t = \sqrt{(n/2)\log(4/\delta)}$ and taking a union bound, it follows that with probability at least $1 - \delta$, we have

$$\left||S_1| - \frac{n}{2}\right| \leq \sqrt{\frac{n}{2}\log\left(\frac{4}{\delta}\right)}, \ \left||S_{-1}| - \frac{n}{2}\right| \leq \sqrt{\frac{n}{2}\log\left(\frac{4}{\delta}\right)}.$$

Therefore, as long as $n \geq 8\log(4/\delta)$, we have $\sqrt{n\log(4/\delta)/2} \leq n/4$ and hence $n/4 \leq |S_1|, |S_{-1}| \leq 3n/4$. $\square$

The following lemma estimates the norms of the noise vectors $\boldsymbol{\xi}_i$ for all $i \in [n]$.

**Lemma C.4.** *Suppose $\delta > 0$ and $s = \Omega(\log(4n/\delta))$. Then with probability at least $1 - \delta$,*

$$\sigma_p^2 s/2 \leq \|\boldsymbol{\xi}_i\|_2^2 \leq 3\sigma_p^2 s/2,$$
$$\sigma_p s/\sqrt{2} \leq \|\boldsymbol{\xi}_i\|_1 \leq \sigma_p s,$$
$$\|\boldsymbol{\xi}_i\|_1 = \sqrt{\frac{2}{\pi}}\sigma_p s \pm O(\sqrt{log(4n/\delta)}s^{-1/2}),$$

*for all $i \in [n]$.*

*Proof.* By Bernstein's inequality, with probability at least $1 - \delta/(2n)$, we have

$$\left|\|\boldsymbol{\xi}_i\|_2^2 - \sigma_p^2 s\right| = O(\sigma_p^2\sqrt{s\log(4n/\delta)}).$$

Therefore, if we set appropriately $s = \Omega(\log(4n/\delta))$, we get

$$\sigma_p^2 s/2 \leq \|\boldsymbol{\xi}_i\|_2^2 \leq 3\sigma_p^2 s/2.$$

Let $k \in \mathcal{B}_i$. Since $\boldsymbol{\xi}_i[k]$ is Gaussian, we have $|\boldsymbol{\xi}_i[k]|$ is sub-gaussian satisfying

$$\||\boldsymbol{\xi}_i[k]| - \mathbb{E}[|\boldsymbol{\xi}_i[k]|]\|_{\psi_2} \leq 2\||\boldsymbol{\xi}_i[k]|\|_{\psi_2}$$
$$= 2\|\boldsymbol{\xi}_i[k]\|_{\psi_2}$$
$$\leq C\sigma_p.$$

By sub-gaussian tail bounds, with probability at least $1 - \delta/2n$, we have

$$\left|\|\boldsymbol{\xi}_i\|_1 - \sqrt{\frac{2}{\pi}}\sigma_p s\right| = O(\sigma_p\sqrt{s\log(4n/\delta)}).$$

Therefore, if we set appropriately $s = \Omega(\log(4n/\delta))$, we get

$$\sigma_p s/\sqrt{2} \leq \|\boldsymbol{\xi}_i\|_1 \leq \sigma_p s.$$

By union bound, we complete the proof. $\square$

**Lemma C.5.** *Suppose $\delta > 0$ and $s = \Omega(n^2 \log(4n/\delta))$. Let $S_1, S_2 \subset [n]$ satisfying $S_1 \cup S_2 = [n]$, $S_1$ and $S_2$ are disjoint, and $|S_1| - |S_2| = c > 0$ for some constant $c$, then*

$$c\sigma_p s/\sqrt{2} \leq \sum_{i \in S_1} \|\boldsymbol{\xi}_i\|_1 - \sum_{i \in S_2} \|\boldsymbol{\xi}_i\|_1 \leq c\sigma_p s.$$

*Proof.* By sub-gaussian tail bounds, with probability at least $1 - \delta/2n$, we have

$$\left| \|\boldsymbol{\xi}_i\|_1 - \sqrt{\frac{2}{\pi}} \sigma_p s \right| = O(\sigma_p \sqrt{s \log(4n/\delta)}).$$

Then we have

$$\sum_{i \in S_1} \|\boldsymbol{\xi}_i\|_1 \geq |S_1| \left( \sqrt{\frac{2}{\pi}} \sigma_p s - O(\sigma_p \sqrt{s \log(4n/\delta)}) \right),$$

$$\sum_{i \in S_2} \|\boldsymbol{\xi}_i\|_1 \leq |S_2| \left( \sqrt{\frac{2}{\pi}} \sigma_p s + O(\sigma_p \sqrt{s \log(4n/\delta)}) \right),$$

which imply

$$\sum_{i \in S_1} \|\boldsymbol{\xi}_i\|_1 - \sum_{i \in S_2} \|\boldsymbol{\xi}_i\|_1 \geq c\sqrt{\frac{2}{\pi}} \sigma_p s - O(n\sigma_p \sqrt{s \log(4n/\delta)}).$$

If we set appropriately $s = \Omega(n^2 \log(4n/\delta))$, we have

$$\sum_{i \in S_1} \|\boldsymbol{\xi}_i\|_1 - \sum_{i \in S_2} \|\boldsymbol{\xi}_i\|_1 \geq c\sigma_p s/\sqrt{2}.$$

Similarly, we have

$$\sum_{i \in S_1} \|\boldsymbol{\xi}_i\|_1 - \sum_{i \in S_2} \|\boldsymbol{\xi}_i\|_1 \leq c\sigma_p s.$$

$\square$

Now turning to network initialization, the following lemmas study the inner product between a randomly initialized value parameter vector $\mathbf{w}_{V,j,r}$ for $j \in \{\pm 1\}$ and $r \in [m_v]$ or query/key parameter vector $\mathbf{w}_{Q,s}/\mathbf{w}_{K,s}$ for $s \in [m_k]$ in attention layer, and the signal/noise vectors in the training data. The calculations characterize how the neural network at initialization randomly captures signal and noise information.

**Lemma C.6.** *Suppose that $s = \Omega(\log(m_v n/\delta))$, $m_v = \Omega(\log(1/\delta))$. Then with probability at least $1 - \delta$,*

$$\sigma_0^2 d/2 \leq \left\| \mathbf{w}_{V,j,r}^{(0)} \right\|_2^2 \leq 3\sigma_0^2 d/2,$$

$$\left| \langle \mathbf{w}_{V,j,r}^{(0)}, \boldsymbol{\mu} \rangle \right| \leq \sqrt{2 \log(12m_v/\delta)} \cdot \sigma_0 \|\boldsymbol{\mu}\|_2,$$

$$\left| \langle \mathbf{w}_{V,j,r}^{(0)}, \boldsymbol{\xi}_i \rangle \right| \leq 2\sqrt{\log(12m_v n/\delta)} \cdot \sigma_0 \sigma_p \sqrt{s},$$

*for all $r \in [m_v]$, $j \in \{\pm 1\}$ and $i \in [n]$.*

**Lemma C.7.** *Suppose that $s = \Omega(\log(m_v n/\delta))$, $m_v = \Omega(\log(1/\delta))$. Then with probability at least $1 - \delta$,*

$$\left| \langle \mathbf{v}^{(0)}, \boldsymbol{\mu} \rangle \right| \leq 2\sqrt{\log(12m_v/\delta)} \cdot m_v^{-1/2} \sigma_0 \|\boldsymbol{\mu}\|_2,$$

$$\left| \langle \mathbf{v}^{(0)}, \boldsymbol{\xi}_i \rangle \right| \leq 2\sqrt{2 \log(12m_v n/\delta)} \cdot m_v^{-1/2} \sigma_0 \sigma_p \sqrt{s},$$

*for all $i \in [n]$.*

**Lemma C.8.** *Suppose that $s = \Omega(\log(m_k n/\delta))$, $m_k = \Omega(\log(1/\delta))$. Then with probability at least $1 - \delta$,*

$$\sigma_0^2 d/2 \leq \left\| \mathbf{w}_{Q,s}^{(0)} \right\|_2^2 \leq 3\sigma_0^2 d/2,$$

$$\sigma_0 \|\boldsymbol{\mu}\|/2 \leq \max_{s \in [m_k]} \left| \langle \mathbf{w}_{Q,s}^{(0)}, \boldsymbol{\mu} \rangle \right| \leq \sqrt{2\log(12m_k/\delta)} \cdot \sigma_0 \|\boldsymbol{\mu}\|_2,$$

$$\sigma_0 \sigma_p \sqrt{s}/4 \leq \max_{s \in [m_k]} \left| \langle \mathbf{w}_{Q,s}^{(0)}, \boldsymbol{\xi}_i \rangle \right| \leq 2\sqrt{\log(12m_k n/\delta)} \cdot \sigma_0 \sigma_p \sqrt{s},$$

$$\sigma_0^2 d/2 \leq \left\| \mathbf{w}_{K,s}^{(0)} \right\|_2^2 \leq 3\sigma_0^2 d/2,$$

$$\sigma_0 \|\boldsymbol{\mu}\|/2 \leq \max_{s \in [m_k]} \left| \langle \mathbf{w}_{K,s}^{(0)}, \boldsymbol{\mu} \rangle \right| \leq \sqrt{2\log(12m_k/\delta)} \cdot \sigma_0 \|\boldsymbol{\mu}\|_2,$$

$$\sigma_0 \sigma_p \sqrt{s}/4 \leq \max_{s \in [m_k]} \left| \langle \mathbf{w}_{K,s}^{(0)}, \boldsymbol{\xi}_i \rangle \right| \leq 2\sqrt{\log(12m_k n/\delta)} \cdot \sigma_0 \sigma_p \sqrt{s},$$

*for all $s \in [m_k]$, and $i \in [n]$.*

*Proof.* The proof of Lemma C.6, C.7 and C.8 are the same as the Lemma B.3 in Cao et al. (2022). $\square$

Define

$$\beta_{\boldsymbol{\xi}} = \max_{i,s,j,r} \left\{ \left| \langle \mathbf{w}_{Q,s}^{(0)}, \boldsymbol{\xi}_i \rangle \right|, \left| \langle \mathbf{w}_{K,s}^{(0)}, \boldsymbol{\xi}_i \rangle \right|, \left| \langle \mathbf{w}_{V,j,r}^{(0)}, \boldsymbol{\xi}_i \rangle \right| \right\}, \tag{2}$$

$$\beta_{\boldsymbol{\mu}} = \max_{s,j,r} \left\{ \left| \langle \mathbf{w}_{Q,s}^{(0)}, \boldsymbol{\mu} \rangle \right|, \left| \langle \mathbf{w}_{K,s}^{(0)}, \boldsymbol{\mu} \rangle \right|, \left| \langle \mathbf{w}_{V,j,r}^{(0)}, \boldsymbol{\mu} \rangle \right| \right\}. \tag{3}$$

The following lemmas study the anti-concentration behaviour in a randomly initialized neural network.

**Lemma C.9.** *Let $x \sim N(0, \sigma^2)$. Then*

$$\mathbb{P}(|x| \leq c) \leq \text{erf}\left( \frac{c}{\sqrt{2}\sigma} \right) \leq \sqrt{1 - \exp(-\frac{2c^2}{\pi\sigma^2})}.$$

*Proof.* The probability density function for $x$ is given by

$$f(x) = \frac{1}{\sqrt{2\pi}\sigma} \exp(-\frac{x^2}{2\sigma^2}).$$

Then we know that

$$\mathbb{P}(|x| \leq c) = \frac{1}{\sqrt{2\pi}\sigma} \int_{-c}^{c} \exp(-\frac{x^2}{2\sigma^2}) dx.$$

By the definition of erf function

$$\text{erf}(c) = \frac{2}{\sqrt{\pi}} \int_0^c \exp(-x^2) dx,$$

and variable substitution yields

$$\text{erf}(\frac{c}{\sqrt{2}\sigma}) = \frac{1}{\sqrt{2\pi}\sigma} \int_0^c \exp(-\frac{x^2}{2\sigma^2}) dx.$$

Therefore, we first conclude $\mathbb{P}(|x| \leq c) = 2\text{erf}(\frac{c}{\sqrt{2}\sigma})$.

Next, by the inequality $\text{erf}(x) \leq \sqrt{1 - \exp(-4x^2/\pi)}$, we finally obtain

$$\mathbb{P}(|x| \leq c) \leq 2\sqrt{1 - \exp(-\frac{2c^2}{\sigma^2\pi})}.$$

$\square$

**Lemma C.10.** *Suppose the results in Lemma C.8 and Lemma C.4 hold. Suppose that $\sigma_p^2 s \geq \frac{32\|\boldsymbol{\mu}\|^2 \log(12m_k/\delta)}{-\pi \log(1-\delta^2/4m_k^2 n^2)}$. Then with probability at least $1 - \delta$, we have*

$$\left|\langle \mathbf{w}_{Q,s}^{(0)}, \boldsymbol{\xi}_i\rangle\right| \geq 2\beta_{\boldsymbol{\mu}}, \left|\langle \mathbf{w}_{K,s}^{(0)}, \boldsymbol{\xi}_i\rangle\right| \geq 2\beta_{\boldsymbol{\mu}},$$

*for all $s \in [m_k]$ and $i \in [n]$.*

*Proof.* Given $\boldsymbol{\xi}_i$, $\langle \mathbf{w}_{Q,s}^{(0)}, \boldsymbol{\xi}_i\rangle \sim N(0, \sigma_0^2 \|\boldsymbol{\xi}_i\|^2)$. By Lemma C.9, we have

$$\mathbb{P}\left(\left|\langle \mathbf{w}_{Q,s}^{(0)}, \boldsymbol{\xi}_i\rangle\right| \leq c\right) \leq \sqrt{1 - \exp(-\frac{2c^2}{\pi\sigma_0^2 \|\boldsymbol{\xi}_i\|^2})}.$$

Let $c = 2\beta_{\boldsymbol{\mu}}$. By Lemma C.8, $\beta_{\boldsymbol{\mu}}$ can be upper bounded by $\sqrt{2\log(12m_k/\delta)} \cdot \sigma_0 \|\boldsymbol{\mu}\|$. By Lemma C.4,

$$\sigma_p^2 s/2 \leq \|\boldsymbol{\xi}_i\|^2 \leq 3\sigma_p^2 s/2,$$

for all $i \in [n]$. Therefore, if we set appropriately

$$\sigma_p^2 s \geq \frac{32 \|\boldsymbol{\mu}\|^2 \log(12m_k/\delta)}{-\pi \log(1 - \delta^2/4m_k^2 n^2)},$$

then we have

$$\mathbb{P}\left(\left|\langle \mathbf{w}_{Q,s}^{(0)}, \boldsymbol{\xi}_i\rangle\right| \leq 2\beta_{\boldsymbol{\mu}}\right) \leq \frac{\delta}{2m_k n}.$$

The analysis for $\langle \mathbf{w}_{K,s}^{(0)}, \boldsymbol{\xi}_i\rangle$ is exactly the same. By union bound, we get the final result. $\square$

**Lemma C.11.** *Suppose the results in Lemma C.8 and Lemma C.4 hold. Suppose that $m_v \geq \frac{256 \log^2(12m_k n/\delta)}{\pi^2 \log^2(1-\delta^2/4m_k^2 n^2)}$. Then with probability at least $1 - \delta$, we have*

$$\left|\langle \mathbf{w}_{Q,s}^{(0)}, \boldsymbol{\xi}_i\rangle\right| \geq \beta_{\boldsymbol{\xi}} m_v^{-1/4}, \left|\langle \mathbf{w}_{K,s}^{(0)}, \boldsymbol{\xi}_i\rangle\right| \geq \beta_{\boldsymbol{\xi}} m_v^{-1/4},$$

*for all $s \in [m_k]$ and $i \in [n]$.*

*Proof.* Given $\boldsymbol{\xi}_i$, $\langle \mathbf{w}_{Q,s}^{(0)}, \boldsymbol{\xi}_i\rangle \sim N(0, \sigma_0^2 \|\boldsymbol{\xi}_i\|^2)$. By Lemma C.9, we have

$$\mathbb{P}\left(\left|\langle \mathbf{w}_{Q,s}^{(0)}, \boldsymbol{\xi}_i\rangle\right| \leq c\right) \leq \sqrt{1 - \exp(-\frac{2c^2}{\pi\sigma_0^2 \|\boldsymbol{\xi}_i\|^2})}.$$

Let $c = \beta_{\boldsymbol{\xi}} m_v^{-1/4}$. By Lemma C.8, $\beta_{\boldsymbol{\xi}}$ can be upper bounded by $2\sqrt{\log(12m_k n/\delta)} \cdot \sigma_0 \sigma_p \sqrt{s}$. By Lemma C.4,

$$\sigma_p^2 s/2 \leq \|\boldsymbol{\xi}_i\|^2 \leq 3\sigma_p^2 s/2,$$

for all $i \in [n]$. Therefore, if we set appropriately

$$m_v \geq \frac{256 \log^2(12m_k n/\delta)}{\pi^2 \log^2(1 - \delta^2/4m_k^2 n^2)},$$

then we have

$$\mathbb{P}\left(\left|\langle \mathbf{w}_{Q,s}^{(0)}, \boldsymbol{\xi}_i\rangle\right| \leq \beta_{\boldsymbol{\xi}} m_v^{-1/4}\right) \leq \frac{\delta}{2m_k n}.$$

The analysis for $\langle \mathbf{w}_{K,s}^{(0)}, \boldsymbol{\xi}_i\rangle$ is exactly the same. By union bound, we get the final result. $\square$

**Lemma C.12.** *Suppose that* $s \geq n^3 \left( \frac{2}{-\pi \log(1-\delta^2/4m_k^2)} \right)^{3/2}$*. Then with probability at least* $1 - \delta$*, we have*

$$\left| \langle \mathbf{w}_{Q,s}^{(0)}, \boldsymbol{\mu} \rangle \right|, \left| \langle \mathbf{w}_{K,s}^{(0)}, \boldsymbol{\mu} \rangle \right| \geq \sigma_0 n s^{-1/3} \|\boldsymbol{\mu}\|,$$

*for all* $s \in [m_k]$*.*

*Proof.* Note that $\langle \mathbf{w}_{Q,s}^{(0)}, \boldsymbol{\mu} \rangle \sim N(0, \sigma_0^2 \|\boldsymbol{\mu}\|^2)$. By Lemma C.9, we have

$$\mathbb{P} \left( \left| \langle \mathbf{w}_{Q,s}^{(0)}, \boldsymbol{\mu} \rangle \right| \leq c \right) \leq \sqrt{1 - \exp(-\frac{2c^2}{\pi \sigma_0^2 \|\boldsymbol{\mu}\|^2})}.$$

Let $c = \sigma_0 n s^{-1/3} \|\boldsymbol{\mu}\|$. If we set appropriately

$$s \geq n^3 \left( \frac{2}{-\pi \log(1 - \delta^2/4m_k^2)} \right)^{3/2},$$

then we have

$$\mathbb{P} \left( \left| \langle \mathbf{w}_{Q,s}^{(0)}, \boldsymbol{\mu} \rangle \right| \leq \sigma_0 n s^{-1/3} \|\boldsymbol{\mu}\| \right) \leq \frac{\delta}{2m_k}.$$

The analysis for $\langle \mathbf{w}_{K,s}^{(0)}, \boldsymbol{\mu} \rangle$ is exactly the same. By union bound, we get the final result. $\qquad \square$

The following lemma studies the magnitude of output of softmax operation.

**Lemma C.13.** *Suppose that* $s = \Omega(\log(m_k n/\delta))$*,* $m_k = \Omega(\log(1/\delta))$*. Then with probability at least* $1 - \delta$*,*

$$\left| s_{i,11}^{(0)} - 1/2 \right| \leq 16 m_k \log(12 m_k n/\delta) \cdot \sigma_0^2 \max \left\{ \sigma_p^2 s, \|\boldsymbol{\mu}^2\| \right\},$$

$$\left| s_{i,21}^{(0)} - 1/2 \right| \leq 16 m_k \log(12 m_k n/\delta) \cdot \sigma_0^2 \max \left\{ \sigma_p^2 s, \|\boldsymbol{\mu}^2\| \right\},$$

*for all* $i \in [n]$*. When* $m_k \log(12 m_k n/\delta) \cdot \sigma_0^2 \max \left\{ \sigma_p^2 s, \|\boldsymbol{\mu}^2\| \right\} = o(1)$*, we can simplify the results into*

$$s_{i,11}^{(0)}, s_{i,21}^{(0)} = 1/2 + o(1),$$

*for all* $i \in [n]$*.*

*Proof.* We directly have

$$\left| \sum_{s=1}^{m_k} \langle \mathbf{w}_{Q,s}^{(0)}, \boldsymbol{\mu} \rangle \langle \mathbf{w}_{K,s}^{(0)}, \boldsymbol{\mu} \rangle \right| \leq m_k \beta_{\boldsymbol{\mu}}^2,$$

$$\left| \sum_{s=1}^{m_k} \langle \mathbf{w}_{Q,s}^{(0)}, y_i \boldsymbol{\xi}_i \rangle \langle \mathbf{w}_{K,s}^{(0)}, y_i \boldsymbol{\xi}_i \rangle \right| \leq m_k \beta_{\boldsymbol{\xi}}^2,$$

$$\left| \sum_{s=1}^{m_k} \langle \mathbf{w}_{Q,s}^{(0)}, \boldsymbol{\mu} \rangle \langle \mathbf{w}_{K,s}^{(0)}, y_i \boldsymbol{\xi}_i \rangle \right|, \left| \sum_{s=1}^{m_k} \langle \mathbf{w}_{Q,s}^{(0)}, y_i \boldsymbol{\xi}_i \rangle \langle \mathbf{w}_{K,s}^{(0)}, \boldsymbol{\mu} \rangle \right| \leq m_k \beta_{\boldsymbol{\mu}} \beta_{\boldsymbol{\xi}},$$

for all $i \in [n]$. Then we have

$$\begin{aligned}
s_{i,21}^{(0)} &\leq \frac{\exp \left( m_k \beta_{\boldsymbol{\xi}} \beta_{\boldsymbol{\mu}} \right)}{\exp \left( m_k \beta_{\boldsymbol{\xi}} \beta_{\boldsymbol{\mu}} \right) + \exp \left( -m_k \beta_{\boldsymbol{\xi}}^2 \right)} \\
&\leq \frac{1}{1 + \exp \left( -m_k \beta_{\boldsymbol{\xi}} (\beta_{\boldsymbol{\xi}} + \beta_{\boldsymbol{\mu}}) \right)} \\
&\leq 1/2 + 2 m_k \beta_{\boldsymbol{\xi}} (\beta_{\boldsymbol{\xi}} + \beta_{\boldsymbol{\mu}}),
\end{aligned}$$

and

$$s_{i,21}^{(0)} \geq \frac{\exp\left(-m_k \beta_{\boldsymbol{\xi}} \beta_{\boldsymbol{\mu}}\right)}{\exp\left(-m_k \beta_{\boldsymbol{\xi}} \beta_{\boldsymbol{\mu}}\right) + \exp\left(m_k \beta_{\boldsymbol{\xi}}^2\right)}$$

$$\geq \frac{1}{1 + \exp\left(m_k \beta_{\boldsymbol{\xi}}(\beta_{\boldsymbol{\xi}} + \beta_{\boldsymbol{\mu}})\right)}$$

$$\geq 1/2 - 2 m_k \beta_{\boldsymbol{\xi}}(\beta_{\boldsymbol{\xi}} + \beta_{\boldsymbol{\mu}}),$$

for all $i \in [n]$. Similarly, we have

$$1/2 - 2 m_k \beta_{\boldsymbol{\mu}}(\beta_{\boldsymbol{\xi}} + \beta_{\boldsymbol{\mu}}) \leq s_{i,11}^{(0)} \leq 1/2 + 2 m_k \beta_{\boldsymbol{\mu}}(\beta_{\boldsymbol{\xi}} + \beta_{\boldsymbol{\mu}})$$

for all $i \in [n]$. By Lemma C.8, with probability $1 - \delta$, $\beta_{\boldsymbol{\xi}}$ can be bounded by $2\sqrt{\log(12 m_k n/\delta)} \cdot \sigma_0 \sigma_p \sqrt{s}$ and $\beta_{\boldsymbol{\mu}}$ can be bounded by $\sqrt{2\log(12 m_k/\delta)} \cdot \sigma_0 \|\boldsymbol{\mu}\|$, and

$$\max\{\beta_{\boldsymbol{\mu}}, \beta_{\boldsymbol{\xi}}\} \leq 2\sqrt{\log(12 m_k n/\delta)} \cdot \sigma_0 \max\{\sigma_p \sqrt{s}, \|\boldsymbol{\mu}\|\},$$

which completes the proof. $\qquad \square$

**Lemma C.14.** *Suppose that $\sigma_p s / \|\boldsymbol{\mu}\| = \Omega(d^{1/5})$. Let*

$$X_i := \langle \mathbf{w}_{Q,s}^{(0)}, y_i \boldsymbol{\xi}_i \rangle + \langle \mathbf{w}_{Q,s}^{(0)}, \boldsymbol{\mu} \rangle,$$

$$Y_i := -\langle \mathbf{w}_{K,s}^{(0)}, y_i \boldsymbol{\xi}_i \rangle + \langle \mathbf{w}_{K,s}^{(0)}, \boldsymbol{\mu} \rangle.$$

*Then with probability at least $1 - nd^{-1/5}$, for all $i \in [n]$, we have*

$$\frac{\sigma_p s - \|\boldsymbol{\mu}\|}{\sigma_p s + \|\boldsymbol{\mu}\|} \cdot \left(\langle \mathbf{w}_{Q,s}^{(0)}, y_i \boldsymbol{\xi}_i \rangle + \langle \mathbf{w}_{Q,s}^{(0)}, \boldsymbol{\mu} \rangle\right) \geq -\langle \mathbf{w}_{K,s}^{(0)}, y_i \boldsymbol{\xi}_i \rangle + \langle \mathbf{w}_{K,s}^{(0)}, \boldsymbol{\mu} \rangle,$$

*or*

$$-\langle \mathbf{w}_{K,s}^{(0)}, y_i \boldsymbol{\xi}_i \rangle + \langle \mathbf{w}_{K,s}^{(0)}, \boldsymbol{\mu} \rangle \geq \frac{\sigma_p s + \|\boldsymbol{\mu}\|}{\sigma_p s - \|\boldsymbol{\mu}\|} \cdot \left(\langle \mathbf{w}_{Q,s}^{(0)}, y_i \boldsymbol{\xi}_i \rangle + \langle \mathbf{w}_{Q,s}^{(0)}, \boldsymbol{\mu} \rangle\right).$$

*Proof.* We only need to show that for all $i \in [n]$ the probability of following event can be controlled.

$$\frac{\sigma_p s - \|\boldsymbol{\mu}\|}{\sigma_p s + \|\boldsymbol{\mu}\|} X_i < Y_i < \frac{\sigma_p s + \|\boldsymbol{\mu}\|}{\sigma_p s - \|\boldsymbol{\mu}\|} X_i.$$

Conditioning on $y_i \boldsymbol{\xi}_i$, we have $X_i \overset{d}{=} Y_i \sim N(0, \sigma_0^2(\|\boldsymbol{\mu}\|^2 + \|\boldsymbol{\xi}_i\|^2))$, this is because $\langle \boldsymbol{\xi}_i, \boldsymbol{\mu} \rangle = 0$. Denote

$$\sigma^2 := \sigma_0^2(\|\boldsymbol{\mu}\|^2 + \|\boldsymbol{\xi}_i\|^2),$$

$$D := \left\{(x, y) \in \mathbb{R}^2 : \frac{\sigma_p s - \|\boldsymbol{\mu}\|}{\sigma_p s + \|\boldsymbol{\mu}\|} x < y < \frac{\sigma_p s + \|\boldsymbol{\mu}\|}{\sigma_p s - \|\boldsymbol{\mu}\|} x\right\}.$$

Therefore, we have

$$\mathbb{P}\left[\frac{\sigma_p s - \|\boldsymbol{\mu}\|}{\sigma_p s + \|\boldsymbol{\mu}\|} X_i < Y_i < \frac{\sigma_p s + \|\boldsymbol{\mu}\|}{\sigma_p s - \|\boldsymbol{\mu}\|} X_i | y_i \boldsymbol{\xi}_i\right]$$

$$= \iint_D 1 \cdot dP_{X_i} dP_{Y_i}$$

$$= \iint_D \frac{1}{2\pi\sigma^2} \exp\left(-\frac{x^2 + y^2}{2\sigma^2}\right) dx dy$$

$$= \iint_D \frac{1}{2\pi\sigma^2} r \exp\left(-\frac{r^2}{2\sigma^2}\right) dr d\theta$$

$$= \frac{\pi}{2} - 2 \arctan\left(\frac{\sigma_p s - \|\boldsymbol{\mu}\|}{\sigma_p s + \|\boldsymbol{\mu}\|}\right)$$

$$\leq \frac{4}{3} \frac{2\|\boldsymbol{\mu}\|}{\sigma_p s + \|\boldsymbol{\mu}\|}$$

$$\leq O(d^{-1/5}).$$

where the fifth step is by $\arctan(1+x) \geq \frac{\pi}{4} + \frac{2}{3}(x-1)$ when $x = 1 - o(1)$, and the last step is by the condition $\sigma_p s / \|\boldsymbol{\mu}\| = \Omega(d^{1/5})$. Then, by integrating over $y_i \boldsymbol{\xi}_i$ and union bound, we have

$$\mathbb{P}\left[\exists\, i \in [n],\; \frac{\sigma_p s - \|\boldsymbol{\mu}\|}{\sigma_p s + \|\boldsymbol{\mu}\|} X_i < Y_i < \frac{\sigma_p s + \|\boldsymbol{\mu}\|}{\sigma_p s - \|\boldsymbol{\mu}\|} X_i\right] \leq O(nd^{-1/5}).$$

$\square$

Next, let

$$S^{(0)}_{s,1,K+,Q-} = \left\{ i \in [n] : \langle \mathbf{w}^{(0)}_{K,s}, \boldsymbol{\xi}_i \rangle > 0, \langle \mathbf{w}^{(0)}_{Q,s}, \boldsymbol{\xi}_i \rangle < 0, y_i = 1 \right\}, \tag{4}$$

$$S^{(0)}_{i,K+,Q-} = \left\{ s \in [m_k] : \langle \mathbf{w}^{(0)}_{K,s}, \boldsymbol{\xi}_i \rangle > 0, \langle \mathbf{w}^{(0)}_{Q,s}, \boldsymbol{\xi}_i \rangle < 0 \right\}, \tag{5}$$

$$S^{(0)}_{\boldsymbol{\mu},K+,Q-} = \left\{ s \in [m_k] : \langle \mathbf{w}^{(0)}_{K,s}, \boldsymbol{\mu} \rangle > 0, \langle \mathbf{w}^{(0)}_{Q,s}, \boldsymbol{\mu} \rangle < 0 \right\}, \tag{6}$$

$$S^{(0)}_{s,K+,Q-} = \left\{ i \in [n] : \langle \mathbf{w}^{(0)}_{K,s}, y_i \boldsymbol{\xi}_i \rangle > 0, \langle \mathbf{w}^{(0)}_{Q,s}, y_i \boldsymbol{\xi}_i \rangle < 0, \right\} = S^{(0)}_{s,1,K+,Q-} \cup S^{(0)}_{s,-1,K-,Q+}. \tag{7}$$

## D  GRADIENT

In this section, we present the gradients of the loss $L_S$ with respect to $\mathbf{w}_{V,j,r}$, $\mathbf{w}_{Q,s}$ and $\mathbf{w}_{K,s}$ for $s \in [m_k]$, $j \in \{\pm 1\}$ and $r \in [m_v]$. Our analysis starts from a more general form:

$$F_j(\mathbf{W}, \mathbf{X}) := \frac{1}{m_v} \sum_{l=1}^{L} \mathbf{1}_{m_v}^\top \sigma\left(\mathbf{W}_{V,j} \mathbf{X} \operatorname{softmax}\left(\mathbf{X}^\top \mathbf{W}_K^\top \mathbf{W}_Q \mathbf{x}^{(l)}\right)\right),$$

and finally focus on $L = 2$ and $\sigma(x) = x$ case, which is exactly the model defined in Eq. (1).

**Lemma D.1** (softmax derivative). *Let $(s_1, \ldots, s_n) = \operatorname{softmax}(z_1, \ldots, z_n)$, then we have*

$$\frac{\partial s_i}{\partial z_j} = s_i \left(\mathbb{1}\left\{i = j\right\} - s_j\right).$$

**Lemma D.2** (Gradient w.r.t. value parameters). *Consider the transformer model defined in Eq. (1). Let $(\mathbf{X}, y)$ be a data point. The derivative of $F_j(\mathbf{W}, \mathbf{X})$ with respect to $\mathbf{w}_{V,j,r}$ is:*

$$\nabla_{\mathbf{w}_{V,j,r}} F_j(\mathbf{W}, \mathbf{X}) = \frac{1}{m_v} \left[(s_{1,1} + s_{2,1}) \mathbf{x}^{(1)} + (s_{1,2} + s_{2,2}) \mathbf{x}^{(2)}\right].$$

*Proof.* We have

$$\nabla_{\mathbf{w}_{V,j,r}} F_j(\mathbf{W}, \mathbf{X}) = \frac{1}{m_v} \sum_{l \in [L]} \sum_{a=1}^{L} s_{l,a} \sigma'\left(\left\langle \mathbf{w}_{V,j,r}, \mathbf{x}^{(a)} \right\rangle\right) \mathbf{x}^{(a)}$$

$$= \frac{1}{m_v} \left[(s_{1,1} + s_{2,1}) \sigma'\left(\left\langle \mathbf{w}_{V,j,r}, \mathbf{x}^{(1)} \right\rangle\right) \mathbf{x}^{(1)} + (s_{1,2} + s_{2,2}) \sigma'\left(\left\langle \mathbf{w}_{V,j,r}, \mathbf{x}^{(2)} \right\rangle\right) \mathbf{x}^{(2)}\right]$$

$$= \frac{1}{m_v} \left[(s_{1,1} + s_{2,1}) \mathbf{x}^{(1)} + (s_{1,2} + s_{2,2}) \mathbf{x}^{(2)}\right],$$

where the second step is by $L = 2$, the third step is by $\sigma(x) = x$. $\qquad \square$

**Lemma D.3** (Gradient w.r.t. pre-softmax quantity). *Consider the transformer model defined in Eq. (1). Let $(\mathbf{X}, y)$ be a data point. The derivative of $F_j(\mathbf{W}, \mathbf{X})$ with respect to $z_{l,a}$ is:*

$$\frac{\partial F_j(\mathbf{W}, \mathbf{X})}{\partial z_{l,a}} = \frac{1}{m_v} \sum_{r \in [m_v]} s_{l,1} s_{l,2} \left[(\langle \mathbf{w}_{V,j,r}, \mathbf{x}_a \rangle) - (\langle \mathbf{w}_{V,j,r}, \mathbf{x}_{3-a} \rangle)\right].$$

*Proof.* We have

$$\frac{\partial F_j(\mathbf{W}, \mathbf{x})}{\partial z_{l,a}} = \frac{1}{m_v} \sum_{r \in [m_v]} \sum_{b=1}^{L} \frac{\partial s_{l,b}}{\partial z_{l,a}} \sigma\left(\left\langle \mathbf{w}_{V,j,r}, \mathbf{x}^{(b)} \right\rangle\right)$$

$$= \frac{1}{m_v} \sum_{r \in [m_v]} \sum_{b \neq a} s_{l,a} s_{l,b} \left[\sigma\left(\left\langle \mathbf{w}_{V,j,r}, \mathbf{x}^{(a)} \right\rangle\right) - \sigma\left(\left\langle \mathbf{w}_{V,j,r}, \mathbf{x}^{(b)} \right\rangle\right)\right]$$

$$= \frac{1}{m_v} \sum_{r \in [m_v]} s_{l,a} \sum_{b \in [L]} s_{l,b} \left[\sigma\left(\left\langle \mathbf{w}_{V,j,r}, \mathbf{x}^{(a)} \right\rangle\right) - \sigma\left(\left\langle \mathbf{w}_{V,j,r}, \mathbf{x}^{(b)} \right\rangle\right)\right]$$

$$= \frac{1}{m_v} \sum_{r \in [m_v]} s_{l,1} s_{l,2} \left[\sigma\left(\langle \mathbf{w}_{V,j,r}, \mathbf{x}_a \rangle\right) - \sigma\left(\langle \mathbf{w}_{V,j,r}, \mathbf{x}_{3-a} \rangle\right)\right]$$

$$= \frac{1}{m_v} \sum_{r \in [m_v]} s_{l,1} s_{l,2} \left[(\langle \mathbf{w}_{V,j,r}, \mathbf{x}_a \rangle) - (\langle \mathbf{w}_{V,j,r}, \mathbf{x}_{3-a} \rangle)\right],$$

where the penultimate step is by $L = 2$, the final step is by $\sigma(x) = x$. $\qquad \square$

**Lemma D.4** (Gradient w.r.t. query parameters). *Consider the transformer model defined in Eq.* (1).
*Let* $(\mathbf{X}, y)$ *be a data point. The derivative of* $F_j(\mathbf{W}, \mathbf{X})$ *with respect to* $\mathbf{w}_{Q,s}$ *is:*

$$\nabla_{\mathbf{w}_{Q,s}} F_j(\mathbf{W}, \mathbf{X}) = \frac{1}{m_v} \sum_{r \in [m_v]} \left\langle \mathbf{w}_{V,j,r}, \mathbf{x}^{(1)} - \mathbf{x}^{(2)} \right\rangle \cdot \left\langle \mathbf{w}_{K,s}, \mathbf{x}^{(1)} - \mathbf{x}^{(2)} \right\rangle \cdot \left( s_{1,1} s_{1,2} \mathbf{x}^{(1)} + s_{2,1} s_{2,2} \mathbf{x}^{(2)} \right)$$

$$= \left\langle \bar{\mathbf{w}}_{V,j}, \mathbf{x}^{(1)} - \mathbf{x}^{(2)} \right\rangle \cdot \left\langle \mathbf{w}_{K,s}, \mathbf{x}^{(1)} - \mathbf{x}^{(2)} \right\rangle \cdot \left( s_{1,1} s_{1,2} \mathbf{x}^{(1)} + s_{2,1} s_{2,2} \mathbf{x}^{(2)} \right).$$

*Proof.* By Lemma D.3, we have

$$\nabla_{\mathbf{w}_{Q,s}} F_j(\mathbf{W}, \mathbf{X}) = \sum_{l,a} \frac{\partial F_j(\mathbf{W}, \mathbf{X})}{\partial z_{l,a}} \nabla_{\mathbf{w}_{Q,s}} z_{l,a}$$

$$= \sum_{l,a} \left[ \frac{1}{m_v} \sum_{m \in [d_v]} s_{l,a} \sum_{b \in [L]} s_{l,b} \left[ \sigma \left( \left\langle \mathbf{w}_{V,j,r}, \mathbf{x}^{(a)} \right\rangle \right) - \sigma \left( \left\langle \mathbf{w}_{V,j,r}, \mathbf{x}^{(b)} \right\rangle \right) \right] \right] \cdot \left\langle \mathbf{w}_{K,s}, \mathbf{x}^{(a)} \right\rangle \mathbf{x}^{(l)}$$

$$= \sum_{l,a} \frac{1}{m_v} \sum_{r \in [m_v]} s_{l,1} s_{l,2} \left[ \sigma \left( \left\langle \mathbf{w}_{V,j,r}, \mathbf{x}^{(a)} \right\rangle \right) - \sigma \left( \left\langle \mathbf{w}_{V,j,r}, \mathbf{x}^{(3-a)} \right\rangle \right) \right] \cdot \left\langle \mathbf{w}_{K,s}, \mathbf{x}^{(a)} \right\rangle \mathbf{x}^{(l)}$$

$$= \frac{1}{m_v} \sum_{r \in [m_v]} s_{1,1} s_{1,2} \left[ \sigma \left( \left\langle \mathbf{w}_{V,j,r}, \mathbf{x}^{(1)} \right\rangle \right) - \sigma \left( \left\langle \mathbf{w}_{V,j,r}, \mathbf{x}^{(2)} \right\rangle \right) \right] \cdot \left\langle \mathbf{w}_{K,s}, \mathbf{x}^{(1)} \right\rangle \mathbf{x}^{(1)}$$

$$+ s_{1,1} s_{1,2} \left[ \sigma \left( \left\langle \mathbf{w}_{V,j,r}, \mathbf{x}^{(2)} \right\rangle \right) - \sigma \left( \left\langle \mathbf{w}_{V,j,r}, \mathbf{x}^{(1)} \right\rangle \right) \right] \cdot \left\langle \mathbf{w}_{K,s}, \mathbf{x}^{(2)} \right\rangle \mathbf{x}^{(1)}$$

$$+ s_{2,1} s_{2,2} \left[ \sigma \left( \left\langle \mathbf{w}_{V,j,r}, \mathbf{x}^{(1)} \right\rangle \right) - \sigma \left( \left\langle \mathbf{w}_{V,j,r}, \mathbf{x}^{(2)} \right\rangle \right) \right] \cdot \left\langle \mathbf{w}_{K,s}, \mathbf{x}^{(1)} \right\rangle \mathbf{x}^{(2)}$$

$$+ s_{2,1} s_{2,2} \left[ \sigma \left( \left\langle \mathbf{w}_{V,j,r}, \mathbf{x}^{(2)} \right\rangle \right) - \sigma \left( \left\langle \mathbf{w}_{V,j,r}, \mathbf{x}^{(1)} \right\rangle \right) \right] \cdot \left\langle \mathbf{w}_{K,s}, \mathbf{x}^{(2)} \right\rangle \mathbf{x}^{(2)}$$

$$= \frac{1}{m_v} \sum_{r \in [m_v]} \left[ \sigma \left( \left\langle \mathbf{w}_{V,j,r}, \mathbf{x}^{(1)} \right\rangle \right) - \sigma \left( \left\langle \mathbf{w}_{V,j,r}, \mathbf{x}^{(2)} \right\rangle \right) \right]$$

$$\cdot \left\langle \mathbf{w}_{K,s}, \mathbf{x}^{(1)} - \mathbf{x}^{(2)} \right\rangle \cdot \left( s_{1,1} s_{1,2} \mathbf{x}^{(1)} + s_{2,1} s_{2,2} \mathbf{x}^{(2)} \right)$$

$$= \frac{1}{m_v} \sum_{r \in [m_v]} \left\langle \mathbf{w}_{V,j,r}, \mathbf{x}^{(1)} - \mathbf{x}^{(2)} \right\rangle \cdot \left\langle \mathbf{w}_{K,s}, \mathbf{x}^{(1)} - \mathbf{x}^{(2)} \right\rangle \cdot \left( s_{1,1} s_{1,2} \mathbf{x}^{(1)} + s_{2,1} s_{2,2} \mathbf{x}^{(2)} \right)$$

$$= \left\langle \bar{\mathbf{w}}_{V,j}, \mathbf{x}^{(1)} - \mathbf{x}^{(2)} \right\rangle \cdot \left\langle \mathbf{w}_{K,s}, \mathbf{x}^{(1)} - \mathbf{x}^{(2)} \right\rangle \cdot \left( s_{1,1} s_{1,2} \mathbf{x}^{(1)} + s_{2,1} s_{2,2} \mathbf{x}^{(2)} \right),$$

where the third step is by $L = 2$, the last step is by $\sigma(x) = x$. $\square$

**Lemma D.5** (Gradient w.r.t. key parameters). *Consider the transformer model defined in Eq.* (1).
*Let* $(\mathbf{X}, y)$ *be a data point. Similarly, The derivative of* $F_j(\mathbf{W}, \mathbf{X})$ *with respect to* $\mathbf{w}_{K,s}$ *is:*

$$\nabla_{\mathbf{w}_{K,s}} F_j(\mathbf{W}, \mathbf{X}) = \left\langle \bar{\mathbf{w}}_{V,j}, \mathbf{x}^{(1)} - \mathbf{x}^{(2)} \right\rangle \cdot \left\langle \mathbf{w}_{Q,s}, s_{1,1} s_{1,2} \mathbf{x}^{(1)} + s_{2,1} s_{2,2} \mathbf{x}^{(2)} \right\rangle \cdot \left( \mathbf{x}^{(1)} - \mathbf{x}^{(2)} \right).$$

*Proof.* By Lemma D.3, we have

$$\nabla_{\mathbf{w}_{K,s}} F_j(\mathbf{W}, \mathbf{X}) = \sum_{l,a} \left[ \frac{1}{m_v} \sum_{r \in [m_v]} s_{l,a} \sum_{b \in [L]} s_{l,b} \left[ \sigma \left( \left\langle \mathbf{w}_{V,j,r}, \mathbf{x}^{(a)} \right\rangle \right) - \sigma \left( \left\langle \mathbf{w}_{V,j,r}, \mathbf{x}^{(b)} \right\rangle \right) \right] \right] \cdot \left\langle \mathbf{w}_{Q,s}, \mathbf{x}^{(l)} \right\rangle \mathbf{x}^{(a)}$$

$$= \frac{1}{m_v} \sum_{r \in [m_v]} \left[ \sigma \left( \left\langle \mathbf{w}_{V,j,r}, \mathbf{x}^{(1)} \right\rangle \right) - \sigma \left( \left\langle \mathbf{w}_{V,j,r}, \mathbf{x}^{(2)} \right\rangle \right) \right]$$

$$\cdot \left\langle \mathbf{w}_{Q,s}, s_{1,1} s_{1,2} \mathbf{x}^{(1)} + s_{2,1} s_{2,2} \mathbf{x}^{(2)} \right\rangle \cdot \left( \mathbf{x}^{(1)} - \mathbf{x}^{(2)} \right)$$

$$= \left\langle \bar{\mathbf{w}}_{V,j}, \mathbf{x}^{(1)} - \mathbf{x}^{(2)} \right\rangle \cdot \left\langle \mathbf{w}_{Q,s}, s_{1,1} s_{1,2} \mathbf{x}^{(1)} + s_{2,1} s_{2,2} \mathbf{x}^{(2)} \right\rangle \cdot \left( \mathbf{x}^{(1)} - \mathbf{x}^{(2)} \right),$$

where the second step is by $L = 2$, the third step is by $\sigma(x) = x$. $\square$

**Lemma D.6.** *Consider the transformer model defined in Eq.* (1). *Let* $\mathbf{w}_{V,j,r}^{(t)}$ *for* $j \in \{\pm 1\}$, $r \in [m_v]$ *be the value parameters of the TF at the* $t$-*th iteration. If we use sign gradient descent, the update formula of* $\mathbf{w}_{V,j,r}^{(t)}$ *is*

$$\mathbf{w}_{V,j,r}^{(t+1)} = \mathbf{w}_{V,j,r}^{(t)} - \eta \operatorname{sgn}\left(\sum_{i \in [n]} \ell_i'^{(t)} y_i j \left[(s_{i,11}^{(t)} + s_{i,21}^{(t)})y_i \boldsymbol{\mu} + (s_{i,12}^{(t)} + s_{i,22}^{(t)})\boldsymbol{\xi}_i\right]\right).$$

*Proof.* This is by Lemma D.2. □

**Lemma D.7.** *Consider the transformer model defined in Eq.* (1). *Let* $\mathbf{w}_{Q,s}^{(t)}$ *and* $\mathbf{w}_{K,s}^{(t)}$ *for* $s \in [m_k]$ *be the query and key parameters of the TF at the* $t$-*th iteration. If we use sign gradient descent, the update formula of* $\mathbf{w}_{Q,s}^{(t)}$ *and* $\mathbf{w}_{K,s}^{(t)}$ *is*

$$\mathbf{w}_{Q,s}^{(t+1)} = \mathbf{w}_{Q,s}^{(t)} - \eta \operatorname{sgn}\left[\sum_{i \in [n]} \ell_i'^{(t)} y_i \langle \bar{\mathbf{w}}_{V,1}^{(t)} - \bar{\mathbf{w}}_{V,-1}^{(t)}, y_i \boldsymbol{\mu} - \boldsymbol{\xi}_i \rangle \langle \mathbf{w}_{K,s}^{(t)}, y_i \boldsymbol{\mu} - \boldsymbol{\xi}_i \rangle (s_{11}^{(t)} s_{12}^{(t)} y_i \boldsymbol{\mu} + s_{21}^{(t)} s_{22}^{(t)} \boldsymbol{\xi}_i)\right].$$

$$\mathbf{w}_{K,s}^{(t+1)} = \mathbf{w}_{K,s}^{(t)} - \eta \operatorname{sgn}\left[\sum_{i \in [n]} \ell_i'^{(t)} y_i \langle \bar{\mathbf{w}}_{V,1}^{(t)} - \bar{\mathbf{w}}_{V,-1}^{(t)}, y_i \boldsymbol{\mu} - \boldsymbol{\xi}_i \rangle \langle \mathbf{w}_{Q,s}^{(t)}, s_{11}^{(t)} s_{12}^{(t)} y_i \boldsymbol{\mu} + s_{21}^{(t)} s_{22}^{(t)} \boldsymbol{\xi}_i \rangle (y_i \boldsymbol{\mu} - \boldsymbol{\xi}_i)\right].$$

*Proof.* This is by Lemma D.4, Lemma D.5. □

# E  PROOFS

In this section, we give a detailed analysis about the dynamics of sign gradient descent on our transformer models Eq. (1) with our data model in Def. 2.1. These results are based on Condition 3.1 and the conclusions in Appendix C, which hold with high probability. Denote by $\mathcal{E}_{\text{prelim}}$ the event that all the results in Appendix C hold (for a given $\delta$, we see $\mathbb{P}(\mathcal{E}_{\text{prelim}}) \geq 1 - 10\delta - n^{-2} - nd^{-1/5}$ by a union bound). For simplicity and clarity, we state all the results in this and the following sections conditional on $\mathcal{E}_{\text{prelim}}$.

## E.1  TECHNIQUE OVERVIEW

For value, the dynamics is linear under sign gradient descent, which means the increment is exactly constant across different iterations and different neurons, and up to a constant multiplier across different samples.

For query and key, the dynamics is complicated. The key point is that we utilize the different increasing speed of different core quantities to divide the whole timeline into many stages where we can care about only one of or some of quantities in each stage while the dynamics of other quantities can be ignored or is very simple. The main reason why the speed is different is attributed to the scale of hyperparameters including $\sigma_p s, \sigma_p \sqrt{s}, \sigma_0$.

We will define some key timesteps in following analysis, including

$$T_1 := \tilde{\Theta}(\sigma_0 \eta^{-1} s^{-1/2} m_v^{-1/2}),$$

$$T_2 := 50\sqrt{2} n \beta_{\boldsymbol{\xi}} \eta^{-1} \sigma_p^{-1} s^{-1} = \tilde{O}(\sigma_0 s^{-1/2} \eta^{-1}),$$

$$T_2^+ := C_1 \sqrt{\log(12 m_k n/\delta)} \sigma_0 n s^{-1/2} \eta^{-1} = \tilde{O}(\sigma_0 s^{-1/2} \eta^{-1}),$$

$$T_3 := 3\beta_{\boldsymbol{\mu}} \eta^{-1} \|\boldsymbol{\mu}\|^{-1} = \tilde{O}(\sigma_0 \eta^{-1}),$$

$$T_3^+ := C_2 \sqrt{\log(12 m_k/\delta)} \sigma_0 \eta^{-1} = \tilde{O}(\sigma_0 \eta^{-1}),$$

$$T_4^- \geq \tilde{T}_4^- := \sqrt{\frac{0.99\pi}{2}} \sqrt{\log\left(\frac{\sigma_p s}{3\sqrt{2} n \|\boldsymbol{\mu}\|}\right)} \eta^{-1} m_k^{-1/2} \sigma_p^{-1} s^{-1} = \tilde{O}(m_k^{-1/2} \sigma_p^{-1} s^{-1} \eta^{-1}),$$

$$T_4^- \leq \hat{T}_4^- := \sqrt{\frac{1.01\pi}{2}} \sqrt{\log\left(\frac{\sigma_p s}{\|\boldsymbol{\mu}\|}\right)} \eta^{-1} m_k^{-1/2} \sigma_p^{-1} s^{-1} = \tilde{O}(m_k^{-1/2} \sigma_p^{-1} s^{-1} \eta^{-1}),$$

$$T_4^+ \leq (1 + \theta_c) T_2^-,$$

$$T_4 := C_3 \log\left(\frac{C_3 \sigma_p s}{\|\boldsymbol{\mu}\|}\right) \eta^{-1} m_k^{-1/2} \sigma_p^{-1} s^{-1} = \tilde{O}(m_k^{-1/2} \sigma_p^{-1} s^{-1} \eta^{-1}),$$

where $C_1, C_2, C_3 = \Theta(1)$ are large constants. The order among these timesteps is

$$T_1 \ll T_2 < T_2^+ \ll T_3 < T_3^+ \ll T_4^- < T_4^+ < T_4.$$

Our analysis basically consists of following parts:

- **Mean value noise.** Let

$$T_1 := \tilde{\Theta}(\sigma_0 \eta^{-1} s^{-1/2} m_v^{-1/2}).$$

  For $t \in [0, T_1]$, the mean value across all samples align quickly while other inner product barely move. For $t \geq T_1$, we have the mean value noise is linear with time

$$\sqrt{2} t \eta \sigma_p s \leq \langle \mathbf{v}^{(t)}, y_i \boldsymbol{\xi}_i \rangle \leq 2 t \eta \sigma_p s,$$

  for all $i \in [n]$, and mean value signal $\langle \mathbf{v}^{(t)}, \boldsymbol{\mu} \rangle$ can be ignored compared with mean value noise $\langle \mathbf{v}^{(t)}, y_i \boldsymbol{\xi}_i \rangle$. Moreover, for $t \geq T_2$, we show the mean value noise across different samples are almost the same

$$\langle \mathbf{v}^{(t)}, y_i \boldsymbol{\xi}_i \rangle = \sqrt{\frac{2}{\pi}} t \eta \sigma_p s (1 + \tilde{O}(s^{-1/2})),$$

  for all $i \in [n]$.

- **Useful bounds.** For $t \in [0, T_3^+]$, noise-signal softmax outputs are close to initialization, i.e., for all $i \in [n]$

$$s_{i,21}^{(t)} = 1/2 + o(1).$$

For $t \in [0, T_4]$, signal-signal softmax outputs are close to initialization, i.e., for all $i \in [n]$

$$s_{i,11}^{(t)} = 1/2 + o(1).$$

For $t \in [0, T_2^+]$, query/key signal are close to initialization, i.e., for all $s \in [m_k]$

$$\left| \langle \mathbf{w}_{K,s}^{(t)}, \boldsymbol{\mu} \rangle \right| = \left| \langle \mathbf{w}_{K,s}^{(0)}, \boldsymbol{\mu} \rangle \right| (1 + o(1)),$$

$$\left| \langle \mathbf{w}_{Q,s}^{(t)}, \boldsymbol{\mu} \rangle \right| = \left| \langle \mathbf{w}_{Q,s}^{(0)}, \boldsymbol{\mu} \rangle \right| (1 + o(1)).$$

For $t \in [0, T_4]$, the loss derivative are close to initialization, i.e., for all $i \in [n]$

$$\ell_i'^{(t)} = 1/2 + o(1).$$

For $t \in [0, T_4^-]$, the ratio of noise-signal softmax outputs are close to 1, i.e., for all $i, k \in [n]$

$$\frac{s_{i,21}^{(t)}}{s_{k,21}^{(t)}} = 1 + o(1).$$

- **Query/Key noise dynamics part i.** Let

$$T_2^+ := C_1 \sqrt{\log(12 m_k n / \delta)} \sigma_0 n s^{-1/2} \eta^{-1},$$
$$T_2 := 50 \sqrt{2} n \beta_{\boldsymbol{\xi}} \eta^{-1} \sigma_p^{-1} s^{-1} = \tilde{O}(\sigma_0 s^{-1/2} \eta^{-1}).$$

where $C_1 = \Theta(1)$ is a large constant. For $t \in [0, T_2]$, we focus on the dynamics of query/key noise. As mentioned above, softmax outputs and query/key signal are stuck at initialization, so we can ignore the variation of query/key signal and softmax outputs before $T_2$. All query/key noise can be divided into two groups. Given $s \in [m_k]$ and $i \in [n]$,

1. if $\langle \mathbf{w}_{Q,s}^{(0)}, y_i \boldsymbol{\xi}_i \rangle =_{\text{sgn}} \langle \mathbf{w}_{K,s}^{(0)}, y_i \boldsymbol{\xi}_i \rangle$, then they encourage each other to increase (in magnitude).
2. if $\langle \mathbf{w}_{Q,s}^{(0)}, y_i \boldsymbol{\xi}_i \rangle =_{\text{sgn}} -\langle \mathbf{w}_{K,s}^{(0)}, y_i \boldsymbol{\xi}_i \rangle$, then they first encourage each other to decrease in magnitude, but rebound quickly and reduce to the first case.

For $t \in [T_2, T_2^+]$, we have some nice properties about query/key noise

$$\langle \mathbf{w}_{Q,s}^{(t)}, y_i \boldsymbol{\xi}_i \rangle =_{\text{sgn}} \langle \mathbf{w}_{K,s}^{(t)} y_i \boldsymbol{\xi}_i \rangle,$$
$$\langle \mathbf{w}_{Q,s}^{(t+1)}, y_i \boldsymbol{\xi}_i \rangle = \langle \mathbf{w}_{Q,s}^{(t)}, y_i \boldsymbol{\xi}_i \rangle + \text{sgn}(\langle \mathbf{w}_{Q,s}^{(t)}, y_i \boldsymbol{\xi}_i \rangle) \eta \| \boldsymbol{\xi}_i \|_1,$$
$$\langle \mathbf{w}_{K,s}^{(t+1)}, y_i \boldsymbol{\xi}_i \rangle = \langle \mathbf{w}_{K,s}^{(t)}, y_i \boldsymbol{\xi}_i \rangle + \text{sgn}(\langle \mathbf{w}_{K,s}^{(t)}, y_i \boldsymbol{\xi}_i \rangle) \eta \| \boldsymbol{\xi}_i \|_1,$$
$$t \eta \sigma_p s / 2 \le \left| \langle \mathbf{w}_{Q,s}^{(t)}, y_i \boldsymbol{\xi}_i \rangle \right|, \left| \langle \mathbf{w}_{K,s}^{(t)}, y_i \boldsymbol{\xi}_i \rangle \right| \le 2 t \eta \sigma_p s,$$

for all $s \in [m_k]$ and $i \in [n]$. Also we have some nice properties about the sum of query/key noise $\sum_{i \in [n]} \langle \mathbf{w}_{Q,s}^{(t)}, y_i \boldsymbol{\xi}_i \rangle, \sum_{i \in [n]} \langle \mathbf{w}_{K,s}^{(t)}, y_i \boldsymbol{\xi}_i \rangle$

$$\left| \sum_{i=1}^n \langle \mathbf{w}_{Q,s}^{(t)}, y_i \boldsymbol{\xi}_i \rangle \right|, \left| \sum_{i=1}^n \langle \mathbf{w}_{K,s}^{(t)}, y_i \boldsymbol{\xi}_i \rangle \right| \ge 2 n \beta_{\boldsymbol{\mu}},$$

$$\sum_{i=1}^n \langle \mathbf{w}_{Q,s}^{(t+1)}, y_i \boldsymbol{\xi}_i \rangle \ge \sum_{i=1}^n \langle \mathbf{w}_{Q,s}^{(t)}, y_i \boldsymbol{\xi}_i \rangle + \text{sgn}(\sum_{i=1}^n \langle \mathbf{w}_{Q,s}^{(t)}, y_i \boldsymbol{\xi}_i \rangle) \frac{1}{\sqrt{2}} \eta \sigma_p s,$$

$$\sum_{i=1}^n \langle \mathbf{w}_{K,s}^{(t+1)}, y_i \boldsymbol{\xi}_i \rangle \ge \sum_{i=1}^n \langle \mathbf{w}_{K,s}^{(t)}, y_i \boldsymbol{\xi}_i \rangle + \text{sgn}(\sum_{i=1}^n \langle \mathbf{w}_{K,s}^{(t)}, y_i \boldsymbol{\xi}_i \rangle) \frac{1}{\sqrt{2}} \eta \sigma_p s,$$

for all $s \in [m_k]$.

- **Query/Key signal dynamics part i.** Let

$$T_3 := 3\beta_{\boldsymbol{\mu}}\eta^{-1}\|\boldsymbol{\mu}\|^{-1} = \tilde{O}(\sigma_0\eta^{-1}),$$
$$T_3^+ := C_2\sqrt{\log(12m_k/\delta)}\sigma_0\eta^{-1},$$

where $C_2 = \Theta(1)$ is a large constant. For $t \in [T_2, T_3]$, we focus on the dynamics of query/key signal. In this period, softmax outputs are stuck at initialization, and the dynamics of query/key noise keep unchanged

$$\langle \mathbf{w}_{Q,s}^{(t)}, y_i\boldsymbol{\xi}_i \rangle =_{\mathrm{sgn}} \langle \mathbf{w}_{K,s}^{(t)} y_i\boldsymbol{\xi}_i \rangle,$$
$$\langle \mathbf{w}_{Q,s}^{(t+1)}, y_i\boldsymbol{\xi}_i \rangle = \langle \mathbf{w}_{Q,s}^{(t)}, y_i\boldsymbol{\xi}_i \rangle + \mathrm{sgn}(\langle \mathbf{w}_{Q,s}^{(t)}, y_i\boldsymbol{\xi}_i \rangle)\eta\|\boldsymbol{\xi}_i\|_1,$$
$$\langle \mathbf{w}_{K,s}^{(t+1)}, y_i\boldsymbol{\xi}_i \rangle = \langle \mathbf{w}_{K,s}^{(t)}, y_i\boldsymbol{\xi}_i \rangle + \mathrm{sgn}(\langle \mathbf{w}_{K,s}^{(t)}, y_i\boldsymbol{\xi}_i \rangle)\eta\|\boldsymbol{\xi}_i\|_1,$$
$$\sum_{i=1}^{n}\langle \mathbf{w}_{Q,s}^{(t+1)}, y_i\boldsymbol{\xi}_i \rangle \geq \sum_{i=1}^{n}\langle \mathbf{w}_{Q,s}^{(t)}, y_i\boldsymbol{\xi}_i \rangle + \mathrm{sgn}(\sum_{i=1}^{n}\langle \mathbf{w}_{Q,s}^{(t)}, y_i\boldsymbol{\xi}_i \rangle)\frac{1}{\sqrt{2}}\eta\sigma_p s,$$
$$\sum_{i=1}^{n}\langle \mathbf{w}_{K,s}^{(t+1)}, y_i\boldsymbol{\xi}_i \rangle \geq \sum_{i=1}^{n}\langle \mathbf{w}_{K,s}^{(t)}, y_i\boldsymbol{\xi}_i \rangle + \mathrm{sgn}(\sum_{i=1}^{n}\langle \mathbf{w}_{K,s}^{(t)}, y_i\boldsymbol{\xi}_i \rangle)\frac{1}{\sqrt{2}}\eta\sigma_p s,$$

for all $s \in [m_k]$ and $i \in [n]$. We show that the update direction of query/key noise is determined by the summation of query/key noise. For $t \in [T_2, T_3]$, for all $s \in [m_k]$ we have

$$\langle \mathbf{w}_{Q,s}^{(t+1)}, \boldsymbol{\mu} \rangle = \langle \mathbf{w}_{Q,s}^{(t)}, \boldsymbol{\mu} \rangle + \mathrm{sgn}(\sum_{i=1}^{n}\langle \mathbf{w}_{Q,s}^{(T_2)}, y_i\boldsymbol{\xi}_i \rangle)\eta\|\boldsymbol{\mu}\|,$$

$$\langle \mathbf{w}_{K,s}^{(t+1)}, \boldsymbol{\mu} \rangle = \langle \mathbf{w}_{K,s}^{(t)}, \boldsymbol{\mu} \rangle - \mathrm{sgn}(\sum_{i=1}^{n}\langle \mathbf{w}_{Q,s}^{(T_2)}, y_i\boldsymbol{\xi}_i \rangle)\eta\|\boldsymbol{\mu}\|.$$

For $t \in [T_3, T_3^+]$, we have some nice properties about query/key signal

$$\langle \mathbf{w}_{Q,s}^{(t)}, \boldsymbol{\mu} \rangle =_{\mathrm{sgn}} -\langle \mathbf{w}_{K,s}^{(t)}, \boldsymbol{\mu} \rangle =_{\mathrm{sgn}} \sum_{i \in [n]}\langle \mathbf{w}_{K,s}^{(t)}, y_i\boldsymbol{\xi}_i \rangle,$$

$$t\eta\|\boldsymbol{\mu}\|/2 \leq \left|\langle \mathbf{w}_{Q,s}^{(t)}, \boldsymbol{\mu} \rangle\right|, \left|\langle \mathbf{w}_{K,s}^{(t)}, \boldsymbol{\mu} \rangle\right| \leq 2t\eta\|\boldsymbol{\mu}\|,$$

$$\langle \mathbf{w}_{Q,s}^{(t+1)}, \boldsymbol{\mu} \rangle = \langle \mathbf{w}_{Q,s}^{(t)}, \boldsymbol{\mu} \rangle + \mathrm{sgn}(\langle \mathbf{w}_{Q,s}^{(t)}, \boldsymbol{\mu} \rangle)\eta\|\boldsymbol{\mu}\|,$$

$$\langle \mathbf{w}_{K,s}^{(t+1)}, \boldsymbol{\mu} \rangle = \langle \mathbf{w}_{K,s}^{(t)}, \boldsymbol{\mu} \rangle + \mathrm{sgn}(\langle \mathbf{w}_{K,s}^{(t)}, \boldsymbol{\mu} \rangle)\eta\|\boldsymbol{\mu}\|,$$

for all $s \in [m_k]$.

- **Query/Key signal and noise dynamics part ii.** Let $T_4^- \geq T_3$ be the first time the following condition does not hold, for all $s \in [m_k]$

$$\left|\sum_{i=1}^{n}s_{i,21}^{(t)}s_{i,22}^{(t)}\langle \mathbf{w}_{Q,s}^{(t)}, y_i\boldsymbol{\xi}_i \rangle\right| \geq \frac{1}{2}n\left|\langle \mathbf{w}_{Q,s}^{(t)}, \boldsymbol{\mu} \rangle\right|,$$

$$s_{i,21}^{(t)}s_{i,22}^{(t)}\left|\langle \mathbf{w}_{Q,s}^{(t)}, y_i\boldsymbol{\xi}_i \rangle\right| \geq 2s_{i,11}^{(t)}s_{i,12}^{(t)}\left|\langle \mathbf{w}_{Q,s}^{(t)}, \boldsymbol{\mu} \rangle\right|.$$

Let $T_4^+ \geq T_4^-$ be the first time the following condition holds, for all $s \in [m_k]$ and $i \in [n]$

$$s_{i,21}^{(t)}s_{i,22}^{(t)}\left|\langle \mathbf{w}_{Q,s}^{(t)}, y_i\boldsymbol{\xi}_i \rangle\right| \leq \frac{1}{2}s_{i,11}^{(t)}s_{i,12}^{(t)}\left|\langle \mathbf{w}_{Q,s}^{(t)}, \boldsymbol{\mu} \rangle\right|.$$

Let

$$T_4 = C_3\log\left(\frac{C_3\sigma_p s}{\|\boldsymbol{\mu}\|}\right)\eta^{-1}m_k^{-1/2}\sigma_p^{-1}s^{-1},$$

where $C_3 = \Theta(1)$ is a large constant We discuss the dynamics by three parts. For $t \in [T_3, T_4^-]$, see stage IV.a in Appendix E.7.1. For $t \in [T_4^-, T_4^+]$, see stage IV.b in Appendix E.7.2. For

$t \in [T_4^+, T_4]$, see stage IV.c in Appendix E.7.1. For $t \geq T_4$, we have for all $s \in [m_k]$ and $i \in [n]$

$$\langle \mathbf{w}_{Q,s}^{(t+1)}, \boldsymbol{\mu} \rangle = \langle \mathbf{w}_{Q,s}^{(t)}, \boldsymbol{\mu} \rangle + \mathrm{sgn}(\sum_{i=1}^{n} \langle \mathbf{w}_{Q,s}^{(T_3)}, y_i \boldsymbol{\xi}_i \rangle) \eta \|\boldsymbol{\mu}\| ,$$

$$\langle \mathbf{w}_{K,s}^{(t+1)}, \boldsymbol{\mu} \rangle = \langle \mathbf{w}_{K,s}^{(t)}, \boldsymbol{\mu} \rangle - \mathrm{sgn}(\sum_{i=1}^{n} \langle \mathbf{w}_{Q,s}^{(T_3)}, y_i \boldsymbol{\xi}_i \rangle) \eta \|\boldsymbol{\mu}\| ,$$

$$\langle \mathbf{w}_{Q,s}^{(t+1)}, y_i \boldsymbol{\xi}_i \rangle = \langle \mathbf{w}_{Q,s}^{(t)}, y_i \boldsymbol{\xi}_i \rangle + \mathrm{sgn}(\langle \mathbf{w}_{Q,s}^{(t)}, y_i \boldsymbol{\xi}_i \rangle) \eta \|\boldsymbol{\xi}_i\|_1 ,$$

$$\langle \mathbf{w}_{K,s}^{(t+1)}, y_i \boldsymbol{\xi}_i \rangle = \langle \mathbf{w}_{K,s}^{(t)}, y_i \boldsymbol{\xi}_i \rangle + \mathrm{sgn}(\langle \mathbf{w}_{K,s}^{(t)}, y_i \boldsymbol{\xi}_i \rangle) \eta \|\boldsymbol{\xi}_i\|_1 ,$$

$$\langle \mathbf{w}_{Q,s}^{(t)}, y_i \boldsymbol{\xi}_i \rangle =_{\mathrm{sgn}} \langle \mathbf{w}_{K,s}^{(t)}, y_i \boldsymbol{\xi}_i \rangle =_{\mathrm{sgn}} \langle \mathbf{w}_{Q,s}^{(t)}, \boldsymbol{\mu} \rangle =_{\mathrm{sgn}} - \langle \mathbf{w}_{K,s}^{(t)}, \boldsymbol{\mu} \rangle.$$

Note that we have following bounds for $T_4^-$ and $T_4^+$

$$T_4^- \geq \tilde{T}_4^- := \sqrt{\frac{0.99\pi}{2}} \sqrt{\log\left(\frac{\sigma_p s}{3\sqrt{2}n \|\boldsymbol{\mu}\|}\right)} \eta^{-1} m_k^{-1/2} \sigma_p^{-1} s^{-1},$$

$$T_4^- \leq \hat{T}_4^- := \sqrt{\frac{1.01\pi}{2}} \sqrt{\log\left(\frac{\sigma_p s}{\|\boldsymbol{\mu}\|}\right)} \eta^{-1} m_k^{-1/2} \sigma_p^{-1} s^{-1},$$

$$T_4^+ \leq (1 + \theta_c) T_4^-.$$

## E.2 THE DYNAMICS OF VALUE

Write down the inner product between gradient of value parameters and signal & noise, we have

$$\langle \mathrm{sgn}\left(\nabla_{\mathbf{w}_{V,j,r}} L\right), \boldsymbol{\mu} \rangle = \mathrm{sgn}\left(\sum_{i \in [n]} l_i'^{(t)} j(s_{i,11}^{(t)} + s_{i,21}^{(t)}) \boldsymbol{\mu}[1]\right) \boldsymbol{\mu}[1] = -\mathrm{sgn}(j) \|\boldsymbol{\mu}\| ,$$

for all $j \in \{\pm 1\}, r \in [m_v]$ and

$$\langle \mathrm{sgn}\left(\nabla_{\mathbf{w}_{V,j,r}} L\right), \boldsymbol{\xi}_i \rangle = \sum_{k \in [d]} \mathrm{sgn}\left(\sum_{i' \in [n]} l_{i'}'^{(t)} y_{i'} j(s_{i',12}^{(t)} + s_{i',22}^{(t)}) \boldsymbol{\xi}_{i'}[k]\right) \boldsymbol{\xi}_i[k],$$

for all $i \in [n], j \in \{\pm 1\}, r \in [m_v]$, and simplified to sparse setting

$$\langle \mathrm{sgn}\left(\nabla_{\mathbf{w}_{V,j,r}} L\right), \boldsymbol{\xi}_i \rangle = \sum_{k \in \mathcal{B}_i} \mathrm{sgn}\left(l_i'^{(t)} y_i j(s_{i,12}^{(t)} + s_{i,22}^{(t)}) \boldsymbol{\xi}_i[k]\right) \boldsymbol{\xi}_i[k],$$

for all $i \in [n], j \in \{\pm 1\}, r \in [m_v]$.

Consider the sparse property of the data, we have

$$\langle \mathrm{sgn}\left(\nabla_{\mathbf{w}_{V,j,r}} L\right), \boldsymbol{\xi}_i \rangle = \sum_{k \in \mathcal{B}_i} \mathrm{sgn}\left(l_i'^{(t)} y_i j(s_{i,12}^{(t)} + s_{i,22}^{(t)}) \boldsymbol{\xi}_i[k]\right) \boldsymbol{\xi}_i[k]$$

$$= -\mathrm{sgn}(y_i j) \sum_{k \in \mathcal{B}_i} |\boldsymbol{\xi}_i[k]|$$

$$= -\mathrm{sgn}(y_i j) \|\boldsymbol{\xi}_i\|_1 ,$$

where the last equality is due to $l' < 0$ and $s_{i,ab} > 0$ for all $i \in [n], a, b \in [2]$. Note the update to value parameters of is irrelevant to the magnitude of query, key, and attention weights. Then, we have

$$\langle \mathbf{w}_{V,y_i,r}^{(t+1)}, y_i \boldsymbol{\mu} \rangle = \langle \mathbf{w}_{V,y_i,r}^{(t)}, y_i \boldsymbol{\mu} \rangle + \eta \|\boldsymbol{\mu}\| ,$$

$$\langle \mathbf{w}_{V,-y_i,r}^{(t+1)}, y_i \boldsymbol{\mu} \rangle = \langle \mathbf{w}_{V,-y_i,r}^{(t)}, y_i \boldsymbol{\mu} \rangle - \eta \|\boldsymbol{\mu}\| ,$$

$$\langle \mathbf{w}_{V,y_i,r}^{(t+1)}, \boldsymbol{\xi}_i \rangle = \langle \mathbf{w}_{V,y_i,r}^{(t)}, \boldsymbol{\xi}_i \rangle + \eta \|\boldsymbol{\xi}_i\|_1 ,$$

$$\langle \mathbf{w}_{V,-y_i,r}^{(t+1)}, \boldsymbol{\xi}_i \rangle = \langle \mathbf{w}_{V,-y_i,r}^{(t)}, \boldsymbol{\xi}_i \rangle - \eta \|\boldsymbol{\xi}_i\|_1 .$$

As $t$ become large, both $\langle \mathbf{w}_{V,y_i,r}^{(t)}, y_i \boldsymbol{\mu} \rangle$ and $\langle \mathbf{w}_{V,y_i,r}^{(t)}, \boldsymbol{\xi}_i \rangle$ increase. Also, $\langle \mathbf{w}_{V,y_i,r}^{(t)}, \boldsymbol{\xi}_i \rangle$ increases faster since we have $s\sigma_p \|\boldsymbol{\mu}\|^{-1} = \tilde{\Omega}(1)$.

### E.3 THE DYNAMICS OF QUERY AND KEY: PREPARATIONS

**Additional Notations.** Let $\mathbf{v}^{(t)} := \bar{\mathbf{w}}_{V,1}^{(t)} - \bar{\mathbf{w}}_{V,-1}^{(t)}$. When we talk about query and key parameters, we refer to $\mathbf{w}_{Q,s}, \mathbf{w}_{K,s}$ for all $s \in [m_k]$. When we talk about query/key signal (inner products), we refer to $\langle \mathbf{w}_{Q,s}, \boldsymbol{\mu} \rangle, \langle \mathbf{w}_{K,s}, \boldsymbol{\mu} \rangle$ for all $s \in [m_k]$. When we talk about query/key noise (inner products), we refer to $\langle \mathbf{w}_{Q,s}, y_i \boldsymbol{\xi}_i \rangle, \langle \mathbf{w}_{K,s}, y_i \boldsymbol{\xi}_i \rangle$ for all $s \in [m_k]$ and $i \in [n]$. When we talk about (mean) value (noise), we refer to $\langle \mathbf{v}, y_i \boldsymbol{\xi}_i \rangle$ for all $i \in [n]$. We use $a =_{\text{sgn}} b$ to denote $\text{sgn}(a) = \text{sgn}(b)$. Let $\mathcal{B}_i$ be the support of sample $\boldsymbol{\xi}_i$ in the training dataset.

**Some Facts.** Based on gradient of query and key parameters and the sparse property of data, we firstly write down the inner product between gradient and signal & noise

$$
\langle \text{sgn}\left( \nabla_{\mathbf{w}_{Q,s}} L \right), \boldsymbol{\mu} \rangle = \text{sgn}\left( \sum_{i \in [n]} l_i'^{(t)} y_i \langle \bar{\mathbf{w}}_{V,1}^{(t)} - \bar{\mathbf{w}}_{V,-1}^{(t)}, y_i \boldsymbol{\mu} - \boldsymbol{\xi}_i \rangle \langle \mathbf{w}_{K,s}^{(t)}, y_i \boldsymbol{\mu} - \boldsymbol{\xi}_i \rangle s_{i,11}^{(t)} s_{i,12}^{(t)} y_i \right),
$$

$$
\langle \text{sgn}\left( \nabla_{\mathbf{w}_{Q,s}} L \right), \boldsymbol{\xi}_i \rangle = \sum_{k \in \mathcal{B}_i} \text{sgn}\left( l_i'^{(t)} y_i \langle \bar{\mathbf{w}}_{V,1}^{(t)} - \bar{\mathbf{w}}_{V,-1}^{(t)}, y_i \boldsymbol{\mu} - \boldsymbol{\xi}_i \rangle \langle \mathbf{w}_{K,s}^{(t)}, y_i \boldsymbol{\mu} - \boldsymbol{\xi}_i \rangle s_{i,21}^{(t)} s_{i,22}^{(t)} \boldsymbol{\xi}_i[k] \right) \boldsymbol{\xi}_i[k],
$$

$$
\langle \text{sgn}\left( \nabla_{\mathbf{w}_{K,s}} L \right), \boldsymbol{\mu} \rangle = \text{sgn}\left( \sum_{i \in [n]} l_i'^{(t)} y_i \langle \bar{\mathbf{w}}_{V,1}^{(t)} - \bar{\mathbf{w}}_{V,-1}^{(t)}, y_i \boldsymbol{\mu} - \boldsymbol{\xi}_i \rangle \langle \mathbf{w}_{Q,s}^{(t)}, s_{i,11}^{(t)} s_{i,12}^{(t)} y_i \boldsymbol{\mu} + s_{i,21}^{(t)} s_{i,22}^{(t)} \boldsymbol{\xi}_i \rangle y_i \right),
$$

$$
\langle \text{sgn}\left( \nabla_{\mathbf{w}_{K,s}} L \right), \boldsymbol{\xi}_i \rangle = \sum_{k \in \mathcal{B}_i} \text{sgn}\left( l_i'^{(t)} y_i \langle \bar{\mathbf{w}}_{V,1}^{(t)} - \bar{\mathbf{w}}_{V,-1}^{(t)}, y_i \boldsymbol{\mu} - \boldsymbol{\xi}_i \rangle \langle \mathbf{w}_{Q,s}^{(t)}, s_{i,11}^{(t)} s_{i,12}^{(t)} y_i \boldsymbol{\mu} + s_{i,21}^{(t)} s_{i,22}^{(t)} \boldsymbol{\xi}_i \rangle (-\boldsymbol{\xi}_i[k]) \right) \boldsymbol{\xi}_i[k].
$$

Before the formal analysis for the query/key dynamics, we have some observations to simplify the gradient update formula. Firstly, we note that the update magnitude of query/key inner product is constant at all iterations. Formally, for all $t \geq 0$ we have

$$
\left| \langle \mathbf{w}_{K,s}^{(t+1)} - \mathbf{w}_{K,s}^{(t)}, \boldsymbol{\mu} \rangle \right|, \left| \langle \mathbf{w}_{Q,s}^{(t+1)} - \mathbf{w}_{Q,s}^{(t)}, \boldsymbol{\mu} \rangle \right| = \eta \|\boldsymbol{\mu}\|, \tag{8}
$$

$$
\left| \langle \mathbf{w}_{K,s}^{(t+1)} - \mathbf{w}_{K,s}^{(t)}, y_i \boldsymbol{\xi}_i \rangle \right|, \left| \langle \mathbf{w}_{Q,s}^{(t+1)} - \mathbf{w}_{Q,s}^{(t)}, y_i \boldsymbol{\xi}_i \rangle \right| = \eta \|\boldsymbol{\xi}_i\|_1, \tag{9}
$$

which means that we only need to analyze the sign of update direction for all quantities of interest.

Secondly, we take a look at the key part in gradient formula for different quantities:

$$
\langle \mathbf{v}^{(t)}, y_i \boldsymbol{\mu} - \boldsymbol{\xi}_i \rangle \langle \mathbf{w}_{K,s}^{(t)}, y_i \boldsymbol{\mu} - \boldsymbol{\xi}_i \rangle
$$
$$
= \langle \mathbf{v}^{(t)}, \boldsymbol{\mu} \rangle \langle \mathbf{w}_{K,s}^{(t)}, \boldsymbol{\mu} \rangle - y_i \langle \mathbf{v}^{(t)}, \boldsymbol{\mu} \rangle \langle \mathbf{w}_{K,s}^{(t)}, \boldsymbol{\xi}_i \rangle - y_i \langle \mathbf{v}^{(t)}, \boldsymbol{\xi}_i \rangle \langle \mathbf{w}_{K,s}^{(t)}, \boldsymbol{\mu} \rangle + \langle \mathbf{v}^{(t)}, \boldsymbol{\xi}_i \rangle \langle \mathbf{w}_{K,s}^{(t)}, \boldsymbol{\xi}_i \rangle,
$$
$$
\langle \mathbf{v}^{(t)}, y_i \boldsymbol{\mu} - \boldsymbol{\xi}_i \rangle \langle \mathbf{w}_{Q,s}^{(t)}, y_i \boldsymbol{\mu} + \boldsymbol{\xi}_i \rangle
$$
$$
= \langle \mathbf{v}^{(t)}, \boldsymbol{\mu} \rangle \langle \mathbf{w}_{Q,s}^{(t)}, \boldsymbol{\mu} \rangle + y_i \langle \mathbf{v}^{(t)}, \boldsymbol{\mu} \rangle \langle \mathbf{w}_{Q,s}^{(t)}, \boldsymbol{\xi}_i \rangle - y_i \langle \mathbf{v}^{(t)}, \boldsymbol{\xi}_i \rangle \langle \mathbf{w}_{Q,s}^{(t)}, \boldsymbol{\mu} \rangle - \langle \mathbf{v}^{(t)}, \boldsymbol{\xi}_i \rangle \langle \mathbf{w}_{Q,s}^{(t)}, \boldsymbol{\xi}_i \rangle.
$$

By the subsequent analysis in the stage I, we have that the $\left| \langle \mathbf{v}^{(t)}, \boldsymbol{\mu} \rangle \right| = o(1) \cdot \langle \mathbf{v}^{(t)}, y_i \boldsymbol{\xi} \rangle$ such that we can approximately neglect the effect of $\langle \mathbf{v}^{(t)}, \boldsymbol{\mu} \rangle$. Also, we have $\langle \mathbf{v}^{(t)}, y_i \boldsymbol{\xi} \rangle \geq 0$. Then, combined with the gradient formula, we have

- For $\langle \mathbf{w}_{Q,s}, y_i \boldsymbol{\xi}_i \rangle$, the sign of increment is aligned with

$$
\langle \mathbf{w}_{Q,s}^{(t+1)} - \mathbf{w}_{Q,s}^{(t)}, y_i \boldsymbol{\xi}_i \rangle
$$
$$
=_{\text{sgn}} - \langle \mathbf{v}^{(t)}, y_i \boldsymbol{\xi}_i \rangle \langle \mathbf{w}_{K,s}^{(t)}, \boldsymbol{\mu} \rangle + \langle \mathbf{v}^{(t)}, y_i \boldsymbol{\xi}_i \rangle \langle \mathbf{w}_{K,s}^{(t)}, y_i \boldsymbol{\xi}_i \rangle
$$
$$
=_{\text{sgn}} - \langle \mathbf{w}_{K,s}^{(t)}, \boldsymbol{\mu} \rangle + \langle \mathbf{w}_{K,s}^{(t)}, y_i \boldsymbol{\xi}_i \rangle. \tag{10}
$$

- For $\langle \mathbf{w}_{Q,s}, \boldsymbol{\mu} \rangle$, the sign of increment is aligned with

$$
\langle \mathbf{w}_{Q,s}^{(t+1)} - \mathbf{w}_{Q,s}^{(t)}, \boldsymbol{\mu} \rangle
$$
$$
=_{\text{sgn}} \sum_{i \in [n]} (-l_i'^{(t)} s_{i,11}^{(t)} s_{i,12}^{(t)}) \cdot \left( -\langle \mathbf{v}^{(t)}, y_i \boldsymbol{\xi}_i \rangle \langle \mathbf{w}_{K,s}^{(t)}, \boldsymbol{\mu} \rangle + \langle \mathbf{v}^{(t)}, y_i \boldsymbol{\xi}_i \rangle \langle \mathbf{w}_{K,s}^{(t)}, y_i \boldsymbol{\xi}_i \rangle \right). \tag{11}
$$

- For $\langle \mathbf{w}_{K,s}, y_i \boldsymbol{\xi}_i \rangle$, the sign of increment is aligned with

$$
\begin{aligned}
&\langle \mathbf{w}_{K,s}^{(t+1)} - \mathbf{w}_{K,s}^{(t)}, y_i \boldsymbol{\xi}_i \rangle \\
=_{\mathrm{sgn}} &- \left( -s_{i,11}^{(t)} s_{i,12}^{(t)} \langle \mathbf{v}^{(t)}, y_i \boldsymbol{\xi}_i \rangle \langle \mathbf{w}_{Q,s}^{(t)}, \boldsymbol{\mu} \rangle - s_{i,21}^{(t)} s_{i,22}^{(t)} \langle \mathbf{v}^{(t)}, y_i \boldsymbol{\xi}_i \rangle \langle \mathbf{w}_{Q,s}^{(t)}, y_i \boldsymbol{\xi}_i \rangle \right) \\
=_{\mathrm{sgn}} &s_{i,11}^{(t)} s_{i,12}^{(t)} \langle \mathbf{w}_{Q,s}^{(t)}, \boldsymbol{\mu} \rangle + s_{i,21}^{(t)} s_{i,22}^{(t)} \langle \mathbf{w}_{Q,s}^{(t)}, y_i \boldsymbol{\xi}_i \rangle.
\end{aligned} \tag{12}
$$

- For $\langle \mathbf{w}_{K,s}, \boldsymbol{\mu} \rangle$, the sign of increment is aligned with

$$
\begin{aligned}
&\langle \mathbf{w}_{K,s}^{(t+1)} - \mathbf{w}_{K,s}^{(t)}, \boldsymbol{\mu} \rangle \\
=_{\mathrm{sgn}} &\sum_{i \in [n]} (-l_i'^{(t)}) \cdot \left( -s_{i,11}^{(t)} s_{i,12}^{(t)} \langle \mathbf{v}^{(t)}, y_i \boldsymbol{\xi}_i \rangle \langle \mathbf{w}_{Q,s}^{(t)}, \boldsymbol{\mu} \rangle - s_{i,21}^{(t)} s_{i,22}^{(t)} \langle \mathbf{v}^{(t)}, y_i \boldsymbol{\xi}_i \rangle \langle \mathbf{w}_{Q,s}^{(t)}, y_i \boldsymbol{\xi}_i \rangle \right).
\end{aligned} \tag{13}
$$

### E.4 STAGE I

In this part, we consider the dynamics of mean value signal and noise, i.e. $\langle \mathbf{v}^{(0)}, y_i \boldsymbol{\xi}_i \rangle$ and $\langle \mathbf{v}^{(0)}, \boldsymbol{\mu} \rangle$. Roughly, at the end of this stage we want to show that mean value noise evolve fast and can be viewed as a linear variable with time, mean value signal is small and negligible afterwards, while query/key noise are very close to initialization.

From the analysis for value above, we directly have

$$
\langle \mathbf{v}^{(t+1)}, \boldsymbol{\mu} \rangle = \langle \mathbf{v}^{(t)}, \boldsymbol{\mu} \rangle + 2\eta \|\boldsymbol{\mu}\|, \tag{14}
$$

$$
\langle \mathbf{v}^{(t+1)}, y_i \boldsymbol{\xi}_i \rangle = \langle \mathbf{v}^{(t)}, y_i \boldsymbol{\xi}_i \rangle + 2\eta \|\boldsymbol{\xi}_i\|_1. \tag{15}
$$

Let

$$
T_1 := 4\beta_{\boldsymbol{\xi}} m_v^{-1/2} \eta^{-1} \sigma_p^{-1} s^{-1}.
$$

Note that we have $\left| \langle \mathbf{v}^{(0)}, y_i \boldsymbol{\xi}_i \rangle \right| \leq \sqrt{2} \beta_{\boldsymbol{\xi}} m_v^{-1/2}$, then we have

$$
t\eta \sigma_p s \leq \langle \mathbf{v}^{(t)}, y_i \boldsymbol{\xi}_i \rangle \leq 2t\eta \sigma_p s,
$$

for all $t \geq T_1$ and $i \in [n]$. However, the query/key noise do not deviate too much from the initialization at $T_1$ since the deviation is at most

$$
\frac{\left| \langle \mathbf{w}_{Q,s}^{(T_1)}, \boldsymbol{\xi}_i \rangle \right|}{\left| \langle \mathbf{w}_{Q,s}^{(0)}, \boldsymbol{\xi}_i \rangle \right|} \leq 1 + \frac{T_1 \eta \|\boldsymbol{\xi}\|_1}{\left| \langle \mathbf{w}_{Q,s}^{(0)}, \boldsymbol{\xi}_i \rangle \right|} \leq 1 + \frac{O(\beta_{\boldsymbol{\xi}} m_v^{-1/2})}{\Omega(\beta_{\boldsymbol{\xi}} m_v^{-1/4})} = 1 + o(1),
$$

where the second step is by Lemma C.11 and magnitude of $T_1$. Besides, the deviation of $\langle \mathbf{v}^{(T_1)}, \boldsymbol{\mu} \rangle$ from initialization is also small. Formally, we have

$$
\frac{\left| \langle \mathbf{v}^{(T_1)}, \boldsymbol{\mu} \rangle \right|}{\left| \langle \mathbf{v}^{(0)}, \boldsymbol{\mu} \rangle \right|} \leq 1 + \frac{T_1 \eta \|\boldsymbol{\mu}\|}{\left| \langle \mathbf{v}^{(0)}, \boldsymbol{\mu} \rangle \right|} \leq 1 + \frac{\tilde{O}(\sigma_0 \|\boldsymbol{\mu}\| s^{-1/2} m_v^{-1/2})}{\Omega(\sigma_0 \|\boldsymbol{\mu}\| n s^{-1/3} m_v^{-1/2})} = 1 + o(1),
$$

where the second step is by Lemma C.12 Therefore, we have

$$
\langle \mathbf{v}^{(t)}, \boldsymbol{\mu} \rangle = o(\langle \mathbf{v}^{(t)}, y_i \boldsymbol{\xi}_i \rangle),
$$

for all $i \in [n]$ and $t \geq T_1$, where this holds at $t = T_1$ due to Lemma C.8 and the definition of $T_1$. Combined with the form of gradient of query/key parameters, this makes us able to ignore the effect of $\langle \mathbf{v}^{(t)}, \boldsymbol{\mu} \rangle$ afterwards.

### E.5 STAGE II

In this part, we consider the dynamics of all query/key noise. Roughly, at the end of this stage we want to show for all neurons $s \in [m_k]$ and samples $i \in [n]$, query/key noise have the same sign and linear with $t$.

Let

$$T_2^+ := C_1\sqrt{\log(12m_k n/\delta)}\sigma_0 n s^{-1/2}\eta^{-1},$$

where $C_1 = \Theta(1)$ is a large constant. In this section, we will analyze the dynamics of $\langle \mathbf{w}_{K,s}^{(t)}, y_i\boldsymbol{\xi}_i \rangle$ and $\langle \mathbf{w}_{Q,s}^{(t)}, y_i\boldsymbol{\xi}_i \rangle$ for all $s \in [m_k]$, $i \in [n]$ and $t \le T_2^+$.

We first consider the samples $i \in S_{s,K+,Q-}^{(0)} \cup S_{s,K-,Q+}^{(0)}$ at initialization. Note that without loss of generality, we can assume that $i \in S_{s,K-,Q+}^{(0)}$. Then, by Lemma C.14, we only need to consider two cases where the one is when we have

$$-\langle \mathbf{w}_{K,s}^{(0)}, y_i\boldsymbol{\xi}_i \rangle + \langle \mathbf{w}_{K,s}^{(0)}, \boldsymbol{\mu} \rangle \le \frac{\sigma_p s - \|\boldsymbol{\mu}\|}{\sigma_p s + \|\boldsymbol{\mu}\|} \cdot (\langle \mathbf{w}_{Q,s}^{(0)}, y_i\boldsymbol{\xi}_i \rangle + \langle \mathbf{w}_{Q,s}^{(0)}, \boldsymbol{\mu} \rangle),$$

and the other one is when we have

$$-\langle \mathbf{w}_{K,s}^{(0)}, y_i\boldsymbol{\xi}_i \rangle + \langle \mathbf{w}_{K,s}^{(0)}, \boldsymbol{\mu} \rangle \ge \frac{\sigma_p s + \|\boldsymbol{\mu}\|}{\sigma_p s - \|\boldsymbol{\mu}\|} \cdot (\langle \mathbf{w}_{Q,s}^{(0)}, y_i\boldsymbol{\xi}_i \rangle + \langle \mathbf{w}_{Q,s}^{(0)}, \boldsymbol{\mu} \rangle).$$

Finally, we give an upper bound for $T_{2,s,i}^3$ (defined later) for all $s \in [m_k]$ and $i \in S_{s,K+,Q-}^{(0)} \cup S_{s,K-,Q+}^{(0)}$, and bound the magnitude of $\langle \mathbf{w}_{K,s}^{(t)}, y_i\boldsymbol{\xi}_i \rangle$ and $\langle \mathbf{w}_{Q,s}^{(t)}, y_i\boldsymbol{\xi}_i \rangle$ before $T_2^+$.

We define following useful timesteps. Let $T_{2,s,i}^1$ be the first time satisfying

$$\langle \mathbf{w}_{Q,s}^{(t)}, y_i\boldsymbol{\xi}_i \rangle =_{\text{sgn}} \langle \mathbf{w}_{K,s}^{(t)}, y_i\boldsymbol{\xi}_i \rangle. \tag{16}$$

Let $T_{2,s,i}^2$ be the first time satisfying

$$\langle \mathbf{w}_{Q,s}^{(t+1)} - \mathbf{w}_{Q,s}^{(t)}, y_i\boldsymbol{\xi}_i \rangle =_{\text{sgn}} \langle \mathbf{w}_{K,s}^{(t+1)} - \mathbf{w}_{K,s}^{(t)} y_i\boldsymbol{\xi}_i \rangle. \tag{17}$$

Let $T_{2,s,i}^3$ be the first time satisfying Eq. (16), (17) and

$$\langle \mathbf{w}_{Q,s}^{(t+1)} - \mathbf{w}_{Q,s}^{(t)}, y_i\boldsymbol{\xi}_i \rangle =_{\text{sgn}} \langle \mathbf{w}_{Q,s}^{(t)} y_i\boldsymbol{\xi}_i \rangle. \tag{18}$$

Let $T_2' := \sqrt{2}\beta_{\boldsymbol{\xi}}\eta^{-1}\sigma_p^{-1}s^{-1}$.

The following lemma studies the order between $T_{2,s,i}^1, T_{2,s,i}^2, T_{2,s,i}^3$.

**Lemma E.1.** *For all $s \in [m_k]$ and $i \in S_{s,K+,Q-}^{(0)} \cup S_{s,K-,Q+}^{(0)}$, we have*

$$0 < T_{2,s,i}^1 \le T_{2,s,i}^3,$$
$$0 < T_{2,s,i}^2 \le T_{2,s,i}^3.$$

*Proof of Lemma E.1.* WLOG, we assume that $i \in S_{s,K-,Q+}^{(0)}$. By the definition of $S_{s,K-,Q+}^{(0)}$, we have

$$-\langle \mathbf{w}_{K,s}^{(0)}, y_i\boldsymbol{\xi}_i \rangle =_{\text{sgn}} \langle \mathbf{w}_{Q,s}^{(0)}, y_i\boldsymbol{\xi}_i \rangle =_{\text{sgn}} 1.$$

Then, by the definition of $T_{2,s,i}^1$, we have $0 < T_{2,s,i}^1$.

By the definition of $T_{2,s,i}^3$, we naturally have $T_{2,s,i}^1 < T_{2,s,i}^3$ and $T_{2,s,i}^2 < T_{2,s,i}^3$.

At $t = 0$, we have

$$\langle \mathbf{w}_{Q,s}^{(t+1)} - \mathbf{w}_{Q,s}^{(t)}, y_i\boldsymbol{\xi}_i \rangle =_{\text{sgn}} -\langle \mathbf{w}_{K,s}^{(t)}, \boldsymbol{\mu} \rangle + \langle \mathbf{w}_{K,s}^{(t)}, y_i\boldsymbol{\xi}_i \rangle =_{\text{sgn}} \langle \mathbf{w}_{K,s}^{(t)}, y_i\boldsymbol{\xi}_i \rangle,$$

where the first step is by Eq. (10) and the second step is by Lemma C.10, and

$$\langle \mathbf{w}_{K,s}^{(t+1)} - \mathbf{w}_{K,s}^{(t)}, y_i\boldsymbol{\xi}_i \rangle =_{\text{sgn}} s_{i,11}^{(t)}s_{i,12}^{(t)}\langle \mathbf{w}_{Q,s}^{(t)}, \boldsymbol{\mu} \rangle + s_{i,21}^{(t)}s_{i,22}^{(t)}\langle \mathbf{w}_{Q,s}^{(t)}, y_i\boldsymbol{\xi}_i \rangle =_{\text{sgn}} \langle \mathbf{w}_{Q,s}^{(t)}, y_i\boldsymbol{\xi}_i \rangle,$$

where the first step is by Eq. (12) and the second step is by Lemma C.10 and C.13, which implies $0 < T_{2,s,i}^2$.

$\square$

Note that by Lemma E.14, we have that for all $i \in [n]$ and $t \le T_2^+$,

$$s_{i,11}^{(t)}, s_{i,21}^{(t)} = s_{i,11}^{(0)}(1 + o(1)), s_{i,21}^{(0)}(1 + o(1)) = 1/2 + o(1).$$

In the following analysis, we will use this fact many times.

WHEN $-\langle \mathbf{w}_{K,s}^{(0)}, y_i \boldsymbol{\xi}_i \rangle + \langle \mathbf{w}_{K,s}^{(0)}, \boldsymbol{\mu} \rangle \le \frac{\sigma_p s - \|\boldsymbol{\mu}\|}{\sigma_p s + \|\boldsymbol{\mu}\|} \cdot (\langle \mathbf{w}_{Q,s}^{(0)}, y_i \boldsymbol{\xi}_i \rangle + \langle \mathbf{w}_{Q,s}^{(0)}, \boldsymbol{\mu} \rangle)$, THE UPDATE DIRECTION ARE DETERMINED BY QUERY NOISE. By the definition of $T_{2,s,i}^2$, we must have that

$$(\langle \mathbf{w}_{Q,s}^{(T_{2,s,i}^2+1)} - \mathbf{w}_{Q,s}^{(T_{2,s,i}^2)}, y_i \boldsymbol{\xi}_i \rangle) \cdot (\langle \mathbf{w}_{K,s}^{(T_{2,s,i}^2+1)} - \mathbf{w}_{K,s}^{(T_{2,s,i}^2)}, y_i \boldsymbol{\xi}_i \rangle) > 0.$$

Note that we have

$$\langle \mathbf{w}_{Q,s}^{(T_{2,s,i}^2+1)} - \mathbf{w}_{Q,s}^{(T_{2,s,i}^2)}, y_i \boldsymbol{\xi}_i \rangle =_{\text{sgn}} - \langle \mathbf{w}_{K,s}^{(T_{2,s,i}^2)}, \boldsymbol{\mu} \rangle + \langle \mathbf{w}_{K,s}^{(T_{2,s,i}^2)}, y_i \boldsymbol{\xi}_i \rangle$$
$$\ge - \langle \mathbf{w}_{K,s}^{(0)}, \boldsymbol{\mu} \rangle + \langle \mathbf{w}_{K,s}^{(0)}, y_i \boldsymbol{\xi}_i \rangle + T_{2,s,i}^2 \eta (\|\boldsymbol{\xi}_i\|_1 - \|\boldsymbol{\mu}\|) \ge 0,$$
$$\langle \mathbf{w}_{K,s}^{(T_{2,s,i}^2+1)} - \mathbf{w}_{K,s}^{(T_{2,s,i}^2)}, y_i \boldsymbol{\xi}_i \rangle =_{\text{sgn}} s_{i,11}^{(T_{2,s,i}^2)} s_{i,12}^{(T_{2,s,i}^2)} \langle \mathbf{w}_{Q,s}^{(T_{2,s,i}^2)}, \boldsymbol{\mu} \rangle + s_{i,21}^{(T_{2,s,i}^2)} s_{i,22}^{(T_{2,s,i}^2)} \langle \mathbf{w}_{Q,s}^{(T_{2,s,i}^2)}, y_i \boldsymbol{\xi}_i \rangle$$
$$\ge \frac{1}{4}(1 \pm o(1))(\langle \mathbf{w}_{Q,s}^{(0)}, \boldsymbol{\mu} \rangle + \langle \mathbf{w}_{Q,s}^{(0)}, y_i \boldsymbol{\xi}_i \rangle - T_{2,s,i}^2 \eta (\|\boldsymbol{\xi}_i\|_1 + \|\boldsymbol{\mu}\|)) \ge 0,$$
$$(19)$$

where the last steps in both lines are by the condition

$$-\langle \mathbf{w}_{K,s}^{(0)}, y_i \boldsymbol{\xi}_i \rangle + \langle \mathbf{w}_{K,s}^{(0)}, \boldsymbol{\mu} \rangle \le \frac{\sigma_p s - \|\boldsymbol{\mu}\|}{\sigma_p s + \|\boldsymbol{\mu}\|} \cdot (\langle \mathbf{w}_{Q,s}^{(0)}, y_i \boldsymbol{\xi}_i \rangle + \langle \mathbf{w}_{Q,s}^{(0)}, \boldsymbol{\mu} \rangle).$$

We claim that the sign of update would not change for $t \in [T_{2,s,i}^2, T_2^+]$. Suppose at $t \le \tilde{t} \in [T_{2,s,i}^2, T_2^+]$, the induction hypothesis holds, then we have

$$\langle \mathbf{w}_{K,s}^{(t'+1)}, y_i \boldsymbol{\xi}_i \rangle > \langle \mathbf{w}_{K,s}^{(t')}, y_i \boldsymbol{\xi}_i \rangle,$$
$$\langle \mathbf{w}_{Q,s}^{(t'+1)}, y_i \boldsymbol{\xi}_i \rangle > \langle \mathbf{w}_{Q,s}^{(t')}, y_i \boldsymbol{\xi}_i \rangle,$$

which imply

$$- \langle \mathbf{w}_{K,s}^{(t'+1)}, \boldsymbol{\mu} \rangle + \langle \mathbf{w}_{K,s}^{(t'+1)}, y_i \boldsymbol{\xi}_i \rangle \ge -\langle \mathbf{w}_{K,s}^{(t')}, \boldsymbol{\mu} \rangle + \langle \mathbf{w}_{K,s}^{(t')}, y_i \boldsymbol{\xi}_i \rangle \ge 0,$$
$$s_{i,11}^{(t'+1)} s_{i,12}^{(t'+1)} \langle \mathbf{w}_{Q,s}^{(t'+1)}, \boldsymbol{\mu} \rangle + s_{i,21}^{(t'+1)} s_{i,22}^{(t'+1)} \langle \mathbf{w}_{Q,s}^{(t'+1)}, y_i \boldsymbol{\xi}_i \rangle \ge \frac{1}{4}(1 \pm o(1))(\langle \mathbf{w}_{Q,s}^{(t')}, \boldsymbol{\mu} \rangle + \langle \mathbf{w}_{Q,s}^{(t')}, y_i \boldsymbol{\xi}_i \rangle) \ge 0,$$

then the conclusion holds and thus $\langle \mathbf{w}_{K,s}^{(T_{2,s,i}^3)}, y_i \boldsymbol{\xi}_i \rangle, \langle \mathbf{w}_{Q,s}^{(T_{2,s,i}^3)}, y_i \boldsymbol{\xi}_i \rangle > 0$.

Moreover, we have an upper bound for $T_{2,s,i}^2$ that

$$T_{2,s,i}^2 \le \frac{\langle \mathbf{w}_{Q,s}^{(0)}, \boldsymbol{\mu} \rangle + \langle \mathbf{w}_{Q,s}^{(0)}, y_i \boldsymbol{\xi}_i \rangle}{\eta(\|\boldsymbol{\xi}_i\|_1 + \|\boldsymbol{\mu}\|)} \le \frac{1.5 \langle \mathbf{w}_{Q,s}^{(0)}, y_i \boldsymbol{\xi}_i \rangle}{\eta \|\boldsymbol{\xi}_i\|_1} \le 1.5 T_2',$$

where the first step is by Eq. (19), the second step is by Lemma C.10, the last step is by the definition of $T_2'$ and Lemma C.4.

WHEN $-\langle \mathbf{w}_{K,s}^{(0)}, y_i \boldsymbol{\xi}_i \rangle + \langle \mathbf{w}_{K,s}^{(0)}, \boldsymbol{\mu} \rangle \ge \frac{\sigma_p s + \|\boldsymbol{\mu}\|}{\sigma_p s - \|\boldsymbol{\mu}\|} \cdot (\langle \mathbf{w}_{Q,s}^{(0)}, y_i \boldsymbol{\xi}_i \rangle + \langle \mathbf{w}_{Q,s}^{(0)}, \boldsymbol{\mu} \rangle)$, THE UPDATE DIRECTION ARE DETERMINED BY KEY NOISE. At $t = T_{2,s,i}^2$ we have

$$\langle \mathbf{w}_{Q,s}^{(T_{2,s,i}^2+1)} - \mathbf{w}_{Q,s}^{(T_{2,s,i}^2)}, y_i \boldsymbol{\xi}_i \rangle =_{\text{sgn}} - \langle \mathbf{w}_{K,s}^{(T_{2,s,i}^2)}, \boldsymbol{\mu} \rangle + \langle \mathbf{w}_{K,s}^{(T_{2,s,i}^2)}, y_i \boldsymbol{\xi}_i \rangle$$
$$\le - \langle \mathbf{w}_{K,s}^{(0)}, \boldsymbol{\mu} \rangle + \langle \mathbf{w}_{K,s}^{(0)}, y_i \boldsymbol{\xi}_i \rangle + T_{2,s,i}^2 \eta (\|\boldsymbol{\xi}_i\|_1 + \|\boldsymbol{\mu}\|) \le 0,$$
$$(20)$$
$$\langle \mathbf{w}_{K,s}^{(T_{2,s,i}^2+1)} - \mathbf{w}_{K,s}^{(T_{2,s,i}^2)}, y_i \boldsymbol{\xi}_i \rangle =_{\text{sgn}} s_{i,11}^{(T_{2,s,i}^2)} s_{i,12}^{(T_{2,s,i}^2)} \langle \mathbf{w}_{Q,s}^{(T_{2,s,i}^2)}, \boldsymbol{\mu} \rangle + s_{i,21}^{(T_{2,s,i}^2)} s_{i,22}^{(T_{2,s,i}^2)} \langle \mathbf{w}_{Q,s}^{(T_{2,s,i}^2)}, y_i \boldsymbol{\xi}_i \rangle$$
$$\le \frac{1}{4}(1 \pm o(1))(\langle \mathbf{w}_{Q,s}^{(0)}, \boldsymbol{\mu} \rangle + \langle \mathbf{w}_{Q,s}^{(0)}, y_i \boldsymbol{\xi}_i \rangle - T_{2,s,i}^2 \eta (\|\boldsymbol{\xi}_i\|_1 - \|\boldsymbol{\mu}\|)) \le 0,$$

where the last steps in both lines are by the condition

$$-\langle \mathbf{w}_{K,s}^{(0)}, y_i \boldsymbol{\xi}_i \rangle + \langle \mathbf{w}_{K,s}^{(0)}, \boldsymbol{\mu} \rangle \ge \frac{\sigma_p s + \|\boldsymbol{\mu}\|}{\sigma_p s - \|\boldsymbol{\mu}\|} \cdot (\langle \mathbf{w}_{Q,s}^{(0)}, y_i \boldsymbol{\xi}_i \rangle + \langle \mathbf{w}_{Q,s}^{(0)}, \boldsymbol{\mu} \rangle).$$

Then, similar to previous analysis, we have for $t \in [T^2_{2,s,i}, T^+_2]$, we have

$$- \langle \mathbf{w}^{(t)}_{K,s}, \boldsymbol{\mu} \rangle + \langle \mathbf{w}^{(t)}_{K,s}, y_i \boldsymbol{\xi}_i \rangle < 0,$$

$$s^{(t)}_{i,11} s^{(t)}_{i,12} \langle \mathbf{w}^{(t)}_{Q,s}, \boldsymbol{\mu} \rangle + s^{(t)}_{i,21} s^{(t)}_{i,22} \langle \mathbf{w}^{(t)}_{Q,s}, y_i \boldsymbol{\xi}_i \rangle < 0,$$

which implies $\langle \mathbf{w}^{(T^3_{2,s,i})}_{K,s}, y_i \boldsymbol{\xi}_i \rangle, \langle \mathbf{w}^{(T^3_{2,s,i})}_{Q,s}, y_i \boldsymbol{\xi}_i \rangle < 0$, and

$$T^2_{2,s,i} \le \frac{\langle \mathbf{w}^{(0)}_{K,s}, \boldsymbol{\mu} \rangle - \langle \mathbf{w}^{(0)}_{K,s}, y_i \boldsymbol{\xi}_i \rangle}{\eta (\|\boldsymbol{\xi}_i\|_1 + \|\boldsymbol{\mu}\|)} \le \frac{1.5 \langle \mathbf{w}^{(0)}_{K,s}, y_i \boldsymbol{\xi}_i \rangle}{\eta \|\boldsymbol{\xi}_i\|_1} \le 1.5 T'_2,$$

where the first step is by Eq. (20), the second step is by Lemma C.10, the last step is by the definition of $T'_2$ and Lemma C.4.

Next, we are ready to bound $T^3_{2,s,i}$. Note that during $[0, T^3_{2,s,i}]$, only one of query and key noise change its update direction. WLOG, assume query noise changes the update direction while key noise doesn't. For the key noise, we have with at most $T'_2$ steps, its sign can align with the sign of its update. For the query noise, we possibly have that the sign at $T^2_{2,s,i}$ can be contrary with the sign at initialization, but with at most another $T^2_{2,s,i}$ steps, its sign can align with the sign of its update (and at initialization). By these analyses, we have

$$T^3_{2,s,i} \le \max\left\{T'_2, 2T^2_{2,s,i}\right\} \le 3T'_2,$$

for all $s \in [m_k]$ and $i \in S^{(0)}_{s,K+,Q-} \cup S^{(0)}_{s,K-,Q+}$.

In the main text, we define $T^{\mathrm{SGN}}_2 := 3T'_2$.

The following lemma gives bounds for the magnitude of query/key noise before $T^+_2$.

**Lemma E.2.** *For all $s \in [m_k]$, $i \in S^{(0)}_{s,K+,Q-} \cup S^{(0)}_{s,K-,Q+}$ and $t \in [3T'_2, T^+_2]$, we have*

$$\langle \mathbf{w}^{(t+1)}_{Q,s}, y_i \boldsymbol{\xi}_i \rangle = \langle \mathbf{w}^{(t)}_{Q,s}, y_i \boldsymbol{\xi}_i \rangle + \mathrm{sgn}(\langle \mathbf{w}^{(t)}_{Q,s}, y_i \boldsymbol{\xi}_i \rangle)\eta \|\boldsymbol{\xi}_i\|_1,$$

$$\langle \mathbf{w}^{(t+1)}_{K,s}, y_i \boldsymbol{\xi}_i \rangle = \langle \mathbf{w}^{(t)}_{K,s}, y_i \boldsymbol{\xi}_i \rangle + \mathrm{sgn}(\langle \mathbf{w}^{(t)}_{K,s}, y_i \boldsymbol{\xi}_i \rangle)\eta \|\boldsymbol{\xi}_i\|_1,$$

*and for all $t \in [12T'_2, T^+_2]$*

$$t\eta\sigma_p s/2 \le \left|\langle \mathbf{w}^{(t)}_{Q,s}, y_i \boldsymbol{\xi}_i \rangle\right|, \left|\langle \mathbf{w}^{(t)}_{K,s}, y_i \boldsymbol{\xi}_i \rangle\right| \le 2t\eta\sigma_p s.$$

*Proof.* The first two lines are shown in the above discussion. For the last line, for the lower bound, suppose $\langle \mathbf{w}^{(T^3_{2,s,i})}_{Q,s}, y_i \boldsymbol{\xi}_i \rangle > 0$, we have

$$\langle \mathbf{w}^{(t)}_{Q,s}, y_i \boldsymbol{\xi}_i \rangle$$
$$\ge (t - 3T'_2)\eta\sigma_p s/\sqrt{2} + \langle \mathbf{w}^{(3T'_2)}_{Q,s}, y_i \boldsymbol{\xi}_i \rangle$$
$$\ge (t - 3T'_2)\eta\sigma_p s/\sqrt{2}$$
$$\ge t\eta\sigma_p s/2,$$

where the first step is by Lemma C.4, the second step is by $3T'_2 \ge T^3_{2,s,i}$, the third step is by $t \ge 12T'_2$. For the upper bound, we have

$$\left|\langle \mathbf{w}^{(t)}_{Q,s}, y_i \boldsymbol{\xi}_i \rangle\right| \le \left|\langle \mathbf{w}^{(0)}_{Q,s}, y_i \boldsymbol{\xi}_i \rangle\right| + t\eta\sigma_p s \le 2t\eta\sigma_p s,$$

where the second step is by $t \ge T'_2$. $\qquad\square$

We next consider the samples $i \in S^{(0)}_{s,K+,Q+} \cup S^{(0)}_{s,K-,Q-}$ at initialization. The following lemma studies the update direction and magnitude of these neurons.

**Lemma E.3.** *For all $s \in [m_k]$, $i \in S_{s,K+,Q+}^{(0)} \cup S_{s,K-,Q-}^{(0)}$ and $t \in [0, T_2^+]$, we have*

$$\langle \mathbf{w}_{Q,s}^{(t+1)}, y_i \boldsymbol{\xi}_i \rangle = \langle \mathbf{w}_{Q,s}^{(t)}, y_i \boldsymbol{\xi}_i \rangle + \mathrm{sgn}(\langle \mathbf{w}_{Q,s}^{(t)}, y_i \boldsymbol{\xi}_i \rangle) \eta \|\boldsymbol{\xi}_i\|_1 \,,$$

$$\langle \mathbf{w}_{K,s}^{(t+1)}, y_i \boldsymbol{\xi}_i \rangle = \langle \mathbf{w}_{K,s}^{(t)}, y_i \boldsymbol{\xi}_i \rangle + \mathrm{sgn}(\langle \mathbf{w}_{K,s}^{(t)}, y_i \boldsymbol{\xi}_i \rangle) \eta \|\boldsymbol{\xi}_i\|_1 \,,$$

*and for all $t \in [T_2', T_2^+]$*

$$t\eta\sigma_p s/2 \le \left| \langle \mathbf{w}_{Q,s}^{(t)}, y_i \boldsymbol{\xi}_i \rangle \right|, \left| \langle \mathbf{w}_{K,s}^{(t)}, y_i \boldsymbol{\xi}_i \rangle \right| \le 2t\eta\sigma_p s.$$

*Proof.* The first two lines are shown in the above discussion. For the last line, note that without loss of generality, we can assume that $i \in S_{s,K+,Q+}^{(0)}$. We can show that the sign of update of $\langle \mathbf{w}_{Q,s}^{(t)}, y_i \boldsymbol{\xi}_i \rangle$ and $\langle \mathbf{w}_{K,s}^{(t)}, y_i \boldsymbol{\xi}_i \rangle$ are the same as their sign at initialization and would not change for $t \in [0, T_2^+]$. Formally

$$\langle \mathbf{w}_{K,s}^{(t+1)}, y_i \boldsymbol{\xi}_i \rangle - \langle \mathbf{w}_{K,s}^{(t)}, y_i \boldsymbol{\xi}_i \rangle =_{\mathrm{sgn}} \langle \mathbf{w}_{K,s}^{(0)}, y_i \boldsymbol{\xi}_i \rangle,$$

$$\langle \mathbf{w}_{Q,s}^{(t+1)}, y_i \boldsymbol{\xi}_i \rangle - \langle \mathbf{w}_{Q,s}^{(t)}, y_i \boldsymbol{\xi}_i \rangle =_{\mathrm{sgn}} \langle \mathbf{w}_{Q,s}^{(0)}, y_i \boldsymbol{\xi}_i \rangle,$$

for all $0 \le t \le T_2^+$. Similar to the proof in Lemma E.1, the claim holds at $t = 0$. Suppose at $t \le t' \in [0, T_2^+]$, the induction hypothesis holds, then we have

$$\langle \mathbf{w}_{K,s}^{(t'+1)}, y_i \boldsymbol{\xi}_i \rangle = \langle \mathbf{w}_{K,s}^{(t')}, y_i \boldsymbol{\xi}_i \rangle + \eta \|\boldsymbol{\xi}_i\|_1 \ge \langle \mathbf{w}_{K,s}^{(t')}, y_i \boldsymbol{\xi}_i \rangle,$$

$$\langle \mathbf{w}_{Q,s}^{(t'+1)}, y_i \boldsymbol{\xi}_i \rangle = \langle \mathbf{w}_{Q,s}^{(t')}, y_i \boldsymbol{\xi}_i \rangle + \eta \|\boldsymbol{\xi}_i\|_1 \ge \langle \mathbf{w}_{Q,s}^{(t')}, y_i \boldsymbol{\xi}_i \rangle,$$

which imply

$$- \langle \mathbf{w}_{K,s}^{(t'+1)}, \boldsymbol{\mu} \rangle + \langle \mathbf{w}_{K,s}^{(t'+1)}, y_i \boldsymbol{\xi}_i \rangle$$
$$\ge - \langle \mathbf{w}_{K,s}^{(t')}, \boldsymbol{\mu} \rangle + \langle \mathbf{w}_{K,s}^{(t')}, y_i \boldsymbol{\xi}_i \rangle + \eta(\|\boldsymbol{\xi}_i\|_1 - \|\boldsymbol{\mu}\|) \ge 0,$$

and

$$s_{i,11}^{(t'+1)} s_{i,12}^{(t'+1)} \langle \mathbf{w}_{Q,s}^{(t'+1)}, \boldsymbol{\mu} \rangle + s_{i,21}^{(t'+1)} s_{i,22}^{(t'+1)} \langle \mathbf{w}_{Q,s}^{(t'+1)}, y_i \boldsymbol{\xi}_i \rangle$$
$$\ge \frac{1}{4}(1 \pm o(1)) \cdot (\langle \mathbf{w}_{Q,s}^{(t')}, \boldsymbol{\mu} \rangle (1 + o(1)) + \langle \mathbf{w}_{Q,s}^{(t')}, y_i \boldsymbol{\xi}_i \rangle \eta(\|\boldsymbol{\xi}_i\|_1 - \|\boldsymbol{\mu}\|)) \ge 0,$$

where the first step is by Lemma E.14. Then we show the result for $t = t' + 1$ and prove the claim. Therefore, for the lower bound, we have

$$\langle \mathbf{w}_{K,s}^{(t)}, y_i \boldsymbol{\xi}_i \rangle = \langle \mathbf{w}_{K,s}^{(0)}, y_i \boldsymbol{\xi}_i \rangle + t\eta \|\boldsymbol{\xi}_i\|_1 \ge t\eta \|\boldsymbol{\xi}_i\|_1 \ge t\eta\sigma_p s/2,$$

where the first step and second step follow from the claim above, the last step is by Lemma C.4, and for the upper bound, we have

$$\langle \mathbf{w}_{K,s}^{(t)}, y_i \boldsymbol{\xi}_i \rangle = \langle \mathbf{w}_{K,s}^{(0)}, y_i \boldsymbol{\xi}_i \rangle + t\eta \|\boldsymbol{\xi}_i\|_1 \le T_2' \eta\sigma_p s + t\eta \|\boldsymbol{\xi}_i\|_1 \le 2t\eta\sigma_p s,$$

where the first step is by the claim above, the second step is by the definition of $T_2'$, the last step is by $t \ge T_2'$ and Lemma C.4. $\square$

**Remark E.4.** *Combining Lemma E.2 and E.3, we have for all $t \in [12T_2', T_2^+]$, $s \in [m_k]$ and $i \in [n]$*

$$t\eta\sigma_p s/2 \le \left| \langle \mathbf{w}_{Q,s}^{(t)}, y_i \boldsymbol{\xi}_i \rangle \right|, \left| \langle \mathbf{w}_{K,s}^{(t)}, y_i \boldsymbol{\xi}_i \rangle \right| \le 2t\eta\sigma_p s. \tag{21}$$

*Note that the upper bound also hold for all $t \ge T_2^+$ since the magnitude of update of query/key noise is always $\eta \|\boldsymbol{\xi}_i\|_1$, i.e., Eq. (9) always holds.*

Next, for given neuron $s$, we study the magnitude of the summation of query/key noise over all samples. The following lemma studies the dynamics and magnitude of query/key noise summation.

**Lemma E.5.** *Let $T_2 := 50nT_2'$. We have*

1. *(update direction). Let $\epsilon_{s,i} := \text{sgn}(\langle \mathbf{w}_{Q,s}^{(3T_2')}, y_i \boldsymbol{\xi}_i \rangle)$. We have the sign of the update of the summation of query/key noise would keep unchanged for $[3T_2', T_2^+]$. Formally, for all $s \in [m_k]$ and $t \in [3T_3', T_3^+]$*

$$\sum_{i=1}^{n} \langle \mathbf{w}_{K,s}^{(t+1)}, y_i \boldsymbol{\xi}_i \rangle - \sum_{i=1}^{n} \langle \mathbf{w}_{K,s}^{(t)}, y_i \boldsymbol{\xi}_i \rangle = \sum_{i=1}^{n} \langle \mathbf{w}_{Q,s}^{(t+1)}, y_i \boldsymbol{\xi}_i \rangle - \sum_{i=1}^{n} \langle \mathbf{w}_{Q,s}^{(t)}, y_i \boldsymbol{\xi}_i \rangle = \eta \sum_{i=1}^{n} \epsilon_{s,i} \|\boldsymbol{\xi}_i\|_1 .$$

2. *(magnitude of update). With probability at least $1 - O(m_k/\sqrt{n})$, for all $s \in [m_k]$*

$$\left| \eta \sum_{i=1}^{n} \epsilon_{s,i} \|\boldsymbol{\xi}_i\|_1 \right| \geq \eta \sigma_p s / \sqrt{2}.$$

3. *(magnitude of query/key noise summation). For all $s \in [m_k]$ and $t \in [T_2, T_2^+]$*

$$\text{sgn}(\eta \sum_{i=1}^{n} \epsilon_{s,i} \|\boldsymbol{\xi}_i\|_1) \cdot \sum_{i=1}^{n} \langle \mathbf{w}_{Q,s}^{(t)}, y_i \boldsymbol{\xi}_i \rangle, \text{sgn}(\eta \sum_{i=1}^{n} \epsilon_{s,i} \|\boldsymbol{\xi}_i\|_1) \cdot \sum_{i=1}^{n} \langle \mathbf{w}_{K,s}^{(t)}, y_i \boldsymbol{\xi}_i \rangle \geq 2n\beta_{\boldsymbol{\mu}}.$$

To prove this lemma, we need the following lemma which studies the property of $\epsilon_{s,i}$.

**Lemma E.6.** *Let*

$$E_{s,+} := \left\{ i \in [n] : \langle \mathbf{w}_{Q,s}^{(T_2)}, y_i \boldsymbol{\xi}_i \rangle > 0 \right\},$$
$$E_{s,-} := \left\{ i \in [n] : \langle \mathbf{w}_{Q,s}^{(T_2)}, y_i \boldsymbol{\xi}_i \rangle < 0 \right\}.$$

*Let*

$$A_{s,i} := \left\{ \omega \in \Omega : -\langle \mathbf{w}_{K,s}^{(0)}, y_i \boldsymbol{\xi}_i \rangle + \langle \mathbf{w}_{K,s}^{(0)}, \boldsymbol{\mu} \rangle < \langle \mathbf{w}_{Q,s}^{(0)}, y_i \boldsymbol{\xi}_i \rangle + \langle \mathbf{w}_{Q,s}^{(0)}, \boldsymbol{\mu} \rangle \right\}.$$

*Let*

$$X_{s,i} := \mathbb{1}[i \in S_{s,K+,Q+}] + \mathbb{1}[i \in S_{s,K-,Q+}, A_{s,i}] + \mathbb{1}[i \in S_{s,K+,Q-}, A_{s,i}]$$
$$- \mathbb{1}[i \in S_{s,K-,Q-}] - \mathbb{1}[i \in S_{s,K-,Q+}, A_{s,i}^c] - \mathbb{1}[i \in S_{s,K+,Q-}, A_{s,i}^c].$$

*Then we have*

1. *(The distribution of $X_{s,i}$). For all $s \in [m_k]$ and $i \in [n]$,*

$$X_{s,i} = \begin{cases} 1, w.p. \ 1/2 \\ -1, w.p. \ 1/2 \end{cases} .$$

2. *(The probability of equal set size).*

$$\mathbb{P}[|E_{s,+}| - |E_{s,-}| = 0] \leq O(n^{-1/2}).$$

3. *(The probability of lucky neurons).*

$$\mathbb{P}[X_{s,i} > 0, \sum_{i=1}^{n} X_{s,i} > 0] \geq \frac{1}{4} - O(n^{-1/2}).$$

*Proof.* By the definition of $X_{s,i}$ and $\epsilon_{s,i}$, we have following equivalent formulations

$$|E_{s,+}| = |\{i \in [n] : \epsilon_{s,i} > 0\}| = \sum_{i=1}^{n} \mathbb{1}[i \in S_{s,K+,Q+}] + \mathbb{1}[i \in S_{s,K-,Q+}, A_{s,i}] + \mathbb{1}[i \in S_{s,K+,Q-}, A_{s,i}],$$

$$|E_{s,-}| = |\{i \in [n] : \epsilon_{s,i} < 0\}| = \sum_{i=1}^{n} \mathbb{1}[i \in S_{s,K-,Q-}] + \mathbb{1}[i \in S_{s,K-,Q+}, A_{s,i}^c] + \mathbb{1}[i \in S_{s,K+,Q-}, A_{s,i}^c].$$

Let $Q_{s,i} := \langle \mathbf{w}_{Q,s}^{(0)}, y_i \boldsymbol{\xi}_i \rangle$, $K_{s,i} := \langle \mathbf{w}_{k,s}^{(0)}, y_i \boldsymbol{\xi}_i \rangle$, $\mu_s := \langle \mathbf{w}_{K,s}^{(0)} - \mathbf{w}_{Q,s}^{(0)}, \boldsymbol{\mu} \rangle$. Then we have $\mu_s \sim N(0, \sigma_0^2 \|\boldsymbol{\mu}\|^2)$, and $Q_{s,i}, K_{s,i}|y_i\boldsymbol{\xi}_i \sim N(0, \sigma_0^2 \|\boldsymbol{\xi}_i\|^2)$. Denote the distribution of $\boldsymbol{\xi}$ and $\mu$ by $P_{\boldsymbol{\xi}}$ and $P_\mu$, respectively.

Let $p(\mu, \boldsymbol{\xi}_i) := \mathbb{P}[X_{s,i} = 1|\mu_s = \mu, \boldsymbol{\xi}_i]$, $p_\mu := \mathbb{P}[X_{s,i} = 1|\mu_s = \mu]$.

Let $\phi(\mu, \boldsymbol{\xi}_i) := \mathbb{P}[i \in S_{s,K-,Q+}, A_{s,i}|\mu_s = \mu, \boldsymbol{\xi}_i]$, $\phi_\mu := \mathbb{P}[i \in S_{s,K-,Q+}, A_{s,i}|\mu_s = \mu]$. By the symmetry of $S_{s,K-,Q+}$ and $S_{s,K+,Q-}$, we have $\mathbb{P}[i \in S_{s,K-,Q+}, A_{s,i}|\mu, \boldsymbol{\xi}_i] = \mathbb{P}[i \in S_{s,K+,Q-}, A_{s,i}|\mu, \boldsymbol{\xi}_i]$, which gives

$$p(\mu, \boldsymbol{\xi}_i) = \frac{1}{4} + 2\phi(\mu, \boldsymbol{\xi}_i),$$
$$p_\mu = \int p(\mu, \boldsymbol{\xi}_i)dP_{\boldsymbol{\xi}_i} = \int \frac{1}{4} + 2\phi(\mu, \boldsymbol{\xi}_i)dP_{\boldsymbol{\xi}_i} = \frac{1}{4} + 2\phi_\mu.$$

By the symmetry of the gaussian density, for any $\boldsymbol{\xi}_i$, we have

$$\begin{aligned}
\phi(\mu, \boldsymbol{\xi}_i) &= \int_{q>0,k<0,q+k>\mu} \exp(-\frac{q^2 + k^2}{2\sigma_0^2 \|\boldsymbol{\xi}_i\|^2})dqdk \\
&= \int_{q>0,k<0,q+k<-\mu} \exp(-\frac{q^2 + k^2}{2\sigma_0^2 \|\boldsymbol{\xi}_i\|^2})dqdk \\
&= \frac{1}{4} - \int_{q>0,k<0,q+k>-\mu} \exp(-\frac{q^2 + k^2}{2\sigma_0^2 \|\boldsymbol{\xi}_i\|^2})dqdk \\
&= \frac{1}{4} - \phi(-\mu, \boldsymbol{\xi}_i),
\end{aligned}$$

which implies $\phi(\mu, \boldsymbol{\xi}_i) - \frac{1}{8}$ is an odd function, and thus

$$\mathbb{P}[i \in S_{s,K-,Q+}, A_{s,i}|\boldsymbol{\xi}_i] = \int \phi(\mu, \boldsymbol{\xi}_i)dP_\mu = \frac{1}{8},$$

thus $\mathbb{P}[i \in S_{s,K-,Q+}, A_{s,i}] = \frac{1}{8}$ and $\mathbb{P}[X_{s,i} = 1] = \frac{1}{2}$, which proves the first result.

Note that given $\mu$, $X_{s,i}$ are independent for $i \in [n]$, then

$$\begin{aligned}
\mathbb{P}[|E_{s,+}| - |E_{s,-}| = 0] &= \mathbb{P}[\sum_{i=1}^n X_{s,i} = 0] \\
&= \int \binom{n}{n/2} p_\mu^{n/2}(1 - p_\mu)^{n/2}dP_\mu \\
&\leq \int \binom{n}{n/2} \frac{1}{2^n}dP_\mu \\
&\leq \binom{n}{n/2} \frac{1}{2^n} \leq \frac{10}{\sqrt{n}},
\end{aligned}$$

which proves the second result.

Recall that

$$\begin{aligned}
\phi(\mu, \boldsymbol{\xi}_i) &= 1/4 - \phi(-\mu, \boldsymbol{\xi}_i) \, \forall \boldsymbol{\xi}_i \\
\implies \phi_\mu &= 1/4 - \phi_{-\mu} \\
\implies p_\mu &= 1 - p_{-\mu}.
\end{aligned}$$

The quantity of interest can be written as

$$\underbrace{\mathbb{P}[X_{s,i} > 0, \sum_{j \neq i} X_{s,i} > 0]}_{I_1} = \int \underbrace{\sum_{k=\lfloor (n-1)/2 \rfloor + 1}^{n-1} \binom{n}{k} p_\mu^k (1 - p_\mu)^{n-k}}_{p_{n-1,\mu}} \cdot p_\mu dP_\mu,$$

$$\underbrace{\mathbb{P}[X_{s,i} > 0, \sum_{j \neq i} X_{s,i} < 0]}_{I_2} = \int \underbrace{\sum_{k=\lfloor (n-1)/2 \rfloor + 1}^{n-1} \binom{n}{k} p_\mu^{n-k} (1 - p_\mu)^k}_{q_{n-1,\mu}} \cdot p_\mu dP_\mu,$$

$$\underbrace{\mathbb{P}[X_{s,i} < 0, \sum_{j \neq i} X_{s,i} > 0]}_{I_3} = \int \sum_{k=\lfloor (n-1)/2 \rfloor + 1}^{n-1} \binom{n}{k} p_\mu^k (1 - p_\mu)^{n-k} \cdot (1 - p_\mu) dP_\mu,$$

$$\underbrace{\mathbb{P}[X_{s,i} < 0, \sum_{j \neq i} X_{s,i} < 0]}_{I_4} = \int \sum_{k=\lfloor (n-1)/2 \rfloor + 1}^{n-1} \binom{n}{k} p_\mu^{n-k} (1 - p_\mu)^k \cdot (1 - p_\mu) dP_\mu.$$

By the symmetry of $P_\mu$ and $p_\mu$, we have

$$\mathbb{P}[X_{s,i} > 0, \sum_{j \neq i} X_{s,i} > 0] = \mathbb{P}[X_{s,i} < 0, \sum_{j \neq i} X_{s,i} < 0],$$

$$\mathbb{P}[X_{s,i} > 0, \sum_{j \neq i} X_{s,i} < 0] = \mathbb{P}[X_{s,i} < 0, \sum_{j \neq i} X_{s,i} > 0].$$

Since

$$I_1 + I_4 - I_2 - I_3 = \int (p_\mu - (1 - p_\mu))(p_{n-1,\mu} - q_{n-1,\mu}) dP_\mu \geq 0,$$

where the last step is by $p_{n-1,\mu} > q_{n-1,\mu}$ when $p_\mu \geq \frac{1}{2}$, then we have

$$\mathbb{P}[X_{s,i} > 0, \sum_{i=1}^n X_{s,i} > 0] \geq I_1 \geq \frac{1}{4}(1 - \mathbb{P}[\sum_{j \neq i} X_{s,i} = 0]) \geq \frac{1}{4} - O(n^{-1/2}),$$

where the second step is by $I_1 = I_4 \geq I_2 = I_3$, the third step is by the second result above, which concludes the proof. □

*Proof of Lemma E.5.* By Lemma E.2 and E.3, we have the update direction of each element $\langle \mathbf{w}_{K,s}^{(t)}, y_i \boldsymbol{\xi}_i \rangle$ and $\langle \mathbf{w}_{Q,s}^{(t)}, y_i \boldsymbol{\xi}_i \rangle$ is unchanged for $t \in [3T_2', T_2^+]$, which implies the first result.

By Lemma E.6, with probability $1 - O(m_k/\sqrt{n})$, we have for all $s \in [m_k]$, $|E_{s,+}| - |E_{s,-}| \neq 0$. Below, suppose this fact holds, and all the results are based on this fact which hold with high probability.

WLOG, for fixed neuron $s$, suppose $|E_{s,+}| - |E_{s,-}| \geq 1$, then by Lemma C.5, we have

$$\eta \sum_{i=1}^n \epsilon_{s,i} \|\boldsymbol{\xi}_i\|_1 \geq \eta \sigma_p s / \sqrt{2}. \tag{22}$$

Finally, we note that at $t = 12T_2'$, we have

$$\sum_{i=1}^n \langle \mathbf{w}_{Q,s}^{(12T_3')}, y_i \boldsymbol{\xi}_i \rangle, \sum_{i=1}^n \langle \mathbf{w}_{K,s}^{(12T_3')}, y_i \boldsymbol{\xi}_i \rangle \geq -24T_3' n \eta \sigma_p s.$$

By the definition of $T_2$, we have $T_2 \geq (12 + 24\sqrt{2}n)T_2' + 2n\beta_{\boldsymbol{\mu}} \eta^{-1} \sigma_p^{-1} s^{-1}$. Then, for $t \in [T_2, T_2^+]$, we have

$$\sum_{i=1}^n \langle \mathbf{w}_{Q,s}^{(t)}, y_i \boldsymbol{\xi}_i \rangle, \sum_{i=1}^n \langle \mathbf{w}_{K,s}^{(t)}, y_i \boldsymbol{\xi}_i \rangle \geq 2n\beta_{\boldsymbol{\mu}}. \tag{23}$$

Similarly, if $|E_{s,+}| - |E_{s,-}| \leq -1$ for some other neuron $s$, then for $t \in [T_2, T_2^+]$, we have

$$\sum_{i=1}^{n} \langle \mathbf{w}_{Q,s}^{(t)}, y_i \boldsymbol{\xi}_i \rangle, \sum_{i=1}^{n} \langle \mathbf{w}_{K,s}^{(t)}, y_i \boldsymbol{\xi}_i \rangle \leq -2n\beta_{\boldsymbol{\mu}}.$$

$\square$

### E.6 STAGE III

Let

$$T_3^+ := C_2 \sqrt{\log(12m_k/\delta)} \sigma_0 \eta^{-1},$$

where $C_2 = \Theta(1)$ is a large constant. In this section, we will analyze the dynamics of $\langle \mathbf{w}_{K,s}^{(t)}, y_i \boldsymbol{\xi}_i \rangle$, $\langle \mathbf{w}_{Q,s}^{(t)}, y_i \boldsymbol{\xi}_i \rangle$, $\langle \mathbf{w}_{Q,s}^{(t)}, \boldsymbol{\mu} \rangle$ and $\langle \mathbf{w}_{K,s}^{(t)}, \boldsymbol{\mu} \rangle$ for all $s \in [m_k]$, $i \in [n]$ and $T_2 \leq t \leq T_3^+$. Generally, we already have controlled the direction and magnitude of the quantity $\sum_{i=1}^{n} \langle \mathbf{w}_{Q,s}^{(t)}, y_i \boldsymbol{\xi}_i \rangle$ and $\sum_{i=1}^{n} \langle \mathbf{w}_{K,s}^{(t)}, y_i \boldsymbol{\xi}_i \rangle$ at the timestep $T_2$, we can show the update direction of query/key noise is determined by these quantities.

For $t \leq T_2$, we want to show the change of $\langle \mathbf{w}_{Q,s}^{(t)}, \boldsymbol{\mu} \rangle$ is not much. Actually, we can show that for $t \leq T_2^+$, $\langle \mathbf{w}_{Q,s}^{(t)}, \boldsymbol{\mu} \rangle$ and $\langle \mathbf{w}_{K,s}^{(t)}, \boldsymbol{\mu} \rangle$ are close to initialization. Formally, we have the following lemma.

**Lemma E.7.** *For all $s \in [m_k]$ and $t \leq T_2^+$, we have*

$$\langle \mathbf{w}_{K,s}^{(t)}, \boldsymbol{\mu} \rangle = \langle \mathbf{w}_{K,s}^{(t)}, \boldsymbol{\mu} \rangle \cdot (1 \pm o(1)),$$

$$\langle \mathbf{w}_{Q,s}^{(t)}, \boldsymbol{\mu} \rangle = \langle \mathbf{w}_{Q,s}^{(t)}, \boldsymbol{\mu} \rangle \cdot (1 \pm o(1)).$$

*Proof.* For key signal, we have

$$\frac{\left| \langle \mathbf{w}_{K,s}^{(t)}, \boldsymbol{\mu} \rangle \right|}{\left| \langle \mathbf{w}_{K,s}^{(0)}, \boldsymbol{\mu} \rangle \right|} \leq \frac{\left| \langle \mathbf{w}_{K,s}^{(0)}, \boldsymbol{\mu} \rangle \right| + t\eta \|\boldsymbol{\mu}\|}{\left| \langle \mathbf{w}_{K,s}^{(0)}, \boldsymbol{\mu} \rangle \right|} \leq 1 + \frac{t\eta \|\boldsymbol{\mu}\|}{\Omega(\sigma_0 n s^{-1/3} \|\boldsymbol{\mu}\|)} = 1 + o(1),$$

where the first step is by Eq. (8), the second step is by Lemma C.12, and the last step is by the definition of $T_2^+$. This also imply the sign of $\langle \mathbf{w}_{K,s}^{(t)}, \boldsymbol{\mu} \rangle$ is the same as initialization for all $t \leq T_2^+$. The proof for query signal is similar. $\square$

For $t \geq T_2$, we want to show the update direction of $\langle \mathbf{w}_{Q,s}^{(t)}, \boldsymbol{\mu} \rangle$ or $\langle \mathbf{w}_{K,s}^{(t)}, \boldsymbol{\mu} \rangle$) would be determined by the sign of $\sum_i \langle \mathbf{w}_{K,s}^{(t)}, y_i \boldsymbol{\xi}_i \rangle$ or $\sum_i \langle \mathbf{w}_{Q,s}^{(t)}, y_i \boldsymbol{\xi}_i \rangle$, respectively. To this end, we need to control the weighting term to make the weighted summation of query/key noise similar to the ordinary summation, which enables us to apply the previous analysis. These weighting terms include: (1) The softmax outputs; (2) The loss derivative; (3) The magnitude of mean value noise $\langle \mathbf{v}^{(t)}, y_i \boldsymbol{\xi}_i \rangle$. We want to show that these quantities are almost same across all samples, or even keep constant before $T_4^+$.

The following lemma studies the magnitude of $\langle \mathbf{v}^{(t)}, y_i \boldsymbol{\xi}_i \rangle$ after $T_2$. This lemma also shows $\langle \mathbf{v}^{(t)}, y_i \boldsymbol{\xi}_i \rangle$ are close between different samples.

**Lemma E.8.** *For all $i \in [n]$ and $t \geq T_2$, we have*

$$\langle \mathbf{v}^{(t)}, y_i \boldsymbol{\xi}_i \rangle = 2\sqrt{\frac{2}{\pi}} t\eta \sigma_p s (1 + \tilde{O}(s^{-1/2})).$$

*Proof.* We can show

$$\begin{aligned}
\langle \mathbf{v}^{(t)}, y_i \boldsymbol{\xi}_i \rangle &= \langle \mathbf{v}^{(0)}, y_i \boldsymbol{\xi}_i \rangle + 2t\eta \|\boldsymbol{\xi}_i\|_1 \\
&= 2t\eta \|\boldsymbol{\xi}_i\|_1 (1 + o(1)) \\
&= 2\sqrt{\frac{2}{\pi}} t\eta \sigma_p s (1 + \tilde{O}(s^{-1/2}))(1 + o(1)).
\end{aligned}$$

where the second step is by $\langle \mathbf{v}^{(0)}, y_i \boldsymbol{\xi}_i \rangle = O(\beta_{\boldsymbol{\xi}} m_v^{-1/2}) = o(T_2 \eta \sigma_p s)$, the third step is by Lemma C.4. $\qquad \square$

By Lemma E.14, we have that softmax outputs are concentrated at $1/2$ for all $i \in [n]$ and $t \leq T_3^+$. By Lemma E.15, we have that the derivative of loss of all samples are concentrated at $1/2$ for all $i \in [n]$ and $t \leq T_3^+$.

With these properties, we are ready to show that the query/key noise dynamics keep unchanged for $[T_2, T_3^+]$, and the query/key signal dynamics are dominated by the sign of the summation of query/key noise, i.e., $\langle \mathbf{w}_{Q,s}^{(t)}, \boldsymbol{\mu} \rangle$ and $\langle \mathbf{w}_{K,s}^{(t)}, \boldsymbol{\mu} \rangle$. The next two lemmas characterize the dynamics of query/key noise and query/key signal, respectively.

**Lemma E.9.** *For all $s \in [m_k]$, $i \in [n]$ and $T_2 \leq t \leq T_3^+$, we have*

$$\langle \mathbf{w}_{Q,s}^{(t+1)}, y_i \boldsymbol{\xi}_i \rangle = \langle \mathbf{w}_{Q,s}^{(t)}, y_i \boldsymbol{\xi}_i \rangle + \mathrm{sgn}(\langle \mathbf{w}_{Q,s}^{(t)}, y_i \boldsymbol{\xi}_i \rangle) \eta \, \|\boldsymbol{\xi}_i\|_1 \,,$$
$$\langle \mathbf{w}_{K,s}^{(t+1)}, y_i \boldsymbol{\xi}_i \rangle = \langle \mathbf{w}_{K,s}^{(t)}, y_i \boldsymbol{\xi}_i \rangle + \mathrm{sgn}(\langle \mathbf{w}_{K,s}^{(t)}, y_i \boldsymbol{\xi}_i \rangle) \eta \, \|\boldsymbol{\xi}_i\|_1 \,,$$

*and for all $t \in [T_2, T_3^+]$*

$$t \eta \sigma_p s / 2 \leq \left| \langle \mathbf{w}_{Q,s}^{(t)}, y_i \boldsymbol{\xi}_i \rangle \right|, \left| \langle \mathbf{w}_{K,s}^{(t)}, y_i \boldsymbol{\xi}_i \rangle \right| \leq 2 t \eta \sigma_p s.$$

*Proof.* WLOG, given neuron $s \in [m_k]$ and sample $i \in [n]$, suppose $\langle \mathbf{w}_{Q,s}^{(t)}, y_i \boldsymbol{\xi}_i \rangle > 0$, $\langle \mathbf{w}_{K,s}^{(t)}, y_i \boldsymbol{\xi}_i \rangle > 0$, we claim that $\langle \mathbf{w}_{Q,s}^{(t)}, y_i \boldsymbol{\xi}_i \rangle$ and $\langle \mathbf{w}_{K,s}^{(t)}, y_i \boldsymbol{\xi}_i \rangle$ are increasing for $T_2 \leq t \leq T_3^+$. At $t = T_2$, we have

$$\langle \mathbf{w}_{Q,s}^{(T_2+1)} - \mathbf{w}_{Q,s}^{(T_2)}, y_i \boldsymbol{\xi}_i \rangle =_{\mathrm{sgn}} - \langle \mathbf{w}_{K,s}^{(T_2)}, \boldsymbol{\mu} \rangle + \langle \mathbf{w}_{K,s}^{(T_2)}, y_i \boldsymbol{\xi}_i \rangle \geq \tilde{\Omega}(\sigma_0 \sigma_p \sqrt{s} n) - \tilde{O}(\sigma_0 \|\boldsymbol{\mu}\|) > 0,$$

and

$$\begin{aligned}
\langle \mathbf{w}_{K,s}^{(T_2+1)} - \mathbf{w}_{K,s}^{(T_2)}, y_i \boldsymbol{\xi}_i \rangle &=_{\mathrm{sgn}} s_{i,11}^{(T_2)} s_{i,12}^{(T_2)} \langle \mathbf{w}_{Q,s}^{(T_2)}, \boldsymbol{\mu} \rangle + s_{i,21}^{(T_2)} s_{i,22}^{(T_2)} \langle \mathbf{w}_{Q,s}^{(T_2)}, y_i \boldsymbol{\xi}_i \rangle \\
&= (\frac{1}{4} + o(1))(\langle \mathbf{w}_{Q,s}^{(T_2)}, \boldsymbol{\mu} \rangle + \langle \mathbf{w}_{Q,s}^{(T_2)}, y_i \boldsymbol{\xi}_i \rangle) \\
&= (\frac{1}{4} + o(1))(\tilde{\Omega}(\sigma_0 \sigma_p \sqrt{s} n) - \tilde{O}(\sigma_0 \|\boldsymbol{\mu}\|)) > 0.
\end{aligned}$$

Suppose the induction holds for $t \leq \tilde{t} - 1$, then at $t = \tilde{t}$, we have

$$\begin{aligned}
\langle \mathbf{w}_{Q,s}^{(\tilde{t}+1)} - \mathbf{w}_{Q,s}^{(\tilde{t})}, y_i \boldsymbol{\xi}_i \rangle &=_{\mathrm{sgn}} - \langle \mathbf{w}_{K,s}^{(\tilde{t})}, \boldsymbol{\mu} \rangle + \langle \mathbf{w}_{K,s}^{(\tilde{t})}, y_i \boldsymbol{\xi}_i \rangle \\
&\geq - \langle \mathbf{w}_{K,s}^{(T_3)}, \boldsymbol{\mu} \rangle + \langle \mathbf{w}_{K,s}^{(T_3)}, y_i \boldsymbol{\xi}_i \rangle + (\tilde{t} - T_3) \eta (\|\boldsymbol{\xi}\|_1 - \|\boldsymbol{\mu}\|) > 0,
\end{aligned}$$

and

$$\begin{aligned}
\langle \mathbf{w}_{K,s}^{(\tilde{t}+1)} - \mathbf{w}_{K,s}^{(\tilde{t})}, y_i \boldsymbol{\xi}_i \rangle &=_{\mathrm{sgn}} s_{i,11}^{(\tilde{t})} s_{i,12}^{(\tilde{t})} \langle \mathbf{w}_{Q,s}^{(\tilde{t})}, \boldsymbol{\mu} \rangle + s_{i,21}^{(\tilde{t})} s_{i,22}^{(\tilde{t})} \langle \mathbf{w}_{Q,s}^{(\tilde{t})}, y_i \boldsymbol{\xi}_i \rangle \\
&= (\frac{1}{4} + o(1))(\langle \mathbf{w}_{Q,s}^{(\tilde{t})}, \boldsymbol{\mu} \rangle + \langle \mathbf{w}_{Q,s}^{(\tilde{t})}, y_i \boldsymbol{\xi}_i \rangle) \\
&\geq (\frac{1}{4} + o(1)) \left( \langle \mathbf{w}_{Q,s}^{(T_3)}, \boldsymbol{\mu} \rangle + \langle \mathbf{w}_{Q,s}^{(T_3)}, y_i \boldsymbol{\xi}_i \rangle + (\tilde{t} - T_3) \eta (\|\boldsymbol{\xi}\|_1 - \|\boldsymbol{\mu}\|) \right) > 0.
\end{aligned}$$

Finally, the bounds about magnitude follow from Lemma E.2, E.3 and C.4. $\qquad \square$

**Remark E.10.** *Lemma E.9 implies that the dynamics of $\sum_{i \in [n]} \langle \mathbf{w}_{Q,s}^{(t)}, y_i \boldsymbol{\xi}_i \rangle$, $\sum_{i \in [n]} \langle \mathbf{w}_{K,s}^{(t)}, y_i \boldsymbol{\xi}_i \rangle$ keep unchanged for $T_2 \leq t \leq T_3^+$ and for all $s \in [m_k]$.*

**Lemma E.11.** *For all $s \in [m_k]$, $i \in [n]$ and $T_2 \leq t \leq T_3^+$, we have*

$$\langle \mathbf{w}_{Q,s}^{(t+1)}, \boldsymbol{\mu} \rangle = \langle \mathbf{w}_{Q,s}^{(t)}, \boldsymbol{\mu} \rangle + \mathrm{sgn}(\sum_{i \in [n]} \langle \mathbf{w}_{Q,s}^{(T_2)}, y_i \boldsymbol{\xi}_i \rangle) \eta \, \|\boldsymbol{\mu}\| \,,$$
$$\langle \mathbf{w}_{K,s}^{(t+1)}, \boldsymbol{\mu} \rangle = \langle \mathbf{w}_{K,s}^{(t)}, \boldsymbol{\mu} \rangle - \mathrm{sgn}(\sum_{i \in [n]} \langle \mathbf{w}_{Q,s}^{(T_2)}, y_i \boldsymbol{\xi}_i \rangle) \eta \, \|\boldsymbol{\mu}\| \,.$$

*Proof.* WLOG, given neuron $s \in [m_k]$, suppose we have $\sum_{i \in [n]} \langle \mathbf{w}_{K,s}^{(T_2)}, y_i \boldsymbol{\xi}_i \rangle > 0$, thus $\sum_{i \in [n]} \langle \mathbf{w}_{Q,s}^{(T_2)}, y_i \boldsymbol{\xi}_i \rangle > 0$. Therefore, for $T_2 \leq t \leq T_3^+$, we have

$$
\begin{aligned}
&\langle \mathbf{w}_{Q,s}^{(t+1)} - \mathbf{w}_{Q,s}^{(t)}, \boldsymbol{\mu} \rangle \\
=_{\mathrm{sgn}} &\sum_{i \in [n]} (-l_i'^{(t)} s_{i,11}^{(t)} s_{i,12}^{(t)}) \cdot \left( -y_i \langle \mathbf{v}^{(t)}, \boldsymbol{\xi}_i \rangle \langle \mathbf{w}_{K,s}^{(t)}, \boldsymbol{\mu} \rangle + \langle \mathbf{v}^{(t)}, \boldsymbol{\xi}_i \rangle \langle \mathbf{w}_{K,s}^{(t)}, \boldsymbol{\xi}_i \rangle \right) \\
=_{\mathrm{sgn}} &\sum_{i \in [n]} \frac{\sqrt{2}}{8\sqrt{\pi}} t\eta\sigma_p s(1 + o(1)) \cdot (\langle \mathbf{w}_{K,s}^{(t)}, y_i \boldsymbol{\xi}_i \rangle - \langle \mathbf{w}_{K,s}^{(t)}, \boldsymbol{\mu} \rangle) \\
=_{\mathrm{sgn}} &\sum_{i \in [n]} \langle \mathbf{w}_{K,s}^{(t)}, y_i \boldsymbol{\xi}_i \rangle - n\langle \mathbf{w}_{K,s}^{(t)}, \boldsymbol{\mu} \rangle \\
\geq &\sum_{i \in [n]} \langle \mathbf{w}_{K,s}^{(T_2)}, y_i \boldsymbol{\xi}_i \rangle - n\langle \mathbf{w}_{K,s}^{(T_2)}, \boldsymbol{\mu} \rangle + (t - T_2)(\eta\sigma_p s/\sqrt{2} - \eta n \|\boldsymbol{\mu}\|) > 0,
\end{aligned}
$$

where the first inequality is due to the lower bound Eq. (22), and the second inequality is due to the lower bound Eq. (23) and $n \|\boldsymbol{\mu}\| = o(\sigma_p s)$. Similarly, we have

$$
\begin{aligned}
&\langle \mathbf{w}_{K,s}^{(t+1)} - \mathbf{w}_{K,s}^{(t)}, \boldsymbol{\mu} \rangle \\
=_{\mathrm{sgn}} &\sum_{i \in [n]} (-l_i'^{(t)}) \cdot \left( -y_i s_{i,11}^{(t)} s_{i,12}^{(t)} \langle \mathbf{v}^{(t)}, \boldsymbol{\xi}_i \rangle \langle \mathbf{w}_{Q,s}^{(t)}, \boldsymbol{\mu} \rangle - s_{i,21}^{(t)} s_{i,22}^{(t)} \langle \mathbf{v}^{(t)}, \boldsymbol{\xi}_i \rangle \langle \mathbf{w}_{Q,s}^{(t)}, \boldsymbol{\xi}_i \rangle \right) \\
=_{\mathrm{sgn}} &\sum_{i \in [n]} \frac{\sqrt{2}}{8\sqrt{\pi}} t\eta\sigma_p s(1 + o(1)) \cdot (-\langle \mathbf{w}_{Q,s}^{(t)}, y_i \boldsymbol{\xi}_i \rangle - \langle \mathbf{w}_{Q,s}^{(t)}, \boldsymbol{\mu} \rangle) \\
=_{\mathrm{sgn}} &-\sum_{i \in [n]} \langle \mathbf{w}_{Q,s}^{(t)}, y_i \boldsymbol{\xi}_i \rangle - n\langle \mathbf{w}_{Q,s}^{(t)}, \boldsymbol{\mu} \rangle \\
\leq &-\sum_{i \in [n]} \langle \mathbf{w}_{Q,s}^{(T_2)}, y_i \boldsymbol{\xi}_i \rangle - n\langle \mathbf{w}_{Q,s}^{(T_2)}, \boldsymbol{\mu} \rangle - (t - T_2)(\eta\sigma_p s/\sqrt{2} - \eta n \|\boldsymbol{\mu}\|) < 0.
\end{aligned}
$$

$\square$

**Lemma E.12.** *Let* $T_3 := 3\beta_{\boldsymbol{\mu}} \eta^{-1} \|\boldsymbol{\mu}\|^{-1} = \tilde{O}(\sigma_0 \eta^{-1})$, *then for* $T_3 \leq t \leq T_3^+$, *we have*

$$
t\eta \|\boldsymbol{\mu}\| /2 \leq \mathrm{sgn}(\sum_{i \in [n]} \langle \mathbf{w}_{Q,s}^{(T_2)}, y_i \boldsymbol{\xi}_i \rangle) \cdot \langle \mathbf{w}_{Q,s}^{(t)}, \boldsymbol{\mu} \rangle \leq 4t\eta \|\boldsymbol{\mu}\| /3, \tag{24}
$$

$$
-4t\eta \|\boldsymbol{\mu}\| /3 \leq \mathrm{sgn}(\sum_{i \in [n]} \langle \mathbf{w}_{Q,s}^{(T_2)}, y_i \boldsymbol{\xi}_i \rangle) \cdot \langle \mathbf{w}_{K,s}^{(t)}, \boldsymbol{\mu} \rangle \leq -t\eta \|\boldsymbol{\mu}\| /2. \tag{25}
$$

*Proof.* WLOG, given neuron $s \in [m_k]$, suppose we have $\sum_{i \in [n]} \langle \mathbf{w}_{K,s}^{(T_2)}, y_i \boldsymbol{\xi}_i \rangle > 0$. We only prove Eq. (24). The proof of Eq. (25) is identical. For the upper bound, for $t \in [T_3, T_3^+]$ we have

$$
\langle \mathbf{w}_{Q,s}^{(t)}, \boldsymbol{\mu} \rangle \leq \beta_{\boldsymbol{\mu}} + t\eta \|\boldsymbol{\mu}\| \leq (t + T_3/3)\eta \|\boldsymbol{\mu}\| \leq 4t\eta \|\boldsymbol{\mu}\| /3,
$$

where the first step is by Eq. (8), the second step is by the definition of $T_3$. For the lower bound, for $t \in [T_3, T_3^+]$ we have

$$
\langle \mathbf{w}_{Q,s}^{(t)}, \boldsymbol{\mu} \rangle \geq -\beta_{\boldsymbol{\mu}} - T_2\eta \|\boldsymbol{\mu}\| + (t - T_2)\eta \|\boldsymbol{\mu}\| \geq (t - T_3/3 - 2T_2)\eta \|\boldsymbol{\mu}\| \geq t\eta \|\boldsymbol{\mu}\| /2,
$$

where the first step is by Eq. (8), the second step is by definition of $T_3$, the third step is by $T_2 T_3^{-1} = o(1)$. $\square$

**Remark E.13.** *Note that the upper bound in Eq. (24) and lower bound in Eq. (25) hold for all $t \geq T_3$ since the magnitude of update of query/key signal is always $\eta \|\boldsymbol{\mu}\|$, i.e., Eq. (8) always holds.*

### E.7 STAGE IV

This section study the concentration behavior of softmax outputs and loss derivative. We first show the magnitude of softmax outputs at initialization $s_{i,11}^{(t)}$ and $s_{i,21}^{(t)}$ are concentrated at 1/2 for all $i \in [n]$ before $T_4$ and $T_3^+$, respectively.

**Lemma E.14.** *Let*

$$T_4 := C_3 \log \left( \frac{C_3 \sigma_p s}{\|\boldsymbol{\mu}\|} \right) \eta^{-1} m_k^{-1/2} \sigma_p^{-1} s^{-1} \tag{26}$$

*where $C_3 = \Theta(1)$ is a large constant. Then, for all $i \in [n]$, we have*

$$s_{i,11}^{(t)} = 1/2 \pm o(1), \ \forall t \le T_4,$$
$$s_{i,21}^{(t)} = 1/2 \pm o(1), \ \forall t \le T_3^+.$$

*Proof of Lemma E.14.* Note that for $t \ge 0$,

$$\left| \langle \mathbf{w}_{Q,s}^{(t)}, y_i \boldsymbol{\xi}_i \rangle \right| \le \left| \langle \mathbf{w}_{Q,s}^{(0)}, y_i \boldsymbol{\xi}_i \rangle \right| + 2t\eta\sigma_p s$$
$$\le 4 \max \{ \beta_{\boldsymbol{\xi}}, t\eta\sigma_p s \},$$

for all $i \in [n]$ and $s \in [m_k]$ and

$$\left| \langle \mathbf{w}_{Q,s}^{(t)}, \boldsymbol{\mu} \rangle \right| \le \left| \langle \mathbf{w}_{Q,s}^{(0)}, \boldsymbol{\mu} \rangle \right| + t\eta \|\boldsymbol{\mu}\|$$
$$\le 4 \max \{ \beta_{\boldsymbol{\mu}}, t\eta \|\boldsymbol{\mu}\| \},$$

for all $s \in [m_k]$. Similarly, we also have

$$\left| \langle \mathbf{w}_{K,s}^{(t)}, \boldsymbol{\mu} \rangle \right| \le 4 \max \{ \beta_{\boldsymbol{\mu}}, t\eta \|\boldsymbol{\mu}\| \},$$
$$\left| \langle \mathbf{w}_{K,s}^{(t)}, y_i \boldsymbol{\xi}_i \rangle \right| \le 4 \max \{ \beta_{\boldsymbol{\xi}}, t\eta\sigma_p s \}.$$

When $t \le T_3^+$, we have for all $i \in [n]$

$$\left| \langle \mathbf{w}_{Q,s}^{(t)}, y_i \boldsymbol{\xi}_i \rangle \right|, \left| \langle \mathbf{w}_{K,s}^{(t)}, y_i \boldsymbol{\xi}_i \rangle \right| \le O(\sqrt{\log(12m_k/\delta)}\sigma_0\sigma_p s),$$
$$\left| \langle \mathbf{w}_{Q,s}^{(t)}, \boldsymbol{\mu} \rangle \right|, \left| \langle \mathbf{w}_{K,s}^{(t)}, \boldsymbol{\mu} \rangle \right| \le O(\sqrt{\log(12m_k/\delta)}\sigma_0 \|\boldsymbol{\mu}\|).$$

Then we have for all $i \in [n]$,

$$s_{i,21}^{(t)} = \frac{\exp \left( \sum_{s \in [m_k]} \langle \mathbf{w}_{Q,s}^{(t)}, y_i \boldsymbol{\xi}_i \rangle \langle \mathbf{w}_{K,s}^{(t)}, \boldsymbol{\mu} \rangle \right)}{\exp \left( \sum_{s \in [m_k]} \langle \mathbf{w}_{Q,s}^{(t)}, y_i \boldsymbol{\xi}_i \rangle \langle \mathbf{w}_{K,s}^{(t)}, \boldsymbol{\mu} \rangle \right) + \exp \left( \sum_{s \in [m_k]} \langle \mathbf{w}_{Q,s}^{(t)}, y_i \boldsymbol{\xi}_i \rangle \langle \mathbf{w}_{K,s}^{(t)}, y_i \boldsymbol{\xi}_i \rangle \right)}$$

$$\le \frac{\exp \left( O(m_k \log(12m_k/\delta)\sigma_0^2\sigma_p s \|\boldsymbol{\mu}\|) \right)}{\exp \left( O(m_k \log(12m_k/\delta)\sigma_0^2\sigma_p s \|\boldsymbol{\mu}\|) \right) + \exp \left( -O(m_k \log(12m_k/\delta)\sigma_0^2\sigma_p^2 s^2) \right)}$$

$$= \frac{1}{1 + \exp \left( -O(m_k \log(12m_k/\delta)\sigma_0^2(\sigma_p^2 s^2 + \sigma_p s \|\boldsymbol{\mu}\|)) \right)}$$

$$\le \frac{1}{2} + O(m_k \log(12m_k/\delta)\sigma_0^2(\sigma_p^2 s^2 + \sigma_p s \|\boldsymbol{\mu}\|)),$$

and

$$s_{i,21}^{(t)} = \frac{\exp \left( \sum_{s \in [m_k]} \langle \mathbf{w}_{Q,s}^{(t)}, y_i \boldsymbol{\xi}_i \rangle \langle \mathbf{w}_{K,s}^{(t)}, \boldsymbol{\mu} \rangle \right)}{\exp \left( \sum_{s \in [m_k]} \langle \mathbf{w}_{Q,s}^{(t)}, y_i \boldsymbol{\xi}_i \rangle \langle \mathbf{w}_{K,s}^{(t)}, \boldsymbol{\mu} \rangle \right) + \exp \left( \sum_{s \in [m_k]} \langle \mathbf{w}_{Q,s}^{(t)}, y_i \boldsymbol{\xi}_i \rangle \langle \mathbf{w}_{K,s}^{(t)}, y_i \boldsymbol{\xi}_i \rangle \right)}$$

$$\ge \frac{\exp \left( -O(m_k \log(12m_k/\delta)\sigma_0^2\sigma_p s \|\boldsymbol{\mu}\|) \right)}{\exp \left( -O(m_k \log(12m_k/\delta)\sigma_0^2\sigma_p s \|\boldsymbol{\mu}\|) \right) + \exp \left( O(m_k \log(12m_k/\delta)\sigma_0^2\sigma_p^2 s^2) \right)}$$

$$= \frac{1}{1 + \exp \left( O(m_k \log(12m_k/\delta)\sigma_0^2(\sigma_p^2 s^2 + \sigma_p s \|\boldsymbol{\mu}\|)) \right)}$$

$$\ge \frac{1}{2} - O(m_k \log(12m_k/\delta)\sigma_0^2(\sigma_p^2 s^2 + \sigma_p s \|\boldsymbol{\mu}\|)).$$

Similarly, when $t \leq T_4$, we have for all $i \in [n]$

$$\left|\langle \mathbf{w}_{Q,s}^{(t)}, y_i \boldsymbol{\xi}_i \rangle\right|, \left|\langle \mathbf{w}_{K,s}^{(t)}, y_i \boldsymbol{\xi}_i \rangle\right| \leq O(\log(C_3 \sigma_p s / \|\boldsymbol{\mu}\|) m_k^{-1/2}), \tag{27}$$

$$\left|\langle \mathbf{w}_{Q,s}^{(t)}, \boldsymbol{\mu} \rangle\right|, \left|\langle \mathbf{w}_{K,s}^{(t)}, \boldsymbol{\mu} \rangle\right| \leq O(\log(C_3 \sigma_p s / \|\boldsymbol{\mu}\|) m_k^{-1/2} \sigma_p^{-1} s^{-1} \|\boldsymbol{\mu}\|). \tag{28}$$

Then we have for all $i \in [n]$,

$$
\begin{aligned}
s_{i,11}^{(t)} &= \frac{\exp\left(\sum_{s \in [m_k]} \langle \mathbf{w}_{Q,s}^{(t)}, \boldsymbol{\mu} \rangle \langle \mathbf{w}_{K,s}^{(t)}, \boldsymbol{\mu} \rangle\right)}{\exp\left(\sum_{s \in [m_k]} \langle \mathbf{w}_{Q,s}^{(t)}, \boldsymbol{\mu} \rangle \langle \mathbf{w}_{K,s}^{(t)}, \boldsymbol{\mu} \rangle\right) + \exp\left(\sum_{s \in [m_k]} \langle \mathbf{w}_{Q,s}^{(t)}, \boldsymbol{\mu} \rangle \langle \mathbf{w}_{K,s}^{(t)}, y_i \boldsymbol{\xi}_i \rangle\right)} \\
&\leq \frac{\exp\left(O(\log^2(C_3 \sigma_p s / \|\boldsymbol{\mu}\|) \sigma_p^{-2} s^{-2} \|\boldsymbol{\mu}\|^2)\right)}{\exp\left(O(\log^2(C_3 \sigma_p s / \|\boldsymbol{\mu}\|) \sigma_p^{-2} s^{-2} \|\boldsymbol{\mu}\|^2)\right) + \exp\left(-O(\log^2(C_3 \sigma_p s / \|\boldsymbol{\mu}\|) \sigma_p^{-1} s^{-1} \|\boldsymbol{\mu}\|)\right)} \\
&= \frac{1}{1 + \exp\left(-O(\log^2(C_3 \sigma_p s / \|\boldsymbol{\mu}\|) \sigma_p^{-1} s^{-1} \|\boldsymbol{\mu}\| (1 + \sigma_p^{-1} s^{-1} \|\boldsymbol{\mu}\|))\right)} \\
&\leq \frac{1}{2} + O(\log^2(C_3 \sigma_p s / \|\boldsymbol{\mu}\|) \sigma_p^{-1} s^{-1} \|\boldsymbol{\mu}\| (1 + \sigma_p^{-1} s^{-1} \|\boldsymbol{\mu}\|)),
\end{aligned}
$$

and

$$
\begin{aligned}
s_{i,11}^{(t)} &= \frac{\exp\left(\sum_{s \in [m_k]} \langle \mathbf{w}_{Q,s}^{(t)}, \boldsymbol{\mu} \rangle \langle \mathbf{w}_{K,s}^{(t)}, \boldsymbol{\mu} \rangle\right)}{\exp\left(\sum_{s \in [m_k]} \langle \mathbf{w}_{Q,s}^{(t)}, \boldsymbol{\mu} \rangle \langle \mathbf{w}_{K,s}^{(t)}, \boldsymbol{\mu} \rangle\right) + \exp\left(\sum_{s \in [m_k]} \langle \mathbf{w}_{Q,s}^{(t)}, \boldsymbol{\mu} \rangle \langle \mathbf{w}_{K,s}^{(t)}, y_i \boldsymbol{\xi}_i \rangle\right)} \\
&\geq \frac{\exp\left(-O(\log^2(C_3 \sigma_p s / \|\boldsymbol{\mu}\|) \sigma_p^{-2} s^{-2} \|\boldsymbol{\mu}\|^2)\right)}{\exp\left(-O(\log^2(C_3 \sigma_p s / \|\boldsymbol{\mu}\|) \sigma_p^{-2} s^{-2} \|\boldsymbol{\mu}\|^2)\right) + \exp\left(O(\log^2(C_3 \sigma_p s / \|\boldsymbol{\mu}\|) \sigma_p^{-1} s^{-1} \|\boldsymbol{\mu}\|)\right)} \\
&= \frac{1}{1 + \exp\left(O(\log^2(C_3 \sigma_p s / \|\boldsymbol{\mu}\|) \sigma_p^{-1} s^{-1} \|\boldsymbol{\mu}\| (1 + \sigma_p^{-1} s^{-1} \|\boldsymbol{\mu}\|))\right)} \\
&\geq \frac{1}{2} - O(\log^2(C_3 \sigma_p s / \|\boldsymbol{\mu}\|) \sigma_p^{-1} s^{-1} \|\boldsymbol{\mu}\| (1 + \sigma_p^{-1} s^{-1} \|\boldsymbol{\mu}\|)).
\end{aligned}
$$

$\square$

The next lemma shows that loss derivative $\ell_i'^{(t)}$ are concentrated at $1/2$ for all $i \in [n]$ before $T_4$.

**Lemma E.15.** *Recall the $T_4$ defined in Eq. (26). For all $i \in [n]$, we have*

$$\ell_i'^{(t)} = 1/2 \pm o(1), \ \forall t \leq T_4.$$

*Proof of Lemma E.15.* Note that $\ell(z) = \log(1 + \exp(-z))$ and $-\ell'(z) = 1/(1 + \exp(z))$. For $t \leq T_3^+$ we have

$$
\begin{aligned}
\ell_i'^{(t)} &= \frac{1}{1 + \exp\left((s_{i,11}^{(t)} + s_{i,21}^{(t)})\langle \mathbf{v}^{(t)}, y_i \boldsymbol{\mu} \rangle + (s_{i,12}^{(t)} + s_{i,22}^{(t)})\langle \mathbf{v}^{(t)}, \boldsymbol{\xi}_i \rangle\right)} \\
&= \frac{1}{1 + \exp\left(y_i(1 \pm o(1))(\langle \mathbf{v}^{(t)}, \boldsymbol{\mu} \rangle + \langle \mathbf{v}^{(t)}, y_i \boldsymbol{\xi}_i \rangle)\right)} \\
&= \frac{1}{1 + \exp\left(y_i(1 \pm o(1))\tilde{O}(\sigma_0 \sigma_p s)\right)} \\
&= \frac{1}{2} \pm o(1),
\end{aligned}
$$

where the first step is by definition, the second step is by Lemma E.14, the third step is by Lemma E.8, the last step is due to $\sigma_0 \sigma_p s = o(1)$. For $t \leq T_4$, suppose $y_i = 1$, we have $\langle \mathbf{v}^{(t)}, y_i \boldsymbol{\mu} \rangle, \langle \mathbf{v}^{(t)}, \boldsymbol{\xi}_i \rangle > 0$,

then

$$\ell_i'^{(t)} = \frac{1}{1 + \exp\left((s_{i,11}^{(t)} + s_{i,21}^{(t)})\langle \mathbf{v}^{(t)}, y_i\boldsymbol{\mu}\rangle + (s_{i,12}^{(t)} + s_{i,22}^{(t)})\langle \mathbf{v}^{(t)}, \boldsymbol{\xi}_i\rangle\right)}$$

$$\leq \frac{1}{2} + o(1),$$

where the first step is by definition, the second step is by $\langle \mathbf{v}^{(t)}, y_i\boldsymbol{\mu}\rangle, \langle \mathbf{v}^{(t)}, \boldsymbol{\xi}_i\rangle > 0$, and

$$\ell_i'^{(t)} = \frac{1}{1 + \exp\left((s_{i,11}^{(t)} + s_{i,21}^{(t)})\langle \mathbf{v}^{(t)}, y_i\boldsymbol{\mu}\rangle + (s_{i,12}^{(t)} + s_{i,22}^{(t)})\langle \mathbf{v}^{(t)}, \boldsymbol{\xi}_i\rangle\right)}$$

$$\geq \frac{1}{1 + \exp\left(y_i(3/2 \pm o(1))(\langle \mathbf{v}^{(t)}, \boldsymbol{\mu}\rangle + \langle \mathbf{v}^{(t)}, y_i\boldsymbol{\xi}_i\rangle)\right)}$$

$$= \frac{1}{1 + \exp\left(y_i(3/2 \pm o(1))\langle \mathbf{v}^{(t)}, y_i\boldsymbol{\xi}_i\rangle\right)}$$

$$= \frac{1}{1 + \exp\left(y_i(3/2 \pm o(1))\tilde{O}(m_k^{-1/2})\right)}$$

$$\geq \frac{1}{2} - o(1),$$

where the first step is by definition, the second step is by Lemma E.14, the third step is by Lemma E.8, the fourth and last step is by $\langle \mathbf{v}^{(T_4)}, y_i\boldsymbol{\xi}_i\rangle = \tilde{O}(m_k^{-1/2}) = o(1)$. □

### E.7.1 STAGE IV.A

This stage studies the dynamics in the interval $[T_3, T_4^-]$, show the fast decay of post-softmax weights and their impact. Roughly, everything keeps unchanged in this stage. The idea is that everything keeps going on as usual before $T_4^-$, which is defined later. And we don't control the behaviour between the interval $[T_4^-, T_4^+]$, since $T_4^-$ is not too far from $T_4^+$, we still can keep the good property at $T_4^+$ even in the worst case. Finally, after $T_4^+$, the good behaviour continues. (Also at $T_4^+$, we have the softmax ratio is basically concentrated at 1.)

Let $T_4^- \geq T_3$ be the last time the such that for all $t \in [T_3, T_4^-]$, $s \in [m_k]$ and $i \in [n]$, following conditions hold,

$$\left|\sum_{i=1}^n s_{i,21}^{(t)} s_{i,22}^{(t)}\langle \mathbf{w}_{Q,s}^{(t)}, y_i\boldsymbol{\xi}_i\rangle\right| \geq \frac{1}{2}n\left|\langle \mathbf{w}_{Q,s}^{(t)}, \boldsymbol{\mu}\rangle\right|, \tag{29}$$

$$s_{i,21}^{(t)} s_{i,22}^{(t)}\left|\langle \mathbf{w}_{Q,s}^{(t)}, y_i\boldsymbol{\xi}_i\rangle\right| \geq 2s_{i,11}^{(t)} s_{i,12}^{(t)}\left|\langle \mathbf{w}_{Q,s}^{(t)}, \boldsymbol{\mu}\rangle\right|. \tag{30}$$

Note that at $T_3$ we have

$$\sum_{i\in[n]} s_{i,21}^{(t)} s_{i,22}^{(t)}\langle \mathbf{w}_{Q,s}^{(t)}, y_i\boldsymbol{\xi}_i\rangle - \frac{1}{2}n\langle \mathbf{w}_{Q,s}^{(t)}, \boldsymbol{\mu}\rangle$$

$$= \frac{1}{4}\sum_{i\in[n]}\langle \mathbf{w}_{Q,s}^{(t)}, y_i\boldsymbol{\xi}_i\rangle - \frac{1}{2}n\langle \mathbf{w}_{Q,s}^{(t)}, \boldsymbol{\mu}\rangle$$

$$\geq \frac{1}{4}\sum_{i\in[n]}\langle \mathbf{w}_{Q,s}^{(T_2)}, y_i\boldsymbol{\xi}_i\rangle - \frac{1}{2}n\langle \mathbf{w}_{Q,s}^{(T_2)}, \boldsymbol{\mu}\rangle - (t - T_2)\eta(\frac{\sigma_p s}{4\sqrt{2}} - \frac{n\|\boldsymbol{\mu}\|}{2}) > 0,$$

and

$$s_{i,21}^{(t)} s_{i,22}^{(t)}\langle \mathbf{w}_{Q,s}^{(t)}, y_i\boldsymbol{\xi}_i\rangle - 2s_{i,11}^{(t)} s_{i,12}^{(t)}\langle \mathbf{w}_{Q,s}^{(t)}, \boldsymbol{\mu}\rangle$$

$$= \frac{1}{4}(1 + o(1))\mathbf{w}_{Q,s}^{(t)}, y_i\boldsymbol{\xi}_i\rangle - \frac{1}{2}(1 + o(1))\langle \mathbf{w}_{Q,s}^{(t)}, \boldsymbol{\mu}\rangle$$

$$\geq \frac{1}{4}(1 + o(1))\langle \mathbf{w}_{Q,s}^{(T_2)}, y_i\boldsymbol{\xi}_i\rangle - \frac{1}{2}(1 + o(1))\langle \mathbf{w}_{Q,s}^{(T_2)}, \boldsymbol{\mu}\rangle - (t - T_2)\eta(\frac{\sigma_p s}{4\sqrt{2}} - \frac{\|\boldsymbol{\mu}\|}{2}) > 0,$$

which implies $T_4^-$ is well-defined.

We use induction on $t$ to simultaneously prove the following properties for all $t = T_3, \ldots, T_4^-$:

- $\mathcal{A}(t)$, the linear-with-$t$ estimation: for all $i \in [n]$ and $s \in [m_k]$

$$\left| \langle \mathbf{w}_{Q,s}^{(t)}, y_i \boldsymbol{\xi}_i \rangle \right| = t\eta \sqrt{\frac{2}{\pi}} \sigma_p s (1 \pm \tilde{O}(s^{-1/2})),$$

$$\left| \langle \mathbf{w}_{K,s}^{(t)}, y_i \boldsymbol{\xi}_i \rangle \right| = t\eta \sqrt{\frac{2}{\pi}} \sigma_p s (1 \pm \tilde{O}(s^{-1/2})).$$

- $\mathcal{D}(t)$, query noise is increasing at $t$: for all $i \in [n]$ and $s \in [m_k]$

$$\langle \mathbf{w}_{Q,s}^{(t+1)}, y_i \boldsymbol{\xi}_i \rangle = \langle \mathbf{w}_{Q,s}^{(t)}, y_i \boldsymbol{\xi}_i \rangle + \mathrm{sgn}(\langle \mathbf{w}_{Q,s}^{(t)}, y_i \boldsymbol{\xi}_i \rangle) \eta \|\boldsymbol{\xi}_i\|_1 .$$

- $\mathcal{E}(t)$, key noise is increasing at $t$: for all $i \in [n]$ and $s \in [m_k]$

$$\langle \mathbf{w}_{K,s}^{(t+1)}, y_i \boldsymbol{\xi}_i \rangle = \langle \mathbf{w}_{K,s}^{(t)}, y_i \boldsymbol{\xi}_i \rangle + \mathrm{sgn}(\langle \mathbf{w}_{K,s}^{(t)}, y_i \boldsymbol{\xi}_i \rangle) \eta \|\boldsymbol{\xi}_i\|_1 .$$

- $\mathcal{F}(t)$, query/key noise have the same sign at $t$: for all $i \in [n]$ and $s \in [m_k]$

$$\langle \mathbf{w}_{Q,s}^{(t)}, y_i \boldsymbol{\xi}_i \rangle =_{\mathrm{sgn}} \langle \mathbf{w}_{K,s}^{(t)}, y_i \boldsymbol{\xi}_i \rangle.$$

**Claim E.16.** $\mathcal{F}(t) \implies \mathcal{E}(t)$.

**Claim E.17.** $\mathcal{D}(T_3), \mathcal{E}(T_3), \ldots, \mathcal{E}(t-1), \mathcal{F}(T_3), \ldots, \mathcal{F}(t-1) \implies \mathcal{D}(t)$.

**Claim E.18.** $\mathcal{A}(t), \mathcal{D}(t), \mathcal{E}(t) \implies \mathcal{A}(t+1)$.

**Claim E.19.** $\mathcal{F}(t), \mathcal{D}(t), \mathcal{E}(t) \implies \mathcal{F}(t+1)$.

Note that we have $\mathcal{A}(T_3), \mathcal{D}(T_3), \mathcal{F}(T_3)$ hold. $\mathcal{D}(T_3)$ and $\mathcal{F}(T_3)$ are already shown in the stage III.

*Proof of $\mathcal{A}(T_3)$.* At $t = T_3$, we have

$$\langle \mathbf{w}_{Q,s}^{(t)}, y_i \boldsymbol{\xi}_i \rangle = (t - T_2)\eta \|\boldsymbol{\xi}_i\|_1 + \langle \mathbf{w}_{Q,s}^{(T_2)}, y_i \boldsymbol{\xi}_i \rangle = t\eta \|\boldsymbol{\xi}_i\|_1 (1 \pm O(T_2 t^{-1}))$$

$$= t\eta \sqrt{\frac{2}{\pi}} \sigma_p s (1 \pm \tilde{O}(s^{-1/2}))(1 \pm O(T_2 t^{-1})) = t\eta \sqrt{\frac{2}{\pi}} \sigma_p s (1 \pm \tilde{O}(s^{-1/2})),$$

where the first step follows from the monotonicity of $\langle \mathbf{w}_{Q,s}^{(t)}, y_i \boldsymbol{\xi}_i \rangle$ for $[T_2, T_3]$, the second step follows from Eq. (21), the estimate of the magnitude of $\langle \mathbf{w}_{Q,s}^{(T_2)}, y_i \boldsymbol{\xi}_i \rangle$, the third step follows from Lemma C.4, the concentration of $\|\boldsymbol{\xi}_i\|_1$, the last step follows from $T_2/T_3 = \tilde{O}(s^{-1/2})$. The analysis is similar for $\langle \mathbf{w}_{K,s}^{(t)}, y_i \boldsymbol{\xi}_i \rangle$. $\qquad\square$

*Proof of Claim E.16.*

$$\langle \mathbf{w}_{K,s}^{(t+1)} - \mathbf{w}_{K,s}^{(t)}, y_i \boldsymbol{\xi}_i \rangle$$
$$=_{\mathrm{sgn}} s_{i,11}^{(t)} s_{i,12}^{(t)} \langle \mathbf{w}_{Q,s}^{(t)}, \boldsymbol{\mu} \rangle + s_{i,21}^{(t)} s_{i,22}^{(t)} \langle \mathbf{w}_{Q,s}^{(t)}, y_i \boldsymbol{\xi}_i \rangle$$
$$=_{\mathrm{sgn}} \langle \mathbf{w}_{Q,s}^{(t)}, y_i \boldsymbol{\xi}_i \rangle,$$

where the first step is by Eq. (12), the second step is by the definition of $T_4^-$, i.e., Eq. (30). $\qquad\square$

*Proof of Claim E.17.*

$$\langle \mathbf{w}_{Q,s}^{(t+1)} - \mathbf{w}_{Q,s}^{(t)}, y_i \boldsymbol{\xi}_i \rangle$$
$$=_{\mathrm{sgn}} - \langle \mathbf{w}_{K,s}^{(t)}, \boldsymbol{\mu} \rangle + \langle \mathbf{w}_{K,s}^{(t)}, y_i \boldsymbol{\xi}_i \rangle$$
$$> - \langle \mathbf{w}_{K,s}^{(T_3)}, \boldsymbol{\mu} \rangle + \langle \mathbf{w}_{K,s}^{(T_3)}, y_i \boldsymbol{\xi}_i \rangle + (t - T_3)\eta (\|\boldsymbol{\xi}_i\|_1 - \|\boldsymbol{\mu}\|) > 0,$$

where the first step is by Eq. (10), the second step is due to $\mathcal{E}(T_3), \ldots, \mathcal{E}(t-1), \mathcal{F}(T_3), \ldots, \mathcal{F}(t-1)$, the third step is by $\mathcal{D}(T_3)$. $\qquad\square$

*Proof of Claim E.18.*

$$\left|\langle \mathbf{w}_{Q,s}^{(t+1)}, y_i\boldsymbol{\xi}_i\rangle\right| = \left|\langle \mathbf{w}_{Q,s}^{(t)}, y_i\boldsymbol{\xi}_i\rangle\right| + \eta\,\|\boldsymbol{\xi}_i\|_1$$

$$= t\eta\sqrt{\frac{2}{\pi}}\sigma_p s(1 \pm \tilde{O}(s^{-1/2})) + \eta\sqrt{\frac{2}{\pi}}\sigma_p s(1 \pm \tilde{O}(s^{-1/2}))$$

$$= (t+1)\eta\sqrt{\frac{2}{\pi}}\sigma_p s(1 \pm \tilde{O}(s^{-1/2})),$$

where the first step is by $\mathcal{D}(t)$, the second step is by $\mathcal{A}(t)$ and Lemma C.4. The proof is same for $\langle \mathbf{w}_{K,s}^{(t)}, y_i\boldsymbol{\xi}_i\rangle$. $\qquad\square$

*Proof of Claim E.19.* Suppose $\langle \mathbf{w}_{Q,s}^{(t)}, y_i\boldsymbol{\xi}_i\rangle > 0$, then by $\mathcal{D}(t)$ we have

$$\langle \mathbf{w}_{Q,s}^{(t+1)}, \boldsymbol{\mu}\rangle = \langle \mathbf{w}_{Q,s}^{(t)}, \boldsymbol{\mu}\rangle + \operatorname{sgn}(\langle \mathbf{w}_{Q,s}^{(t)}, \boldsymbol{\mu}\rangle)\eta\,\|\boldsymbol{\mu}\| > 0.$$

By $\mathcal{F}(t)$, we have $\langle \mathbf{w}_{K,s}^{(t)}, y_i\boldsymbol{\xi}_i\rangle > 0$, then by $\mathcal{E}(t)$ we have

$$\langle \mathbf{w}_{K,s}^{(t+1)}, \boldsymbol{\mu}\rangle = \langle \mathbf{w}_{K,s}^{(t)}, \boldsymbol{\mu}\rangle + \operatorname{sgn}(\langle \mathbf{w}_{K,s}^{(t)}, \boldsymbol{\mu}\rangle)\eta\,\|\boldsymbol{\mu}\| > 0,$$

which gives $\mathcal{F}(t+1)$. $\qquad\square$

Now we study the ratio of softmax outputs across different samples. The following lemma is crucial to prove the lower bound of $T_4^-$.

**Lemma E.20.** *Let*

$$t^* := \min\left\{T_4^-, T_4\right\}. \tag{31}$$

*Then, for all $T_3 \le t \le t^*$ and all $i, k \in [n]$, we have*

$$\frac{s_{i,21}^{(t)}}{s_{k,21}^{(t)}} = 1 \pm o(1), \ \forall i, k \in [n].$$

*Proof of Lemma E.20.* By $\mathcal{A}(t)$ and $\mathcal{F}(t)$ we have for $t \in [T_3, T_4^-]$ and all $i \in [n]$

$$\sum_{s\in[m_k]} \langle \mathbf{w}_{Q,s}^{(t)}, \boldsymbol{\xi}_i\rangle\langle \mathbf{w}_{K,s}^{(t)}, \boldsymbol{\xi}_i\rangle = \frac{2}{\pi}m_k t^2\eta^2\sigma_p^2 s^2(1 \pm \tilde{O}(s^{-1/2})). \tag{32}$$

Then, for all $T_3 \le t \le t^*$ and $i \in [n]$

$$s_{i,21}^{(t)} = \frac{\exp\left(\sum_{s\in[m_k]}\langle \mathbf{w}_{Q,s}^{(t)}, \boldsymbol{\xi}_i\rangle\langle \mathbf{w}_{K,s}^{(t)}, y_i\boldsymbol{\mu}\rangle\right)}{\exp\left(\sum_{s\in[m_k]}\langle \mathbf{w}_{Q,s}^{(t)}, \boldsymbol{\xi}_i\rangle\langle \mathbf{w}_{K,s}^{(t)}, y_i\boldsymbol{\mu}\rangle\right) + \exp\left(\sum_{s\in[m_k]}\langle \mathbf{w}_{Q,s}^{(t)}, \boldsymbol{\xi}_i\rangle\langle \mathbf{w}_{K,s}^{(t)}, \boldsymbol{\xi}_i\rangle\right)}$$

$$= \frac{1 \pm o(1)}{1 \pm o(1) + \exp\left(\frac{2}{\pi}m_k t^2\eta^2\sigma_p^2 s^2(1 \pm \tilde{O}(s^{-1/2}))\right)}$$

$$= \frac{1 \pm o(1)}{1 \pm o(1) + \exp(\frac{2}{\pi}m_k t^2\eta^2\sigma_p^2 s^2)\exp(\pm\tilde{O}(\frac{2}{\pi}m_k t^2\eta^2\sigma_p^2 s^{3/2}))}$$

$$= \frac{1 \pm o(1)}{1 \pm o(1) + \exp(\frac{2}{\pi}m_k t^2\eta^2\sigma_p^2 s^2)(1 \pm o(1))}$$

$$= \frac{1 \pm o(1)}{1 + \exp(\frac{2}{\pi}m_k t^2\eta^2\sigma_p^2 s^2)}, \tag{33}$$

where the first step is by definition, the second step is by Eq. (28), (27), and Eq. (32), the fourth step is by $m_k T_4^2\eta^2\sigma_p^2 s^2 = o(1)$. Then we have for all $T_3 \le t \le t^*$ and all $i, k \in [n]$

$$\frac{s_{i,21}^{(t)}}{s_{k,21}^{(t)}} = (1 \pm o(1))\frac{(1 \pm o(1))(1 + \exp(\frac{2}{\pi}m_k t^2\eta^2\sigma_p^2 s^2))}{(1 \pm o(1))(1 + \exp(\frac{2}{\pi}m_k t^2\eta^2\sigma_p^2 s^2))} = 1 \pm o(1),$$

where the first step is by Eq. (33). $\qquad\square$

The following lemma studies the lower bound for $T_4^-$.

**Lemma E.21.** *We have*

$$T_4^- \geq \tilde{T}_4^- := \sqrt{\frac{0.99\pi}{2}} \sqrt{\log\left(\frac{\sigma_p s}{3\sqrt{2}n\|\boldsymbol{\mu}\|}\right)} \eta^{-1} m_k^{-1/2} \sigma_p^{-1} s^{-1}. \tag{34}$$

*Proof.* Suppose $T_4^- < \tilde{T}_4^-$, thus we have $T_4^- + 1 < \tilde{T}_4^- + 1 < T_4$. By Claim E.18 and E.19, we have $\mathcal{A}(T_4^- + 1)$ and $\mathcal{F}(T_4^- + 1)$ hold, which implies Eq. (32) holds at $t = T_4^- + 1$. Then, for all $i \in [n]$, we have

$$
\begin{aligned}
s_{i,21}^{(T_4^- +1)} &= \frac{1 \pm o(1)}{1 \pm o(1) + \exp\left(\frac{2}{\pi}m_k(T_4^- + 1)^2\eta^2\sigma_p^2 s^2(1 \pm \tilde{O}(s^{-1/2}))\right)} \\
&= \frac{1 \pm o(1)}{1 + \exp(\frac{2}{\pi}m_k(T_4^- + 1)^2\eta^2\sigma_p^2 s^2)} \\
&\geq \frac{1 \pm o(1)}{1 + \exp(\frac{2}{\pi}m_k(\tilde{T}_4^- + 1)^2\eta^2\sigma_p^2 s^2)} \\
&\geq \frac{1 \pm o(1)}{1.01\exp(\frac{2}{\pi}m_k(\tilde{T}_4^- + 1)^2\eta^2\sigma_p^2 s^2)} \\
&\geq \frac{1 \pm o(1)}{1.01}\left(\frac{3\sqrt{2}n\|\boldsymbol{\mu}\|}{\sigma_p s}\right)^{0.99+o(1)} \\
&\geq \frac{(1 \pm o(1))3\sqrt{2}n\|\boldsymbol{\mu}\|}{1.01\sigma_p s},
\end{aligned}
\tag{35}
$$
$$\tag{36}$$

where the first step is by Eq. (28), (27), and Eq. (32), the second step is by $T_4^- + 1 < T_4$, the fifth step is by $\eta = o(m_k^{-1/2}\sigma_p^{-1}s^{-1})$, the last step is by $n\|\boldsymbol{\mu}\|\sigma_p^{-1}s^{-1} = o(1)$, Then we estimate the magnitude of $\sum_{i=1}^n\langle\mathbf{w}_{Q,s}^{(t)}, y_i\boldsymbol{\xi}_i\rangle$. WLOG, given neuron $s$, suppose $\sum_{i=1}^n\langle\mathbf{w}_{Q,s}^{(T_3)}, y_i\boldsymbol{\xi}_i\rangle > 0$. For $T_3 < t \leq T_4^- + 1$

$$
\begin{aligned}
\sum_{i=1}^n\langle\mathbf{w}_{Q,s}^{(t)}, y_i\boldsymbol{\xi}_i\rangle &\geq \frac{1}{\sqrt{2}}(t - T_2)\eta\sigma_p s + \sum_{i=1}^n\langle\mathbf{w}_{Q,s}^{(T_2)}, y_i\boldsymbol{\xi}_i\rangle \geq \frac{1}{\sqrt{2}}(t - T_2)\eta\sigma_p s + 2n\beta_{\boldsymbol{\mu}} \\
&\geq \frac{1}{\sqrt{2}}t\eta\sigma_p s(1 \pm O(T_2 t^{-1})) = \frac{1}{\sqrt{2}}t\eta\sigma_p s(1 \pm o(1)),
\end{aligned}
\tag{37}
$$

where the first step follows from $\mathcal{D}(t)$ and Eq. (22), the second step follows from the lower bound of $\sum_{i\in[n]}\langle\mathbf{w}_{Q,s}^{(T_2)}, y_i\boldsymbol{\xi}_i\rangle$ Eq. (23), the last step follows from $T_2T_3^{-1} = o(1)$. The analysis is similar for $\langle\mathbf{w}_{K,s}^{(t)}, y_i\boldsymbol{\xi}_i\rangle$. Now we can show that

$$\sum_{i=1}^n s_{i,21}^{(T_4^- +1)}s_{i,22}^{(T_4^- +1)}\langle\mathbf{w}_{Q,s}^{(T_4^- +1)}, y_i\boldsymbol{\xi}_i\rangle$$

$$= \frac{1 \pm o(1)}{1 + \exp(\frac{2}{\pi}m_k(T_4^- + 1)^2\eta^2\sigma_p^2 s^2)} \cdot \left(1 - \frac{1 \pm o(1)}{1 + \exp(\frac{2}{\pi}m_k(T_4^- + 1)^2\eta^2\sigma_p^2 s^2)}\right) \cdot \sum_{i=1}^n\langle\mathbf{w}_{Q,s}^{(T_4^- +1)}, y_i\boldsymbol{\xi}_i\rangle$$

$$\geq \frac{(1 \pm o(1))3\sqrt{2}n\|\boldsymbol{\mu}\|}{1.01\sigma_p s} \cdot 0.5 \cdot \frac{1}{\sqrt{2}}(T_4^- + 1)\eta\sigma_p s(1 \pm o(1))$$

$$\geq 1.4n(T_4^- + 1)\eta\|\boldsymbol{\mu}\|$$

$$\geq \frac{1}{2}n\langle\mathbf{w}_{Q,s}^{(T_4^- +1)}, \boldsymbol{\mu}\rangle,$$

where the first step is by Eq. (35), the second step is by Eq. (36), (37) and $T_4^- \geq T_3$, the last step is by the upper bound in Eq. (24). Similarly, we can show that for all $s \in [m_k]$ and $i \in [n]$

$$s_{i,21}^{(T_4^- +1)}s_{i,22}^{(T_4^- +1)}\left|\langle\mathbf{w}_{Q,s}^{(T_4^- +1)}, y_i\boldsymbol{\xi}_i\rangle\right| \geq 2s_{i,11}^{(T_4^- +1)}s_{i,12}^{(T_4^- +1)}\left|\langle\mathbf{w}_{Q,s}^{(T_4^- +1)}, \boldsymbol{\mu}\rangle\right|,$$

which contradicts with the definition of $T_4^-$. Therefore, we have $T_4^- \geq \tilde{T}_4^-$. $\qquad\square$

**Remark E.22.** *Note the constant 0.99 in the definition of $\tilde{T}_4^-$ is crucial and its impact is on the exponent. When this constant is greater than 1, then the bound does not hold. Furthermore, we have $s_{i,21}^{(t)} \leq 0.1$ for $t \geq T_4^-$.*

Next, the following lemmas study the behaviour of query/key signal.

**Lemma E.23.** *The query signal are monotonic for $t \in [T_3, t^*]$. Formally, for all $t \in [T_3, t^*]$ and $s \in [m_k]$,*

$$\langle \mathbf{w}_{Q,s}^{(t+1)}, \boldsymbol{\mu} \rangle = \langle \mathbf{w}_{Q,s}^{(t)}, \boldsymbol{\mu} \rangle + \mathrm{sgn}(\langle \mathbf{w}_{Q,s}^{(t)}, \boldsymbol{\mu} \rangle)\eta \|\boldsymbol{\mu}\|,$$

*where $t^*$ is defined in Eq. (31).*

**Lemma E.24.** *The key signal are monotonic for $t \in [T_3, t^*]$. Formally, for all $t \in [T_3, t^*]$ and $s \in [m_k]$,*

$$\langle \mathbf{w}_{K,s}^{(t+1)}, \boldsymbol{\mu} \rangle = \langle \mathbf{w}_{K,s}^{(t)}, \boldsymbol{\mu} \rangle + \mathrm{sgn}(\langle \mathbf{w}_{K,s}^{(t)}, \boldsymbol{\mu} \rangle)\eta \|\boldsymbol{\mu}\|,$$

*where $t^*$ is defined in Eq. (31).*

*Proof of Lemma E.23.* WLOG, given neuron $s$, suppose $\langle \mathbf{w}_{Q,s}^{(t)}, \boldsymbol{\mu} \rangle > 0$ and $\langle \mathbf{w}_{K,s}^{(t)}, \boldsymbol{\mu} \rangle < 0$. For query signal, similar to previous analysis, we have

$$\langle \mathbf{w}_{Q,s}^{(t+1)} - \mathbf{w}_{Q,s}^{(t)}, \boldsymbol{\mu} \rangle$$

$$=_{\mathrm{sgn}} \sum_{i \in [n]} (-\ell_i'^{(t)} s_{i,11}^{(t)} s_{i,12}^{(t)}) \cdot \left( -y_i \langle \mathbf{v}^{(t)}, \boldsymbol{\xi}_i \rangle \langle \mathbf{w}_{K,s}^{(t)}, \boldsymbol{\mu} \rangle + \langle \mathbf{v}^{(t)}, \boldsymbol{\xi}_i \rangle \langle \mathbf{w}_{K,s}^{(t)}, \boldsymbol{\xi}_i \rangle \right)$$

$$=_{\mathrm{sgn}} \sum_{i \in [n]} \frac{\sqrt{2}}{8\sqrt{\pi}} t\eta\sigma_p s(1 \pm o(1)) \cdot (\langle \mathbf{w}_{K,s}^{(t)}, y_i\boldsymbol{\xi}_i \rangle - \langle \mathbf{w}_{K,s}^{(t)}, \boldsymbol{\mu} \rangle)$$

$$=_{\mathrm{sgn}} \sum_{i \in [n]} \langle \mathbf{w}_{K,s}^{(t)}, y_i\boldsymbol{\xi}_i \rangle - n\langle \mathbf{w}_{K,s}^{(t)}, \boldsymbol{\mu} \rangle$$

$$\geq \sum_{i \in [n]} \langle \mathbf{w}_{K,s}^{(T_2)}, y_i\boldsymbol{\xi}_i \rangle - n\langle \mathbf{w}_{K,s}^{(T_2)}, \boldsymbol{\mu} \rangle + (t - T_2)(\eta\sigma_p s/\sqrt{2} - \eta n \|\boldsymbol{\mu}\|) > 0,$$

where the first step is by Eq. (11), the second step is by the concentration of $\ell_i'$ and $s_{i,11}^{(t)}$ for $t \leq T_4$, i.e., Lemma E.14 and E.15, the fourth step is by $\mathcal{E}(t)$ and the lower bound Eq. (22), and the last step is by lower bound Eq. (23). $\qquad\square$

*Proof of Lemma E.24.* For key signal, we have

$$\langle \mathbf{w}_{K,s}^{(t+1)} - \mathbf{w}_{K,s}^{(t)}, \boldsymbol{\mu} \rangle$$

$$=_{\mathrm{sgn}} \sum_{i \in [n]} (-l_i'^{(t)}) \cdot \left( -y_i s_{i,11}^{(t)} s_{i,12}^{(t)} \langle \mathbf{v}^{(t)}, \boldsymbol{\xi}_i \rangle \langle \mathbf{w}_{Q,s}^{(t)}, \boldsymbol{\mu} \rangle - s_{i,21}^{(t)} s_{i,22}^{(t)} \langle \mathbf{v}^{(t)}, \boldsymbol{\xi}_i \rangle \langle \mathbf{w}_{Q,s}^{(t)}, \boldsymbol{\xi}_i \rangle \right)$$

$$=_{\mathrm{sgn}} \sum_{i \in [n]} \frac{\sqrt{2}}{2\sqrt{\pi}} t\eta\sigma_p s(1 \pm o(1)) \cdot (-s_{i,21}^{(t)} s_{i,22}^{(t)} \langle \mathbf{w}_{Q,s}^{(t)}, y_i\boldsymbol{\xi}_i \rangle - \frac{1}{4}(1 \pm o(1)) \langle \mathbf{w}_{Q,s}^{(t)}, \boldsymbol{\mu} \rangle)$$

$$=_{\mathrm{sgn}} -\sum_{i \in [n]} s_{i,21}^{(t)} s_{i,22}^{(t)} \langle \mathbf{w}_{Q,s}^{(t)}, y_i\boldsymbol{\xi}_i \rangle - \frac{1}{4}(1 \pm o(1))n\langle \mathbf{w}_{Q,s}^{(t)}, \boldsymbol{\mu} \rangle)$$

$$=_{\mathrm{sgn}} -\sum_{i \in [n]} s_{i,21}^{(t)} s_{i,22}^{(t)} \langle \mathbf{w}_{Q,s}^{(t)}, y_i\boldsymbol{\xi}_i \rangle$$

$$=_{\mathrm{sgn}} -s_{21}^{(t)} s_{22}^{(t)}(1 \pm o(1)) \sum_{i \in [n]} \langle \mathbf{w}_{Q,s}^{(t)}, y_i\boldsymbol{\xi}_i \rangle < 0,$$

where the first step is by Eq. (11), the second step is by the concentration of $\ell_i'$ and $s_{i,11}^{(t)}$ for $t \leq T_4$, i.e., Lemma E.14 and E.15, the fourth step is by the definition of $T_4^-$, i.e., Eq. (29), the fifth step is by the $\mathcal{B}(t)$, the last step is due to the upper bound Eq. (23) and $\mathcal{D}(T_3), \ldots \mathcal{D}(t-1)$. $\qquad\square$

At the end of this section, we give an upper bound for $T_4^-$.

**Lemma E.25.** *We have*

$$T_4^- \le \hat{T}_4^- := \sqrt{\frac{1.01\pi}{2}} \sqrt{\log\left(\frac{\sigma_p s}{\|\boldsymbol{\mu}\|}\right)} \eta^{-1} m_k^{-1/2} \sigma_p^{-1} s^{-1}. \tag{38}$$

*Proof.* Suppose $T_4^- > \hat{T}_4^-$. First, we estimate the magnitude of $\sum_{i=1}^n \langle \mathbf{w}_{Q,s}^{(t)}, y_i \boldsymbol{\xi}_i \rangle$ and $\langle \mathbf{w}_{Q,s}^{(t)}, \boldsymbol{\mu} \rangle$ for $T_3 \le t \le T_4^-$. WLOG, given neuron $s$, suppose $\sum_{i=1}^n \langle \mathbf{w}_{Q,s}^{(T_2)}, y_i \boldsymbol{\xi}_i \rangle > 0$. For all $s \in [m_k]$ we have

$$\langle \mathbf{w}_{Q,s}^{(t)}, \boldsymbol{\mu} \rangle \ge \langle \mathbf{w}_{Q,s}^{(T_3)}, \boldsymbol{\mu} \rangle + (t - T_3)\eta\|\boldsymbol{\mu}\| \ge t\eta\|\boldsymbol{\mu}\|/2, \tag{39}$$

where the first step is by Lemma E.23 and $\hat{T}_4^- < t^*$, the second step is by the lower bound in Eq. (24) at $t = T_3$. For all $s \in [m_k]$ we also have

$$\sum_{i=1}^n \langle \mathbf{w}_{Q,s}^{(t)}, y_i \boldsymbol{\xi}_i \rangle \le \sum_{i=1}^n \left| \langle \mathbf{w}_{Q,s}^{(t)}, y_i \boldsymbol{\xi}_i \rangle \right| \le tn\eta \sqrt{\frac{2}{\pi}} \sigma_p s (1 \pm \tilde{O}(s^{-1/2})),$$

where the second step is by $\mathcal{A}(t)$. By the definition of $T_4^-$, at $t = \hat{T}_4^-$ we have

$$\begin{aligned}
tn\eta\|\boldsymbol{\mu}\|/4 &\le \frac{1}{2} n \langle \mathbf{w}_{Q,s}^{(t)}, \boldsymbol{\mu} \rangle \\
&\le \left| \sum_{i=1}^n s_{i,21}^{(t)} s_{i,22}^{(t)} \langle \mathbf{w}_{Q,s}^{(t)}, y_i \boldsymbol{\xi}_i \rangle \right| \\
&\le \frac{1 \pm o(1)}{1 + \exp(\frac{2}{\pi} m_k t^2 \eta^2 \sigma_p^2 s^2)} \cdot tn\eta \sqrt{\frac{2}{\pi}} \sigma_p s (1 \pm \tilde{O}(s^{-1/2})) \\
&\le \frac{1 \pm o(1)}{\exp(\frac{2}{\pi} m_k t^2 \eta^2 \sigma_p^2 s^2)} \cdot tn\eta \sqrt{\frac{2}{\pi}} \sigma_p s (1 \pm \tilde{O}(s^{-1/2})),
\end{aligned}$$

which implies that

$$\hat{T}_4^- \le \sqrt{\frac{\pi}{2}} \sqrt{\log\left(\frac{4\sqrt{2}\sigma_p s}{\sqrt{\pi}\|\boldsymbol{\mu}\|}(1 + o(1))\right)} \eta^{-1} m_k^{-1/2} \sigma_p^{-1} s^{-1},$$

which gives a contradiction. $\qquad\square$

**Remark E.26.** *The upper bound for $T_4^-$ also proves that $t^* = T_4^-$, which gives that for all $t \in [T_3, T_4^-]$, $s \in [m_k]$, and $i, k \in [n]$*

$$\frac{s_{i,21}^{(t)}}{s_{k,21}^{(t)}} = 1 \pm o(1),$$

$$\langle \mathbf{w}_{Q,s}^{(t+1)}, \boldsymbol{\mu} \rangle = \langle \mathbf{w}_{Q,s}^{(t)}, \boldsymbol{\mu} \rangle + \text{sgn}(\langle \mathbf{w}_{Q,s}^{(t)}, \boldsymbol{\mu} \rangle)\eta\|\boldsymbol{\mu}\|,$$

$$\langle \mathbf{w}_{K,s}^{(t+1)}, \boldsymbol{\mu} \rangle = \langle \mathbf{w}_{K,s}^{(t)}, \boldsymbol{\mu} \rangle + \text{sgn}(\langle \mathbf{w}_{K,s}^{(t)}, \boldsymbol{\mu} \rangle)\eta\|\boldsymbol{\mu}\|.$$

*Moreover, for neuron $s$, suppose $\langle \mathbf{w}_{Q,s}^{(T_3)}, \boldsymbol{\mu} \rangle > 0$, then at $t = T_4^-$, for all $s \in [m_k]$, we have*

$$\langle \mathbf{w}_{Q,s}^{(t)}, \boldsymbol{\mu} \rangle = (t - T_3)\eta\|\boldsymbol{\mu}\| + \langle \mathbf{w}_{Q,s}^{(T_3)}, \boldsymbol{\mu} \rangle = t\eta\|\boldsymbol{\mu}\|(1 \pm o(1)), \tag{40}$$

$$\langle \mathbf{w}_{K,s}^{(t)}, \boldsymbol{\mu} \rangle = -(t - T_3)\eta\|\boldsymbol{\mu}\| + \langle \mathbf{w}_{K,s}^{(T_3)}, \boldsymbol{\mu} \rangle = -t\eta\|\boldsymbol{\mu}\|(1 \pm o(1)), \tag{41}$$

*where for query signal the first step is by Lemma E.23 and the second step is by the bound Eq. 24. The proof of statement for key signal is similar.*

### E.7.2 STAGE IV.B

This stage studies the dynamics in the transition interval $[T_4^-, T_4^+]$. For $T_4^- \leq t \leq T_4^+$, given neuron $s$, for those samples where query/key noise align with the query/key signal, query/key noise dynamics keep unchanged, which will be defined later. On the other hands, for those samples where query/key noise do not align with query/key signal, query noise dynamics keep unchanged. However, key noise start to decrease but are still greater than zero at $T_4^+$.

Let $T_4^+$ be the first time the following condition holds

$$s_{i,21}^{(t)} s_{i,22}^{(t)} \left| \langle \mathbf{w}_{Q,s}^{(t)}, y_i \boldsymbol{\xi}_i \rangle \right| \leq \frac{1}{2} s_{i,11}^{(t)} s_{i,12}^{(t)} \left| \langle \mathbf{w}_{Q,s}^{(t)}, \boldsymbol{\mu} \rangle \right|, \tag{42}$$

for all $s \in [m_k]$ and $i \in [n]$. Note that at $T_4^+$, we have

$$\sum_{i=1}^n s_{i,21}^{(t)} s_{i,22}^{(t)} \langle \mathbf{w}_{Q,s}^{(t)}, y_i \boldsymbol{\xi}_i \rangle \leq \sum_{i=1}^n s_{i,21}^{(t)} s_{i,22}^{(t)} \left| \langle \mathbf{w}_{Q,s}^{(t)}, y_i \boldsymbol{\xi}_i \rangle \right| \leq \frac{1}{2} \sum_{i=1}^n s_{i,11}^{(t)} s_{i,12}^{(t)} \left| \langle \mathbf{w}_{Q,s}^{(t)}, \boldsymbol{\mu} \rangle \right|$$

$$= \frac{1}{8}(1 + o(1)) n s_{i,11}^{(t)} s_{i,12}^{(t)} \left| \langle \mathbf{w}_{Q,s}^{(t)}, \boldsymbol{\mu} \rangle \right| < \frac{1}{2} n s_{i,11}^{(t)} s_{i,12}^{(t)} \left| \langle \mathbf{w}_{Q,s}^{(t)}, \boldsymbol{\mu} \rangle \right|,$$

which implies $T_4^+ \geq T_4^-$, $T_4^+$ is well-defined.

Let

$$E_i^{(T_3)} := \left\{ s \in [m_k] : \langle \mathbf{w}_{Q,s}^{(T_3)}, y_i \boldsymbol{\xi}_i \rangle =_{\text{sgn}} \langle \mathbf{w}_{Q,s}^{(T_3)}, \boldsymbol{\mu} \rangle \right\},$$

$$E_s^{(T_3)} := \left\{ i \in [n] : \langle \mathbf{w}_{Q,s}^{(T_3)}, y_i \boldsymbol{\xi}_i \rangle =_{\text{sgn}} \langle \mathbf{w}_{Q,s}^{(T_3)}, \boldsymbol{\mu} \rangle \right\}.$$

all neurons with $s \in [m_k]$ and $i \in E_s^{(T_3)}$ are called aligned neurons. The next lemma studies the concentration behaviour about $\left| E_i^{(T_3)} \right|$. This lemma is crucial to upper bound $s_{i,21}^{(t)}$ for $T_4^- \leq t \leq T_4^+$, and further upper bound $T_4^+$.

**Lemma E.27.** *Suppose that $\delta > 0$ Then with probability at least $1 - \delta$,*

$$\left| E_i^{(T_3)} \right| \geq m_k/2 - \tilde{O}(m_k n^{-1/2} + m_k^{1/2}) \geq m_k/2 - \tilde{O}(\sqrt{m_k}),$$

*for all $i \in [n]$.*

*Proof.* We can write $E_i^{(T_3)}$ as

$$E_i^{(T_3)} = \left\{ s \in [m_k] : \langle \mathbf{w}_{Q,s}^{(T_2)}, y_i \boldsymbol{\xi}_i \rangle =_{\text{sgn}} \sum_{i=1}^n \langle \mathbf{w}_{Q,s}^{(T_2)}, y_i \boldsymbol{\xi}_i \rangle \right\},$$

where this is by the sign of $\langle \mathbf{w}_{Q,s}^{(t)}, y_i \boldsymbol{\xi}_i \rangle$ and $\sum_{i=1}^n \langle \mathbf{w}_{Q,s}^{(t)}, y_i \boldsymbol{\xi}_i \rangle$ does not change for $[T_2, T_3]$ and the sign of $\langle \mathbf{w}_{Q,s}^{(T_3)}, \boldsymbol{\mu} \rangle$ is the same as that of $\sum_{i=1}^n \langle \mathbf{w}_{Q,s}^{(T_3)}, y_i \boldsymbol{\xi}_i \rangle$, i.e., Lemma E.9 and E.11. Then we have

$$p := \frac{1}{m_k} \mathbb{E}[\left| E_i^{(T_3)} \right|]$$

$$= \mathbb{P}[\langle \mathbf{w}_{Q,s}^{(T_2)}, y_i \boldsymbol{\xi}_i \rangle > 0, \sum_{i=1}^n \langle \mathbf{w}_{Q,s}^{(T_2)}, y_i \boldsymbol{\xi}_i \rangle > 0] + \mathbb{P}[\langle \mathbf{w}_{Q,s}^{(T_2)}, y_i \boldsymbol{\xi}_i \rangle < 0, \sum_{i=1}^n \langle \mathbf{w}_{Q,s}^{(T_2)}, y_i \boldsymbol{\xi}_i \rangle < 0]$$

$$= \mathbb{P}[X_{s,i} > 0, \sum_{i=1}^n X_{s,i} > 0] + \mathbb{P}[X_{s,i} < 0, \sum_{i=1}^n X_{s,i} < 0]$$

$$\geq \frac{1}{2} - O(n^{-1/2}),$$

where the third step by that the sign of $X_{s,i}$ determines the sign of $\langle \mathbf{w}_{Q,s}^{(T_2)}, y_i \boldsymbol{\xi}_i \rangle$, the last step is by Lemma E.6. Then, by Hoeffding's inequality, with probability at least $1 - \delta/n$

$$\left| \left| E_i^{(T_3)} \right| - p m_k \right| \leq \sqrt{\frac{m_k}{2} \log(2n/\delta)}.$$

Finally, by the condition $n = \Omega(m_k^2)$, we get the conclusion. $\qquad \square$

We do not directly characterize the dynamics for $[T_4^-, T_4^+]$. Instead, we characterize for $[T_4^-], (1 + \theta)T_4^-$, where $\theta$ is a large constant in $(0, 1)$, and show that the definition of $T_4^+$ are satisfied, which characterize he dynamics before $T_4^+$ and give an upper bound for $T_4^+$ simultaneously. Specifically, we use induction on $t$ to simultaneously prove the following properties for all $t = T_4^-, \ldots, (1 + \theta)T_4^-$:

- $\mathcal{A}(t)$, the monotonicity of query noise aligned with signal: for any $s \in [m_k]$ and $i \in E_s^{(T_3)}$

$$\langle \mathbf{w}_{Q,s}^{(t+1)}, y_i\boldsymbol{\xi}_i \rangle = \langle \mathbf{w}_{Q,s}^{(t)}, y_i\boldsymbol{\xi}_i \rangle + \text{sgn}(\langle \mathbf{w}_{Q,s}^{(t)}, y_i\boldsymbol{\xi}_i \rangle)\eta \, \|\boldsymbol{\xi}_i\|_1 \,.$$

- $\mathcal{B}(t)$, the monotonicity of key noise aligned with signal: for any $s \in [m_k]$ and $i \in E_s^{(T_3)}$

$$\langle \mathbf{w}_{K,s}^{(t+1)}, y_i\boldsymbol{\xi}_i \rangle = \langle \mathbf{w}_{K,s}^{(t)}, y_i\boldsymbol{\xi}_i \rangle + \text{sgn}(\langle \mathbf{w}_{K,s}^{(t)}, y_i\boldsymbol{\xi}_i \rangle)\eta \, \|\boldsymbol{\xi}_i\|_1 \,.$$

- $\mathcal{C}(t)$, the monotonicity of query noise unaligned with signal: for any $s \in [m_k]$ and $i \in [n]/E_s^{(T_3)}$

$$\langle \mathbf{w}_{Q,s}^{(t+1)}, y_i\boldsymbol{\xi}_i \rangle = \langle \mathbf{w}_{Q,s}^{(t)}, y_i\boldsymbol{\xi}_i \rangle + \text{sgn}(\langle \mathbf{w}_{Q,s}^{(t)}, y_i\boldsymbol{\xi}_i \rangle)\eta \, \|\boldsymbol{\xi}_i\|_1 \,.$$

- $\mathcal{D}(t)$, the sign does not change for unaligned key noise: for any $s \in [m_k]$ and $i \in [n]/E_s^{(T_3)}$

$$\langle \mathbf{w}_{K,s}^{(t)}, y_i\boldsymbol{\xi}_i \rangle =_{\text{sgn}} \langle \mathbf{w}_{K,s}^{(T_4^-)}, y_i\boldsymbol{\xi}_i \rangle =_{\text{sgn}} -\langle \mathbf{w}_{Q,s}^{(t)}, y_i\boldsymbol{\xi}_i \rangle.$$

- $\mathcal{E}(t)$, the monotonicity of query signal: for any $s \in [m_k]$

$$\langle \mathbf{w}_{Q,s}^{(t+1)}, \boldsymbol{\mu} \rangle = \langle \mathbf{w}_{Q,s}^{(t)}, \boldsymbol{\mu} \rangle + \text{sgn}(\langle \mathbf{w}_{Q,s}^{(t)}, \boldsymbol{\mu} \rangle)\eta \, \|\boldsymbol{\mu}\| \,.$$

- $\mathcal{F}(t)$, the sign does not change for key signal: for any $s \in [m_k]$

$$\langle \mathbf{w}_{K,s}^{(t)}, \boldsymbol{\mu} \rangle =_{\text{sgn}} \langle \mathbf{w}_{K,s}^{(T_4^-)}, \boldsymbol{\mu} \rangle =_{\text{sgn}} -\langle \mathbf{w}_{Q,s}^{(t)}, y_i\boldsymbol{\xi}_i \rangle.$$

- $\mathcal{G}(t)$, the linear-with-$t$ estimation: for all $s \in [m_k]$ and $i \in [n]$

$$\langle \mathbf{w}_{Q,s}^{(t)}, y_i\boldsymbol{\xi}_i \rangle = \text{sgn}(\langle \mathbf{w}_{Q,s}^{(t)}, \boldsymbol{\mu} \rangle)t\eta\sqrt{\frac{2}{\pi}}\sigma_p s(1 \pm \tilde{O}(s^{-1/2})), \forall i \in E_s^{(T_3)}$$

$$\langle \mathbf{w}_{K,s}^{(t)}, y_i\boldsymbol{\xi}_i \rangle = \text{sgn}(\langle \mathbf{w}_{Q,s}^{(t)}, \boldsymbol{\mu} \rangle)t\eta\sqrt{\frac{2}{\pi}}\sigma_p s(1 \pm \tilde{O}(s^{-1/2})), \forall i \in E_s^{(T_3)}$$

$$\langle \mathbf{w}_{Q,s}^{(t)}, y_i\boldsymbol{\xi}_i \rangle = -\text{sgn}(\langle \mathbf{w}_{Q,s}^{(t)}, \boldsymbol{\mu} \rangle)t\eta\sqrt{\frac{2}{\pi}}\sigma_p s(1 \pm \tilde{O}(s^{-1/2})), \forall i \in [n]/E_s^{(T_3)}$$

$$\langle \mathbf{w}_{Q,s}^{(t)}, \boldsymbol{\mu} \rangle = \text{sgn}(\langle \mathbf{w}_{Q,s}^{(t)}, \boldsymbol{\mu} \rangle)t\eta \, \|\boldsymbol{\mu}\| \, (1 \pm o(1)).$$

**Remark E.28.** *Before stating the results in this section, we first recall and summarize the results until $T_4^-$. WLOG, given neuron $s$, we suppose that $\sum_{i=1}^n \langle \mathbf{w}_{Q,s}^{(T_2)}, y_i\boldsymbol{\xi}_i \rangle > 0$ in this section below. Then we have $\langle \mathbf{w}_{Q,s}^{(T_3)}, \boldsymbol{\mu} \rangle > 0$ and $\langle \mathbf{w}_{K,s}^{(T_3)}, \boldsymbol{\mu} \rangle < 0$. Then at $t = T_4^-$ we have*

$$\langle \mathbf{w}_{Q,s}^{(t)}, y_i\boldsymbol{\xi}_i \rangle = t\eta\sqrt{\frac{2}{\pi}}\sigma_p s(1 \pm \tilde{O}(s^{-1/2})), \forall i \in E_s^{(T_3)}$$

$$\langle \mathbf{w}_{K,s}^{(t)}, y_i\boldsymbol{\xi}_i \rangle = t\eta\sqrt{\frac{2}{\pi}}\sigma_p s(1 \pm \tilde{O}(s^{-1/2})), \forall i \in E_s^{(T_3)}$$

$$\langle \mathbf{w}_{Q,s}^{(t)}, y_i\boldsymbol{\xi}_i \rangle = -t\eta\sqrt{\frac{2}{\pi}}\sigma_p s(1 \pm \tilde{O}(s^{-1/2})), \forall i \in [n]/E_s^{(T_3)}$$

$$\langle \mathbf{w}_{K,s}^{(t)}, y_i\boldsymbol{\xi}_i \rangle = -t\eta\sqrt{\frac{2}{\pi}}\sigma_p s(1 \pm \tilde{O}(s^{-1/2})), \forall i \in [n]/E_s^{(T_3)}$$

$$\langle \mathbf{w}_{Q,s}^{(t)}, \boldsymbol{\mu} \rangle = t\eta \, \|\boldsymbol{\mu}\| \, (1 \pm o(1)),$$

$$\langle \mathbf{w}_{K,s}^{(t)}, \boldsymbol{\mu} \rangle = -t\eta \, \|\boldsymbol{\mu}\| \, (1 \pm o(1)).$$

*This gives that $\mathcal{A}(T_4^-)$, $\mathcal{B}(T_4^-)$, $\mathcal{C}(T_4^-)$, $\mathcal{D}(T_4^-)$, $\mathcal{E}(T_4^-)$, $\mathcal{F}(T_4^-)$ hold.*

To prove these properties, we need some further properties after $T_4^-$.

**Lemma E.29** (Properties after $T_4^-$). *Let $\theta$ be a large constant in $(0,1)$. For all $t \geq T_4^-$, we have*

$$(2T_4^- - t)\eta\sqrt{\frac{2}{\pi}}\sigma_p s(1 - \tilde{O}(s^{-1/2})) \leq \operatorname{sgn}(\langle \mathbf{w}_{K,s}^{(T_4^-)}, y_i\boldsymbol{\xi}_i\rangle) \cdot \langle \mathbf{w}_{K,s}^{(t)}, y_i\boldsymbol{\xi}_i\rangle \leq t\eta\sqrt{\frac{2}{\pi}}\sigma_p s(1 + \tilde{O}(s^{-1/2})),$$

$$(2T_4^- - t)\eta\,\|\boldsymbol{\mu}\|\,(1 - o(1)) \leq \operatorname{sgn}(\langle \mathbf{w}_{K,s}^{(T_4^-)}, \boldsymbol{\mu}\rangle) \cdot \langle \mathbf{w}_{K,s}^{(t)}, \boldsymbol{\mu}\rangle \leq t\eta\,\|\boldsymbol{\mu}\|\,(1 + o(1)),$$

*and for all $T_4^- \leq t \leq (1 + \theta)T_4^-$*

$$\langle \mathbf{w}_{K,s}^{(t)}, y_i\boldsymbol{\xi}_i\rangle =_{\operatorname{sgn}} \langle \mathbf{w}_{K,s}^{(T_4^-)}, y_i\boldsymbol{\xi}_i\rangle,$$

$$\langle \mathbf{w}_{K,s}^{(t)}, \boldsymbol{\mu}\rangle =_{\operatorname{sgn}} \langle \mathbf{w}_{K,s}^{(T_4^-)}, \boldsymbol{\mu}\rangle.$$

*Particularly, For all $0 \leq \theta' \leq \theta$, $s \in [m_k]$ and $i \in [n]$, setting $t = (1 + \theta')T_4^-$, we have*

$$(1 - \theta')T_4^- \eta\sqrt{\frac{2}{\pi}}\sigma_p s(1 - \tilde{O}(s^{-1/2})) \leq \left|\langle \mathbf{w}_{K,s}^{((1+\theta')T_4^-)}, y_i\boldsymbol{\xi}_i\rangle\right| \leq (1 + \theta')T_4^- \eta\sqrt{\frac{2}{\pi}}\sigma_p s(1 + \tilde{O}(s^{-1/2})),$$

$$(1 - \theta')T_4^- \eta\,\|\boldsymbol{\mu}\|\,(1 - o(1)) \leq \left|\langle \mathbf{w}_{K,s}^{((1+\theta')T_4^-)}, \boldsymbol{\mu}\rangle\right| \leq (1 + \theta')T_4^- \eta\,\|\boldsymbol{\mu}\|\,(1 + o(1)).$$

*Proof of Lemma E.29.* Suppose $\langle \mathbf{w}_{K,s}^{(T_4^-)}, y_i\boldsymbol{\xi}\rangle > 0$, then we have for all $t \geq T_4^-$

$$\langle \mathbf{w}_{K,s}^{(t)}, y_i\boldsymbol{\xi}_i\rangle \leq \langle \mathbf{w}_{K,s}^{(T_4^-)}, y_i\boldsymbol{\xi}_i\rangle + (t - T_4^-)\eta\,\|\boldsymbol{\xi}_i\|_1 \leq t\eta\sqrt{\frac{2}{\pi}}\sigma_p s(1 \pm \tilde{O}(s^{-1/2})),$$

where the first step is by that the magnitude of single update of query/key noise is always $\eta\,\|\boldsymbol{\xi}_i\|_1$, the second step is by summary above and Lemma C.4. Similarly, we have

$$\langle \mathbf{w}_{K,s}^{(t)}, y_i\boldsymbol{\xi}_i\rangle \geq \langle \mathbf{w}_{K,s}^{(T_4^-)}, y_i\boldsymbol{\xi}_i\rangle - (t - T_4^-)\eta\,\|\boldsymbol{\xi}_i\|_1 \geq (2T_4^- - t)\eta\sqrt{\frac{2}{\pi}}\sigma_p s(1 \pm \tilde{O}(s^{-1/2})).$$

Set $t = (1 + \theta')T_4^-$ gives the conclusion. Also, the lower bound gives

$$\langle \mathbf{w}_{K,s}^{(t)}, y_i\boldsymbol{\xi}_i\rangle \geq (2T_4^- - t)\eta\sqrt{\frac{2}{\pi}}\sigma_p s(1 \pm \tilde{O}(s^{-1/2}))$$

$$\geq (1 - \theta)T_4^- \eta\sqrt{\frac{2}{\pi}}\sigma_p s(1 \pm \tilde{O}(s^{-1/2})) > 0,$$

where the last step is by $\theta$ is a constant. Similarly, we can prove the statement for $\langle \mathbf{w}_{K,s}^{(t)}, \boldsymbol{\mu}\rangle$. $\quad\square$

**Remark E.30.** *Note that Lemma E.29 proves the $\mathcal{D}(t)$ and $\mathcal{F}(t)$ for all $T_2^- \leq t \leq (1 + \theta)T_4^-$.*

**Claim E.31.** $\mathcal{G}(t), \mathcal{F}(t) \implies \mathcal{A}(t)$.

**Claim E.32.** $\mathcal{G}(t) \implies \mathcal{B}(t)$.

**Claim E.33.** $\mathcal{D}(t), \mathcal{F}(t) \implies \mathcal{C}(t)$.

**Claim E.34.** $\mathcal{B}(t) \implies \mathcal{E}(t)$.

**Claim E.35.** $\mathcal{G}(t), \mathcal{A}(t), \mathcal{B}(t), \mathcal{C}(t), \mathcal{E}(t) \implies \mathcal{G}(t+1)$.

*Proof of Claim E.31.* We have

$$\langle \mathbf{w}_{Q,s}^{(t+1)} - \mathbf{w}_{Q,s}^{(t)}, y_i\boldsymbol{\xi}_i\rangle =_{\operatorname{sgn}} -\langle \mathbf{w}_{K,s}^{(t)}, \boldsymbol{\mu}\rangle + \langle \mathbf{w}_{K,s}^{(t)}, y_i\boldsymbol{\xi}_i\rangle > \langle \mathbf{w}_{K,s}^{(t)}, y_i\boldsymbol{\xi}_i\rangle > 0,$$

where the first step is by Eq. (10), the second step is by $\mathcal{F}(t)$, the last step is by $\mathcal{G}(t)$. $\quad\square$

*Proof of Claim E.32.* We have

$$\langle \mathbf{w}_{K,s}^{(t+1)} - \mathbf{w}_{K,s}^{(t)}, y_i\boldsymbol{\xi}_i\rangle =_{\operatorname{sgn}} s_{i,11}^{(t)} s_{i,12}^{(t)} \langle \mathbf{w}_{Q,s}^{(t)}, \boldsymbol{\mu}\rangle + s_{i,21}^{(t)} s_{i,22}^{(t)} \langle \mathbf{w}_{Q,s}^{(t)}, y_i\boldsymbol{\xi}_i\rangle > 0,$$

where the first step is by Eq. (12), the second step is by $\mathcal{G}(t)$. $\quad\square$

*Proof of Claim E.33.* We have

$$\langle \mathbf{w}_{Q,s}^{(t+1)} - \mathbf{w}_{Q,s}^{(t)}, y_i\boldsymbol{\xi}_i \rangle =_{\text{sgn}} -\langle \mathbf{w}_{K,s}^{(t)}, \boldsymbol{\mu} \rangle + \langle \mathbf{w}_{K,s}^{(t)}, y_i\boldsymbol{\xi}_i \rangle > \langle \mathbf{w}_{K,s}^{(t)}, y_i\boldsymbol{\xi}_i \rangle > 0,$$

where the first step is by Eq. (10), the second step is by $\mathcal{F}(t)$, the last step is by $\mathcal{D}(t)$. □

*Proof of Claim E.34.* We have

$$
\begin{aligned}
&\langle \mathbf{w}_{Q,s}^{(t+1)} - \mathbf{w}_{Q,s}^{(t)}, \boldsymbol{\mu} \rangle \\
=_{\text{sgn}}& \sum_{i\in[n]} (-\ell_i^{\prime(t)} s_{i,11}^{(t)} s_{i,12}^{(t)}) \cdot \left( -y_i\langle \mathbf{v}^{(t)}, \boldsymbol{\xi}_i \rangle \langle \mathbf{w}_{K,s}^{(t)}, \boldsymbol{\mu} \rangle + \langle \mathbf{v}^{(t)}, \boldsymbol{\xi}_i \rangle \langle \mathbf{w}_{K,s}^{(t)}, \boldsymbol{\xi}_i \rangle \right) \\
=_{\text{sgn}}& \sum_{i\in[n]} \frac{\sqrt{2}}{8\sqrt{\pi}} t\eta\sigma_p s(1\pm o(1)) \cdot (\langle \mathbf{w}_{K,s}^{(t)}, y_i\boldsymbol{\xi}_i \rangle - \langle \mathbf{w}_{K,s}^{(t)}, \boldsymbol{\mu} \rangle) \\
=_{\text{sgn}}& \sum_{i\in[n]} \langle \mathbf{w}_{K,s}^{(t)}, y_i\boldsymbol{\xi}_i \rangle - n\langle \mathbf{w}_{K,s}^{(t)}, \boldsymbol{\mu} \rangle \\
=& -n\langle \mathbf{w}_{K,s}^{(t)}, \boldsymbol{\mu} \rangle + \sum_{i\in E_s^{(T3)}} \langle \mathbf{w}_{K,s}^{(t)}, y_i\boldsymbol{\xi}_i \rangle + \sum_{i\in[n]/E_s^{(T3)}} \langle \mathbf{w}_{K,s}^{(t)}, y_i\boldsymbol{\xi}_i \rangle \\
\geq& -n\langle \mathbf{w}_{K,s}^{(t)}, \boldsymbol{\mu} \rangle + \sum_{i\in E_s^{(T3)}} \langle \mathbf{w}_{K,s}^{(T_4^-)}, y_i\boldsymbol{\xi}_i \rangle + (t-T_4^-)\eta\|\boldsymbol{\xi}_i\|_1 + \sum_{i\in[n]/E_s^{(T3)}} \langle \mathbf{w}_{K,s}^{(T_4^-)}, y_i\boldsymbol{\xi}_i \rangle - (t-T_4^-)\eta\|\boldsymbol{\xi}_i\|_1 \\
\geq& -n\langle \mathbf{w}_{K,s}^{(t)}, \boldsymbol{\mu} \rangle + \sum_{i\in[n]} \langle \mathbf{w}_{K,s}^{(T_4^-)}, y_i\boldsymbol{\xi}_i \rangle + (t-T_4^-)\eta\sigma_p s/\sqrt{2} \\
\geq& -n\langle \mathbf{w}_{K,s}^{(T_4^-)}, \boldsymbol{\mu} \rangle + \sum_{i\in[n]} \langle \mathbf{w}_{K,s}^{(T_4^-)}, y_i\boldsymbol{\xi}_i \rangle + (t-T_4^-)\eta(\sigma_p s/\sqrt{2} - n\|\boldsymbol{\mu}\|) > 0,
\end{aligned}
$$

where the first step is by Eq. (11), the second step is by the concentration of $\ell_i'$ and $s_{i,11}^{(t)}$ for $t \leq T_4$, i.e., Lemma E.14 and E.15, the fifth step is by $\mathcal{B}(t)$, the sixth step is by the lower bound Eq. (22), the last step is guaranteed by the dynamics in $[T_3, T_4^-]$. □

*Proof of Claim E.35.* This claim is straightforward. □

The following lemma gives an upper bound for $T_4^+$.

**Lemma E.36.** *Let $\theta_c$ be a small constant satisfying $0.99(1+\theta_c) - O(\theta_c/\sqrt{m_k}) > 1$, then we have*

$$T_4^+ \leq (1+\theta_c)T_4^-, \tag{43}$$

*and for all $t \in [(1+\theta_c)T_4^-, (2-\theta_c)T_4^-]$ we have*

$$s_{i,21}^{(t)} s_{i,22}^{(t)} \left| \langle \mathbf{w}_{Q,s}^{(t)}, y_i\boldsymbol{\xi}_i \rangle \right| \leq \frac{1}{2} s_{i,11}^{(t)} s_{i,12}^{(t)} \left| \langle \mathbf{w}_{Q,s}^{(t)}, \boldsymbol{\mu} \rangle \right|.$$

*Proof.* For fixed $\theta \in [\theta_c, 1 - \theta_c]$, at $t = (1+\theta)T_4^-$, for all $i \in [n]$, we have

$$\sum_{s \in [m_k]} \langle \mathbf{w}_{Q,s}^{(t)}, \boldsymbol{\xi}_i \rangle \langle \mathbf{w}_{K,s}^{(t)}, \boldsymbol{\xi}_i \rangle$$

$$= \sum_{s \in E_i^{(T_3)}} \langle \mathbf{w}_{Q,s}^{(t)}, \boldsymbol{\xi}_i \rangle \langle \mathbf{w}_{K,s}^{(t)}, \boldsymbol{\xi}_i \rangle + \sum_{s \in [m_k]/E_i^{(T_3)}} \langle \mathbf{w}_{Q,s}^{(t)}, \boldsymbol{\xi}_i \rangle \langle \mathbf{w}_{K,s}^{(t)}, \boldsymbol{\xi}_i \rangle$$

$$\geq \left| E_i^{(T_3)} \right| \frac{2}{\pi} t^2 \eta^2 \sigma_p^2 s^2 (1 \pm \tilde{O}(s^{-1/2})) + (m_k - \left| E_i^{(T_3)} \right|) \frac{2}{\pi} t(2T_4^- - t) \eta^2 \sigma_p^2 s^2 (1 \pm \tilde{O}(s^{-1/2}))$$

$$= (\left| E_i^{(T_3)} \right| t^2 + (m_k - \left| E_i^{(T_3)} \right|) t(2T_4^- - t)) \cdot \frac{2}{\pi} \eta^2 \sigma_p^2 s^2 (1 \pm \tilde{O}(s^{-1/2}))$$

$$= (\left| E_i^{(T_3)} \right| (1+\theta)^2 T_4^{-2} + (m_k - \left| E_i^{(T_3)} \right|)(1+\theta)(1-\theta) T_4^{-2} \cdot \frac{2}{\pi} \eta^2 \sigma_p^2 s^2 (1 \pm \tilde{O}(s^{-1/2}))$$

$$\geq ((\frac{m_k}{2} - o(m_k))(1+\theta)^2 + (\frac{m_k}{2} + o(m_k))(1-\theta^2)) \cdot T_4^{-2} \frac{2}{\pi} \eta^2 \sigma_p^2 s^2 (1 \pm \tilde{O}(s^{-1/2}))$$

$$= ((1+\theta)m_k - (\theta^2 + \theta)o(m_k)) \cdot T_4^{-2} \frac{2}{\pi} \eta^2 \sigma_p^2 s^2 (1 \pm \tilde{O}(s^{-1/2})), \tag{44}$$

where the second step is by $\mathcal{G}(t)$ and Lemma E.29, the fifth step is by Lemma E.27.

Then, we have

$$s_{i,21}^{(t)} = \frac{\exp\left(\sum_{s \in [m_k]} \langle \mathbf{w}_{Q,s}^{(t)}, \boldsymbol{\xi}_i \rangle \langle \mathbf{w}_{K,s}^{(t)}, y_i \boldsymbol{\mu} \rangle \right)}{\exp\left(\sum_{s \in [m_k]} \langle \mathbf{w}_{Q,s}^{(t)}, \boldsymbol{\xi}_i \rangle \langle \mathbf{w}_{K,s}^{(t)}, y_i \boldsymbol{\mu} \rangle \right) + \exp\left(\sum_{s \in [m_k]} \langle \mathbf{w}_{Q,s}^{(t)}, \boldsymbol{\xi}_i \rangle \langle \mathbf{w}_{K,s}^{(t)}, \boldsymbol{\xi}_i \rangle \right)}$$

$$\leq \frac{1 \pm o(1)}{1 \pm o(1) + \exp\left(((1+\theta)m_k - (\theta^2 + \theta)O(\sqrt{m_k})) \cdot T_4^{-2} \frac{2}{\pi} \eta^2 \sigma_p^2 s^2 (1 \pm \tilde{O}(s^{-1/2}))\right)}$$

$$= \frac{1 \pm o(1)}{1 \pm o(1) + \exp(((1+\theta)m_k - (\theta^2 + \theta)O(\sqrt{m_k})) \cdot T_4^{-2} \frac{2}{\pi} \eta^2 \sigma_p^2 s^2) \exp(\pm \tilde{O}(\frac{2}{\pi} m_k T_4^{-2} \eta^2 \sigma_p^2 s^{3/2}))}$$

$$= \frac{1 \pm o(1)}{1 \pm o(1) + \exp(((1+\theta)m_k - (\theta^2 + \theta)O(\sqrt{m_k})) \cdot T_4^{-2} \frac{2}{\pi} \eta^2 \sigma_p^2 s^2)(1 \pm o(1))}$$

$$= \frac{1 \pm o(1)}{1 + \exp(((1+\theta)m_k - (\theta^2 + \theta)O(\sqrt{m_k})) \cdot T_4^{-2} \frac{2}{\pi} \eta^2 \sigma_p^2 s^2)}$$

$$\leq \frac{1 \pm o(1)}{\exp(((1+\theta)m_k - (\theta^2 + \theta)O(\sqrt{m_k})) \cdot T_4^{-2} \frac{2}{\pi} \eta^2 \sigma_p^2 s^2)}, \tag{45}$$

where the first step is by definition, the second step is by Eq. (28), (27), and (44), the fourth step is by $T_4^- \lesssim T_4$ and $m_k T_4^2 \eta^2 \sigma_p^2 s^2 = o(1)$.

Therefore, we have

$$s_{i,21}^{((1+\theta)T_4^-)} s_{i,22}^{((1+\theta)T_4^-)} \left| \langle \mathbf{w}_{Q,s}^{((1+\theta)T_4^-)}, y_i \boldsymbol{\xi}_i \rangle \right|$$

$$\leq \frac{1 + o(1)}{\exp(((1+\theta)m_k - (\theta^2 + \theta)O(\sqrt{m_k})) \cdot T_4^{-2} \frac{2}{\pi} \eta^2 \sigma_p^2 s^2)} \cdot 1 \cdot (1+\theta)T_4^- \eta \sqrt{\frac{2}{\pi}} \sigma_p s (1 + \tilde{O}(s^{-1/2}))$$

$$\leq (1+\theta)T_4^- \eta \sqrt{\frac{2}{\pi}} \sigma_p s (1 + \tilde{O}(s^{-1/2})) \cdot \left( \frac{3\sqrt{2}n \|\boldsymbol{\mu}\|}{\sigma_p s} \right)^{0.99(1+\theta) - O(\theta/\sqrt{m_k})}$$

$$\leq \frac{1}{9}(1+\theta)T_4^- \eta \|\boldsymbol{\mu}\|$$

$$\leq \frac{1}{2} s_{i,11}^{((1+\theta)T_4^-)} s_{i,12}^{((1+\theta)T_4^-)} \left| \langle \mathbf{w}_{Q,s}^{((1+\theta)T_4^-)}, \boldsymbol{\mu} \rangle \right|,$$

where the first step is by Eq. (45) and $\mathcal{G}(t)$, the second step is by Lemma E.21, the third step is by $0.99(1+\theta) - O(\theta/\sqrt{m_k}) > 1$, the last step is by $\mathcal{G}(t)$ and the concentration results in Lemma E.14, thus the conclusion holds. $\square$

### E.7.3 STAGE IV.C

$T_4$ is a timestep where $s_{i,11}^{(t)} = 1/2 + o(1)$, we characterize the dynamics in this interval.

WLOG, given neuron $s$, we suppose that $\sum_{i=1}^{n} \langle \mathbf{w}_{Q,s}^{(T_2)}, y_i \boldsymbol{\xi}_i \rangle > 0$ in this section below.

Specifically, we use induction on $t$ to simultaneously prove the following properties for all $t = (1 + \theta_c)T_4^-, \ldots, T_4$:

- $\mathcal{A}(t)$, the monotonicity of query noise aligned with signal: for any $s \in [m_k]$ and $i \in E_s^{(T_3)}$

$$\langle \mathbf{w}_{Q,s}^{(t+1)}, y_i \boldsymbol{\xi}_i \rangle = \langle \mathbf{w}_{Q,s}^{(t)}, y_i \boldsymbol{\xi}_i \rangle + \text{sgn}(\langle \mathbf{w}_{Q,s}^{(t)}, y_i \boldsymbol{\xi}_i \rangle) \eta \|\boldsymbol{\xi}_i\|_1.$$

- $\mathcal{B}(t)$, the monotonicity of key noise aligned with signal: for any $s \in [m_k]$ and $i \in E_s^{(T_3)}$

$$\langle \mathbf{w}_{K,s}^{(t+1)}, y_i \boldsymbol{\xi}_i \rangle = \langle \mathbf{w}_{K,s}^{(t)}, y_i \boldsymbol{\xi}_i \rangle + \text{sgn}(\langle \mathbf{w}_{K,s}^{(t)}, y_i \boldsymbol{\xi}_i \rangle) \eta \|\boldsymbol{\xi}_i\|_1.$$

- $\mathcal{C}(t)$, the fate of query noise unaligned with signal: for any $s \in [m_k]$ and $i \in [n]/E_s^{(T_3)}$, for $t \leq (2 - \theta_c)T_4^-$ we have

$$\langle \mathbf{w}_{Q,s}^{(t+1)}, y_i \boldsymbol{\xi}_i \rangle = \langle \mathbf{w}_{Q,s}^{(t)}, y_i \boldsymbol{\xi}_i \rangle + \text{sgn}(\langle \mathbf{w}_{Q,s}^{(T_4^+)}, y_i \boldsymbol{\xi}_i \rangle) \eta \|\boldsymbol{\xi}_i\|_1,$$

and for $t \geq (2 + 3\theta_c)T_4^-$ we have

$$\langle \mathbf{w}_{Q,s}^{(t+1)}, y_i \boldsymbol{\xi}_i \rangle = \langle \mathbf{w}_{Q,s}^{(t)}, y_i \boldsymbol{\xi}_i \rangle - \text{sgn}(\langle \mathbf{w}_{Q,s}^{(T_4^+)}, y_i \boldsymbol{\xi}_i \rangle) \eta \|\boldsymbol{\xi}_i\|_1.$$

- $\mathcal{D}(t)$, the monotonicity of key noise unaligned with signal: for any $s \in [m_k]$ and $i \in [n]/E_s^{(T_3)}$

$$\langle \mathbf{w}_{K,s}^{(t+1)}, y_i \boldsymbol{\xi}_i \rangle = \langle \mathbf{w}_{K,s}^{(t)}, y_i \boldsymbol{\xi}_i \rangle + \text{sgn}(\langle \mathbf{w}_{Q,s}^{(t)}, \boldsymbol{\mu} \rangle) \eta \|\boldsymbol{\xi}_i\|_1.$$

- $\mathcal{E}(t)$, the monotonicity of query signal: for any $s \in [m_k]$

$$\langle \mathbf{w}_{Q,s}^{(t+1)}, \boldsymbol{\mu} \rangle = \langle \mathbf{w}_{Q,s}^{(t)}, \boldsymbol{\mu} \rangle + \text{sgn}(\langle \mathbf{w}_{Q,s}^{(t)}, \boldsymbol{\mu} \rangle) \eta \|\boldsymbol{\mu}\|.$$

- $\mathcal{F}(t)$: the monotonicity of key signal: for any $s \in [m_k]$

$$\langle \mathbf{w}_{K,s}^{(t+1)}, \boldsymbol{\mu} \rangle = \langle \mathbf{w}_{K,s}^{(t)}, \boldsymbol{\mu} \rangle + \text{sgn}(\langle \mathbf{w}_{K,s}^{(t)}, \boldsymbol{\mu} \rangle) \eta \|\boldsymbol{\mu}\|.$$

- $\mathcal{G}(t)$, the linear-with-$t$ estimation: for all $s \in [m_k]$ and $i \in [n]$

$$\langle \mathbf{w}_{Q,s}^{(t)}, y_i \boldsymbol{\xi}_i \rangle = \text{sgn}(\langle \mathbf{w}_{Q,s}^{(t)}, \boldsymbol{\mu} \rangle) t \eta \sqrt{\frac{2}{\pi}} \sigma_p s (1 \pm \tilde{O}(s^{-1/2})), \forall i \in E_s^{(T_3)}$$

$$\langle \mathbf{w}_{K,s}^{(t)}, y_i \boldsymbol{\xi}_i \rangle = \text{sgn}(\langle \mathbf{w}_{Q,s}^{(t)}, \boldsymbol{\mu} \rangle) t \eta \sqrt{\frac{2}{\pi}} \sigma_p s (1 \pm \tilde{O}(s^{-1/2})), \forall i \in E_s^{(T_3)}$$

$$\langle \mathbf{w}_{Q,s}^{(t)}, \boldsymbol{\mu} \rangle = \text{sgn}(\langle \mathbf{w}_{Q,s}^{(t)}, \boldsymbol{\mu} \rangle) t \eta \|\boldsymbol{\mu}\| (1 \pm o(1)).$$

- $\mathcal{H}(t)$: for any $s \in [m_k]$ and $i \in [n]$

$$s_{i,21}^{(t)} s_{i,22}^{(t)} \left| \langle \mathbf{w}_{Q,s}^{(t)}, y_i \boldsymbol{\xi}_i \rangle \right| \leq \frac{1}{2} s_{i,11}^{(t)} s_{i,12}^{(t)} \left| \langle \mathbf{w}_{Q,s}^{(t)}, \boldsymbol{\mu} \rangle \right|.$$

**Remark E.37.** *By Lemma E.36, we have $\mathcal{H}(t)$ holds for $t \in [(1 + \theta_c)T_4^-, (2 - \theta_c)T_4^-]$. The $\mathcal{A}(t)$, $\mathcal{B}(t)$, $\mathcal{E}(t)$ and $\mathcal{G}(t)$ for $t \in [(1 + \theta_c)T_4^-, (2 - \theta_c)T_4^-]$ are already shown in the induction statement in the last section. Actually, the proof of $\mathcal{A}(t)$, $\mathcal{B}(t)$, $\mathcal{E}(t)$ and $\mathcal{G}(t)$ for all subsequent $t$ are same as those in section E.7.2, thus we omit them in this section.*

**Claim E.38.** $\mathcal{H}(t) \implies \mathcal{D}(t), \mathcal{F}(t).$

*Proof of Claim E.38.* We have

$$\langle \mathbf{w}_{K,s}^{(t+1)} - \mathbf{w}_{K,s}^{(t)}, y_i\boldsymbol{\xi}_i\rangle$$
$$=_{\text{sgn}} s_{i,11}^{(t)}s_{i,12}^{(t)}\langle \mathbf{w}_{Q,s}^{(t)}, \boldsymbol{\mu}\rangle + s_{i,21}^{(t)}s_{i,22}^{(t)}\langle \mathbf{w}_{Q,s}^{(t)}, y_i\boldsymbol{\xi}_i\rangle$$
$$=_{\text{sgn}} \langle \mathbf{w}_{Q,s}^{(t)}, \boldsymbol{\mu}\rangle,$$

where the first step is by Eq. (12), the second step is by $\mathcal{H}(t)$ and

$$\langle \mathbf{w}_{K,s}^{(t+1)} - \mathbf{w}_{K,s}^{(t)}, \boldsymbol{\mu}\rangle$$
$$=_{\text{sgn}} \sum_{i\in[n]}(-l_i'^{(t)})\cdot\left(-s_{i,11}^{(t)}s_{i,12}^{(t)}\langle \mathbf{v}^{(t)}, y_i\boldsymbol{\xi}_i\rangle\langle \mathbf{w}_{Q,s}^{(t)}, \boldsymbol{\mu}\rangle - s_{i,21}^{(t)}s_{i,22}^{(t)}\langle \mathbf{v}^{(t)}, y_i\boldsymbol{\xi}_i\rangle\langle \mathbf{w}_{Q,s}^{(t)}, y_i\boldsymbol{\xi}_i\rangle\right)$$
$$=_{\text{sgn}} \sum_{i\in[n]}(-l_i'^{(t)})\cdot(-s_{i,11}^{(t)}s_{i,12}^{(t)}\langle \mathbf{v}^{(t)}, y_i\boldsymbol{\xi}_i\rangle\langle \mathbf{w}_{Q,s}^{(t)}, \boldsymbol{\mu}\rangle)$$
$$=_{\text{sgn}} -\langle \mathbf{w}_{Q,s}^{(t)}, \boldsymbol{\mu}\rangle =_{\text{sgn}} \langle \mathbf{w}_{K,s}^{(t)}, \boldsymbol{\mu}\rangle,$$

where the first step is by Eq. (13), the second step is by $\mathcal{H}(t)$. $\square$

The next lemma gives an one side bound for unaligned key noise for $t \geq (1+\theta_c)T_4^-$.

**Lemma E.39.** *Let $t \geq (1+\theta_c)T_4^-$, suppose $\mathcal{D}(t)$ holds for $t = (1+\theta_c)T_4^-, \ldots, \tilde{t}$, then for all $s \in [m_k]$, $i \in [n]/E_s^{(T_3)}$ we have*

$$\text{sgn}(\langle \mathbf{w}_{K,s}^{(T_4^-)}, y_i\boldsymbol{\xi}_i\rangle)\cdot\langle \mathbf{w}_{K,s}^{(t)}, y_i\boldsymbol{\xi}\rangle \leq (2(1+\theta_c)T_4^- - t)\eta\sqrt{\frac{2}{\pi}}\sigma_p s(1+\tilde{O}(s^{-1/2})).$$

*Proof.* Since $\sum_{i=1}^n\langle \mathbf{w}_{Q,s}^{(T_2)}, y_i\boldsymbol{\xi}_i\rangle > 0$, we have $\langle \mathbf{w}_{K,s}^{(T_4^-)}, y_i\boldsymbol{\xi}_i\rangle < 0$ for $s \in [m_k]$, $i \in [n]/E_s^{(T_3)}$. We can show

$$\langle \mathbf{w}_{K,s}^{(t)}, y_i\boldsymbol{\xi}\rangle = \langle \mathbf{w}_{K,s}^{(1+\theta_c)T_4^-}, y_i\boldsymbol{\xi}\rangle + (t-(1+\theta_c)T_4^-)\eta\|\boldsymbol{\xi}_i\|_1$$
$$\geq \langle \mathbf{w}_{K,s}^{(T_4^-)}, y_i\boldsymbol{\xi}\rangle + (t-(1+\theta_c)T_4^- - \theta_cT_4^-)\eta\|\boldsymbol{\xi}_i\|_1$$
$$= -T_4^-\eta\sqrt{\frac{2}{\pi}}\sigma_p s(1\pm\tilde{O}(s^{-1/2})) + (t-(1+\theta_c)T_4^- - \theta_cT_4^-)\eta\|\boldsymbol{\xi}_i\|_1$$
$$\geq -(2(1+\theta_c)T_4^- - t)\eta\sqrt{\frac{2}{\pi}}\sigma_p s(1+\tilde{O}(s^{-1/2})),$$

where the first step is by $\mathcal{D}(t)$ for $t \in [(1+\theta_c)T_4^-, \tilde{t}]$, the second step is by Lemma E.29, the third step is by the argument in Remark E.28, the last step is by Lemma C.4. $\square$

Now we only need to and are ready to show $\mathcal{H}(t)$ holds for $t = (2-\theta_c)T_4^- + 1, \ldots, T_4$. Suppose $\mathcal{A}(t'), \ldots, \mathcal{H}(t')$ hold for $t' \leq t$, we want to show $\mathcal{H}(t+1)$ holds. We split to two stages to prove where the first one is $[(2-\theta_c)T_4^-, (2+3\theta_c)T_4^-]$. In this stage, we can bound the magnitude of unaligned key noise $\langle \mathbf{w}_{K,s}^{(t)}, \boldsymbol{\xi}_i\rangle$ by Lemma E.29 and E.39. Note that

$$\sum_{s\in E_i^{(T_3)}}\langle \mathbf{w}_{Q,s}^{(t+1)}, \boldsymbol{\xi}_i\rangle\langle \mathbf{w}_{K,s}^{(t+1)}, \boldsymbol{\xi}_i\rangle - \langle \mathbf{w}_{Q,s}^{(t)}, \boldsymbol{\xi}_i\rangle\langle \mathbf{w}_{K,s}^{(t)}, \boldsymbol{\xi}_i\rangle$$
$$= \sum_{s\in E_i^{(T_3)}}\left(\left|\langle \mathbf{w}_{Q,s}^{(t)}, \boldsymbol{\xi}_i\rangle\right| + \eta\|\boldsymbol{\xi}_i\|_1\right)\left(\left|\langle \mathbf{w}_{K,s}^{(t)}, \boldsymbol{\xi}_i\rangle\right| + \eta\|\boldsymbol{\xi}_i\|_1\right) - \left|\langle \mathbf{w}_{Q,s}^{(t)}, \boldsymbol{\xi}_i\rangle\right|\left|\langle \mathbf{w}_{K,s}^{(t)}, \boldsymbol{\xi}_i\rangle\right|$$
$$= \sum_{s\in E_i^{(T_3)}}\eta^2\|\boldsymbol{\xi}_i\|_1^2 + \eta\|\boldsymbol{\xi}_i\|_1\left(\left|\langle \mathbf{w}_{Q,s}^{(t)}, \boldsymbol{\xi}_i\rangle\right| + \left|\langle \mathbf{w}_{K,s}^{(t)}, \boldsymbol{\xi}_i\rangle\right|\right)$$
$$= \left|E_i^{(T_3)}\right|(\eta^2\|\boldsymbol{\xi}_i\|_1^2 + 2t\|\boldsymbol{\xi}_i\|_1^2\eta^2(1\pm o(1))), \tag{46}$$

where the last step is by $\mathcal{G}(t)$, and

$$\sum_{s\in[n]/E_i^{(T_3)}} \langle\mathbf{w}_{Q,s}^{(t+1)},\boldsymbol{\xi}_i\rangle\langle\mathbf{w}_{K,s}^{(t+1)},\boldsymbol{\xi}_i\rangle - \langle\mathbf{w}_{Q,s}^{(t)},\boldsymbol{\xi}_i\rangle\langle\mathbf{w}_{K,s}^{(t)},\boldsymbol{\xi}_i\rangle$$

$$= \sum_{t\geq T_{4,s,i}^{Q,\text{flip}}} \langle\mathbf{w}_{Q,s}^{(t+1)},\boldsymbol{\xi}_i\rangle\langle\mathbf{w}_{K,s}^{(t+1)},\boldsymbol{\xi}_i\rangle - \langle\mathbf{w}_{Q,s}^{(t)},\boldsymbol{\xi}_i\rangle\langle\mathbf{w}_{K,s}^{(t)},\boldsymbol{\xi}_i\rangle + \sum_{t< T_{4,s,i}^{Q,\text{flip}}} \langle\mathbf{w}_{Q,s}^{(t+1)},\boldsymbol{\xi}_i\rangle\langle\mathbf{w}_{K,s}^{(t+1)},\boldsymbol{\xi}_i\rangle - \langle\mathbf{w}_{Q,s}^{(t)},\boldsymbol{\xi}_i\rangle\langle\mathbf{w}_{K,s}^{(t)},\boldsymbol{\xi}_i\rangle$$

$$= \sum_{t< T_{4,s,i}^{Q,\text{flip}},\langle\mathbf{w}_{Q,s}^{(t)},\boldsymbol{\mu}\rangle>0} -\eta^2\|\boldsymbol{\xi}_i\|_1^2 - \eta\|\boldsymbol{\xi}_i\|_1 (-\langle\mathbf{w}_{Q,s}^{(t)},\boldsymbol{\xi}_i\rangle + \langle\mathbf{w}_{K,s}^{(t)},\boldsymbol{\xi}_i\rangle)$$

$$+ \sum_{t< T_{4,s,i}^{Q,\text{flip}},\langle\mathbf{w}_{Q,s}^{(t)},\boldsymbol{\mu}\rangle<0} -\eta^2\|\boldsymbol{\xi}_i\|_1^2 - \eta\|\boldsymbol{\xi}_i\|_1 (\langle\mathbf{w}_{Q,s}^{(t)},\boldsymbol{\xi}_i\rangle - \langle\mathbf{w}_{K,s}^{(t)},\boldsymbol{\xi}_i\rangle)$$

$$+ \sum_{t\geq T_{4,s,i}^{Q,\text{flip}},\langle\mathbf{w}_{Q,s}^{(t)},\boldsymbol{\mu}\rangle>0} \eta^2\|\boldsymbol{\xi}_i\|_1^2 + \eta\|\boldsymbol{\xi}_i\|_1 (\langle\mathbf{w}_{Q,s}^{(t)},\boldsymbol{\xi}_i\rangle + \langle\mathbf{w}_{K,s}^{(t)},\boldsymbol{\xi}_i\rangle)$$

$$+ \sum_{t\geq T_{4,s,i}^{Q,\text{flip}},\langle\mathbf{w}_{Q,s}^{(t)},\boldsymbol{\mu}\rangle<0} \eta^2\|\boldsymbol{\xi}_i\|_1^2 - \eta\|\boldsymbol{\xi}_i\|_1 (\langle\mathbf{w}_{Q,s}^{(t)},\boldsymbol{\xi}_i\rangle + \langle\mathbf{w}_{K,s}^{(t)},\boldsymbol{\xi}_i\rangle)$$

$$\geq \sum_{s\in[n]/E_i^{(T_3)}} -\eta^2\|\boldsymbol{\xi}_i\|_1^2 - \eta\|\boldsymbol{\xi}_i\|_1 (\left|\langle\mathbf{w}_{Q,s}^{(t)},\boldsymbol{\xi}_i\rangle\right| + \left|\langle\mathbf{w}_{K,s}^{(t)},\boldsymbol{\xi}_i\rangle\right|).$$

Furthermore, when $(2-\theta_c)T_4^- \leq t \leq (2+3\theta_c)T_4^-$, for $s\in[m_k]$ and $i\in[n]/E_s^{(T_3)}$, we have

$$-3\theta_c T_4^- \eta\sqrt{\frac{2}{\pi}}\sigma_p s(1+\tilde{O}(s^{-1/2})) \leq \text{sgn}(\langle\mathbf{w}_{K,s}^{(T_4^-)},y_i\boldsymbol{\xi}\rangle)\cdot\langle\mathbf{w}_{K,s}^{(t)},y_i\boldsymbol{\xi}\rangle \leq 3\theta_c T_4^- \eta\sqrt{\frac{2}{\pi}}\sigma_p s(1+\tilde{O}(s^{-1/2})),$$
(47)

where the upper bound is by Lemma E.39, and the lower bound is by Lemma E.29. Therefore, for $(2-\theta_c)T_4^- t \leq (2+3\theta_c)T_4^-$ we have

$$\sum_{s\in[n]} \langle\mathbf{w}_{Q,s}^{(t+1)},\boldsymbol{\xi}_i\rangle\langle\mathbf{w}_{K,s}^{(t+1)},\boldsymbol{\xi}_i\rangle - \langle\mathbf{w}_{Q,s}^{(t)},\boldsymbol{\xi}_i\rangle\langle\mathbf{w}_{K,s}^{(t)},\boldsymbol{\xi}_i\rangle$$

$$\geq \sum_{s\in E_i^{(T_3)}} \eta^2\|\boldsymbol{\xi}_i\|_1^2 + \eta\|\boldsymbol{\xi}_i\|_1 (\left|\langle\mathbf{w}_{Q,s}^{(t)},\boldsymbol{\xi}_i\rangle\right| + \left|\langle\mathbf{w}_{K,s}^{(t)},\boldsymbol{\xi}_i\rangle\right|)$$

$$- \sum_{s\in[n]/E_i^{(T_3)}} \eta^2\|\boldsymbol{\xi}_i\|_1^2 + \eta\|\boldsymbol{\xi}_i\|_1 (\left|\langle\mathbf{w}_{Q,s}^{(t)},\boldsymbol{\xi}_i\rangle\right| + \left|\langle\mathbf{w}_{K,s}^{(t)},\boldsymbol{\xi}_i\rangle\right|)$$

$$= \left|E_i^{(T_3)}\right| (\eta^2\|\boldsymbol{\xi}_i\|_1^2 + 2t\|\boldsymbol{\xi}_i\|_1^2\eta^2(1\pm o(1))) - (m_k - \left|E_i^{(T_3)}\right|)(\eta^2\|\boldsymbol{\xi}_i\|_1^2 + (t+3\theta_c T_4^-)\|\boldsymbol{\xi}_i\|_1^2\eta^2(1\pm o(1)))$$

$$\geq (\frac{m_k}{2} - O(\sqrt{m_k}))(\eta^2\|\boldsymbol{\xi}_i\|_1^2 + 2t\|\boldsymbol{\xi}_i\|_1^2\eta^2(1\pm o(1))) - (\frac{m_k}{2} + O(\sqrt{m_k}))(\eta^2\|\boldsymbol{\xi}_i\|_1^2 + (t+3\theta_c T_4^-)\|\boldsymbol{\xi}_i\|_1^2\eta^2(1\pm o(1)))$$

$$\geq -O(\sqrt{m_k})\eta^2\|\boldsymbol{\xi}_i\|_1^2 + (\frac{m_k}{2}(t-3\theta_c T_4^-) - O(\sqrt{m_k})(t+\theta_c T_4^-))\eta^2\|\boldsymbol{\xi}_i\|_1^2(1\pm o(1))) > 0,$$

where the second step is by Eq. (46) for aligned part, Lemma E.29 for upper bound of unaligned query noise and Eq. (47) for upper bound of unaligned key noise, the third step is by concentration for $\left|E_i^{(T_3)}\right|$ in Lemma E.27. Combined with $\mathcal{H}(t')$ for $(1+\theta_c)T_4^- \leq t' \leq t$, we have

$$s_{i,21}^{(t+1)}s_{i,22}^{(t+1)}\left|\langle\mathbf{w}_{Q,s}^{(t+1)},y_i\boldsymbol{\xi}_i\rangle\right| \leq s_{i,21}^{((1+\theta)T_4^-)}\left|\langle\mathbf{w}_{Q,s}^{(t+1)},y_i\boldsymbol{\xi}_i\rangle\right|$$

$$\leq \frac{1}{9}(1+t)\eta\|\boldsymbol{\mu}\| \leq \frac{1}{2}s_{i,11}^{(t+1)}s_{i,12}^{(t+1)}\left|\langle\mathbf{w}_{Q,s}^{(t+1)},\boldsymbol{\mu}\rangle\right|,$$

where the first step is by induction, the second step is similar to the proof in Lemma E.36, which implies $\mathcal{H}(t+1)$ holds.

Note that now we have $\mathcal{D}(t')$, $\mathcal{F}(t')$ hold for $t = (2 - \theta_c)T_4^-, \ldots, (2 + 3\theta_c)T_4^-$. Then for $t \geq (2 + 3\theta_c)T_4^-$, suppose $\mathcal{F}(t)$ and $\mathcal{D}(t)$ hold for $t' \leq t$, for all $s \in [m_k]$ and $i \in [n]/E_s^{(T_3)}$, we have

$$
\begin{aligned}
\langle \mathbf{w}_{Q,s}^{(t+1)} - \mathbf{w}_{Q,s}^{(t)}, y_i\boldsymbol{\xi}_i \rangle &=_{\mathrm{sgn}} -\langle \mathbf{w}_{K,s}^{(t)}, \boldsymbol{\mu} \rangle + \langle \mathbf{w}_{K,s}^{(t)}, y_i\boldsymbol{\xi}_i \rangle \\
&\geq \langle \mathbf{w}_{K,s}^{(t)}, y_i\boldsymbol{\xi}_i \rangle \\
&\geq (t - 2(1 + \theta_c)T_4^-)\eta\sqrt{\frac{2}{\pi}}\sigma_p s(1 + \tilde{O}(s^{-1/2})) > 0,
\end{aligned}
$$

where the first step is by Eq. (10), the second step is by $\mathcal{F}(t)$, the last step is by Lemma E.39. Combined with $\mathcal{D}(t)$, when $\langle \mathbf{w}_{Q,s}^{(t+1)}, y_i\boldsymbol{\xi}_i \rangle < 0$ (note that we suppose $\langle \mathbf{w}_{Q,s}^{(T_3)}, \boldsymbol{\mu} \rangle > 0$, in more general case this is $\langle \mathbf{w}_{Q,s}^{(t+1)}, y_i\boldsymbol{\xi}_i \rangle =_{\mathrm{sgn}} -\langle \mathbf{w}_{Q,s}^{(t+1)}, \boldsymbol{\mu} \rangle$), this gives

$$
\left| \langle \mathbf{w}_{Q,s}^{(t+1)}, y_i\boldsymbol{\xi}_i \rangle \right| + \left| \langle \mathbf{w}_{K,s}^{(t+1)}, y_i\boldsymbol{\xi}_i \rangle \right| = \left| \langle \mathbf{w}_{Q,s}^{(t)}, y_i\boldsymbol{\xi}_i \rangle \right| + \left| \langle \mathbf{w}_{K,s}^{(t)}, y_i\boldsymbol{\xi}_i \rangle \right|. \tag{48}
$$

Note that at $t = (2 + 3\theta_c)T_4^-$, we have

$$
\left| \langle \mathbf{w}_{Q,s}^{((2+3\theta_c)T_4^-)}, y_i\boldsymbol{\xi}_i \rangle \right| + \left| \langle \mathbf{w}_{K,s}^{((2+3\theta_c)T_4^-)}, y_i\boldsymbol{\xi}_i \rangle \right| \leq (2 + 6\theta_c)T_4^- \|\boldsymbol{\xi}_i\|_1^2 \eta^2(1 \pm o(1)), \tag{49}
$$

where the upper bound for query noise is by Lemma E.29 and the upper bound for key noise is by Lemma E.39. Then we have

$$
\begin{aligned}
&\sum_{s \in [n]} \langle \mathbf{w}_{Q,s}^{(t+1)}, \boldsymbol{\xi}_i \rangle \langle \mathbf{w}_{K,s}^{(t+1)}, \boldsymbol{\xi}_i \rangle - \langle \mathbf{w}_{Q,s}^{(t)}, \boldsymbol{\xi}_i \rangle \langle \mathbf{w}_{K,s}^{(t)}, \boldsymbol{\xi}_i \rangle \\
&\geq \sum_{s \in E_i^{(T_3)}} \eta^2\|\boldsymbol{\xi}_i\|_1^2 + \eta\|\boldsymbol{\xi}_i\|_1 \left( \left|\langle \mathbf{w}_{Q,s}^{(t)}, \boldsymbol{\xi}_i \rangle\right| + \left|\langle \mathbf{w}_{K,s}^{(t)}, \boldsymbol{\xi}_i \rangle\right| \right) \\
&\quad - \sum_{s \in [n]/E_i^{(T_3)}, \langle \mathbf{w}_{Q,s}^{(t+1)}, y_i\boldsymbol{\xi}_i \rangle =_{\mathrm{sgn}} -\langle \mathbf{w}_{Q,s}^{(t+1)}, \boldsymbol{\mu} \rangle} \eta^2\|\boldsymbol{\xi}_i\|_1^2 + \eta\|\boldsymbol{\xi}_i\|_1 \left( \left|\langle \mathbf{w}_{Q,s}^{(t)}, \boldsymbol{\xi}_i \rangle\right| + \left|\langle \mathbf{w}_{K,s}^{(t)}, \boldsymbol{\xi}_i \rangle\right| \right) \\
&= \left|E_i^{(T_3)}\right| (\eta^2\|\boldsymbol{\xi}_i\|_1^2 + 2t\|\boldsymbol{\xi}_i\|_1^2\eta^2(1 \pm o(1))) - (m_k - \left|E_i^{(T_3)}\right|)(\eta^2\|\boldsymbol{\xi}_i\|_1^2 + (2 + 6\theta_c)T_4^-\|\boldsymbol{\xi}_i\|_1^2\eta^2(1 \pm o(1))) \\
&\geq (\frac{m_k}{2} - O(\sqrt{m_k}))(\eta^2\|\boldsymbol{\xi}_i\|_1^2 + 2t\|\boldsymbol{\xi}_i\|_1^2\eta^2(1 \pm o(1))) - (\frac{m_k}{2} + O(\sqrt{m_k}))(\eta^2\|\boldsymbol{\xi}_i\|_1^2 + (2 + 6\theta_c)T_4^-\|\boldsymbol{\xi}_i\|_1^2\eta^2(1 \pm o(1))) \\
&\geq -O(\sqrt{m_k})\eta^2\|\boldsymbol{\xi}_i\|_1^2 + (m_k(t - (1 + 3\theta_c)T_4^-) - O(\sqrt{m_k})(t + (1 + 3\theta_c)T_4^-))\eta^2\|\boldsymbol{\xi}_i\|_1^2(1 \pm o(1))) > 0,
\end{aligned}
$$

where the second step is by Eq. (46) for aligned part, Eq. (48) and (49) for unaligned part, the third step is by concentration for $\left|E_i^{(T_3)}\right|$ in Lemma E.27. Combined with $\mathcal{H}(t)$, this implies $\mathcal{H}(t + 1)$ holds. Then we have $\mathcal{H}(t)$ holds for all $(1 + \theta_c)T_4^- \leq t \leq T_4$.

**Remark E.40.** *When $t = (4 + 7\theta_c)T_4^-$, we have that for all $s \in [m_k]$ and $i \in [n]$*

$$
\langle \mathbf{w}_{Q,s}^{(t)}, y_i\boldsymbol{\xi}_i \rangle =_{\mathrm{sgn}} \langle \mathbf{w}_{K,s}^{(t)}, y_i\boldsymbol{\xi}_i \rangle =_{\mathrm{sgn}} \langle \mathbf{w}_{Q,s}^{(t)}, \boldsymbol{\mu} \rangle.
$$

*Finally, for all $s \in [m_k]$, $i \in [n]$ and $t \geq T_4$, we have*

$$
\begin{aligned}
\langle \mathbf{w}_{Q,s}^{(t+1)}, y_i\boldsymbol{\xi}_i \rangle &= \langle \mathbf{w}_{Q,s}^{(t)}, y_i\boldsymbol{\xi}_i \rangle + \mathrm{sgn}(\langle \mathbf{w}_{Q,s}^{(t)}, y_i\boldsymbol{\xi}_i \rangle)\eta\|\boldsymbol{\xi}_i\|_1, \\
\langle \mathbf{w}_{K,s}^{(t+1)}, y_i\boldsymbol{\xi}_i \rangle &= \langle \mathbf{w}_{K,s}^{(t)}, y_i\boldsymbol{\xi}_i \rangle + \mathrm{sgn}(\langle \mathbf{w}_{K,s}^{(t)}, y_i\boldsymbol{\xi}_i \rangle)\eta\|\boldsymbol{\xi}_i\|_1, \\
\langle \mathbf{w}_{Q,s}^{(t+1)}, \boldsymbol{\mu} \rangle &= \langle \mathbf{w}_{Q,s}^{(t)}, \boldsymbol{\mu} \rangle + \mathrm{sgn}(\langle \mathbf{w}_{Q,s}^{(t)}, \boldsymbol{\mu} \rangle)\eta\|\boldsymbol{\mu}\|, \\
\langle \mathbf{w}_{K,s}^{(t+1)}, \boldsymbol{\mu} \rangle &= \langle \mathbf{w}_{K,s}^{(t)}, \boldsymbol{\mu} \rangle + \mathrm{sgn}(\langle \mathbf{w}_{K,s}^{(t)}, \boldsymbol{\mu} \rangle)\eta\|\boldsymbol{\mu}\|, \\
\langle \mathbf{w}_{Q,s}^{(t)}, y_i\boldsymbol{\xi}_i \rangle &=_{\mathrm{sgn}} \langle \mathbf{w}_{K,s}^{(t)}, y_i\boldsymbol{\xi}_i \rangle =_{\mathrm{sgn}} \langle \mathbf{w}_{Q,s}^{(t)}, \boldsymbol{\mu} \rangle =_{\mathrm{sgn}} -\langle \mathbf{w}_{K,s}^{(t)}, \boldsymbol{\mu} \rangle.
\end{aligned}
$$

## E.8 Proof of Theorem 3.2

In this section, we aim to prove the Theorem 3.2.

*Proof of Theorem 3.2.* By Lemma E.41 and E.42, we complete the proof. □

**Lemma E.41** (Training and Logistic Test Loss). *For any fixed $\epsilon > 0$, let $T = 2\log(\epsilon^{-1})\eta^{-1}\sigma_p^{-1}s^{-1}$, we have $L_S(\mathbf{W}^{(T)}) \leq \epsilon$ and $L_\mathcal{D}(\mathbf{W}^{(T)}) \geq 0.1$.*

*Proof.* Note that

$$
\begin{aligned}
\langle \mathbf{v}^{(T)}, \boldsymbol{\mu} \rangle &= \langle \mathbf{v}^{(0)}, \boldsymbol{\mu} \rangle + \eta T \|\boldsymbol{\mu}\| \\
&= \pm\tilde{O}(\sigma_0 \|\boldsymbol{\mu}\| m_v^{-1/2}) + 2\log(\epsilon^{-1})\|\boldsymbol{\mu}\|\sigma_p^{-1}s^{-1} \\
&= \Theta(\log(\epsilon^{-1})\|\boldsymbol{\mu}\|\sigma_p^{-1}s^{-1}),
\end{aligned}
$$

where the last step is by $\tilde{O}(\sigma_0\|\boldsymbol{\mu}\|m_v^{-1/2}) = o(\eta T\|\boldsymbol{\mu}\|)$. Then, for all $i \in [n]$, we have

$$
\begin{aligned}
y_i f(\mathbf{W}^{(T)}, \mathbf{X}_i) &= (s_{i,11}^{(T)} + s_{i,21}^{(T)})\langle \mathbf{v}^{(T)}, \boldsymbol{\mu} \rangle + (s_{i,12}^{(T)} + s_{i,22}^{(T)})\langle \mathbf{v}^{(T)}, y_i\boldsymbol{\xi}_i \rangle \\
&\geq (s_{i,12}^{(T)} + s_{i,22}^{(T)})\langle \mathbf{v}^{(T)}, y_i\boldsymbol{\xi}_i \rangle \\
&\geq 0.9\langle \mathbf{v}^{(T)}, y_i\boldsymbol{\xi}_i \rangle \\
&\geq \log(\epsilon^{-1}),
\end{aligned}
$$

where the first step is by the definition of $f$, the second step is by $s_{i,11}^{(T)} \geq 0$, $s_{i,21}^{(T)} \geq 0$ and $\langle \mathbf{v}^{(T)}, \boldsymbol{\mu} \rangle \geq 0$, the third step is by $s_{i,12}^{(T)} \geq 0$ $s_{i,22}^{(T)} \geq 0.9$, the last step is by Lemma E.8 and the definition of $T$. This implies $\ell_i^{(T)} = \log(1 + \exp(-y_i f_i(\mathbf{W}^{(T)}, \mathbf{x}_i))) \leq \epsilon$ and thus $L_S(\mathbf{W}^{(T)}) \leq \epsilon$.

On the other hand, for a new data point $\mathbf{x} = (y\boldsymbol{\mu}, \boldsymbol{\xi})^\top$, let event $\mathcal{E}$ be the event that $\boldsymbol{\xi}$ has disjoint support with $\boldsymbol{\xi}_1, \ldots, \boldsymbol{\xi}_n$. Then when $\mathcal{E}$ holds, we have

$$
\begin{aligned}
yf(\mathbf{W}^{(T)}, \mathbf{X}) &= (s_{11}^{(T)} + s_{21}^{(T)})\langle \mathbf{v}^{(T)}, \boldsymbol{\mu} \rangle + (s_{12}^{(T)} + s_{22}^{(T)})\langle \mathbf{v}^{(T)}, y\boldsymbol{\xi} \rangle \\
&\leq 2\langle \mathbf{v}^{(T)}, \boldsymbol{\mu} \rangle + 2\langle \mathbf{v}^{(T)}, y\boldsymbol{\xi} \rangle \\
&\leq \tilde{O}(\log(\epsilon^{-1})\|\boldsymbol{\mu}\|\sigma_p^{-1}s^{-1} + \sigma_0\sigma_p s^{1/2} m_v^{-1/2}) \\
&\leq 1,
\end{aligned}
$$

where the second step is by $s_{11}^{(T)}, s_{12}^{(T)}, s_{21}^{(T)}, s_{22}^{(T)} \leq 1$, the last step is by $\sigma_0\sigma_p s^{1/2} = o(1)$. And when $\mathcal{E}$ does not hold, we have

$$
\begin{aligned}
yf(\mathbf{W}^{(T)}, \mathbf{X}) &= (s_{11}^{(T)} + s_{21}^{(T)})\langle \mathbf{v}^{(T)}, \boldsymbol{\mu} \rangle + (s_{12}^{(T)} + s_{22}^{(T)})\langle \mathbf{v}^{(T)}, y\boldsymbol{\xi} \rangle \\
&\leq 2\langle \mathbf{v}^{(T)}, \boldsymbol{\mu} \rangle + 2\langle \mathbf{v}^{(T)}, y\boldsymbol{\xi} \rangle \\
&\leq O(\log(\epsilon^{-1})\|\boldsymbol{\mu}\|\sigma_p^{-1}s^{-1} + \log(\epsilon^{-1})) \\
&\leq O(\log(\epsilon^{-1})).
\end{aligned}
$$

Note that $\mathbb{P}[\mathcal{E}] \geq 1 - n^{-2}$, which gives

$$
\begin{aligned}
\mathbb{E}[\ell(yf(\mathbf{W}^{(T)}, \mathbf{X}))] &= \mathbb{E}[\mathbb{1}(\mathcal{E})\ell(yf(\mathbf{W}^{(T)}, \mathbf{x}))] + \mathbb{E}[\mathbb{1}(\mathcal{E}^c)\ell(yf(\mathbf{W}^{(T)}, \mathbf{x}))] \\
&\geq (1 - n^{-2})\log(1 + e^{-1}) \geq 0.1,
\end{aligned}
$$

which completes the proof. □

**Lemma E.42** (Attention layer all attends to noise patch). *Let $T_{\text{attn}} = \Theta(\log(\sigma_p s/\|\boldsymbol{\mu}\|)\eta^{-1}m_k^{-1/2}\sigma_p^{-1/2}s^{-1/2}\|\boldsymbol{\mu}\|^{-1/2})$, we have $s_{i,11}^{(T_{\text{attn}})} = o(1)$, $s_{i,21}^{(T_{\text{attn}})} = o(1)$, for all $i \in [n]$.*

*Proof.* By the previous analysis, we already have $s_{i,21}^{(t)} = o(1)$ for $t \geq T_4^-$. For $t \geq T_{\text{attn}}$

$$
\begin{aligned}
s_{i,11}^{(t)} &= \frac{\exp\left(\sum_{s \in [m_k]} \langle \mathbf{w}_{Q,s}^{(t)}, \boldsymbol{\mu} \rangle \langle \mathbf{w}_{K,s}^{(t)}, \boldsymbol{\mu} \rangle\right)}{\exp\left(\sum_{s \in [m_k]} \langle \mathbf{w}_{Q,s}^{(t)}, \boldsymbol{\mu} \rangle \langle \mathbf{w}_{K,s}^{(t)}, \boldsymbol{\mu} \rangle\right) + \exp\left(\sum_{s \in [m_k]} \langle \mathbf{w}_{Q,s}^{(t)}, \boldsymbol{\mu} \rangle \langle \mathbf{w}_{K,s}^{(t)}, y_i \boldsymbol{\xi}_i \rangle\right)} \\
&\leq \frac{\exp\left(2 m_k t^2 \eta^2 \|\boldsymbol{\mu}\|^2\right)}{\exp\left(2 m_k t^2 \eta^2 \|\boldsymbol{\mu}\|^2\right) + \exp\left(m_k t^2 \eta^2 \sigma_p s \|\boldsymbol{\mu}\| / 4\right)} \\
&\leq \frac{1}{\exp\left(\Omega(m_k t^2 \eta^2 \sigma_p s \|\boldsymbol{\mu}\|)\right)} \\
&\leq o(1),
\end{aligned}
$$

where the first step is by definition, the second step is by $\mathcal{H}(t)$ in Stage IV.c, Remark. E.40, and the concentration in Lemma E.27, the last step is by $t \geq T_{\text{attn}}$. $\square$

# F DISCUSSION ON EXTENSIONS

## F.1 0-1 TEST LOSS

In this section, we talk about the magntiude of 0-1 test loss. In binary classification, constant logistic test loss doesn't necessarily mean constant 0-1 test loss. And actually, we find it depends on the network initialization parameter $\sigma_0$.

The 0-1 Test Loss is defined as

$$L_{\mathcal{D}}^{0-1}(\mathbf{W}) := \mathbb{E}[\mathbb{1}(yf(\mathbf{W}, \mathbf{X}) > 0)] = \mathbb{P}[yf(\mathbf{W}, \mathbf{X}) > 0].$$

The value of $yf(\mathbf{W}, \mathbf{X})$ depends on two components, the signal part $\langle \mathbf{v}, \boldsymbol{\mu} \rangle$, which continuously increases during training, and the noise part $\langle \mathbf{v}, \boldsymbol{\xi} \rangle$, which is random for the unseen noise $\boldsymbol{\xi}_i$. With probability at least $1 - n^{-2}$, the unseen $\boldsymbol{\xi}$ is disjoint with all training data $\boldsymbol{\xi}_i$.

If $\sigma_0$ is small enough, i.e., $\sigma_0 = o(\sigma_p^{-2} s^{-3/2} m_v^{1/2})$, then the noise part can be ignored: By the disjoint property, we can use the estimate at initialization for unseen noise. By Lemma C.6, with probability at least $1 - \delta$, we have

$$\langle \mathbf{v}^{(T)}, \boldsymbol{\mu} \rangle = \Theta(\log(\epsilon^{-1}) \|\boldsymbol{\mu}\| \sigma_p^{-1} s^{-1}) \gg O(m_v^{-1/2} \sigma_0 \sigma_p s^{1/2}) = |\langle \mathbf{v}^{(T)}, \boldsymbol{\xi} \rangle|.$$

This is similar for $\langle \mathbf{w}_{Q,s}^{(T)}, \boldsymbol{\mu} \rangle$ v.s. $\langle \mathbf{w}_{Q,s}^{(T)}, \boldsymbol{\xi} \rangle$, and $\langle \mathbf{w}_{K,s}^{(T)}, \boldsymbol{\mu} \rangle$ v.s. $\langle \mathbf{w}_{K,s}^{(T)}, \boldsymbol{\xi} \rangle$, which gives $s_{11}^{(T)}, s_{21}^{(T)} = \Omega(1)$. Then, we have

$$yf(\mathbf{W}^{(T)}, \mathbf{X}) = (s_{11}^{(T)} + s_{21}^{(T)})\langle \mathbf{v}^{(T)}, \boldsymbol{\mu} \rangle + (s_{12}^{(T)} + s_{22}^{(T)})\langle \mathbf{v}^{(T)}, y\boldsymbol{\xi} \rangle$$
$$\geq \Theta(\log(\epsilon^{-1}) \|\boldsymbol{\mu}\| \sigma_p^{-1} s^{-1}) - \tilde{O}(m_v^{-1/2} \sigma_0 \sigma_p s^{1/2})$$
$$> 0,$$

which implies a small 0-1 loss $\mathbb{P}[yf(\mathbf{W}^{(T)}, \mathbf{X}) > 0] \geq 1 - n^{-O(1)}$.

On the other hand, if $\sigma_0$ is large, i.e., $\sigma_0 = \Omega(\sigma_p^{-2} s^{-3/2} m_v^{1/2})$ but still satisfies Condition 4.1, then we have

$$\langle \mathbf{v}^{(T)}, \boldsymbol{\mu} \rangle = \Theta(\log(\epsilon^{-1}) \|\boldsymbol{\mu}\| \sigma_p^{-1} s^{-1}) \ll \Omega(m_v^{-1/2} \sigma_0 \sigma_p s^{1/2}) = |\langle \mathbf{v}^{(T)}, \boldsymbol{\xi} \rangle|.$$

This is similar for $\langle \mathbf{w}_{Q,s}^{(T)}, \boldsymbol{\mu} \rangle$ v.s. $\langle \mathbf{w}_{Q,s}^{(T)}, \boldsymbol{\xi} \rangle$, and $\langle \mathbf{w}_{K,s}^{(T)}, \boldsymbol{\mu} \rangle$ v.s. $\langle \mathbf{w}_{K,s}^{(T)}, \boldsymbol{\xi} \rangle$, which gives $s_{11}^{(T)}, s_{21}^{(T)} = 1/2 \pm o(1)$ since the symmetry of unseen noise. This means that the softmax outputs are random. Then,

$$yf(\mathbf{W}^{(T)}, \mathbf{X}) = (s_{11}^{(T)} + s_{21}^{(T)})\langle \mathbf{v}^{(T)}, \boldsymbol{\mu} \rangle + (s_{12}^{(T)} + s_{22}^{(T)})\langle \mathbf{v}^{(T)}, y\boldsymbol{\xi} \rangle$$
$$= \Omega(1)\langle \mathbf{v}^{(T)}, y\boldsymbol{\xi} \rangle,$$

which implies $|\mathbb{P}[yf(\mathbf{W}^{(T)}, \mathbf{X}) > 0] - 1/2| \leq 1 - n^{-O(1)}$. Note that the condition regarding $\sigma_0$ here is not included in our Condition 4.1.

## F.2 EXTENSION TO LONGER CONTEXTS

In this section, we talk about how to extend our current theory to longer context lengths with additional assumptions.

We consider two case of extension:

**Noise patches are the same within one sample.**
Note that many works use similar settings, e.g., Tarzanagh et al. (2023a); Vasudeva et al. (2024); Sheen et al. (2024) assume all non-optimal tokens are the same in some way. If the data $\mathbf{X}$ has $L$ patches $\mathbf{X} = [\mathbf{x}^{(1)}, \dots, \mathbf{x}^{(L)}]$ where the first $L/2$ patches are signal vectors $\boldsymbol{\mu}$ and the latter $L/2$ patches are noise vectors $\boldsymbol{\xi}^{(l)}$, and all noise patches are the same, i.e., $\boldsymbol{\xi}^{(l)} = \boldsymbol{\xi}^{(l')}$ for $L/2 < l, l' \leq L$, then the training dynamics are the same as in the $L = 2$ setting in the paper. If we change the ratio of number of signal patches to noise patches, e.g., 1 signal patch and $L - 1$ noise patches, it remains essentially the same except that the signal/noise-signal softmax is around $1/L$ while (the sum of)

signal/noise-noise softmax is around $(L-1)/L$ at initialization. Note that in this case the softmax of all noise patches have the same value (no single patch becomes dominant).

**Noise patches are different within one sample.**

We remark that WLOG we can consider 1 $\boldsymbol{\mu}$ and $(L-1)$ $\boldsymbol{\xi}$ vectors. With high probability, the $n(L-1)$ noise patches are disjoint. The main difficulty here is the **competition between noise patches**, which can be seen in the gradient and increase speed. To address or avoid this issue, one solution is to assume some form of sparsity in the dimension of context length. For example, Tarzanagh et al. (2023a); Vasudeva et al. (2024); Sheen et al. (2024) assume there is only one optimal token in one sample, which has a large gap from the non-optimal ones. Jiang et al. (2024) assumes the second noise vector $\boldsymbol{\xi}_2$ is greater than remain noise $\boldsymbol{\xi}_3, \ldots, \boldsymbol{\xi}_n$ in one data point. In our case, we need to assume for each sample there is a noise patch $L$ (WLOG, we assume it is the last patch) such that $\sum_{l'=2}^{L-1} \|\boldsymbol{\xi}^{(l')}\|_1 = o(\|\boldsymbol{\xi}^{(l')}\|_L)$. We can achieve this by applying large sparsity $s$ and/or large std $\sigma_p$ on this patch. Note that this large patch across different samples can be in different positions. We provide a detailed example here: First, we still have stage I since the gradient of value does not change. The gradient of query is

$$\nabla_{\mathbf{w}_{Q,s}} L_S(\mathbf{W}) = \frac{1}{n} \sum_{i=1}^{n} \ell'_i y_i \sum_{l=1}^{L} \mathbf{x}_i^{(l)} \sum_{a<b} s_{i,la} s_{i,lb} \langle \bar{\mathbf{w}}_{V,j}, \mathbf{x}_i^{(a)} - \mathbf{x}_i^{(b)} \rangle \langle \mathbf{w}_{K,s}, \mathbf{x}_i^{(a)} - \mathbf{x}_i^{(b)} \rangle,$$

and the inner product is

$$\langle \mathrm{sgn}(\nabla_{\mathbf{w}_{Q,s}} L_S), y_i \boldsymbol{\xi}_i^{(l)} \rangle = -\|\boldsymbol{\xi}_i^{(l)}\|_1 \mathrm{sgn}(\sum_{a<b} s_{i,la} s_{i,lb} \langle \mathbf{v}, \mathbf{x}_i^{(a)} - \mathbf{x}_i^{(b)} \rangle \langle \mathbf{w}_{K,s}, \mathbf{x}_i^{(a)} - \mathbf{x}_i^{(b)} \rangle)$$

$$\approx -\|\boldsymbol{\xi}_i^{(l)}\|_1 \mathrm{sgn}(\langle \mathbf{w}_{K,s}, y_i \boldsymbol{\xi}_i^{(L)} - \sum_{a<L} \mathbf{x}_i^{(a)} \rangle).$$

We approximate the inner sum with only one term by 1) concentration of softmax to $1/L$, 2) the assumption $\sum_{l'=2}^{L-1} \|\boldsymbol{\xi}^{(l')}\|_1 = o(\|\boldsymbol{\xi}^{(l')}\|_L)$, which implies the dominance of $\langle \mathbf{v}, \boldsymbol{\xi}_i^{(L)} \rangle$ after stage I, and the dominance of $\langle \mathbf{w}_{K,s}^{(0)}, \boldsymbol{\xi}_i^{(L)} \rangle$. The approximation for the update of key noise is essentially the same, thus ensuring the alignment between the largest query and the largest key, thus extending our result for $L = 2$. Subsequently, due to the gap in the $L_1$ norm, the softmax at the position of the largest noise patch will converge to 1, while the others will go to zero.

## F.3 Extension to Joint Training of Linear Layer

In this section, we talk about how to extend our main theoretical results, which fix the second linear layer, to the joint training of softmax attention layer and linear layer.

Firstly, we would like to remark that the value matrix in our model setting already acts as a linear layer while many works Tarzanagh et al. (2023a;b) on analyzing training dynamics of transformers fix the value matrix. In our specific case, we can extend the current theory to the joint training of the softmax attention layer and linear layer.

We can write the gradient with respect to the parameters in the second layer as follows:

$$\nabla_{\theta_{j,r}} L_S(W) = \frac{1}{n} \sum_{i=1}^{n} y_i \ell'_i j[(s_{i,11} + s_{i,21}) \langle \mathbf{w}_{V,j,r}, y_i \boldsymbol{\mu} \rangle + (s_{i,12} + s_{i,22}) \langle \mathbf{w}_{V,j,r}, \boldsymbol{\xi}_i \rangle],$$

where $\theta_{j,r}$ denotes the parameter for the second layer $F_j$ at the $r$-th entry, initialized with $j/m_v$, for $1 \le r \le m_v$. During stage I, it moves at most $T_1 \eta$ which is negligible compared to the initialization $1/m_v$, and will not affect our analysis for value in stage I. After stage I, the terms $\langle \mathbf{w}_{V,j,r}, jy_i \boldsymbol{\xi}_i \rangle$ become positive for all $1 \le i \le n$ and larger than the value signal term $\langle \mathbf{w}_{V,j,r}, \boldsymbol{\mu} \rangle$, which leads to that $\mathrm{sgn}(\nabla_{\theta_{j,r}} L_S(\mathbf{W})) = -j$. Consequently, all $\theta_{1,r}$ keep increasing while all $\theta_{-1,r}$ keep decreasing. Until then, all $\theta_{j,r}$'s sign are the same as initialization and will not change during the rest of training. Therefore, it will not affect the analysis for other parameters.

In summary, the key to the joint training extension lies in the different magnitudes of initialization between the second layer weights and the value matrix, which is $1/m_v$ and $\sigma_0$, respectively. When value converges to pattern specified in stage I, the second layer weights have almost no change. But after stage I, the sign of gradient of second layer become fixed and aligned with its initialized sign, so that it will have no effect on other parameters.

### F.4 EXTENSION TO MULTI-HEAD ATTENTION

In this section, we discuss how to extend our main theoretical results, which focus on a single-head attention layer, to a multi-head attention layer. We note that our theoretical results can be easily extended to multi-head attention.

For a multi-head attention layer, let the parameters be $\mathbf{W} := \{\mathbf{W}_{Q,h}, \mathbf{W}_{K,h}, \mathbf{W}_{V,j,h}\}_{h=1}^{H}$, where $\mathbf{W}_{Q,h}, \mathbf{W}_{K,h} \in \mathbb{R}^{m_k \times d}$ and $\mathbf{W}_{V,j,h} \in \mathbb{R}^{m_v \times d}$ for $j \in \{\pm 1\}$ and $h \in [H]$. Here, the $H$ is the number of attention head, assumed to be a fixed constant. Then, the network can be written as $f(\mathbf{W}, \mathbf{X}) := F_1(\mathbf{W}, \mathbf{X}) - F_{-1}(\mathbf{W}, \mathbf{X})$, where $F_1(\mathbf{W}, \mathbf{X})$ and $F_{-1}(\mathbf{W}, \mathbf{X})$ are defined as:

$$F_j(\mathbf{W}, \mathbf{X}) := \sum_{h=1}^{H} F_{j,h}(\mathbf{W}, \mathbf{X}),$$

$$F_{j,h}(\mathbf{W}, \mathbf{X}) := \frac{1}{m_v} \sum_{l=1}^{L} \mathbf{1}_{m_v}^{\top} \mathbf{W}_{V,j,h} \mathbf{X} \text{softmax} \left( \mathbf{X}^{\top} \mathbf{W}_{K,h}^{\top} \mathbf{W}_{Q,h} \mathbf{x}^{(l)} \right).$$

**Gradients in Multi-Head Attention.** Regarding the gradients, the partial derivatives of the single-head model outputs with respect to the parameters, i.e., $\frac{\partial F_{j,h}}{\partial \mathbf{W}_{Q,h}}$, $\frac{\partial F_{j,h}}{\partial \mathbf{W}_{K,h}}$, $\frac{\partial F_{j,h}}{\partial \mathbf{W}_{V,h}}$, remain unchanged. However, the gradient of the loss with respect to the single-head model outputs, i.e., $\frac{\partial \ell}{\partial F_{j,h}}$, does change. This change, however, is linear and straightforward to analyze. Intuitively, the model outputs increase approximately $H$ times, causing the magnitude of the loss derivatives $\ell'$ to decrease accordingly.

Formally, our theory shows that in single-head attention, the loss derivative are close to initialization up to $t = 4T_4^-$, where $4T_4^-$ is the time when the sign alignment of negative query noise completes. Specifically, for all $i \in [n]$, $t \leq 4T_4^-$, we have

$$\ell_i'^{(t)} := \frac{\partial \ell}{\partial f(\mathbf{W}^{(t)}, \mathbf{X}_i)} = 1/2 \pm o(1),$$

This implies $f(\mathbf{W}^{(4T_4^-)}, \mathbf{X}_i) = o(1)$. Then, the $H$-fold increase in the multi-head model outputs does not alter this result, so the effect of changes in $\frac{\partial \ell}{\partial F_{j,h}}$ can be neglected.

Consequently, the behavior of signals and noise still follows the four-stage dynamics in the single-head case, with the dynamics of all attention heads being the same.

**Experiments.** In Figure 13, we plot the full dynamics of our simplified transformer but with 4 attention heads. We can see that the dynamics of query noise, key noise, query signals, and key signals are identicalto those in the single-head model (Figure 18). Additionally, the dynamics of the softmax outputs in each head are consistent. These empirical evidences further support that our theory holds in the multi-head attention models.

# G    MORE DETAILED EXPLANATIONS FOR THE FOUR-STAGE DYNAMICS

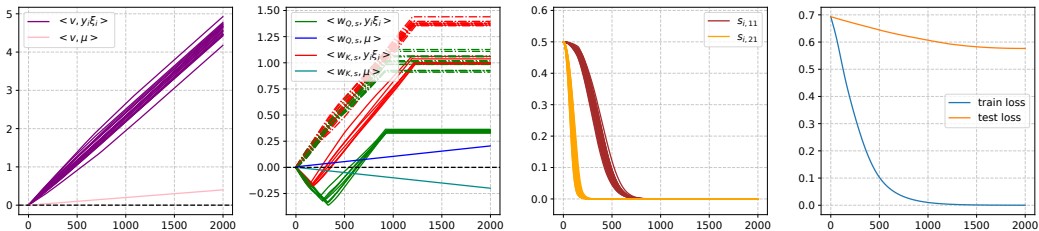

Figure 18: **The full training dynamics of two-layer transformers with SignGD.** This figure shows the dynamics of value, query, and key signals and noise, as well as softmax outputs and loss across all iterations. **(a)** Dynamics of value noise, and value signals. **(b)** Dynamics of query noise, key noise, query signals, and key signals. **(c)** Dynamics of softmax outputs. **(d)** Training and test loss curve. Omitted notations are the same as Fig. 1.

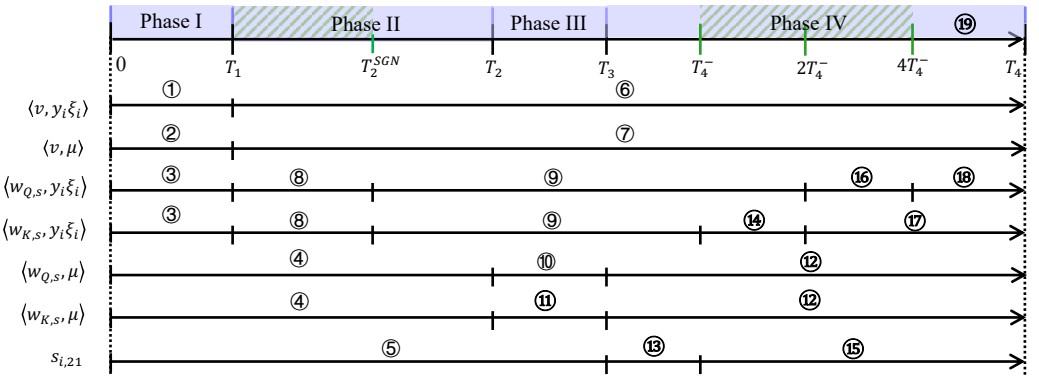

Figure 19: **Timeline of the four-stage training dynamics.** We mark all key points, including $T_1$, $T_2^{\text{SGN}}$, $T_2$, $T_3$, $T_4^-$, $T_4$, along with their corresponding key behaviors in the dynamics: ①: Mean value noise increases rapidly, becomes positive, and grows linearly with $t$. ②: Mean value signals remain stuck at their initialization. ③: Query and key noise are stuck at initialization. ④: Query and key signals are stuck at initialization. ⑤: Noise-signal softmax outputs are stuck at initialization. ⑥: Mean value noise continuously increases. ⑦: Mean value signals are negligible compared to mean value noise. ⑧: Query noise and key noise align their sign with each other. ⑨: Sign alignment completes, query noise and key noise continuously increase, and grow linearly with $t$. ⑩: Query signals leave the initialization, align their sign with the majority of query noise, and grow linearly with $t$. ⑪: Key signals leave the initialization, align their sign with the minority of query noise, and grow linearly with $t$. ⑫: Query signals and key signals continuously increase. ⑬: Noise-signal softmax outputs decrease exponentially. ⑭: Positive key noise continuously increases. Negative key noise flip the update direction, and align with query signals. ⑮: Noise-signal softmax outputs reach $o(1)$ and remain sufficiently small. ⑯: Positive query noise continuously increase; Negative query noise flips update direction and aligns with query signals. ⑰: Negative key noise completes sign alignment, and all key noise continuously increases. ⑱: Negative query noise completes sign alignment, and all query noise continuously increases. ⑲: All signals and noise continuously increase, and the loss decreases exponentially. Note that by "increase", we refer to an increase in magnitude.

## H    DISCUSSION ON DIFFERENCES FROM REAL-WORLD SETUP

We would like to discuss the following key differences between our theoretical setups and the transformers used in real-world applications:

**The gap in the data noise structure.**

- The data model considered in our study is simplified — we assume Gaussian noise that is i.i.d. across data samples and independent of the true features. All these simplifications inevitably create a gap from real-world datasets. Additionally, our data settings are motivated by image datasets. For language data, since language tokens have more dense semantic information, and since the noise must be discrete, the noise in language data could be more structured compared to the image data.

- These differences are acknowledged in the Limitation section of our initial manuscript, and make it difficult to compare the noise in our data model and real-world language or image datasets. However, we would like to highlight again that even in this simplified data model, significant challenges present in the theoretical analysis.

**The gap in tasks regarding optimization.**

- The sensitivity of SignGD to noise, as demonstrated in our findings, is relative to GD and is observed in a task where both optimizers achieve perfect convergence. In contrast, real-world language and vision tasks using transformers often reveal that GD struggles to optimize effectively, with a significant training loss gap compared to Adam (Zhang et al., 2024c).

- This highlights a gap between our task and real-world tasks in terms of optimization. This gap in tasks regarding optimization may also affect the generalization performance and robustness of the optimization methods. Additionally, as GD cannot optimize effectively in real-world transformers, it is difficult to fairly compare the generalization properties of GD and SignGD. Understanding why GD fails to optimize transformers on real-world tasks should be an important prerequisite.

- However, we would like to emphasize that, although our task optimizes easily, our work provides a clear example where SignGD, and by extension Adam, fails in learning transformers, and hence is not always more effective than GD.

- The reason behind this gap: This gap in the optimization may stem from factors such as simplified data structures and models in our setups. However, since our experiments on two-layer and three-layer transformers demonstrate consistent results regarding training dynamics (See Appendix B.6), we suspect this gap is more attributable to differences in data.

**How to define robustness in the real-world transformers, particularly language models.**    The concept of robustness in real-world language models requires careful consideration. In practice, when training transformers on language tasks with Adam, training instability and loss spikes are often observed, usually due to low-quality data batches (Chowdhery et al., 2023). Given that transformer training typically involves billions of tokens, the generalization performance is often closely tied to the training performance. Thus, from this viewpoint, training instability and loss spikes can reasonably be viewed as a form of non-robustness. The message we aim to convey is: To fairly connect our theoretically motivated findings to real-world applications, it is crucial to carefully define what robustness means in practical transformers like language models.

In the above, we discuss the gap from the perspectives of data noise, optimization complexity, and the definition of robustness. We hope this discussion helps clarify the differences between our theoretical findings and real-world scenarios.

