# OpenReview forum: "On the Optimization and Generalization of Two-layer Transformers with Sign Gradient Descent"
_ICLR.cc/2025/Conference — ICLR 2025 Spotlight_

### Official Review · Reviewer_YN62 · 2024-11-01

**Soundness:** 3
**Presentation:** 4
**Contribution:** 3
**Rating:** 8
**Confidence:** 4

**Summary:**

This manuscript provides a deep theoretical analysis of the training dynamics of a simplified two-layer transformer using signSGD on a synthetic dataset. It explicitly identifies four complex but distinct stages in the training dynamics. Furthermore, it empirically uncovers that Adam exhibits similar behavior under the same conditions as signSGD. The analysis demonstrates that while signSGD typically converges quickly, it generalizes poorly and requires high-quality data compared to SGD.

**Strengths:**

To the best of my knowledge, this manuscript is the first to theoretically analyze the detailed training dynamics of the transformer and softmax attention layer with trainable query-key parameterization using signSGD. It breaks down the process into four stages, capturing the complex and subtle behaviors, which deepens our theoretical understanding of the optimization process for Transformers.

**Weaknesses:**

- Some assumptions in the analysis are overly strong and unrealistic. For example, the manuscript assumes that the data dimension satisfies  $ d = \Omega(\text{poly}(n)) $, and that the variance of noise satisfies $\sigma_p = \Omega(d^{\frac{1}{4}} n^3)$, where $ n $ is the number of data samples, which is typically much larger in practice.

- Although the manuscript attempts to analyze the complex softmax attention with trainable query-key parameterization, it simplifies the attention block to a single-head variant rather than using the original multi-head version. Additionally, the formulation of $ v $ is somewhat unusual, defined as $v = \bar{w} _{v,1} - \bar{w} _{v,-1}$ which undermines the theoretical insights in relation to real-world scenarios.

- While the empirical results validate the theoretical analyses for training a two-layer transformer using signSGD, it would be valuable to investigate whether the empirical training dynamics of a multi-layer transformer, or even the original transformer, align with the theoretical findings from the simplified two-layer model.  I would like to see whether the results  can extend to deeper transformers, or implement more experiments to test if the key behaviors you identified persist in more complex models.

**Minor Issues**
- At Line 168, the variable $ i $ in the equation is not explicitly defined, though it can be inferred that $ i$ represents the index of data samples.

- In Figure 2, the manuscript suggests distinct differences between the training dynamics of signSGD and Adam, yet does not provide sufficient explanations. For instance, in (a) and (b), it is noted that the negative term $\langle w_{Q,s}, y_i \epsilon_{i} \rangle$ for signSGD ultimately becomes positive, while for Adam, it remains consistently negative. In (c), signSGD converges at a linear rate, whereas Adam converges at a sublinear rate. I think it needs to provide more detailed explanations for these differences, and to discuss their implications for understanding Adam's behavior in practice.

**Questions:**

Please see the weaknesses.

---

> ### Author Response · Authors · 2024-11-21
> **Response to Reviewer YN62 (1/3)**
>
> We thank the reviewer for the thoughtful comment and valuable feedback. Regarding your questions:
>
> ## W1: Assumptions about $d$ and $\sigma_p$ in the analysis are overly strong and unrealistic.
>
> For the assumption $d = \text{poly}(n)$:
> - We need a large enough $d$ to make the network overparameterized. Note that our hidden widths $m_v$ and $m_k$ are less than $n$, so a large $d$ is the only assumption we make for overparameterization. A large $d$, and hence overparameterization, is necessary to obtain concentration results and to show perfectly fitting on the training data.
>
> - The use of a large $d$ originates from the analysis of signal-noise models and is standard. Similar assumptions on $d$ are made in the literature [1, 2].
>
> For the assumption about $\sigma_p$:
> - There seems to be a typo in the review. The assumption we used is $\sigma_p = \Omega(d^{-1/4}n^3)$. The lower bound is $o(1)$ since $d = \text{poly}(n)$, which means our theory allows for small noise of $o(1)$.
>
>
>
> ---
>
> ## W2, part 1: Extension to multi-head attention
>
> Fortunately, we note that our theoretical results can be easily extended to multi-head attention. We discuss the extension below and have also included it in **Appendix F.4** of our updated manuscript.
>
>
> For a multi-head attention layer, let the parameters be  $W := (W_{Q,h}, W_{K,h}, W_{V,j,h})^H_{h=1}$, where $W_{Q,h}, W_{K,h} \in \mathbb{R}^{m_k \times d}$ and $W_{V,j,h} \in \mathbb{R}^{m_v \times d}$ for $j \in \{\pm 1\}$ and $h \in [H]$. Here, $H$ is the number of attention heads, assumed to be a fixed constant. Then, the network can be written as  $f(W, X) := F_{1}(W, X) - F_{-1}(W, X)$, where $F_{1}(W, X)$ and $F_{-1}(W, X)$ are defined as:
> $$
> F_j(W, X) := \sum_{h=1}^{H} F_{j,h}(W, X), \quad
> F_{j,h}(W, X) := \frac{1}{m_v} \sum_{l=1}^{L} \mathbf{1}^\top_{m_v}
> W_{V,j,h} X \text{softmax}(X^\top W_{K,h}^\top W_{Q,h} x^{(l)}).
> $$
>
> **Gradients in Multi-Head Attention.**
> Regarding the gradients, the partial derivatives of the single-head model outputs with respect to the parameters, i.e.,  $\frac{\partial F_{j,h}}{\partial W_{Q,h}}, \quad \frac{\partial F_{j,h}}{\partial W_{K,h}}, \quad \frac{\partial F_{j,h}}{\partial W_{V,h}}$, remain unchanged. However, the gradient of the loss with respect to the single-head model outputs, i.e.,  $
> \frac{\partial \ell}{\partial F_{j,h}}$, does change. This change, however, is linear and straightforward to analyze. Intuitively, the model outputs increase approximately $H$-fold, causing the magnitude of the loss derivatives $\ell^{\prime}$ to decrease accordingly.
>
> Formally, our theory shows that in single-head attention, the loss derivatives remain close to initialization up to $t = 4T_{4}^{-}$, where $4T_{4}^{-}$ is the time when the sign alignment of negative query noise completes. Specifically, for all $i \in [n]$ and $t \leq 4T_{4}^{-}$, we have:
> $$
> \ell_{i}^{\prime(t)} := \frac{\partial \ell}{\partial f(W^{(t)}, X_i)} = 1/2 + o(1).
> $$
> This implies $f(W^{(4T_4^-)}, X_i) = o(1)$. The $H$-fold increase in the multi-head model outputs does not alter this result, so the effect of changes in $\frac{\partial \ell}{\partial F_{j,h}}$ can be neglected.
>
> Consequently, the behavior of signals and noise still follows the four-stage dynamics observed in the single-head case, with the dynamics of all attention heads being the same.
>
> **Experiments.**
> In **Figure 12**, we plot the full dynamics of our simplified transformer with 4 attention heads. We can see that the dynamics of query noise, key noise, query signals, and key signals are identical to those in the single-head model (**Figure 18**). Additionally, the dynamics of the softmax outputs in each head are consistent.
>
> These empirical observations further support that our theory holds in multi-head attention models.

---

> ### Author Response · Authors · 2024-11-21
> **Response to Reviewer YN62 (2/3)**
>
> ## W3: Empirical results on more complex transformers (with multiple layers, MLP components, and/or residual connections)
>
> We have conducted additional experiments on deeper transformers using our synthetic dataset with SignGD, exploring various settings. Specifically, we extend our analysis to models with additional attention layers, MLP layers, and residual connections, which are essential components of modern transformer architectures. Since our theory primarily predicts the behavior of data-parameter inner products, for transformers with multiple attention layers, we focus on the dynamics of the first layer. **Our main finding is that the key behaviors identified by our theory do persist in multi-layer transformers with MLPs if residual connections are added.** Without residual connections, the dynamics could become wild.
>
> To examine how well the key behaviors identified by our theory persist in more complex models, we performed an ablation study. We provide the full dynamics of all relevant quantities in **Figures 13 and 14** and augment these results with Tables 4-11, which illustrate the sign alignment behavior during Stage II. **We include this part in Appendix B.6 in the revised manuscript.**
>
> **Transformers with Residual Connections.**
> Firstly, on transformers with residual connections, across all model configurations we tested—including 2-layer transformers without MLPs, 3-layer transformers without MLPs, 2-layer transformers with MLPs, and 3-layer transformers with MLPs—we observe the following behaviors, consistent with our theoretical predictions:
> - Stage I: Value noise increases faster than query and key noise, and the value signal remains small relative to the value noise.
> - Stage II: Query and key noise exhibit sign alignment behavior early in training.
> - Stage III: The query and key signals have opposite signs, determined by query (and key) noise via a majority-voting mechanism.
> - Stage IV: Noise-feature softmax outputs firstly decay and decay exponentially, and both negative query and key noise align with the query signal.
>
> However, we remark that in more complex models, the final alignment observed in Stage IV—i.e., the flip of negative query and key noise—often halts midway. This phenomenon becomes more pronounced with the addition of MLP layers, where the final alignment stops earlier. We attribute this behavior to the \textit{rapid shrinking of query and key gradients}. This is partly driven by the decay of softmax outputs (as shown in Lemma D.7). Furthermore, as the number of layers increases and/or MLP layers are introduced, additional layers significantly contribute to this gradient shrinkage, as illustrated in the last column of Figure 13.
> It is worth noting that this gradient shrinking is a numerical precision issue unrelated to our theory. In theory, the sign operation maps gradients to ±1 regardless of their magnitude. However, in practice, extremely small gradients are rounded to zero, disrupting the alignment process. Despite this, we conclude that the key behaviors predicted by our theory persist in deeper transformers with residual connections.
>
> **Transformers without Residual Connections.**
> On the other hand, in deeper transformers lacking residual connections, the dynamics become erratic. While some short-term behaviors (e.g., sign alignment between query and key noise in Stage II, and the opposing signs between query and key signal) are preserved (see Tables 8-11, and Figure 14), long-term behaviors deviate significantly from theoretical predictions. For instance:
> - Feature-feature softmax outputs start to increase instead of decreasing.
> - The dynamics of positive key noise become non-monotonic.
> - Value noise exhibits irregular patterns rather than increasing consistently.
>
> Additionally, we remark that the training dynamics of transformers without residual connections are less stable and more irregular compared to those with residual connections. This instability may be linked to the phenomenon of rank collapse in transformers, as discussed in prior works [5, 6].
>
> Based on these findings, we conclude that the key behaviors predicted by our theory persist in deeper transformers with residual connections. Without the residual connections, the key behaviors outlined in our theory are only partially preserved.

---

> ### Author Response · Authors · 2024-11-21
> **Response to Reviewer YN62 (3/3)**
>
> ## W2, part 2: Clarification on the formulation of v
>
> - In our model definition, we average over the value dimension $m_v$ to output an one-dim scalar for classification. As a result, the model outputs during the forward pass depend on $w_{v,j,r}$ only through $v$. We remark that this type of model definition is common in theoretical analyses of classification tasks, including works on transformers [3, 4], and CNNs. In our theory, the specific form of v comes from the fact that half of the parameters in the fixed linear head are 1/m_v, and the other half is -1/m_v.
> - This model definition has implications for the backward process as well. Specifically, all $w_{v,1,r}$ has exactly the same gradient, and the gradients of $w_{v,-1,r}$ are exactly the negatives of those for $w_{v,1,r}$, which is given in Lemma D.6 of the appendix. Consequently, all $w_{v,j,r}$ for a given j update in the same direction and with the same step size, which corresponds to how  $v$  changes (or half of it).
> - Therefore, from the viewpoint of theoretical analysis, one benefit of the $v$-formulation is that:
> both in the forward and backward processes, analyzing $v$ is sufficient to understand $w_v$. The update direction and magnitude of $w_v$ can be directly inferred from the dynamics of $v$.
>
> ---
>
> ## W5: Explanations for differences between SignGD and Adam
>
> Although SignGD can serve as a proxy for understanding Adam, our experiments reveal notable differences between the two. In Figure 2(a) and 2(b), SignGD causes the negative query to eventually become positive, whereas it remains negative with Adam. Additionally, in Figure 2(c), the training loss of SignGD converges linearly, while Adam exhibits sublinear convergence. While we previously suggested that these differences might arise from Adam’s momentum term, we did not provide detailed evidence. Here, we try to explain these differences in terms of training dynamics and convergence rates. **We also include this part in Appendix B.7 in the revised manuscript.**
>
> To investigate factors influencing Adam’s behavior, we vary its $\beta$ parameters and conduct experiments under the same model and dataset as in Figure 2. In **Figure 2**, we observe that $\beta_1$ values ranging from 0 (no first moment) to 0.9 (commonly used in practice) do not significantly impact training speed. Similarly, in **Figure 6** of Appendix B.3, changes in $\beta_1$ have little effect on training dynamics. Thus, our focus shifts to the role of $\beta_2$.
>
> **Convergence rate.**
> In **Figure 15**, we observe that when $\beta_2 > 0.9$, the training loss exhibits a sublinear convergence rate. We remark that when the $\beta_2 < 0.9$, the loss curve closely resembles that of SignGD, thus we use a range of $[0.9, 0.999]$ for $\beta_2$.
> Since the training loss convergence is primarily driven by the growth of mean value noise, we believe this behavior can be approximated through the analysis of a linear model fitting the noise.
>
> **Training Dynamics.**
> **Figure 16** (first row) shows that only small values of $\beta_2$ prevent the negative query noise from turning positive. As $\beta_2$ increases, the dynamics become smoother, and the evolution of query noise halts earlier.
>
> To understand this, we examine the mean gradient and update magnitude in the second and last rows of Figure 16. Unlike multi-layer transformers, the query and key gradients do not shrink faster. Instead, Adam’s update magnitude for query parameters decays to zero before the gradients approach zero. This early decay of the update magnitude (or effective step size) can be attributed to $\beta_2$.
> As $\beta_2$ increases, the update magnitude decreases earlier, while the gradient shrinkage occurs at the same point.
>
> These observations suggest that $\beta_2$ plays a crucial role in both the convergence rate and training dynamics of Adam, highlighting key differences from SignGD.
>
> ---
>
> ## W4: Missed definitions at line 168
> Thank you for your kind reminder. We have revised the manuscript and defined the variable $i$ at line 168.
>
>
> [1] Cao et al., Benign Overfitting in Two-layer Convolutional Neural Networks, 2022.
>
> [2] Allen-Zhu et al., Towards Understanding Ensemble, Knowledge Distillation and Self-Distillation in Deep Learning, 2023
>
> [3] Li et al., A Theoretical Understanding of Shallow Vision Transformers: Learning, Generalization, and Sample Complexity, 2023.
>
> [4] Jiang et al., Unveil Benign Overfitting for Transformer in Vision: Training Dynamics, Convergence, and Generalization, 2024.
>
> [5] Dong et al., Attention Is Not All You Need: Pure Attention Loses Rank Doubly Exponentially with Depth, 2021.
>
> [6] Noci et al., Signal Propagation in Transformers: Theoretical Perspectives and the Role of Rank Collapse, 2022.

---

> > ### Comment · Reviewer_YN62 · 2024-11-25
> >
> > Thank you for providing detailed responses to my comments and for conducting additional experiments. My concerns have been thoroughly addressed, and I have decided to raise my score accordingly.

---

### Official Review · Reviewer_hqL4 · 2024-11-04

**Soundness:** 3
**Presentation:** 3
**Contribution:** 3
**Rating:** 6
**Confidence:** 3

**Summary:**

The paper investigates the optimization and generalization properties of Sign Gradient Descent (SignGD) for transformers, focusing on binary classification tasks involving noisy, linearly separable data. The authors claim that SignGD is an effective surrogate for the Adam optimizer. They identify four stages in the training dynamics of SignGD and observe that it achieves fast convergence in training loss but generalizes poorly, especially on noisy data.

**Strengths:**

The paper provides a multi-stage framework to analyze the transformer training using SignGD, making this complex behaviour into small interpretable stages.

By establishing that SignGD is a proxy for Adam, the paper is capable of offering new perspectives on the reasons why Adam present some generalization problems.

The combination of theoretical proofs and experimental validation strengthens the proposed analysis and its overall message.

**Weaknesses:**

The main weakness of the method is its limited applicability to real-world scenarios. The reliance on assumptions such as linearly separable data and the use of only two-layer transformers restricts its effectiveness when dealing with more complex datasets and modern, state-of-the-art transformer architectures.

**Questions:**

Can the framework be extended to deeper transformers or multi-head attention mechanisms? I ask that because no method is currently using two layers transformer, so its extension to SOTA networks could provide broader applicability.

From the current work what strategies could the authors envision can be used to improve generalization of SignGD and consequently Adam in data noisy settings? Identifying such strategies would be valuable for increasing robustness in real-world applications, where noise is often unavoidable.

---

> ### Author Response · Authors · 2024-11-21
> **Response to Reviewer hqL4 (1/2)**
>
> We thank the reviewer for the thoughtful comment and valuable feedback. Regarding your questions:
>
> ## Q1: Extension to deeper networks and multi-head attention
>
> ### **Extension to deeper networks:**
>
> - While we acknowledge that our current theory does not extend to deeper transformers, we would like to kindly highlight that, from a theoretical perspective, analyzing deeper networks is a highly complex and challenging task.
> - We also would like to clarify that our theoretical setting for learning transformers is more practical and challenging compared to existing literature. Many works on gradient-based optimization analysis for transformers use linear attention (e.g., [1, 2]) or combined query-key parameterization (e.g., [3, 4]) to simplify the analysis. In contrast, our work employs softmax attention and trainable query-key parameterization, which are more aligned with practical implementations but significantly increase the complexity of the analysis. The softmax activation and the intertwined dynamics of query and key parameters make this setting highly challenging and non-trivial to analyze.
> - Additionally, even for simpler models such as MLPs, extending analysis to deeper networks often necessitates non-standard and less practical training strategies, such as layer-wise training [5, 6].
> - We sincerely believe that extending our theory to deeper networks is an important and meaningful direction for future research, and we are optimistic that with further effort, these challenges can be addressed.
>
>
> ### **Extension to multi-head attention:**
>
> Fortunately, we note that our theoretical results can be easily extended to multi-head attention. We discuss the extension below and have also included it in **Appendix F.4** of our updated manuscript.
>
>
> For a multi-head attention layer, let the parameters be  $W := (W_{Q,h}, W_{K,h}, W_{V,j,h})^H_{h=1}$, where $W_{Q,h}, W_{K,h} \in \mathbb{R}^{m_k \times d}$ and $W_{V,j,h} \in \mathbb{R}^{m_v \times d}$ for $j \in \{\pm 1\}$ and $h \in [H]$. Here, $H$ is the number of attention heads, assumed to be a fixed constant. Then, the network can be written as  $f(W, X) := F_{1}(W, X) - F_{-1}(W, X)$, where $F_{1}(W, X)$ and $F_{-1}(W, X)$ are defined as:
> $$
> F_j(W, X) := \sum_{h=1}^{H} F_{j,h}(W, X), \quad
> F_{j,h}(W, X) := \frac{1}{m_v} \sum_{l=1}^{L} \mathbf{1}^\top_{m_v}
> W_{V,j,h} X \text{softmax}(X^\top W_{K,h}^\top W_{Q,h} x^{(l)}).
> $$
>
> **Gradients in Multi-Head Attention.**
> Regarding the gradients, the partial derivatives of the single-head model outputs with respect to the parameters, i.e.,  $\frac{\partial F_{j,h}}{\partial W_{Q,h}}, \quad \frac{\partial F_{j,h}}{\partial W_{K,h}}, \quad \frac{\partial F_{j,h}}{\partial W_{V,h}}$, remain unchanged. However, the gradient of the loss with respect to the single-head model outputs, i.e.,  $
> \frac{\partial \ell}{\partial F_{j,h}}$, does change. This change, however, is linear and straightforward to analyze. Intuitively, the model outputs increase approximately $H$-fold, causing the magnitude of the loss derivatives $\ell^{\prime}$ to decrease accordingly.
>
> Formally, our theory shows that in single-head attention, the loss derivatives remain close to initialization up to $t = 4T_{4}^{-}$, where $4T_{4}^{-}$ is the time when the sign alignment of negative query noise completes. Specifically, for all $i \in [n]$ and $t \leq 4T_{4}^{-}$, we have:
> $$
> \ell_{i}^{\prime(t)} := \frac{\partial \ell}{\partial f(W^{(t)}, X_i)} = 1/2 + o(1).
> $$
> This implies $f(W^{(4T_4^-)}, X_i) = o(1)$. The $H$-fold increase in the multi-head model outputs does not alter this result, so the effect of changes in $\frac{\partial \ell}{\partial F_{j,h}}$ can be neglected.
>
> Consequently, the behavior of signals and noise still follows the four-stage dynamics observed in the single-head case, with the dynamics of all attention heads being the same.
>
> **Experiments.**
> In **Figure 12**, we plot the full dynamics of our simplified transformer with 4 attention heads. We can see that the dynamics of query noise, key noise, query signals, and key signals are identical to those in the single-head model (**Figure 18**). Additionally, the dynamics of the softmax outputs in each head are consistent.
>
> These empirical observations further support that our theory holds in multi-head attention models.

---

> ### Author Response · Authors · 2024-11-21
> **Response to Reviewer hqL4 (2/2)**
>
> ## Q2: Potential strategies to increase robustness for SignGD and Adam in real-world applications
>
> Thank you for raising this concern. We would like to emphasize that the main contribution of our work is on theoretical analysis, but we are happy to discuss the potential applications and further insights conveyed by our theory.
>
> **Firstly, We would like to clarify our main contribution.**
> We would like to emphasize that the main contribution of our work is the theoretical analysis of the optimization and generalization of SignGD. We provably characterize the optimization dynamics and poor generalization (or say, non-robustness to noise) in a noisy dataset.
> The experiments conducted provide evidence that our theory hold in more general settings
> - We observed that the four-stage behaviors could extend to more complex models (e.g., transformers with multiple layers, multi-head attention and/or MLP components, as shown in Appendix B.5, B.6), and extend to more complex optimizers (e.g. Adam, as shown in Appendix B.7) on our theoretical data model.
> - The non-robustness to data noise of SignGD is shown in the real-world MNIST datasets.
> These experimental validation indicates that our results possess a certain degree of generality and could offer valuable insights into real-world tasks.
>
> **Potential applications inspired by our theory and findings.**
>
> We propose two potential applications based on our theory and findings:
> 1. Mixed Optimization Strategy. Our theory and experiments show that while GD is relatively more robust, SignGD and Adam achieve faster optimization. To leverage the strengths of both approaches, a mixed or adaptive optimization strategy can be used during training. For instance, Adam could be employed in the early stages to accelerate the decay of training loss, and the optimizer could switch to GD in the middle or final stages to enhance stability and improve robustness.
> 2. Data pruning. In data pruning, researchers typically quantify the importance of each training data point for model generalization, with many metrics relying on checkpoints or trained ensembles [7, 8]. If these checkpoints are obtained using different optimizers, the resulting data pruning metric can vary. By incorporating multiple optimizers into the process, we can mitigate bias introduced by relying on a single optimization method, leading to a more robust and reliable data pruning metric.
>
> These examples illustrate how insights from our work could inspire real-world applications.
>
>
> [1] Zhang et al., Trained transformers learn linear models in-context, 2024
>
> [2] Kim et al., Transformers learn nonlinear features in context: Nonconvex mean-field
> dynamics on the attention landscape, 2024
>
> [3] Tarzanagh et al., Transformers as support vector machines, 2023
>
> [4] Tian et al., Scan and snap: Understanding training dynamics and token composition in 1-layer transformer, 2023
>
> [5] Nichani et al., Provable Guarantees for Nonlinear Feature Learning in Three-Layer Neural Networks, 2023
>
> [6] Wang et al., Learning Hierarchical Polynomials with Three-Layer Neural Networks, 2024
>
> [7] Paul et al., Deep Learning on a Data Diet: Finding Important Examples Early in Training, 2021
>
> [8] Agarwal et al., Estimating Example Difficulty using Variance of Gradients, 2022

---

> ### Author Response · Authors · 2024-11-26
> **Looking Forward to Your Feedback**
>
> Dear Reviewer hqL4,
>
> Thank you very much for your thoughtful feedback and valuable insights. We hope that our responses address the concerns you raised. If you have any further questions or suggestions, we warmly welcome further discussion. We would greatly appreciate it if you would consider raising the score for our work.
>
> Thank you once again for your time and effort.
>
> Best regards, \
> Authors of submission 10289

---

> > ### Comment · Reviewer_hqL4 · 2024-11-30
> >
> > Thank you for the clarification. I've raised my score.

---

### Official Review · Reviewer_w8BA · 2024-11-04

**Soundness:** 3
**Presentation:** 2
**Contribution:** 3
**Rating:** 8
**Confidence:** 2

**Summary:**

This paper studies the learning dynamics of two-layer transformers optimized with SignGD. It introduces a theoretical framework that categorizes the learning process into four distinct stages, providing an analytical tool to study these dynamics. The study demonstrates that while SignGD achieves fast convergence of the transformer model, it leads to poor generalization on noisy datasets.

**Strengths:**

1. The paper offers a comprehensive four-stage analysis of transformer training dynamics, which is a promising contribution to understanding the behavior of transformers.

2. The demonstration of fast convergence but poor generalization with SignGD on noisy data is important. These findings are beneficial for practical applications as they guide the selection of optimizers and the preparation of datasets in real-world tasks.

3. The theoretical approach is novel.

**Weaknesses:**

1) The paper claims that both SignGD and Adam require high-quality data to perform well. However, an apparent contradiction arises in Figure 2(d), where the model with less noisy data (SNR=1.0) performs worse than that with noisier data (SNR=0.5). This discrepancy challenges the theoretical claims and needs to be addressed or explained to validate the model’s consistency across different noise levels.

2)  The description of the training dynamics in Section 3.1 lacks clarity, making it difficult to follow through the stages. It would be beneficial to explicitly map the training steps of the analysis to each stage in the diagrams (e.g., steps 1-2 corresponding to Stage 1). Additionally, providing data plots before and after switching steps of different stages, such as providing the mean noise values before and after the transition steps for 1st stage and 2nd stage in Fig 1(a), would greatly enhance understanding.

3) In many real-world applications of Adam in transformer, such as those used in vision and language tasks, often exhibit robustness to noise in practical applications, which is contrary to the findings presented. Understanding whether this discrepancy is due to lower real-world noise levels or other factors that contradict the assumptions made in the study is essential for aligning theoretical insights with empirical observations.

**Questions:**

1. In Figure 2(d), the test loss for the less noisy case (SNR=1.0) is higher than for the noisier case (SNR=0.5) when using SignGD, which contradicts theoretical expectations about the benefits of higher-quality data. Could you provide an analysis of the noise characteristics or data distribution across different SNR levels that might explain this behavior? Additional experiments or analyses could also be valuable in investigating this phenomenon further. For instance, it’s possible that entirely clean data may reduce generalization performance compared to slightly noisy data, though too much noise also significantly hinders generalization.

2. Clarity and Detail in Training Dynamics Analysis (Section 3.1): Could you offer a clearer mapping between training steps and each stage of the training dynamics? More detailed explanations of the transitions between each of the four stages would be helpful. Additionally, including data plots that capture the state of relevant variables before and after critical transition points would improve clarity (rather than focusing solely on within-stage details, like in Figure 1(a)).

3. Real-world applications of Transformers, such as language modeling, demonstrate robustness to noise. How do the noise levels assumed in your theoretical model compare to those typically encountered in practical Transformer applications? Could you discuss any factors or experiments that might explain the gap between your theoretical findings and the observed robustness of Transformers to noise in real-world scenarios?

---

> ### Author Response · Authors · 2024-11-21
> **Response to Reviewer w8BA (1/2)**
>
> We thank the reviewer for the thoughtful comment and valuable feedback. Regarding your questions:
>
> ## W1 + Q1: High test loss of SignGD in less noisy case in Figure 2(d)
>
> We carefully re-examined the experiments presented in Figure 2(d) and identified issues with our original experimental settings. Specifically, we used only 128 training samples to train the two-layer transformers with deterministic optimizers and evaluated the model on the entire test dataset. In this scenario, even in a noiseless setting, SignGD tends to overfit to the training data, resulting in poor generalization. This phenomenon arises due to the effect of empirical risk minimization (ERM). In our theoretical analysis, we assume identical features in both training and test datasets, which implicitly avoids the effect of ERM, hence the observed results do not contradict our theory.
>
> We have re-run the MNIST experiments using 2000 training samples with GD, SignGD, and Adam. In this updated setting, all optimizers achieve good generalization at SNR=1. Consistently, we observe that as SNR decreases, the test loss for SignGD and Adam increases more rapidly compared to GD, and GD consistently demonstrates better generalization across various levels of data noise. In all experiments, we ensured a training loss below 0.05.
>
> We have updated **Figure 2** and revised the experimental details in Appendix B.1. We sincerely thank your review again, which helped us identify and address this issue in our manuscript.
>
> ---
>
> ##  W2 + Q2: Clarity and detail in training dynamics analysis
>
> **Figure 17 added for clearer mapping and cross-stage details:**
>
> We have re-plotted Figure 1 to provide greater clarity and detail regarding the training dynamics. Due to space constraints, this updated figure is temporarily included as **Figure 17** in Appendix G. In future revisions, we plan to move it to Section 3.1 of the main text and revise the corresponding descriptions accordingly.
>
> Figure 17 provides a clearer mapping between training steps and each stage:
> - Stage I: \(t = 0\) to \(t = 2\)
> - Stage II: \(t = 2\) to \(t = 10\)
> - Stage III: \(t = 10\) to \(t = 40\)
> - Stage IV: \(t = 40\) to \(t = 2000\)
>
> The figure also includes detailed cross-stage dynamics for all relevant signals and noise:
> - **Figure 17 (a):** Dynamics of mean value noise and mean value signals in Stages I and II.
> - **Figure 17 (b):** Dynamics of key noise in Stages I and II.
> - **Figure 17 (c):** Dynamics of query noise, key noise, query signals, and key signals in Stages II and III.
> - **Figure 17 (d):** Dynamics of query noise, key noise, query signals, and key signals in Stages III and IV.
> - **Figure 17 (e):** Dynamics of softmax outputs across all stages (I–IV).
>
> **Additional revision for clarity:**
>
> We rewrote the text explaining Figure 1 in Section 3.1 of the original manuscript and included it in Appendix G, below Figure 17.
>
> We note that Figure 17 focuses on the details of the key behaviors identified by our theory, while some quantities with unchanged dynamics at certain stages may not be displayed. To address this, we also provide Figure 18, which illustrates the dynamics of all quantities across the full time horizon but inevitably lacks finer details.
> Additionally, we include **Figure 19** in Appendix G, as an illustration diagram, to provide a detailed explanation of the behaviors of all quantities across all stages.
>
> We recommend simultaneously reviewing Figures 17, 18, and 19 for a comprehensive understanding of the four-stage training dynamics.
> We hope these additions provide a clearer understanding of the four-stage dynamics.

---

> ### Author Response · Authors · 2024-11-21
> **Response to Reviewer w8BA (2/2)**
>
> ## W3+Q3: Discussion on Differences Between Our Theoretical Setups and Real-World Transformers
>
> Thank you for raising this concern. Your questions have helped us reflect on this distinction, and we are grateful for the chance to discuss it further. We would like to discuss the following key differences between our theoretical setups and the transformers used in real-world applications:
>
> **The gap in the data noise structure.**
> - The data model considered in our study is simplified – we assume Gaussian noise that is i.i.d. across data samples and independent of the true features. All these simplifications inevitably create a gap from real-world datasets. Additionally, our data settings are motivated by image datasets. For language data, since language tokens have more dense semantic information, and since the noise must be discrete, the noise in language data could be more structured compared to the image data.
> - These differences are acknowledged in the Limitation section of our initial manuscript, and make it difficult to compare the noise in our data model and real-world language or image datasets. However, we would like to highlight again that even in this simplified data model, significant challenges present in the theoretical analysis.
>
> **The gap in tasks regarding optimization.**
>
> - The sensitivity of SignGD to noise, as demonstrated in our findings, is relative to GD and is observed in a task where both optimizers achieve perfect convergence. In contrast, real-world language and vision tasks using transformers often reveal that GD struggles to optimize effectively, with a significant training loss gap compared to Adam [1].
> - This highlights a gap between our task and real-world tasks in terms of optimization.
> This gap in tasks regarding optimization may also affect the generalization performance and robustness of the optimization methods. Additionally, as GD cannot optimize effectively in real-world transformers, it is difficult to fairly compare the generalization properties of GD and SignGD. Understanding why GD fails to optimize transformers on real-world tasks should be an important prerequisite.
> - However, we would like to emphasize that, although our task optimizes easily, our work provides a clear example where SignGD, and by extension Adam, fails in learning transformers, and hence is not always more effective than GD.
> - **The reason behind this gap:** This gap in the optimization may stem from factors such as simplified data structures and models in our setups. However, since our experiments on two-layer and three-layer transformers demonstrate consistent results regarding training dynamics (See Appendix B.6), we suspect this gap is more attributable to differences in data.
>
>
> **How to define robustness in the real-world transformers, particularly language models.**
>
> The concept of robustness in real-world language models requires careful consideration. In practice, when training transformers on language tasks with Adam, training instability and loss spikes are often observed, usually due to low-quality data batches [2]. Given that transformer training typically involves billions of tokens, the generalization performance is often closely tied to the training performance. Thus, from this viewpoint, training instability and loss spikes can reasonably be viewed as a form of non-robustness.
> The message we aim to convey is: To fairly connect our theoretically motivated findings to real-world applications, it is crucial to carefully define what robustness means in practical transformers like language models.
>
> In the above, we discuss the gap from the perspectives of data noise, optimization complexity, and the definition of robustness. We hope this discussion helps clarify the differences between our theoretical findings and real-world scenarios.
>
>
> [1] Zhang et al., Why Transformers Need Adam: A Hessian Perspective, 2024
>
> [2] Chowdhery et al., Palm: Scaling language modeling with pathways, 2023

---

> > ### Comment · Reviewer_w8BA · 2024-11-25
> > **Post rebuttal comment**
> >
> > Thank you for your detailed response. The authors have addressed most of the concerns. I recommend including these discussions from the rebuttal in the paper. I consider this work to be a good contribution, and I will maintain my original score of 8 (accept).

---

> > > ### Author Response · Authors · 2024-11-26
> > >
> > > Thank you for your suggestion. We will incorporate these discussions into our subsequent revisions.

---

### Author Response · Authors · 2024-11-21
**Global Response**

Dear AC and reviewers,

We thank all the reviewers for the thoughtful reviews. We are excited that all the reviewers acknowledged the novelty and contribution of our theoretical analysis. We have individually responded to each reviewer and updated a revision of our paper. All changes are highlighted in **red** in the revised manuscript. Here, we provide a summary of our revisions to the manuscript.

- In **Appendix B.5** and **Appendix B.6**, we add **more experiments on more complex transformer models**, including transformers with multiple layers, multi-head attention, MLP components, and/or residual connections. The experiments verify that the key behaviors identified by our theory persist in multi-layer, MLP-augmented, multi-head attention transformers if residual connections are added. These experimental validations indicate that our theory possesses a certain degree of generality and could offer valuable insights into the optimization dynamics of real-world tasks.
- In **Appendix B.7**, we add **more experiments on Adam** to explore the differences between SignGD and Adam. In our settings, Adam differs from SignGD mainly in two aspects: (1) the convergence rate of the training loss and (2) the dynamics of negative query noise at the final stage. We provide more detailed explanations for the reasons behind these differences, specifically the role of $\beta_2$.
- In **Appendix F.4**, we discuss the **extension of our theory to multi-head attention**. Our theory can be easily extended to multi-head attention, with all characterized dynamics remaining unchanged. Empirical validation is provided in **Appendix B.5**.
- In **Appendix G**, we add **more figures to clarify our four-stage behaviors**. We add **Figure 17**, a refined version of Figure 1, to provide a clearer mapping between training steps and each stage and to show more cross-stage dynamics. We also add **Figure 18**, which illustrates the dynamics across the full time horizon to complement Figure 17. Finally, we include **Figure 19**, an illustrative diagram detailing the timeline and behavior of all quantities across all stages. We hope these figures and explanations bring more clarity to our theory's characterization of the training dynamics.
- We update **Figure 2** to make the experimental settings on MNIST more reasonable. In the updated figure, we clearly observe that the test loss for SignGD and Adam increases more rapidly compared to GD, and GD consistently demonstrates better generalization across various levels of data noise. All optimizers achieve good generalization at SNR=1.

We sincerely appreciate the reviewers’ valuable feedback and constructive suggestions, which helped improve the quality and presentation of our work. We are happy to provide further clarifications or address additional concerns to strengthen the understanding of our contributions. Thank you for your time and effort in reviewing our submission.

Best regards,
Authors of submission 10289

---

### Meta-Review · Area_Chair_Noaj · 2024-12-21

**Metareview:**

This work presents a theoretical analysis of the training dynamics of a simplified two-layer transformer using sign gradient descent on a synthetic dataset. They identify four stages in the training dynamics observe that it achieves fast convergence in training loss but generalizes poorly. The paper received a mix of reviews, but some reviewers raised their score after rebuttal. I recommended an acceptance.

**Additional Comments On Reviewer Discussion:**

The reviewers agreed that the authors' rebuttal addressed their concerns and raised their scores. All reviewers agree on acceptance in the end.

---

### Decision · Program_Chairs · 2025-01-22

Accept (Spotlight)